# Learning single-index models via harmonic decomposition

**Nirmit Joshi**
Toyota Technological Institute at Chicago
nirmit@ttic.edu

**Hugo Koubbi**[*]
CEREMADE, UMR 7534,
Université Paris Dauphine PSL, Paris , France
koubbi@ceremade.fr

**Theodor Misiakiewicz**
Department of Statistics and Data Science
Yale University
theodor.misiakiewicz@yale.edu

**Nathan Srebro**
Toyota Technological Institute at Chicago
nati@ttic.edu

## Abstract

We study the problem of learning single-index models, where the label $y \in \mathbb{R}$ depends on the input $\boldsymbol{x} \in \mathbb{R}^d$ only through an unknown one-dimensional projection $\langle \boldsymbol{w}_*, \boldsymbol{x} \rangle$. Prior work has shown that under Gaussian inputs, the statistical and computational complexity of recovering $\boldsymbol{w}_*$ is governed by the Hermite expansion of the link function. In this paper, we propose a new perspective: we argue that *spherical harmonics*—rather than *Hermite polynomials*—provide the natural basis for this problem, as they capture its intrinsic *rotational symmetry*. Building on this insight, we characterize the complexity of learning single-index models under arbitrary spherically symmetric input distributions. We introduce two families of estimators—based on tensor-unfolding and online SGD—that respectively achieve either optimal sample complexity or optimal runtime, and argue that estimators achieving both may not exist in general. When specialized to Gaussian inputs, our theory not only recovers and clarifies existing results but also reveals new phenomena that had previously been overlooked.

## 1   Introduction

Single-index models (SIMs)—also known as *generalized linear models*—are among the most widely studied models in statistics [20, 64, 49, 40]. They generalize linear regression by introducing nonlinearity through a one-dimensional projection of the input. Due to their simplicity and flexibility, SIMs have become foundational in both (semi)parametric statistics [44, 58, 26] and machine learning [52, 51, 21]. In recent years, SIMs have also emerged as prototypical models for exploring several key phenomena in modern high-dimensional learning, including: (1) *Statistical-to-computational gaps*, with close ties to problems such as phase retrieval [12, 66, 61, 62] and tensor PCA [67, 28]; (2) *Non-convex optimization*, including multi-phase dynamics [76, 10, 79, 15] and landscape concentration [65]; and (3) *Representation learning* in neural networks trained via gradient descent, where SIMs offer a simplified yet informative setting for studying feature learning [68, 18, 79, 30, 31, 6].

Spurred by this growing interest, a recent line of work has investigated the fundamental limits of learning SIMs in high dimensions under Gaussian assumption [12, 66, 61, 10, 28, 29, 21]. In this setting, referred to as the *Gaussian single-index model*, one observes i.i.d. samples $(y_i, \boldsymbol{x}_i) \sim \mathbb{P}_{\boldsymbol{w}_*}$, where

$$(y, \boldsymbol{x}) \sim \mathbb{P}_{\boldsymbol{w}_*}: \quad \boldsymbol{x} \sim \mathsf{N}(\boldsymbol{0}, \mathbf{I}_d) \quad \text{and} \quad y | \boldsymbol{x} \sim \rho(\cdot | \langle \boldsymbol{w}_*, \boldsymbol{x} \rangle), \tag{1}$$

---

[*]Part of this work was done while HK was a Visiting Assistant Researcher in the Department of Statistics and Data Science, Yale University.

39th Conference on Neural Information Processing Systems (NeurIPS 2025).

for an unknown unit vector $\boldsymbol{w}_* \in \mathbb{S}^{d-1}$ and a fixed link distribution $\rho \in \mathcal{P}(\mathcal{Y} \times \mathbb{R})$, modeling the pair $(Y, G)$ with $G \sim \mathsf{N}(0, 1)$. Thus, the label $y$ depends only on the one-dimensional projection $\langle \boldsymbol{w}_*, \boldsymbol{x} \rangle$ of the input. The goal is to recover the latent direction $\boldsymbol{w}_*$ from these observations.

In a remarkable work, Damian et al. [29] provided a sharp characterization of the statistical and computational complexity of learning in this model. Their analysis relies on expanding the link distribution $\rho$ in the orthonormal basis of Hermite polynomials $\{\mathrm{He}_k\}_{k \geq 0}$; that is, $\mathbb{E}[\mathrm{He}_k(G)\mathrm{He}_j(G)] = \delta_{kj}$ for $G \sim \mathsf{N}(0, 1)$. They defined[2] the *generative exponent* (GE) of $\rho$ as

$$\mathsf{k}_\star(\rho) = \arg\min\{k \geq 1 \ : \ \|\zeta_k\|_{L^2(\rho)} > 0\} \quad \text{where} \quad \zeta_k(Y) := \mathbb{E}_\rho[\mathrm{He}_k(G)|Y]\}, \tag{2}$$

and showed that the optimal sample size $\mathsf{m}$ and runtime $\mathsf{T}$ for recovering $\boldsymbol{w}_*$ from data (1) scales as[3] (writing $\mathsf{k}_\star = \mathsf{k}_\star(\rho)$ and assuming $\mathsf{k}_\star > 1$ for simplicity)

$$\mathsf{m} = \Theta_d(d^{\mathsf{k}_\star/2}), \qquad \mathsf{T} = \widetilde{\Theta}_d(d^{\mathsf{k}_\star/2+1}). \tag{3}$$

The sample complexity is optimal among statistical query (SQ) and low-degree polynomial (LDP) algorithms (up to some additional sample-runtime trade-offs, see Remark 3.2), while the runtime is necessary simply to process $\mathsf{m}$ samples in $d$ dimensions.

Several works have developed algorithms that progressively closed the gap to these optimal rates. In a seminal contribution, Ben Arous et al. [10] analyzed online stochastic gradient descent (SGD) on the single neuron model $\mathrm{He}_{\mathsf{k}_\star}(\langle \boldsymbol{w}, \boldsymbol{x} \rangle)$, and showed it recovers $\boldsymbol{w}_*$ with suboptimal $\mathsf{m} = \widetilde{\Theta}_d(d^{\mathsf{k}_\star-1})$ and $\mathsf{T} = \widetilde{\Theta}_d(d^{\mathsf{k}_\star})$. To close this gap, Damian et al. [28] proposed a smoothing-based modification of SGD inspired by the tensor PCA literature [19], which locally averages the loss landscape and achieves near-optimal $\mathsf{m} = \widetilde{\Theta}_d(d^{\mathsf{k}_\star/2})$ and $\mathsf{T} = \widetilde{\Theta}_d(d^{\mathsf{k}_\star/2+1})$. Finally, the polylogarithmic factor in sample complexity was removed by Damian et al. [29] via a partial trace estimator—again inspired by tensor PCA [47]—which achieves $\mathsf{m} = \Theta_d(d^{\mathsf{k}_\star/2})$ and $\mathsf{T} = \Theta_d(d^{\mathsf{k}_\star/2+1} \log(d))$, thereby matching the optimal rates in (3).

While these results yield a sharp characterization of learning Gaussian SIMs in high dimensions, several conceptual gaps remain:

> *Why is the vanilla SGD algorithm suboptimal, with runtime $d^{\mathsf{k}_\star}$ instead of the optimal $d^{\mathsf{k}_\star/2+1}$? Why do methods such as landscape smoothing and partial trace estimators—both borrowed from the tensor PCA literature—achieve optimal complexity[4]? And what role does the Gaussian assumption play in these results?*

In this paper, we propose simple—and perhaps surprising—answers to these questions. Our key observation is that the complexity of learning SIMs is governed not by Gaussianity itself, but by the problem's *rotational symmetry*. Specifically, the family $\{\mathbb{P}_{\boldsymbol{w}} \ : \ \boldsymbol{w} \in \mathbb{S}^{d-1}\}$ consists of all pushforwards under orthogonal transformations of the input, suggesting that optimal algorithms should respect this symmetry—that is, be equivariant under the action of the orthogonal group $\mathcal{O}_d$.

This symmetry-based perspective naturally leads to *spherical harmonics*—which arise as irreducible representations of $\mathcal{O}_d$—as the appropriate basis for this problem, instead of Hermite polynomials. Adopting this basis not only clarifies the above questions, but also extends the theory beyond the Gaussian setting to arbitrary spherically symmetric distributions.

## 1.1 Summary of main results

In this paper, we characterize the sample and computational complexity of learning single-index models under general spherically symmetric input distributions. Let $\mu \in \mathcal{P}(\mathbb{R}^d)$ be a distribution invariant under orthogonal transformations, i.e., $\boldsymbol{R}_{\#}\mu = \mu$ for all $\boldsymbol{R} \in \mathcal{O}_d$. Such distributions admit a polar decomposition $\boldsymbol{x} = r\boldsymbol{z}$, where the radius $r = \|\boldsymbol{x}\|_2 \sim \mu_r$ is independent of the direction $\boldsymbol{z} = \boldsymbol{x}/\|\boldsymbol{x}\|_2 \sim \tau_d := \mathrm{Unif}(\mathbb{S}^{d-1})$. We define a natural generalization of Gaussian SIMs (1), which we call *spherical single-index models*, specified by a joint distribution $\nu_d \in \mathcal{P}(\mathbb{R}^3)$:

$$(Y, R, Z) \sim \nu_d : \quad R \sim \mu_r \ \perp \ Z \sim \tau_{d,1} \quad \text{and} \quad Y|(R, Z) \sim \nu_d(\cdot|R, Z), \tag{4}$$

---

[2] An earlier notion—the *information exponent*—was proposed in [40, 10]. See Appendix A.5 for discussion.

[3] Throughout, $\tilde{\Theta}_d(\cdot)$ hides polylogarithmic factors in $d$.

[4] Let us emphasize here that these algorithms fail to achieve optimal complexity for (slightly) more general SIMs (see Section A.3). Thus an analogy to tensor PCA is not enough to explain their success in this setting.

| Subspace $V_{d,\ell}$ | Sample optimal | Runtime optimal |
|---|---|---|
| | Spectral algorithm | |
| $\ell = 1$ | $\mathsf{m} \asymp d \vee \dfrac{d^{1/2}}{\|\xi_{d,1}\|_{L^2}^2}, \quad \mathsf{T} \asymp d^2 \vee \dfrac{d^{3/2}}{\|\xi_{d,1}\|_{L^2}^2}$ | |
| $\ell = 2$ | $\mathsf{m} \asymp \dfrac{d}{\|\xi_{d,2}\|_{L^2}^2}, \quad \mathsf{T} \asymp \dfrac{d^2}{\|\xi_{d,2}\|_{L^2}^2} \log d$ | |
| | Harmonic tensor unfolding | Online SGD |
| $\ell \geq 3$ | $\mathsf{m} \asymp \dfrac{d^{\ell/2}}{\|\xi_{d,\ell}\|_{L^2}^2}, \quad \mathsf{T} \asymp \dfrac{d^{\ell+1}}{\|\xi_{d,\ell}\|_{L^2}^2} \log d$ | $\mathsf{m} \asymp \dfrac{d^{\ell-1}}{\|\xi_{d,\ell}\|_{L^2}^2}, \quad \mathsf{T} \asymp \dfrac{d^{\ell}}{\|\xi_{d,\ell}\|_{L^2}^2}$ |

Table 1: Summary of algorithms for learning spherical SIMs on each irreducible subspace $V_{d,\ell}$, with their sample complexity $\mathsf{m}$ and runtime $\mathsf{T}$. Here, the notation $\asymp$ hides constants that depend on $\ell$ and assumptions on $\nu_d$. The estimator in the left (resp. right) column matches the optimal sample complexity (resp. optimal runtime) predicted by the LDP (resp. SQ) lower bound (8). See Section 3 for details and formal statements.

where $\tau_{d,1}$ is the distribution of the first coordinate of $\boldsymbol{z} \sim \tau_d$. Samples are drawn according to:

$$(y, \boldsymbol{x}) \sim \mathbb{P}_{\boldsymbol{w}_*}: \quad \boldsymbol{x} = (r, \boldsymbol{z}) \sim \mu = \mu_r \otimes \tau_d \quad \text{and} \quad y|(r, \boldsymbol{z}) \sim \nu_d(\cdot|r, \langle \boldsymbol{w}_*, \boldsymbol{z} \rangle), \tag{5}$$

for an unknown unit vector $\boldsymbol{w}_* \in \mathbb{S}^{d-1}$. Thus, the label $y$ may now depend on both $(r, \langle \boldsymbol{w}_*, \boldsymbol{z} \rangle)$, rather than only on the scalar projection $\langle \boldsymbol{w}_*, \boldsymbol{x} \rangle$. Unlike the Gaussian case, the conditional distribution $\nu_d(\cdot|R, Z)$ is allowed to depend on the ambient dimension $d$: our learning guarantees will hold for fixed $\nu_d$, with explicit (up to universal constants), non-asymptotic bounds.

We now summarize our main results on estimating $\boldsymbol{w}_*$ from i.i.d. samples drawn from the spherical single-index model (5):

**Harmonic decomposition and lower bounds.** To characterize the complexity of learning in this setting, we exploit the decomposition of $L^2(\mathbb{S}^{d-1})$ into harmonic subspaces:

$$L^2(\mathbb{S}^{d-1}) = \bigoplus_{\ell=0}^{\infty} V_{d,\ell}, \qquad n_{d,\ell} = \dim(V_{d,\ell}) = \Theta_d(d^\ell), \tag{6}$$

where $V_{d,\ell}$ denotes the space of degree-$\ell$ spherical harmonics. For each $\ell \geq 1$, we define the $\ell$-th Gegenbauer coefficient of $\nu_d$ to be

$$\xi_{d,\ell}(Y, R) := \mathbb{E}_{\nu_d}[Q_\ell(Z)|Y, R], \tag{7}$$

where $Q_\ell \in V_{d,\ell}$ is the (normalized) degree-$\ell$ Gegenbauer polynomial in $L^2([-1, 1], \tau_{d,1})$, with $\mathbb{E}_{\tau_{d,1}}[Q_\ell(Z)Q_k(Z)] = \delta_{kl}$. We establish the following lower bounds on the sample complexity $\mathsf{m}$ and runtime $\mathsf{T}$ for recovering $\boldsymbol{w}_*$ using the low-degree polynomial (LDP) and statistical query (SQ) frameworks:

$$\mathsf{m} \gtrsim \left\{ d \vee \frac{d^{1/2}}{\|\xi_{d,1}\|_{L^2}^2} \right\} \wedge \inf_{\ell \geq 2} \frac{d^{\ell/2}}{\|\xi_{d,\ell}\|_{L^2}^2}, \qquad \mathsf{T} \gtrsim \left\{ d^2 \vee \frac{d^{3/2}}{\|\xi_{d,1}\|_{L^2}^2} \right\} \wedge \inf_{\ell \geq 2} \frac{d^\ell}{\|\xi_{d,\ell}\|_{L^2}^2}. \tag{8}$$

These bounds effectively decouple across irreducible subspaces: each term $\ell \geq 1$ in the infimum corresponds to a lower bound for learning spherical SIMs using estimators restricted to the harmonic subspace $V_{d,\ell}$, with matching upper bounds summarized in Table 1. Note that $\|\xi_{d,\ell}\|_{L^2} \leq 1$ and can decay with $d$: thus, these lower bounds capture the competition between the dimensionality $\Theta_d(d^\ell)$ of $V_{d,\ell}$ and the 'signal strength' $\|\xi_{d,\ell}\|_{L^2}^2$ it carries about $\mathbb{P}_{\nu_d, \boldsymbol{w}_*}$.

**Optimal algorithms and trade-offs.** For each harmonic subspace $V_{d,\ell}$, we construct: (1) a sample-optimal[5] estimator based either on spectral methods for $\ell \in \{1, 2\}$ (that is also runtime (near-)optimal)

---

[5]Throughout the paper, we refer to *sample-optimal* as the optimal conjectural sample complexity for polynomial time algorithms (see Remark 3.2), and refer to the *information-theoretic optimal* sample complexity otherwise.

| Gaussian SIMs | Sample optimal | Runtime optimal |
|---|---|---|
| With $\|\boldsymbol{x}\|_2$ | \multicolumn — Spectral algorithm ($\mathsf{l}_\star = 1$ if $\mathsf{k}_\star$ odd, $\mathsf{l}_\star = 2$ if $\mathsf{k}_\star$ even) $\quad$ $\mathsf{m} \asymp d^{\mathsf{k}_\star/2}, \quad \mathsf{T} \asymp d^{\mathsf{k}_\star/2+1} \log d$ | |
| Without $\|\boldsymbol{x}\|_2$ | Harmonic tensor unfolding ($\mathsf{l}_\star = \mathsf{k}_\star$) $\mathsf{m} \asymp d^{\mathsf{k}_\star/2}, \quad \mathsf{T} \asymp d^{\mathsf{k}_\star+1} \log d$ | Online SGD ($\mathsf{l}_\star = \mathsf{k}_\star$) $\mathsf{m} \asymp d^{\mathsf{k}_\star-1}, \quad \mathsf{T} \asymp d^{\mathsf{k}_\star}$ |

Table 2: Summary of algorithms for learning Gaussian SIMs with generative exponent $\mathsf{k}_\star > 1$, with or without using the radial component $r = \|\boldsymbol{x}\|_2$. Here, the notation $\asymp$ hides constants that depend only on the link distribution $\rho$. See Section 4 for details and formal statements.

or on tensor-unfolding of reproducing harmonic operators for $\ell \geq 3$; and (2) a runtime-optimal estimator based on online SGD for $\ell \geq 3$. Their complexities are summarized in Table 1.

This leads to a simple strategy for learning spherical SIMs (5): identify $\mathsf{l}_{\mathsf{m},\star}$ (sample-optimal) or $\mathsf{l}_{\mathsf{T},\star}$ (runtime-optimal) as the degree minimizing the corresponding term in the lower bound (8), and apply the matching algorithm from Table 1. Note that we always have $\mathsf{l}_{\mathsf{m},\star} \geq \mathsf{l}_{\mathsf{T},\star}$. If $\mathsf{l}_{\mathsf{m},\star} = \mathsf{l}_{\mathsf{T},\star} \in \{1, 2\}$, the spectral algorithm achieves both optimal sample and runtime complexity. If $\mathsf{l}_{\mathsf{m},\star} = \mathsf{l}_{\mathsf{T},\star} > 2$, it remains open whether a single estimator can achieve both[6]. More generally, one can construct distributions for which $\mathsf{l}_{\mathsf{m},\star} \gg \mathsf{l}_{\mathsf{T},\star}$, suggesting that *no algorithm can simultaneously achieve optimal sample and runtime complexity in these cases*. This stands in sharp contrast to Gaussian SIMs, where both complexities are always jointly achievable. Thus,

*Additional sample-runtime trade-offs appear when learning SIMs beyond the Gaussian setting.*

**The case of Gaussian inputs.** We now specialize our results to the Gaussian single-index model (1), where $\mu = \mathsf{N}(0, \mathbf{I}_d)$ and $\nu_d(Y, R, Z) = \rho(Y, R \cdot Z)$ with generative exponent $\mathsf{k}_\star > 1$. This yields a particularly transparent picture:

(1) The optimal degrees $\mathsf{l}_{\mathsf{m},\star} = \mathsf{l}_{\mathsf{T},\star}$ are always either 1 (if $\mathsf{k}_\star$ is odd) or 2 (if $\mathsf{k}_\star$ is even), with the spectral algorithm from Table 1 achieving both optimal sample and runtime complexity (3). Thus, *for any Gaussian SIMs, optimal algorithms lie in the harmonic subspaces $V_{d,1}$ and $V_{d,2}$, corresponding to degree-1 or 2 spherical harmonics in $\boldsymbol{z} = \boldsymbol{x}/\|\boldsymbol{x}\|_2$.*

(2) The SGD algorithm of [10] is dominated by the high-frequency harmonics $V_{d,\mathsf{k}_\star}$, while smoothing [28] reweights the loss landscape toward low-frequency harmonics $V_{d,1}$ or $V_{d,2}$. The partial trace estimator [29] explicitly projects onto them. *Both methods achieve optimal complexity by effectively exploiting these low-frequency components.*

(3) We provide an alternative perspective for understanding the suboptimality of SGD: its optimization dynamics remains essentially unchanged when $\boldsymbol{x}$ is replaced by its direction $\boldsymbol{z}$, implying it does not exploit the radial component $r = \|\boldsymbol{x}\|_2$. We show that algorithms that ignore $r$ must incur a runtime complexity of $\Omega_d(d^{\mathsf{k}_\star})$. In this sense, SGD is runtime-optimal among methods that rely solely on the directional component. *To achieve the optimal runtime $\Theta_d(d^{\mathsf{k}_\star/2+1})$, one must exploit the radial component—even though it carries no information about $\boldsymbol{w}_*$ and asymptotically concentrates around $r/\sqrt{d} \to 1$.*

These results are summarized in Table 2.

## 2 Setting and definitions

Throughout the paper, we assume our link functions $\nu_d$ are drawn from the following class:

**Definition 1** (Spherical link functions). *Let $\mathfrak{L}_d$ denote the set of joint distributions $\nu_d$ on $\mathcal{Y} \times \mathbb{R}_{\geq 0} \times [-1, 1]$, where $\mathcal{Y}$ is an arbitrary measurable space, such that the following hold:*

---

[6]In fact, we show that for $\ell$ even, harmonic tensor unfolding achieves both optimal sample and runtime complexity, with a potential additional $\log(d)$ factor in sample complexity.

(i) *The marginals of $(Y, R, Z) \sim \nu_d$ satisfy $Z \sim \tau_{d,1}$ (the distribution of the first coordinate of $\boldsymbol{z} \sim \tau_d = \mathrm{Unif}(\mathbb{S}^{d-1})$) and $R \sim \nu_{d,R} \in \mathcal{P}(\mathbb{R}_{\geq 0})$ not concentrated at 0.*

(ii) *Let $\nu_{d,0} = \nu_{d,Y,R} \otimes \tau_{d,1}$ be the product of marginals $(Y, R)$ and $Z \sim \tau_{d,1}$. Then $\nu_d \ll \nu_{d,0}$ and the Radon-Nikodym derivative satisfy $\frac{\mathrm{d}\nu_d}{\mathrm{d}\nu_{d,0}} \in L^2(\nu_{d,0})$.*

The corresponding spherical single-index model over $(y, \boldsymbol{x}) \in \mathcal{Y} \times \mathbb{R}^d$ are then given by:

$$(y, \boldsymbol{x}) \sim \mathbb{P}_{\nu_d, \boldsymbol{w}_*}: \quad \boldsymbol{x} = (r, \boldsymbol{z}) \sim \mu := \nu_{d,R} \otimes \tau_d \quad \text{and} \quad y|(r, \boldsymbol{z}) \sim \nu_d(\cdot | r, \langle \boldsymbol{w}_*, \boldsymbol{z} \rangle), \quad (9)$$

where $\boldsymbol{x} = r\boldsymbol{z}$ is the polar decomposition with $r = \|\boldsymbol{x}\|_2 \sim \nu_{d,R}$ and $\boldsymbol{z} = \boldsymbol{x}/\|\boldsymbol{x}\|_2 \sim \tau_d$.

From $\nu_{d,0} = \nu_{d,Y,R} \otimes \tau_{d,1}$, we define the null model $\mathbb{P}_{\nu_d,0} := \nu_{d,Y,R} \otimes \tau_d$, where $(y, r)$ is independent of the direction $\boldsymbol{z}$. Note that assumption $\frac{\mathrm{d}\nu_d}{\mathrm{d}\nu_{d,0}} \in L^2(\nu_{d,0})$ is equivalent to $\frac{\mathrm{d}\mathbb{P}_{\nu_d, \boldsymbol{w}_*}}{\mathrm{d}\mathbb{P}_{\nu_d,0}} \in L^2(\mathbb{P}_{\nu_d,0})$. This ensures that the model has 'enough noise' in the label and excludes non-robust algorithms that can beat the lower bounds (3) in the noise-free setting (e.g., see [74]).

**Remark 2.1.** Throughout this paper, we assume $\nu_d$ to be known. This assumption is mild in high dimensions, where the primary challenge lies in recovering $\boldsymbol{w}_*$. When $\nu_d$ is unknown, one can modify our algorithms and use random nonlinearities. See discussion in Appendix A.2.

**Harmonic decomposition.** Let $L^2(\mathbb{S}^{d-1}) := L^2(\mathbb{S}^{d-1}, \tau_d)$ denote the space of squared integrable functions on the unit sphere, with inner-product $\langle f, g \rangle_{L^2} = \mathbb{E}_{\boldsymbol{z} \sim \tau_d}[f(\boldsymbol{z})g(\boldsymbol{z})]$ and norm $\|f\|_{L^2} := \langle f, f \rangle_{L^2}^{1/2}$. This space admits the orthogonal decomposition (6), with $V_{d,\ell}$ the subspace of degree-$\ell$ spherical harmonics, that is, degree-$\ell$ polynomials that are orthogonal (with respect to $\langle \cdot, \cdot \rangle_{L^2}$) to all polynomials with degree less than $\ell$. We refer to Appendix B for background on spherical harmonics.

We use this harmonic decomposition to expand the likelihood ratio of the model in $L^2(\mathbb{P}_{\nu_d,0})$:

$$\frac{\mathrm{d}\mathbb{P}_{\nu_d, \boldsymbol{w}_*}}{\mathrm{d}\mathbb{P}_{\nu_d,0}}(y, \boldsymbol{x}) = 1 + \sum_{\ell=1}^{\infty} \xi_{d,\ell}(y, r) Q_\ell(\langle \boldsymbol{w}_*, \boldsymbol{z} \rangle), \qquad \xi_{d,\ell}(Y, R) := \mathbb{E}_{\nu_d}[Q_\ell(Z) | Y, R], \quad (10)$$

where $Q_\ell : [-1, 1] \to \mathbb{R}$ is the normalized degree-$\ell$ Gegenbauer polynomial. We denote $\|\xi_{d,\ell}\|_{L^2}$ the $L^2$-norm with respect to $\nu_d$. The $\chi^2$- mutual information of $((Y, R), Z)$ under $\nu_d$ is given by $I_{\chi^2}[\nu_d] := \mathsf{D}_{\chi^2}[\nu_d \| \nu_{d,0}] = \sum_{\ell=1}^{\infty} \|\xi_{d,\ell}\|_{L^2}^2$. See Appendix A.2 for further details.

**Lower bounds.** We establish lower bounds within two standard frameworks: statistical query (SQ) and low-degree polynomials (LDP) algorithms. Our lower bounds will hold for the weaker task of distinguishing the planted model $\mathbb{P}_{\nu_d, \boldsymbol{w}_*}$ from the null $\mathbb{P}_{\nu_d,0}$, i.e., the 'detection' problem. These lower bounds directly imply lower bounds on our estimation problem.

- For SQ algorithms $\mathcal{A} \in \mathsf{SQ}(q, \tau)$, with $q$ query calls of tolerance $\tau$, we derive lower bounds on the query complexity $q/\tau^2$. Heuristically, this corresponds to a runtime lower bound of $\mathsf{T} \geq q/\tau^2$, under the standard assumption that each query requires at least $\Omega(1/\tau^2)$ samples to implement. While this connection is informal, it is validated by our matching upper bounds: the actual runtime of our proposed estimators meets these SQ-based lower bounds, except for a minor discrepancy when $\ell = 1$, where a tighter bound $\mathsf{m}d$ (the cost of processing $\mathsf{m}$ samples in $d$ dimensions) applies.
- For LDP lower bounds on sample complexity, we work in an asymptotic setting $d \to \infty$. In this case, the bounds hold uniformly over sequences $\{\nu_d\}_{d \geq 1}$, with $\nu_d \in \mathfrak{L}_d$ satisfying the following mild condition (the model is 'solvable in polynomial time'):

**Assumption 1.** *There exists $p \in \mathbb{N}$ such that the sequence $\{\nu_d\}_{d \geq 1}$ satisfies $\mathsf{M}_\star(\nu_d) = O_d(d^{p/2})$.*

We refer to Appendix C for background on SQ and LDP algorithms.

## 3 Learning spherical single-index models

Let $\{\nu_d\}_{d \geq 1}$ be a sequence of spherical SIMs with $\nu_d \in \mathfrak{L}_d$ (Definition 1). Under mild conditions, the information-theoretic sample complexity for recovering $\boldsymbol{w}_*$ is $\Theta_d(d)$ (see Appendix H). However,

polynomial-time algorithms may require significantly more samples—that is, the model exhibits a so-called *computational-to-statistical gap*. We introduce the complexity parameters:

$$\mathsf{M}_\star(\nu_d) := \inf_{\ell \geq 1} \frac{\sqrt{n_{d,\ell}}}{\|\xi_{d,\ell}\|_{L^2}^2}, \qquad \mathsf{Q}_\star(\nu_d) := \inf_{\ell \geq 1} \frac{n_{d,\ell}}{\|\xi_{d,\ell}\|_{L^2}^2}, \qquad (11)$$

where we recall that $n_{d,\ell} = \dim(V_{d,\ell}) = \Theta_d(d^\ell)$ and $\xi_{d,\ell}$ is the $\ell$-th Gegenbauer coefficient of the likelihood ratio (10). These quantities govern the sample and query complexity lower bounds within the LDP and SQ frameworks respectively, as formalized in the following theorem:

**Theorem 1** (General lower bounds). *Let $\{\nu_d\}_{d \geq 1}$ be a sequence of spherical SIMs with $\nu_d \in \mathfrak{L}_d$.*

    *(i) (SQ runtime lower bound.) Any algorithm $\mathcal{A} \in \mathsf{SQ}(q, \tau)$ that distinguishes between $\mathbb{P}_{\nu_d, \boldsymbol{w}_*}$ and $\mathbb{P}_{\nu_d, 0}$ satisfies $q/\tau^2 \geq \mathsf{Q}_\star(\nu_d)$.*

    *(ii) (LDP sample lower bound.) Assume $\{\nu_d\}_{d \geq 1}$ satisfy Assumption 1. Then, under the low-degree conjecture, no polynomial-time algorithm can distinguish $\mathbb{P}_{\nu_d, \boldsymbol{w}_*}$ from $\mathbb{P}_{\nu_d, 0}$ unless $\mathsf{m} = \Omega_d(\mathsf{M}_\star(\nu_d) \log(d)^{-Cp})$ for some universal constant $C > 0$.*

The proof and detailed statements are provided in Appendix C.

**Remark 3.1.** The lower bounds in Theorem 1 are for the detection problem. A detection-recovery gap can appear in this model when the infinum is achieved at $\ell = 1$ and $\sqrt{d}/\|\xi_{d,1}\|_{L^2}^2 \ll d$. In this case, the information-theoretic lower bound $\mathsf{m} = \Omega_d(d)$ is tighter (Appendix H). Thus, the informal complexity lower bounds (8) stated in the introduction are obtained as follows: the sample complexity lower bound is the maximum of the information-theoretic bound $\Omega(d)$ and $\mathsf{M}_\star(\nu_d)$, while the runtime lower bound is the maximum of $\mathsf{Q}_\star(\nu_d)$ and $d$ times the sample complexity lower bound—that is, the cost of processing this many samples.

**Remark 3.2.** Similarly to tensor PCA [16, 84], one can trade-off a factor $D^{-\Theta(1)}$ less sample complexity for $d^{\tilde{\Theta}(D)}$ more runtime. See Appendix C for the LDP lower bound with this explicit trade-off, and [29] for a discussion on how to construct higher-order tensors to match it. In this paper, we ignore these additional sample-runtime trade-offs and focus on matching the exponent in $d$ in the sample complexity. We leave finer-grained analyses to future work.

Intuitively, these lower bounds decompose the problem into separate subproblems associated with each harmonic subspace $V_{d,\ell}$. Each $\ell \geq 1$ in (11) corresponds to the complexity of algorithms restricted to degree-$\ell$ spherical harmonics in $\boldsymbol{z}$ (see Appendix C for further discussion). Below, we introduce matching upper bounds for each subspace $V_{d,\ell}$, summarized earlier in Table 1. These algorithms are stated with general transformations $\mathcal{T}_\ell : \mathcal{Y} \times \mathbb{R}_{\geq 0} \to \mathbb{R}$. For simplicity, we present our learning guarantees below under the following assumption:

**Assumption 2.** *For $\nu_d \in \mathfrak{L}_d$ and $\ell \geq 1$, there exist $\mathcal{T}_\ell : \mathcal{Y} \times \mathbb{R}_{\geq 0} \to \mathbb{R}$ and $\kappa_\ell > 1$ such that $\|\mathcal{T}_\ell\|_{L^2} = 1$, $\|\mathcal{T}_\ell\|_\infty \leq \kappa_\ell$ and $\mathbb{E}_{\nu_d}[\mathcal{T}_\ell(Y, R) Q_\ell(Z)] \geq \kappa_\ell^{-1} \|\xi_{d,\ell}\|_{L^2}$.*

For Gaussian inputs (next section), Assumption 2 is satisfied with $\kappa_\ell$ only depending on $\rho$. In the appendices, we state our learning guarantees under weaker assumptions (with possible additional log factors), e.g., using $\mathcal{T}_\ell := \xi_{d,\ell}/\|\xi_{d,\ell}\|_{L^2}$ under moment condition on $\xi_{d,\ell}$.

**Remark 3.3** (Weak to strong recovery). We further state our results below for the weak recovery task $|\langle \hat{\boldsymbol{w}}, \boldsymbol{w}_* \rangle| \geq 1/4$. Having achieved weak recovery, one can achieve strong recovery $|\langle \hat{\boldsymbol{w}}, \boldsymbol{w}_* \rangle| \geq 1 - \varepsilon$, with arbitrary $\varepsilon > 0$, using an additional $\Theta_d(d/\varepsilon)$ samples (information-theoretic optimal) and $\Theta_d(d^2/\varepsilon)$ runtime—similarly to prior works [10, 88, 28]—under mild assumptions. See discussion in Appendix A.2.

**Spectral algorithm:** For $\ell \in \{1, 2\}$, we present spectral estimators—similar to [61, 66, 62, 29]—that achieve both optimal sample and runtime complexity. Given $\mathsf{m}$ samples $(y_i, \boldsymbol{x}_i)$, these estimators are defined as:

$$\ell = 1: \quad \hat{\boldsymbol{w}}_0 = \frac{\hat{\boldsymbol{v}}}{\|\hat{\boldsymbol{v}}\|_2}, \qquad \hat{\boldsymbol{v}} := \frac{1}{\mathsf{m}} \sum_{i \in [\mathsf{m}]} \mathcal{T}_1(y_i, r_i) \sqrt{d}\, \boldsymbol{z}_i,$$

$$\ell = 2: \quad \hat{\boldsymbol{w}} = \arg\max_{\boldsymbol{w} \in \mathbb{S}^{d-1}} \boldsymbol{w}^\mathsf{T} \hat{\boldsymbol{M}} \boldsymbol{w}, \qquad \hat{\boldsymbol{M}} := \frac{1}{\mathsf{m}} \sum_{i \in [\mathsf{m}]} \mathcal{T}_2(y_i, r_i) \left[ d \cdot \boldsymbol{z}_i \boldsymbol{z}_i^\mathsf{T} - \mathbf{I}_d \right]. \qquad \text{(SP-Alg)}$$

For $\ell = 1$, the estimator $\hat{w}_0$ either achieves weak recovery directly or requires boosting as in [88, 28, 29]; we leave the description of the full algorithm to Appendix D. For $\ell = 2$, the leading eigenvector can be efficiently computed in runtime $\Theta_d(\mathsf{m}d \log(d))$ via power iteration.

**Theorem 2** (Spectral algorithm). *Let $\nu_d \in \mathfrak{L}_d$ and set $\mathcal{T}_\ell$ as in Assumption 2. There exists $C_\ell \geq 0$ that only depends on $\ell$ such that for any $\delta > 0$, the output $\hat{w}$ of the spectral estimator (SP-Alg) achieves $|\langle \hat{w}, w_* \rangle| \geq 1/4$ with probability $1 - \delta$ when*

$$\ell = 1: \quad \mathsf{m} \geq C_\ell \kappa_\ell^2 \frac{d}{\|\xi_{d,1}\|_{L^2}^2} \sqrt{\log(1/\delta)}, \qquad\qquad and \quad \mathsf{T} \geq C_\ell \mathsf{m} \cdot d,$$

$$\ell = 2: \quad \mathsf{m} \leq C_\ell \kappa_\ell^2 \frac{d}{\|\xi_{d,2}\|_{L^2}^2} \left(1 + \|\xi_{d,2}\|_{L^2} \log^2(d/\delta)\right), \qquad and \quad \mathsf{T} \geq C_\ell \mathsf{m} \cdot d \log(d). \tag{12}$$

*Furthermore, for $\ell = 1$, one can achieve better sample complexity under an additional condition: the boosted estimator achieves $|\langle \hat{w}, w_* \rangle| \geq 1/4$ with probability $1 - \delta$ when*

$$\ell = 1: \quad \mathsf{m} \geq C_\ell \kappa_\ell^2 \frac{\sqrt{d}}{\|\xi_{d,1}\|_{L^2}^2} \sqrt{\log(1/\delta)}, \qquad and \qquad \mathsf{T} \geq C_\ell \mathsf{m} \cdot d. \tag{13}$$

The proof and detailed statement of this theorem can be found in Appendix D. For $\ell = 2$ and $\|\xi_{d,2}\|_{L^2} = \Omega_d(\log(d)^{-2})$, the additional factor $\log^2(d)$ can be removed by following a similar argument as in [66].

**Online SGD algorithm:** For $\ell \geq 3$, we propose an online SGD algorithm inspired by [10] that achieves optimal runtime. We run projected online SGD on the population loss

$$\min_{w \in \mathbb{S}^{d-1}} \mathbb{E}_{\mathbb{P}_{\nu_d, w_*}} \left[ (\mathcal{T}_\ell(y, r) - Q_\ell(\langle w, z \rangle))^2 \right], \tag{SGD-Alg}$$

with a carefully chosen step size [10, 88]. The number of samples in this algorithm corresponds to the number of SGD iterations, and the total runtime is $\Theta_d(\mathsf{m}d)$—the cost of computing a $d$-dimensional gradient at each iteration. Details can be found in Appendix E.

**Theorem 3** (Online SGD algorithm). *Let $\nu_d \in \mathfrak{L}_d$ and set $\mathcal{T}_\ell$ as in Assumption 2. There exists $C_\ell \geq 0$ that only depends on $\ell$ such that for any $\delta > 0$, the output $\hat{w}$ of the online SGD estimator (SGD-Alg) achieves $|\langle \hat{w}, w_* \rangle| \geq 1/4$ with probability $1 - \delta$ when*

$$\ell \geq 3: \quad \mathsf{m} \geq C_\ell \kappa_\ell^2 \frac{d^{\ell-1}}{\|\xi_{d,\ell}\|_{L^2}^2} \log(1/\delta), \qquad and \qquad \mathsf{T} \geq C_\ell \mathsf{m} \cdot d. \tag{14}$$

The proof of this theorem follows by adapting the arguments in [10, 88] and can be found in Appendix E.

**Harmonic tensor unfolding.** For $\ell \geq 3$, we present a tensor-unfolding algorithm inspired by the seminal work [67] on Tensor PCA, that achieves optimal sample complexity. We introduce a degree-$\ell$ harmonic tensor $\mathcal{H}_\ell(z) \in \text{Sym}((\mathbb{R}^d)^{\otimes \ell})$ (the space of symmetric $\ell$-tensors in $d$-dimensions). It is defined via the degree-$\ell$ Gegenbauer polynomial as

$$Q_\ell(\langle w, z \rangle) = \langle w^{\otimes \ell}, \mathcal{H}_\ell(z) \rangle, \qquad \text{for all } w \in \mathbb{S}^{d-1}. \tag{15}$$

We provide an explicit expression of $\mathcal{H}_\ell$ in Appendix F. This tensor can be seen as the projection of $z^{\otimes \ell}$ into the space of symmetric, trace-less tensors. In particular, it has the reproducing property

$$\mathbb{E}_z \left[ Q_k(\langle w, z \rangle) \mathcal{H}_\ell(z) \right] = \frac{\delta_{\ell k}}{\sqrt{n_{d,\ell}}} \mathcal{H}_\ell(w). \tag{16}$$

Given $\mathsf{m}$ samples $(y_i, x_i) \sim \mathbb{P}_{\nu_d, w_*}$, we compute the empirical tensor

$$\hat{T} := \frac{1}{\mathsf{m}} \sum_{i \in [\mathsf{m}]} \mathcal{T}_\ell(y_i, r_i) \mathcal{H}_\ell(z_i). \tag{17}$$

By the reproducing property (16), the expectation under $\mathbb{P}_{\nu_d, w_*}$ of this tensor is proportional to $\mathcal{H}_\ell(w_*)$, which has principal component $w_*^{\otimes \ell}$. To extract this component, we will consider two

different estimators. We follow [67] and construct the unfolded matrix of $\hat{T} \in (\mathbb{R}^d)^{\otimes(I+J)}$, denoted $\mathbf{Mat}_{I,J}(\hat{T}) \in \mathbb{R}^{d^I \times d^J}$, with entries

$$\mathbf{Mat}_{I,J}(\hat{T})_{(i_1,\ldots,i_I),(j_1,\ldots,j_J)} = \hat{T}_{i_1,\ldots,i_I,j_1,\ldots,j_J},$$

where we identify $(i_1, \ldots, i_I)$ with $1 + \sum_{j=1}^{I}(i_j - 1)d^{j-1}$.

For $\ell \geq 3$ even, our first estimator computes $\boldsymbol{s}_1(\mathbf{Mat}_{\ell/2,\ell/2}(\hat{T})) \in \mathbb{R}^{d^{\lfloor \ell/2 \rfloor}}$, the top left singular vector of $\mathbf{Mat}_{\ell/2,\ell/2}(\hat{T}) \in \mathbb{R}^{d^{\ell/2} \times d^{\ell/2}}$ via power iteration, and return

$$\hat{w} := \mathbf{Vec}_{\ell/2}\left(\boldsymbol{s}_1\left(\mathbf{Mat}_{\ell/2,\ell/2}(\hat{T})\right)\right), \tag{TU-Alg-b}$$

where the mapping $\mathbf{Vec}_k : \mathbb{R}^{d^k} \to \mathbb{S}^{d-1}$ applied to $\boldsymbol{u} \in \mathbb{R}^{d^k}$ returns the top left singular vector of the matrix $\boldsymbol{U} \in \mathbb{R}^{d \times d^{k-1}}$ with entries $\boldsymbol{U}_{i_1,(i_2,\ldots,i_k)} = u_{(i_1,\ldots,i_k)}$.

When $\ell$ is odd, however, the above estimator (TU-Alg-b) (now with $I = \lfloor \ell/2 \rfloor$ and $J = \lceil \ell/2 \rceil$) requires $\mathsf{m} \asymp d^{\lceil \ell/2 \rceil}/\|\xi_{d,\ell}\|_{L^2}^2$ samples, which is suboptimal by a factor $d^{1/2}$. This is due to the covariance structure of the Harmonic tensor $\mathcal{H}_\ell(\boldsymbol{z})$: a similar problem, with same suboptimality, arises for tensor PCA with symmetric noise [67] (if the noise is not symmetric and all entries are assumed independent, this algorithm achieves the optimal threshold in tensor PCA [48]).

Here, we modify (TU-Alg-b) by removing the diagonal elements. We set $I = \lfloor (\ell - 1)/2 \rfloor$ and $J = \lceil (\ell + 1)/2 \rceil$, and introduce the matrix

$$\hat{M} = \mathbf{Mat}_{I,J}(\hat{T})\mathbf{Mat}_{I,J}(\hat{T})^\mathsf{T} - \frac{1}{m^2}\sum_{i=1}^{m}\mathcal{T}_\ell(y_i,r_i)^2\mathbf{Mat}_{I,J}(\mathcal{H}_\ell(\boldsymbol{z}_i))\mathbf{Mat}_{I,J}(\mathcal{H}_\ell(\boldsymbol{z}_i))^\mathsf{T}$$

$$= \frac{1}{m^2}\sum_{i \neq j}\mathcal{T}_\ell(y_i,r_i)\mathcal{T}_\ell(y_j,r_j)\mathbf{Mat}_{I,J}(\mathcal{H}_\ell(\boldsymbol{z}_i))\mathbf{Mat}_{I,J}(\mathcal{H}_\ell(\boldsymbol{z}_j))^\mathsf{T} \in \mathbb{R}^{d^I \times d^I}. \tag{18}$$

For $\ell \geq 3$, our second estimator computes $\boldsymbol{s}_1(\hat{M}) \in \mathbb{R}^{d^I}$, the top left singular vector of $\hat{M}$, via power iteration, and return

$$\hat{w} := \mathbf{Vec}_I\left(\boldsymbol{s}_1\left(\hat{M}\right)\right), \tag{TU-Alg}$$

where the mapping $\mathbf{Vec}_I$ is as defined above.

For both estimators (TU-Alg-b) and (TU-Alg), we show that given enough samples, the leading eigenvector $\boldsymbol{s}_1$ is well approximated by $\boldsymbol{w}_*^{\otimes I}$, and the vectorization operation returns a good approximation of $\boldsymbol{w}_*$. The runtime of this algorithm is dominated by the computation of the top eigenvector $\boldsymbol{s}_1(\hat{M})$ via power iteration, which requires $\Theta_d(\mathsf{m}(d^I + d^J)\log(d))$ operations.

**Theorem 4** (Harmonic tensor unfolding). *Let $\nu_d \in \mathfrak{L}_d$ and set $\mathcal{T}_\ell$ as in Assumption 2. There exist $c_\ell, C_\ell \geq 0$ that only depend on $\ell$ such that the following holds.*

(i) *For $\ell \geq 3$ even, the output $\hat{w}$ of the balanced harmonic tensor unfolding algorithm (TU-Alg-b) achieves $|\langle \hat{w}, \boldsymbol{w}_* \rangle| \geq 1/4$ with probability $1 - \delta$ when*

$$\ell \text{ even}: \quad \mathsf{m} \geq C_\ell \kappa_\ell^2 \frac{d^{\ell/2}}{\|\xi_{d,\ell}\|_{L^2}^2}\left(1 + \|\xi_{d,\ell}\|_{L^2}\log(d/\delta)\right), \quad \text{and} \quad \mathsf{T} \geq C_\ell \mathsf{m} \cdot d^{\ell/2}\log(d).$$

(ii) *For $\ell \geq 3$, the output $\hat{w}$ of the harmonic tensor unfolding algorithm (TU-Alg) achieves $|\langle \hat{w}, \boldsymbol{w}_* \rangle| \geq 1/4$ with probability $1 - e^{-d^{c_\ell}}$ when*

$$\ell \text{ even}: \qquad \mathsf{m} \geq C_\ell \kappa_\ell^2 \frac{d^{\ell/2}}{\|\xi_{d,\ell}\|_{L^2}^2}, \quad \text{and} \qquad \mathsf{T} \geq C_\ell \mathsf{m} \cdot d^{\ell/2+1}\log(d),$$

$$\ell \text{ odd}: \qquad \mathsf{m} \geq C_\ell \kappa_\ell^2 \frac{d^{\ell/2}}{\|\xi_{d,\ell}\|_{L^2}^2}, \quad \text{and} \qquad \mathsf{T} \geq C_\ell \mathsf{m} \cdot d^{\ell/2+1/2}\log(d).$$

The proof of this theorem can be found in Appendix F.

**Remark 3.4.** A computationally more efficient partial trace estimator was proposed to recover the principal component in symmetric tensor PCA [47, 29]. Here, however, $\mathcal{H}_\ell(\boldsymbol{z})[\mathbf{I}_d]$ (contraction of two indices) projects onto $\mathcal{H}_{\ell-2}(\boldsymbol{z})$ and lower order harmonics, which defeats the purpose of our estimator (and lead to suboptimal performance if $\ell$ is chosen to be the optimal degree).

**Optimal algorithms.** We define the sample-optimal and runtime-optimal degrees as

$$l_{m,\star} = \arg\min_{\ell \geq 1} \frac{\sqrt{n_{d,\ell}}}{\|\xi_{d,\ell}\|_{L^2}^2}, \qquad \text{and} \qquad l_{T,\star} = \arg\min_{\ell \geq 1} \frac{n_{d,\ell}}{\|\xi_{d,\ell}\|_{L^2}^2}. \tag{19}$$

Then choosing the corresponding algorithm associated to degree $l_{m,\star}$ (resp. $l_{T,\star}$) achieves the optimal sample (resp. runtime) complexity among SQ and LDP algorithms. For $l_{m,\star} = l_{T,\star} \geq 3$ odd, we leave open the possibility that an algorithm achieves both sample and runtime complexity. More generally, in Appendix A.2, we show how to construct distributions with arbitrary large gaps $l_{m,\star} \gg l_{T,\star}$. As discussed in the introduction, this suggest that no single estimator can achieve both optimal complexities in this case, and sample-runtime trade-offs appear.

Specifically, our example proceeds as follows. Let $k \geq 1$ be an arbitrary integer. The label $Y$ is a mixture of two single-index models: $Y|R, Z \sim \nu_{1,d}$ with probability $d^{-2k}$ and $Y|R, Z \sim \nu_{2,d}$ with probability $1 - d^{-2k}$. The two models $\nu_{1,d}$ and $\nu_{2,d}$ are chosen such that optimal sample complexity is achieved at $l_{\star,m} = 10k$ (thanks to $\nu_{2,d}$) and optimal runtime is achieved at $l_{\star,T} = 4k$ (thanks to $\nu_{1,d}$). Thus, in this example:

- **Sample-optimal algorithm:** the harmonic tensor unfolding algorithm at $l_{\star,m} = 10k$ achieves
$$m = \widetilde{\Theta}_d\left(d^{5k}\right), \quad \text{and} \quad T = \widetilde{\Theta}_d\left(d^{10k}\right).$$

- **Runtime-optimal algorithm:** the harmonic tensor unfolding algorithm at $l_{\star,T} = 4k$ achieves
$$m = \widetilde{\Theta}_d\left(d^{6k}\right), \quad \text{and} \quad T = \widetilde{\Theta}_d\left(d^{8k}\right).$$

## 4 Learning Gaussian single-index models

We now specialize our results to the case of Gaussian single-index model (1). Recall that $\nu_{d,R} := \chi_d$ and $\nu_d(Y|R, Z) = \rho(Y|R \cdot Z)$. We assume below that $\rho$ is a fixed link distribution of generative exponent $k_\star > 1$ (defined in Eq. (2)). In particular, the notations $\asymp$ and $\lesssim$ only hide constants that depend on $\rho$. From Section 3, we need to decompose $\rho$ into the spherical harmonics basis. Using the decomposition of Hermite polynomials into Gegenbauer polynomials proved in Appendix B:

$$\mathrm{He}_k(\langle \boldsymbol{w}_\star, \boldsymbol{x} \rangle) = \sum_{\substack{\ell=0 \\ \ell \equiv k \bmod 2}}^{k} \beta_{k,\ell}(\|\boldsymbol{x}\|_2) Q_\ell(\langle \boldsymbol{w}_\star, \boldsymbol{z} \rangle), \qquad \|\beta_{k,\ell}\|_{L^2(\chi_d)}^2 = \Theta_d(d^{-(k-\ell)/2}), \tag{20}$$

we show the following bounds on the $\ell$-th Gegenbauer coefficient $\xi_{d,\ell}$ associated to $\rho$:

**Lemma 1.** *For all $\ell \leq k_\star$, we have*

$$\|\xi_{d,\ell}\|_{L^2}^2 \asymp d^{-(k_\star-\ell)/2} \text{ for } \ell \equiv k_\star \bmod 2 \quad \text{and} \quad \|\xi_{d,\ell}\|_{L^2}^2 \lesssim d^{-(k_\star-\ell+1)/2} \text{ for } \ell \not\equiv k_\star \bmod 2.$$

Plugging these estimates in Theorem 1, we deduce the following complexity lower bounds and associated optimal degrees for learning $\rho$. For sample complexity, we obtain

$$m \gtrsim M_\star(\nu_d) \asymp \inf_{\ell \geq 1} \frac{d^{\ell/2}}{\|\xi_{d,\ell}\|_{L^2}^2} \asymp \inf_{\substack{\ell \geq 1 \\ \ell \equiv k_\star \bmod 2}} \frac{d^{\ell/2}}{d^{-(k_\star-\ell)/2}} \asymp \inf_{\substack{\ell \geq 1 \\ \ell \equiv k_\star \bmod 2}} d^{k_\star/2} \asymp d^{k_\star/2},$$

and any $\ell \leq k_\star$ with $\ell \equiv k_\star \bmod 2$ is sample optimal (for asymptotic rates). For runtime, we get

$$T \gtrsim Q_\star(\nu_d) \asymp \inf_{\ell \geq 1} \frac{d^\ell}{\|\xi_{d,\ell}\|_{L^2}^2} \asymp \inf_{\substack{\ell \geq 1 \\ \ell \equiv k_\star \bmod 2}} \frac{d^\ell}{d^{-(k_\star-\ell)/2}} \asymp \inf_{\substack{\ell \geq 1 \\ \ell \equiv k_\star \bmod 2}} d^{(k_\star+\ell)/2} \asymp d^{\lceil (k_\star+1)/2 \rceil},$$

and only $\ell = 1$ ($k_\star$ odd) or $\ell = 2$ ($k_\star$ even) are runtime optimal. Thus, although $\rho$ has vanishing projection on degree $\ell < k_\star$ harmonic spaces, this is offset by smaller $n_{d,\ell} = \dim(V_{d,\ell})$, and we should always choose $l_{T,\star} = l_{m,\star} \in \{1, 2\}$ (depending on $k_\star$ parity). In particular, we can use our general spectral algorithm (SP-Alg) to learn $\rho$ with optimal sample and runtime complexity:

**Corollary 1.** *The estimator (SP-Alg) achieves optimal* $m = \Theta_d(d^{k_\star/2})$ *and* $T = \Theta_d(d^{k_\star/2+1} \log(d))$.

**Estimating using only directional information.** Consider the same Gaussian SIM with link function $\rho$, but suppose we only observe the pair $(y, \boldsymbol{z})$, i.e., the label and the direction of the input. This defines a new spherical SIM $\nu_d$ with fixed radius $\nu_{d,R} = \delta_{R=1}$ and conditional distribution

$$\nu_d(Y|R, Z) = \mathbb{E}_{\tilde{R}}[\rho(Y|\tilde{R}Z)], \quad \text{where} \ \ \tilde{R} \sim \chi_d.$$

For example, this setting corresponds to the common practice in statistics and machine learning of normalizing input vectors to unit norm. We show below that, in Gaussian single-index models, such normalization necessarily leads to a quadratic increase in runtime complexity. Under this model, the Gegenbauer coefficients of $\nu_d$ scale as:

**Lemma 2.** *For all $\ell \leq \mathsf{k}_\star$, we have*

$$\|\xi_{d,\ell}\|_{L^2}^2 \asymp d^{-(\mathsf{k}_\star - \ell)} \textit{ for } \ell \equiv \mathsf{k}_\star \bmod 2 \qquad \textit{and} \qquad \|\xi_{d,\ell}\|_{L^2}^2 \lesssim d^{-(\mathsf{k}_\star - \ell + 1)} \textit{ for } \ell \not\equiv \mathsf{k}_\star \bmod 2\,.$$

Similarly as above, it is easy to verify that the sample-optimal degree is now always $\mathsf{l}_{\mathsf{m},\star} = \mathsf{k}_\star$, while the runtime-optimal degrees are $\ell \equiv \mathsf{k}_\star \bmod 2$, $\ell \leq \mathsf{k}_\star$. The new complexity lower bounds are given by

$$\mathsf{m} \gtrsim d^{\mathsf{k}_\star/2}, \qquad \mathsf{T} \gtrsim d^{\mathsf{k}_\star}.$$

Thus, while the sample complexity stays the same, the optimal runtime goes from $d^{\mathsf{k}_\star/2+1}$ to $d^{\mathsf{k}_\star}$. Again, we can use our general estimators from Section 3 to match these lower bounds:

**Corollary 2.** *For any $3 \leq \ell \leq \mathsf{k}_\star$, $\ell \equiv \mathsf{k}_\star \bmod 2$, estimator (SGD-Alg) achieves $\mathsf{m} = \Theta_d(d^{\mathsf{k}_\star - 1})$ and $\mathsf{T} = \Theta_d(d^{\mathsf{k}_\star})$. For $\ell = \mathsf{k}_\star$, estimator (TU-Alg) achieves $\mathsf{m} = \Theta_d(d^{\mathsf{k}_\star/2})$ and $\mathsf{T} = \Theta_d(d^{\mathsf{k}_\star+1}\log(d))$.*

The proofs of all the results in this section can be found in Appendix G. We also discuss the general phenomenology underlying algorithms for Gaussian SIMs—namely, vanilla SGD [10], landscape smoothing [28], and partial trace [29]—in Appendices A.3 and A.4.

# 5 Conclusion

In this paper, we introduced a generalization of Gaussian single-index models, which we termed *spherical single-index models*, that allows for arbitrary spherically symmetric input distributions and general dependence of the label on the norm of the input. We provided a sharp characterization of both the statistical and computational complexity of learning in these models. A key insight is that the SQ and LDP lower bounds decouple across the irreducible subspaces $V_{d,\ell}$ of degree-$\ell$ spherical harmonics. For each such subspaces, we established two matching estimators: an online SGD algorithm that achieves the optimal runtime among SQ algorithms, and a harmonic tensor unfolding estimators that achieves the optimal sample complexity among LDP algorithms. The optimal algorithm is then obtained by selecting the degree $\mathsf{l}_{\mathsf{m},\star}$ or $\mathsf{l}_{\mathsf{T},\star}$ that minimizes sample or runtime complexity respectively. In general, these may differ—i.e., $\mathsf{l}_{\mathsf{m},\star} \neq \mathsf{l}_{\mathsf{T},\star}$—implying that no single estimator can achieve both optimal sample and runtime complexity. We applied this framework to the Gaussian case, recovering and unifying prior results while clarifying the role of the harmonic decomposition in their performance. Below, we discuss two directions for future work.

**Multi-index models.** A natural extension is to *multi-index models*, where the label depends on a low-dimensional projection $y \sim \rho(\cdot|\boldsymbol{W}_*^\mathsf{T}\boldsymbol{x})$, with $\boldsymbol{W}_* \in \mathbb{R}^{d\times s}$ an unknown rank-$s$ subspace. Recent work [2, 17] has shown that learning in such models proceeds via a sequential recovery of directions in the signal subspace. Unlike the single-index case—where degree-1 and 2 spherical harmonics suffice—multi-index models require higher-order harmonics. For instance, the multivariate Hermite monomial $x_1 x_2 \cdots x_s = r^s z_1 z_2 \cdots z_s$ is a degree-$\ell$ spherical harmonic with no projection onto lower-degree spaces. In such cases, landscape smoothing and partial trace estimators fail, and the harmonic tensor unfolding estimator on $V_{d,s}$ becomes necessary. We expect our harmonic framework to extend naturally to the spherical multi-index setting, and leave this direction to future work.

**General symmetry groups.** Finally, our lower bounds in Theorem 1—and the decoupling into irreducible subspaces—hinges on Schur's orthogonality relations for the action representation of $\mathcal{O}_d$ on $L^2(\mathbb{S}^{d-1})$. More broadly, the Peter–Weyl theorem ensures that such decompositions exist for any compact group. This suggests that our bounds, and the decoupling into irreducible representations, hold beyond SIMs and $\mathcal{O}_d$. We hope to explore this broader setting in future work.

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

# Supplementary Material for the paper
## "Learning single-index models via harmonic decomposition"

## Contents

# A Additional discussions and details from the main text

## A.1 Related work

The problem of learning single and multi-index models has a long history in statistics and machine learning. We refer the reader to the recent survey [21] and references therein for a more thorough account of this rich history. Early work showed that SIMs with monotonic link function only require $n = \Theta_d(d)$ samples to learn even in the distribution-free setting, using perceptron-like algorithms [52, 51]. For non-monotonic link functions with generative exponent less of equal to 2, such as phase retrieval $y = |\langle \boldsymbol{w}_*, \boldsymbol{x} \rangle|^2 + \varepsilon$, [12, 62] established the information-theoretic limits, including the asymptotic MMSE, in the proportional regime $\alpha = n/d$. In particular, using the conjectured optimality of approximate message passing (AMP) algorithm, they showed that statistical–computational gaps appear in this model, that is, there exists an algorithmic threshold $\alpha_{\mathrm{ALG}} > \alpha_{\mathrm{IT}}$—the information-theoretic threshold—such that, conjecturally, no polynomial-time algorithm succeeds when $\alpha_{\mathrm{IT}} < \alpha < \alpha_{\mathrm{ALG}}$. [62, 66, 61] proposed spectral algorithms that match the algorithmic threshold, with [62] explaining how these spectral methods can be viewed as linearizations of AMP algorithms. In parallel lines of work, [23] and references therein proposed a number of algorithms to learn single and multi-index models, [38, 59, 86, 82] studied the problem of robustly learning a single index model, [37, 42] demonstrated near-optimal SQ lower bounds for learning ReLUs and halfspaces under Gaussian marginals, [36, 35] established cryptographic or worst-case hardness of learning a single ReLU neuron, and [5, 74] showed in the noise-free setting that some lattice-based algorithms can vastly outperform SQ lower bounds. We further emphasize that in the multi-index case, the picture is much richer than in the single-index case: optimal learning happens through an adaptive multi-phase process [3, 1, 2, 17]. Recent work have explored optimal learning of Gaussian multi-index models in the proportional scaling [23, 77, 34, 54] and in the polynomial scaling [27]. We leave the application of our harmonic framework to the multi-index case to future work.

A number of works have studied learning SIMs with gradient algorithms, and in particular, gradient-trained neural networks. [72, 76] studied GD and online SGD for phase retrieval. The notion of an information exponent, which characterizes the sample complexity of learning Gaussian single-index models via online SGD on the square loss, was introduced by [10], sparking a series of follow-up studies [43, 79, 8, 88, 6, 70]. A separate line of research investigates how neural networks can learn in just one gradient step [11, 31]. A two-timescale approach has also been proposed in [15], and subsequently applied in [17, 63] to analyze convergence of gradient flow dynamics in learning multi-index models. This information exponent characterizes the complexity of learning SIMs with CSQ algorithms. [32, 56, 7, 50] showed that one can outperform this information exponent by doing multi-pass gradient descent or by changing the loss function. However, we expect these algorithms to still fall under the SQ framework and be lower bounded by the generative exponent.

As pointed out previously [28, 29], learning single-index models is deeply connected to tensor PCA [67]. Tensor PCA exhibits a statistical-computational gap for $k \geq 3$ [67, 69, 40, 41, 48, 46, 55], and its algorithmic picture is remarkably similar to single-index models, with a correspondence between their information-theoretic, algorithmic, and local-search thresholds. Montanari and Richard [67] proposes power iteration and tensor unfolding to solve tensor PCA. They conjectured that the algorithmic threshold should be given by $\beta \gtrsim d^{(k-2)/4}$, and proves algorithmic achievability with $\beta \gtrsim d^{(\lceil k/2 \rceil - 1)/2}$ using tensor unfolding. They show that local algorithms like tensor power iteration may require even higher SNR of $\beta \gtrsim d^{(k-2)/2}$, subsequently investigated in [57, 9]. The optimal algorithmic threshold is achieved by a number of algorithms: Sum-Of-Squares [47, 45], partial trace algorithm [47], homotopy method [4] for $k = 3$, landscape smoothing introduced by [19], and tensor unfolding [87, 14].

Finally, in a concurrent and independent work, [27] introduced an (unbalanced) tensor unfolding estimator very similar to (18)—with Hermite tensor instead of harmonic tensor—in the context of learning Gaussian multi-index models. This method achieves sharp sample complexity of $n \gtrsim d^{1 \wedge k^*/2}$, where $k^*$ is the leap generative exponent of the link function.

## A.2 Discussion on learning spherical SIMs

**Likelihood ratio.** In our Definition 1, we consider spherical SIMs such that $\nu_d \ll \nu_{d,0}$ and the associated Radon-Nikodym derivative satisfies $\|\frac{d\nu_d}{d\nu_{d,0}}\|_{L^2(\nu_{d,0})} < \infty$. We can expand the likelihood ratio in $L^2(\nu_{d,0})$ into the Gegenbauer basis:

$$\frac{d\nu_d}{d\nu_{d,0}}(y, r, z) \overset{L^2(\nu_{d,0})}{=} 1 + \sum_{\ell=1}^{\infty} \xi_{d,\ell}(y, r) Q_\ell(z), \tag{21}$$

where

$$\xi_{d,\ell}(Y, R) := \mathbb{E}_{Z \sim \tau_{d,1}} \left[ \frac{d\nu_d}{d\nu_{d,0}}(Y, R, Z) Q_\ell(Z) \right] = \mathbb{E}_{\nu_d} [Q_\ell(Z) | Y, R].$$

The mutual $\chi^2$-mutual information divergence of $((Y, R), Z)$ is given by

$$I_{\chi^2}[\nu_d] = \mathbb{E}_{\nu_{d,0}} \left[ \left( \frac{d\nu_d}{d\nu_{d,0}} \right)^2 \right] - 1 = \sum_{\ell=1}^{\infty} \|\xi_{d,\ell}\|_{L^2}.$$

**Link function $\nu_d$ unknown.** When $\nu_d$ is unknown, similar to [29], one can still hope to learn the planted direction $\boldsymbol{w}_*$, as long as it is possible to approximate the non-linearity $\mathcal{T}_\ell(y, r)$ using random linear combinations of the first few orthogonal functions in a basis of $L^2(\nu_{d,Y,R})$. Similar to [29, Assumption 4.1] this requires assuming that the expansion of $\mathcal{T}_\ell$ has non-vanishing mass on these first few basis functions of $L^2(\nu_{d,Y,R})$. Intuitively, this amounts to a 'smoothness' condition on the link function $\nu_d$. We leave this as a direction for future work.

**Weak to strong recovery.** In our framework, it is only meaningful to restrict ourselves to the sequence $\{\nu_d\}_{d \geq 1}$ such that $I_{\chi^2}[\nu_d]$ is non-vanishing and there exists a component $\ell \geq 1$ (independent of $d$) such that $\|\xi_{d,\ell}\|_{L^2} > c > 0$. For example, in Gaussian SIMs with the generative exponent $\mathsf{k}_\star$, such an $\ell = \mathsf{k}_\star$ always (both with or without using the norm). Under this mild assumption, we can carry out the online SGD algorithm similar to the final phase algorithms of [10, 28, 88] but now on the frequency $Q_\ell$. As we have non-vanishing signal $\|\xi_{d,\ell}\| = \Omega_d(1)$, we can achieve strong recovery $|\langle \hat{\boldsymbol{w}}, \boldsymbol{w}_* \rangle| \geq 1 - \varepsilon$ using $O(d/\varepsilon)$ samples, and so $O(d^2/\varepsilon)$ runtime hiding constants in $\ell$.

**Spherical SIMs with sample-runtime trade-off.** Below we show how to construct examples of spherical single-index models with $\mathsf{l}_{m,\star} > \mathsf{l}_{\mathsf{T},\star}$. We construct $\nu_d$ such that it is a mixture of two spherical SIMs. Consider $\nu_d^{(1)}$ and $\nu_d^{(2)}$ associated to two Gaussian SIMs with generative exponents $k_1$ and $k_2$, where we marginalized over the norm (that is, $R = 1$). In particular, as shown in [29, Theorem 5.1], we can choose the Gaussian SIMs to be $y^{(j)} = \sigma_j(R \cdot Z) + \tau \mathsf{N}(0, 1)$, with $\|\sigma_j\|_\infty < \infty$ and $\tau$ sufficiently large such that $C^{-1} \leq \nu_d^{(1)}(y)/\nu_d^{(2)}(y) \leq C$.

Assume that $(Y, Z) \sim \nu_d$ (recall $R = 1$ here, and we remove it for clarity) is drawn with probability $d^{-\alpha}$ from $\nu_d^{(1)}$ and with probability $1 - d^{-\alpha}$ from $\nu_d^{(2)}$, where $\alpha > 0$ is a constant chosen later. In

this model, we have

$$\xi_{d,\ell}(Y) = \mathbb{E}_{\nu_d}[Q_\ell(Z)|Y] = d^{-\alpha}C_\alpha^{(1)}(Y)\xi_{d,\ell}^{(1)}(Y) + C_\alpha^{(2)}(Y)\xi_{d,\ell}^{(2)}(Y),$$

where

$$\xi_{d,\ell}^{(1)}(Y) = \mathbb{E}_{\nu_d^{(1)}}[Q_\ell(Z)|Y], \qquad \xi_{d,\ell}^{(2)}(Y) = \mathbb{E}_{\nu_d^{(2)}}[Q_\ell(Z)|Y],$$

and

$$C_\alpha^{(1)}(Y) = \frac{1}{d^{-\alpha} + (d^\alpha - 1)\nu_d^{(2)}(Y)/\nu_d^{(1)}(Y)}, \qquad C_\alpha^{(2)}(Y) = \frac{1}{d^{-\alpha}\nu_d^{(1)}(Y)/\nu_d^{(2)}(Y) + 1 - d^{-\alpha}}.$$

From our choice of $\nu_d^{(j)}$, there exists a constant $C$, such that for $d \geq C$, we have $C^{-1} \leq C_\alpha^{(1)}(y), C_\alpha^{(2)}(y) \leq C$ for all $y \in \mathbb{R}$. We deduce that there exist a constant $\tilde{C}$ sufficiently large but independent of $d$ such that

$$\|\xi_{d,\ell}\|_{L^2}^2 \leq \tilde{C}\max(d^{-2\alpha}\|\xi_{d,\ell}^{(1)}\|_{L^2}^2, \|\xi_{d,\ell}^{(2)}\|_{L^2}^2),$$
$$\|\xi_{d,\ell}\|_{L^2}^2 \geq \tilde{C}^{-1}\max\left(d^{-2\alpha}\|\xi_{d,\ell}^{(1)}\|_{L^2}^2 - \tilde{C}^2\|\xi_{d,\ell}^{(2)}\|_{L^2}^2, \|\xi_{d,\ell}^{(2)}\|_{L^2}^2 - \tilde{C}^2d^{-2\alpha}\|\xi_{d,\ell}^{(1)}\|_{L^2}^2\right), \tag{22}$$

where the $L^2$-norms are with respect to the associated nulls $\nu_{d,0}$, $\nu_{d,0}^{(1)}$, and $\nu_{d,0}^{(2)}$.

Consider $\mathsf{k}_\star$ a multiple of 10 for simplicity, and set $k_2 = \mathsf{k}_\star$, $k_1 = 2\mathsf{k}_\star/5$, and $\alpha = \mathsf{k}_\star/5$. Using Lemma 2, we can bound the contributions from $\nu_d^{(1)}$ and $\nu_d^{(2)}$:

- Consider the contributions of $\nu_d^{(1)}$ to the sample and runtime complexity:
  - Sample complexity:

$$\ell \leq k_1, \ \ell \equiv k_1[2]: \qquad d^{2\alpha}\frac{\sqrt{n_{d,\ell}}}{\|\xi_{d,\ell}^{(1)}\|_{L^2}^2} \asymp d^{2k_1-\ell/2} = d^{4\mathsf{k}_\star/5-\ell/2},$$

$$\ell \leq k_1, \ \ell \not\equiv k_1[2]: \qquad d^{2\alpha}\frac{\sqrt{n_{d,\ell}}}{\|\xi_{d,\ell}^{(1)}\|_{L^2}^2} \asymp d^{2k_1-\ell/2+1} = d^{4\mathsf{k}_\star/5-\ell/2+1}, \tag{23}$$

$$\ell > k_1: \qquad d^{2\alpha}\frac{\sqrt{n_{d,\ell}}}{\|\xi_{d,\ell}^{(1)}\|_{L^2}^2} \gtrsim d^{k_1+\ell/2} = d^{2\mathsf{k}_\star/5+\ell/2}.$$

Thus the optimal sample complexity is achieved at degree $\ell = k_1 = 2\mathsf{k}_\star/5$ with $d^{3\mathsf{k}_\star/5}$ lower bound.
  - Runtime complexity:

$$\ell \leq k_1, \ \ell \equiv k_1[2]: \qquad d^{2\alpha}\frac{n_{d,\ell}}{\|\xi_{d,\ell}^{(1)}\|_{L^2}^2} \asymp d^{2k_1} = d^{4\mathsf{k}_\star/5},$$

$$\ell \leq k_1, \ \ell \not\equiv k_1[2]: \qquad d^{2\alpha}\frac{n_{d,\ell}}{\|\xi_{d,\ell}^{(1)}\|_{L^2}^2} \asymp d^{2k_1+1} = d^{4\mathsf{k}_\star/5+1}, \tag{24}$$

$$\ell > k_1: \qquad d^{2\alpha}\frac{n_{d,\ell}}{\|\xi_{d,\ell}^{(1)}\|_{L^2}^2} \gtrsim d^{k_1+\ell} = d^{2\mathsf{k}_\star/5+\ell}.$$

Thus the optimal runtime complexity is achieved at degrees $\ell \leq k_1 = 2\mathsf{k}_\star/5$, $\ell \equiv k_1[2]$ with $d^{4\mathsf{k}_\star/5}$ lower bound.
- Consider the contributions of $\nu_d^{(2)}$ to the sample and runtime complexity:
  - Sample complexity:

$$\ell \leq k_2, \ \ell \equiv k_2[2]: \qquad \frac{\sqrt{n_{d,\ell}}}{\|\xi_{d,\ell}^{(2)}\|_{L^2}^2} \asymp d^{k_2-\ell/2} = d^{\mathsf{k}_\star-\ell/2},$$

$$\ell \leq k_2, \ \ell \not\equiv k_2[2]: \qquad \frac{\sqrt{n_{d,\ell}}}{\|\xi_{d,\ell}^{(2)}\|_{L^2}^2} \asymp d^{k_1-\ell/2+1} = d^{\mathsf{k}_\star-\ell/2+1}, \tag{25}$$

$$\ell > k_2: \qquad \frac{\sqrt{n_{d,\ell}}}{\|\xi_{d,\ell}^{(2)}\|_{L^2}^2} \gtrsim d^{\ell/2}.$$

Thus the optimal sample complexity is achieved at degree $\ell = k_2 = \mathsf{k}_\star$ with $d^{\mathsf{k}_\star/2}$ lower bound.

– Runtime complexity:

$$
\begin{aligned}
\ell \leq k_2, \; \ell \equiv k_2[2] : & \qquad \frac{n_{d,\ell}}{\|\xi_{d,\ell}^{(2)}\|_{L^2}^2} \asymp d^{k_2} = d^{\mathsf{k}_\star}, \\
\ell \leq k_2, \; \ell \not\equiv k_2[2] : & \qquad \frac{n_{d,\ell}}{\|\xi_{d,\ell}^{(2)}\|_{L^2}^2} \asymp d^{k_2+1} = d^{\mathsf{k}_\star+1}, \\
\ell > k_2 : & \qquad \frac{n_{d,\ell}}{\|\xi_{d,\ell}^{(2)}\|_{L^2}^2} \gtrsim d^{\ell}.
\end{aligned}
\tag{26}
$$

Thus the optimal runtime complexity is achieved at degrees $\ell \leq k_2 = \mathsf{k}_\star, \; \ell \equiv k_2[2]$ with $d^{\mathsf{k}_\star}$ lower bound.

From the bounds (22), the sample or runtime complexity for each $\ell$ is the minimum of the two contributions associated to $\nu_d^{(1)}$ and $\nu_d^{(2)}$. We deduce that in this model:

- Optimal sample complexity is achieved at degree $\mathsf{l}_{\mathsf{m},\star} = \mathsf{k}_\star$, with a matching algorithm that succeeds with $\mathsf{m} = \Theta_d(d^{\mathsf{k}_\star/2})$ samples and $\mathsf{T} = \Theta(d^{\mathsf{k}_\star})$ runtime (thanks to contribution $\nu_d^{(2)}$).
- Optimal runtime is achieved at degrees $\mathsf{l}_{\mathsf{T},\star} = \ell$ with $\ell \leq 2\mathsf{k}_\star/5$ and $\ell \equiv 2\mathsf{k}_\star/5[2]$. For example, choosing $\mathsf{l}_{\mathsf{T},\star} = 2\mathsf{k}_\star/5$, we have a matching algorithm that succeeds with $\mathsf{T} = \Theta_d(d^{4\mathsf{k}_\star/5})$ runtime and $\mathsf{m} = \Theta_d(d^{3\mathsf{k}_\star/5})$ samples (thanks to contribution $\nu_d^{(1)}$).

We conjecture that for this distribution $\nu_d$, no algorithm exists that achieves both optimal sample complexity $\mathsf{m} = \Theta_d(d^{\mathsf{k}_\star/2})$ and optimal runtime complexity $\mathsf{T} = \Theta_d(d^{4\mathsf{k}_\star/5})$. Further note, that by choosing intermediary degrees $\ell$, one can trade-off sample and runtime complexity.

### A.3 Revisiting vanilla SGD, landscape smoothing, and partial trace for Gaussian SIMs

In light of our results in Section 4, we revisit the three algorithms for learning Gaussian SIMs mentioned in the introduction [10, 28, 29], and reinterpret their behavior through the lens of harmonic decomposition. Below, we state informal observations which aim to build intuition rather than make formal statements. We provide supporting computations in Appendix A.4.

Consider a Gaussian single-index model (1) with link function $\rho \in \mathcal{P}(\mathbb{R}^2)$ and generative exponent $\mathsf{k}_\star := \mathsf{k}_\star(\rho)$ as defined in Eq. (2). Throughout, let $\mathcal{T}_\star : \mathbb{R} \to \mathbb{R}$ be a transformation of the label satisfying:

$$
\|\mathcal{T}_\star\|_{L^2} = 1, \qquad \|\mathcal{T}_\star\|_\infty \leq C, \qquad \Gamma_{\mathsf{k}_\star} := \mathbb{E}_\rho[\mathcal{T}_\star(Y)\mathrm{He}_{\mathsf{k}_\star}(G)] \geq \frac{1}{C}\|\zeta_{\mathsf{k}_\star}\|_{L^2}.
$$

Such transformations always exist (see [29, Lemma F.2]). Informally, one can construct $\mathcal{T}_\star$ by truncating $\zeta_{\mathsf{k}_\star}/\|\zeta_{\mathsf{k}_\star}\|_{L^2}$ (with truncation at large enough value as to approximately preserve the correlation with $\mathrm{He}_{\mathsf{k}_\star}$).

**Online SGD with Hermite neuron.** In a seminal paper, Ben Arous et al. [10] studied online SGD on a non-convex loss over $\boldsymbol{w} \in \mathbb{S}^{d-1}$, with planted signal $\boldsymbol{w}_\ast$ and a $\mathsf{k}_\star$-order saddle at the equator $\langle \boldsymbol{w}, \boldsymbol{w}_\ast \rangle = 0$. Adapting their results to the task of learning Gaussian SIMs, their algorithm performs online SGD on the population loss

$$
\min_{\boldsymbol{w} \in \mathbb{S}^{d-1}} \mathcal{L}(\boldsymbol{w}) := \mathbb{E}_{(y,\boldsymbol{x}) \sim \mathbb{P}_{\boldsymbol{w}_\ast}}\left[\left(\mathcal{T}_\star(y) - \mathrm{He}_{\mathsf{k}_\star}(\langle \boldsymbol{w}, \boldsymbol{x} \rangle)\right)^2\right], \tag{HeSGD}
$$

and succeeds with suboptimal $\mathsf{m} = \widetilde{\Theta}_d(d^{\mathsf{k}_\star-1})$ samples and $\mathsf{T} = \widetilde{\Theta}_d(d^{\mathsf{k}_\star})$.

**Observation 1** (Informal, harmonic structure of the loss). *When $|\langle \boldsymbol{w}, \boldsymbol{w}_\ast \rangle| \gtrsim d^{-1/2}$, the loss landscape $\mathcal{L}(\boldsymbol{w})$ is dominated by degree-$\mathsf{k}_\star$ spherical harmonics. The resulting SGD dynamics behaves similarly to online SGD using $Q_{\mathsf{k}_\star}(\langle \boldsymbol{w}, \boldsymbol{z} \rangle)$ in place of $\mathrm{He}_{\mathsf{k}_\star}(\langle \boldsymbol{w}, \boldsymbol{x} \rangle)$.*

This suggests that algorithm (HeSGD) effectively restricts itself to the degree-$k_\star$ subspace $V_{d,k_\star}$. As a consequence, we expect its performance to be constrained by the query complexity lower bound $\Omega_d(d^{k_\star})$, and its behavior to be similar to the degree-$k_\star$ online SGD estimator (SGD-Alg) in Section 3. We further provide an alternative perspective that highlights the role of the norm $\|x\|_2$ of the input data in learning Gaussian SIMs:

**Observation 2** (Informal, norm-invariance of dynamics). *The SGD dynamics of [10] remains essentially the same if the input $x$ is replaced by $\tilde{r} \cdot x/\|x\|_2$, where $\tilde{r} \sim \chi_d$ is sampled independently.*

This indicates that the algorithm does not exploit the radial component $\|x\|_2$, and effectively operates on the normalized direction $z = x/\|x\|_2$. From our theory (Section 4), any such estimator incurs a query complexity of $\Omega_d(d^{k_\star})$. In this sense, algorithm (HeSGD) is runtime-optimal among methods that ignore radial information.

**Landscape smoothing.** To address the suboptimality of (HeSGD), Damian et al. [28] introduced a landscape smoothing operator that averages the loss on a sphere around each parameter $w \in \mathbb{S}^{d-1}$:

$$\min_{w \in \mathbb{S}^{d-1}} \mathbb{E}_{u \sim \mathrm{Unif}(\mathbb{S}^{d-1})} \left[ \mathcal{L} \left( \frac{w + \lambda u}{\|w + \lambda u\|_2} \right) \right]. \tag{SmLD}$$

This modification achieves near-optimal complexities: $m = \widetilde{\Theta}_d(d^{k_\star/2})$ and $\mathsf{T} = \widetilde{\Theta}_d(d^{k_\star/2+1})$.

**Observation 3** (Informal, low-pass filtering effect). *Landscape smoothing suppresses high-frequency components of the loss, effectively amplifying lower-degree harmonics. The initial phase of SGD dynamics behaves essentially like optimization over $Q_1(\langle w, z \rangle)$ and $Q_2(\langle w, z \rangle)$.*

Thus, smoothing can be interpreted as projecting the dynamics onto low-degree harmonic components—specifically, the statistics of the spectral algorithm (SP-Alg) associated to spherical harmonics of degree $\ell \in \{1, 2\}$. In this sense, the first phase of the dynamics on (SmLD) essentially corresponds to running SGD on the optimal spectral estimator (SP-Alg).

**Partial trace estimator.** In a subsequent work, Damian et al. [29] proposed an estimator based on the partial trace of a Hermite tensor, inspired by techniques from tensor PCA. Their construction begins with the empirical Hermite tensor:

$$\hat{T} := \frac{1}{m} \sum_{i \in [m]} \mathcal{T}_\star(y_i) \mathbf{He}_{k_\star}(x_i) \in (\mathbb{R}^d)^{\otimes k}, \tag{27}$$

where $\mathbf{He}_{k_\star}(x)$ denotes the rank-$k_\star$ multivariate Hermite tensor, and $\mathcal{T}_\star$ is the transformation defined earlier. The expectation $\mathbb{E}[\hat{T}]$ is proportional to $w_\star^{\otimes k_\star}$. To extract this principal component, they compute a *partial trace* of the empirical tensor by contracting $\hat{T}$ with identity tensors. This results in an empirical vector or matrix, depending on whether $k_\star$ is odd or even. The resulting estimator is

$$k_\star \text{ odd:} \quad \hat{w}_0 = \frac{\hat{v}}{\|\hat{v}\|_2}, \qquad \hat{v} := \frac{1}{m} \sum_{i \in [m]} \mathcal{T}_\star(y_i) P_{k_\star}(\|x_i\|_2) x_i,$$

$$k_\star \text{ even:} \quad \hat{w} = \arg\max_{w \in \mathbb{S}^{d-1}} w^\mathsf{T} \hat{M} w, \qquad \hat{M} := \frac{1}{m} \sum_{i=1}^m \mathcal{T}_\star(y_i) P_{k_\star}(\|x_i\|_2) \left[ x_i x_i^\mathsf{T} - \mathbf{I}_d \right], \tag{PrTR}$$

where $P_{k_\star}$ is a univariate polynomial derived from the contraction of the Hermite tensor. In the odd case, a second refinement phase (see Section D) is used to boost $\hat{w}_0$ from $d^{-1/4}$ to constant correlation with $w_\star$. This estimator achieves the optimal sample complexity $\Theta_d(d^{k_\star/2})$ and (near) optimal runtime $\Theta_d(d^{k_\star/2+1} \log(d))$, matching the lower bounds for learning Gaussian SIMs (3).

Importantly, estimator (PrTR) corresponds precisely to the general spectral estimator (SP-Alg), associated to the optimal degree $\ell \in \{1, 2\}$ harmonic subspaces, using

$$\mathcal{T}_\ell(y, r) = \mathcal{T}_\star(y) P_{k_\star}(r) \quad \text{with } \ell = 1 \text{ if } k_\star \text{ is odd, } \quad \text{and } \ell = 2 \text{ if } k_\star \text{ is even.}$$

**Observation 4** (Informal, lower-frequency projection). *Partial trace effectively projects the high-degree Hermite tensor onto lower-degree spherical harmonic subspaces ($\ell = 1$ or $2$ for partial trace over all but $1$ or $2$ coordinates).*

Although our estimator recovers (PrTR) in the Gaussian case, we emphasize that its derivation is very different (including constructing the non-linearity $\mathcal{T}_\ell(y, r)$, see Appendix G). We construct it directly from rotational invariance and harmonic decomposition, without relying on prior knowledge of tensor PCA, contractions of Hermite tensors, or Gaussian-specific identities. We believe this alternative, transparent derivation of (PrTR) highlights the advantages of the harmonic perspective when learning single-index models.

**Remark A.1.** We further remark that if we normalize the input $\boldsymbol{x}$ and apply landscape smoothing or partial trace algorithms to the data $(y_i, \sqrt{d} \cdot \boldsymbol{x}_i/\|\boldsymbol{x}_i\|_2)_{i \in [m]}$, the sample complexity increases to at least $d^{\mathsf{k}_\star - 1}$, and these estimators become suboptimal.

**Remark A.2.** Both landscape smoothing [19] and partial trace estimators [47] originate in the tensor PCA literature, where a similar gap between 'local' and optimal algorithms arises—with $d^{\mathsf{k}_\star/2}$ versus $d^{\mathsf{k}_\star - 1}$ gap in signal strength. It is intriguing to connect the phenomena observed in Gaussian single-index models (Observations 3 and 4) to analogous behaviors in tensor PCA. We leave this direction for future work.

### A.4 Discussion on Gaussian SIM phenomenology

We provide below quick computations to justify the observations in Appendix A.3.

**Observation 1: harmonic decomposition of the loss.** First, note that

$$\mathcal{L}(\boldsymbol{w}) = 2 - 2\beta_{\mathsf{k}_\star} \mathbb{E}_{\boldsymbol{x}}[\mathrm{He}_{\mathsf{k}_\star}(\langle \boldsymbol{w}_*, \boldsymbol{x} \rangle)\mathrm{He}_{\mathsf{k}_\star}(\langle \boldsymbol{w}, \boldsymbol{x} \rangle)] = 2 - 2\Gamma_{\mathsf{k}_\star} \langle \boldsymbol{w}_*, \boldsymbol{w} \rangle^k,$$

and it is enough to consider the correlation loss. Let's decompose the landscape into contributions from the different harmonic subspaces: using the Hermite to Gegenbauer polynomial decomposition in Eq. (20), we get

$$\begin{aligned}
\mathbb{E}_{\boldsymbol{x}}[\mathrm{He}_{\mathsf{k}_\star}(\langle \boldsymbol{w}_*, \boldsymbol{x} \rangle)\mathrm{He}_{\mathsf{k}_\star}(\langle \boldsymbol{w}, \boldsymbol{x} \rangle)] &= \sum_{\substack{\ell \leq k \\ \ell \equiv \overline{\mathsf{k}}_\star[2]}} \mathbb{E}[\beta_{\mathsf{k}_\star, \ell}(r)^2]\mathbb{E}[Q_\ell(\langle \boldsymbol{w}_*, \boldsymbol{z} \rangle)Q_\ell(\langle \boldsymbol{w}, \boldsymbol{z} \rangle)] \\
&= \sum_{\substack{\ell \leq k \\ \ell \equiv \overline{k}[2]}} \frac{\|\beta_{\mathsf{k}_\star, \ell}\|_{L^2}^2}{\sqrt{n_{d,\ell}}} Q_\ell(\langle \boldsymbol{w}, \boldsymbol{w}_* \rangle),
\end{aligned} \quad (28)$$

where $\|\beta_{\mathsf{k}_\star, \ell}\|_{L^2}^2/\sqrt{n_{d,\ell}} = \Theta_d(d^{-\mathsf{k}_\star/2})$ (see Appendix B). For $|\langle \boldsymbol{w}_*, \boldsymbol{w} \rangle| \geq C_{\mathsf{k}_\star} d^{-1/2}$, the leading contribution in the loss (and its gradient) is $\ell = \mathsf{k}_\star$ (recall that the leading term in $Q_\ell(\langle \boldsymbol{w}, \boldsymbol{w}_* \rangle)$ is $\Theta_d(d^{\ell/2})\langle \boldsymbol{w}, \boldsymbol{w}_* \rangle^\ell$). Informally, this implies that we could have replaced $\mathrm{He}_{\mathsf{k}_\star}(\langle \boldsymbol{w}_*, \boldsymbol{x} \rangle)$ by $Q_{\mathsf{k}_\star}(\langle \boldsymbol{w}_*, \boldsymbol{z} \rangle)$ in the above loss.

**Observation 2: dynamics with independent norm.** Let's consider the loss (28) when we have independent norms between the input and the signal:

$$\mathbb{E}[\mathrm{He}_{\mathsf{k}_\star}(r \cdot \langle \boldsymbol{w}_*, \boldsymbol{z} \rangle)\mathrm{He}_{\mathsf{k}_\star}(\tilde{r} \cdot \langle \boldsymbol{w}, \boldsymbol{z} \rangle)] = \sum_{\substack{\ell \leq k \\ \ell \equiv \overline{\mathsf{k}}_\star[2]}} \frac{\mathbb{E}[\beta_{\mathsf{k}_\star, \ell}(r)]^2}{\sqrt{n_{d,\ell}}} Q_\ell(\langle \boldsymbol{w}, \boldsymbol{w}_* \rangle), \quad (29)$$

where $\mathbb{E}[\beta_{\mathsf{k}_\star, \ell}(r)]^2/\sqrt{n_{d,\ell}} = \Theta_d(d^{-\mathsf{k}_\star + \ell/2})$ (see Appendix B). In particular, the leading term $\ell = \mathsf{k}_\star$ remain the same between Eq. (29) and Eq. (28). Following the proof in [10] (see Section E), the dynamics with same hyperparameters behaves similarly between the two losses (29) and (28).

**Observation 3: low-pass filtering of landscape smoothing.** Again, it is enough to directly consider the correlation term. Let's decompose

$$\mathbb{E}_{\boldsymbol{u} \sim \tau_d} \mathbb{E}_{\boldsymbol{x}}\left[\mathrm{He}_{\mathsf{k}_\star}(\langle \boldsymbol{w}_*, \boldsymbol{x} \rangle)\mathrm{He}_{\mathsf{k}_\star}\left(\frac{\boldsymbol{w} + \lambda \boldsymbol{u}}{\|\boldsymbol{w} + \lambda \boldsymbol{u}\|_2} \cdot \boldsymbol{x}\right)\right] = \sum_{\substack{\ell \leq k \\ \ell \equiv \overline{\mathsf{k}}_\star[2]}} m_\ell(\lambda) \frac{\|\beta_{\mathsf{k}_\star, \ell}\|_{L^2}^2}{\sqrt{n_{d,\ell}}} Q_\ell(\langle \boldsymbol{w}, \boldsymbol{w}_* \rangle),$$

where each frequency (28) in the original loss $\mathcal{L}(\boldsymbol{w})$ is now reweighted by

$$m_\ell(\lambda) = \frac{1}{\sqrt{n_{d,\ell}}}\mathbb{E}_{\boldsymbol{u}}\left[Q_\ell\left(\frac{\boldsymbol{w} + \lambda \boldsymbol{u}}{\|\boldsymbol{w} + \lambda \boldsymbol{u}\|_2} \cdot \boldsymbol{w}\right)\right] = \frac{1}{\sqrt{n_{d,\ell}}}\mathbb{E}_{Z \sim \tau_{d,1}}\left[Q_\ell\left(\frac{1 + \lambda Z}{\sqrt{1 + 2\lambda Z + \lambda^2}}\right)\right].$$

When $\lambda = 0$, we indeed have $m_\ell(0) = Q_\ell(1)/\sqrt{n_{d,\ell}} = 1$. For $\lambda \gg 1$, we have $m_\ell(\lambda) \asymp 1/\lambda^\ell$, and as long as $|\langle \boldsymbol{w}, \boldsymbol{w}_* \rangle| \ll \lambda^{-1}$, the loss (and its gradient) are dominated by frequencies $\ell \in \{1, 2\}$.

## A.5 Correlation queries and the information exponent

In the main text, we focused on the *generative exponent* introduced by [29]: this notion tightly capture the optimal complexity of learning Gaussian single-index models among Statistical Query and Low-Degree Polynomial algorithms. An earlier notion—the *information exponent*—was proposed in [40, 10]. Specifically, for scalar labels $\mathcal{Y} \subseteq \mathbb{R}$, the *information exponent (IE)* of $\rho$ is defined by

$$\mathsf{k_I}(\rho) = \arg\min\{k \geq 1 \ : \ \mathbb{E}_\rho[Y \mathrm{He}_k(G)] \neq 0\}. \tag{30}$$

This exponent captures the complexity of learning with so-called *correlation statistical query* (CSQ) algorithms, which only access labels through correlation statistics $y\phi(\boldsymbol{x})$. In other words, using the terminology introduced in Appendix C.1, it captures the complexity of $\mathcal{Q}$-restricted SQ algorithms, with

$$\mathcal{Q} = \mathcal{Q}_{\mathsf{CSQ}} := \{\phi(y, \boldsymbol{x}) = y\tilde{\phi}(x) : \ \tilde{\phi} \text{ measurable function}\}.$$

We denote $\mathsf{CSQ}(q, \tau) := \mathcal{Q}_{\mathsf{CSQ}}\text{-}\mathsf{SQ}(q, \tau)$ this restricted class of SQ algorithms.

For CSQ algorithm, Damian et al. [28] showed lower bounds within the $\mathcal{Q}$-SQ framework of

$$\mathsf{m} = \Theta_d(d^{\mathsf{k_I}(\rho)/2}), \qquad \mathsf{T} = \Theta_d(d^{\mathsf{k_I}(\rho)/2+1}). \tag{31}$$

Note however, that only the generative exponent reflects the fundamental hardness of the learning task: indeed, we always have $\mathsf{k_\star}(\rho) \leq \mathsf{k_I}(\rho)$, with $\mathsf{k_\star}(\rho)$ always one or two for all $y$ polynomial function of $\boldsymbol{x}$ (while $\mathsf{k_I}(\rho) = k$ if $y = \mathrm{He}_k(G)$). In the case $\mathsf{k_\star}(\rho) < \mathsf{k_I}(\rho)$, the complexity predicted by the information exponent can be improved upon by using non-correlation queries, such as using a non-correlation loss [29] or by reusing samples [32]. Nonetheless, IE remains relevant in several natural settings, such as online stochastic gradient descent on the squared or cross-entropy loss.

Below, we discuss how to recover this information exponent from our harmonic framework when considering $\mathcal{Q}_{\mathsf{CSQ}}$-SQ algorithms. Introduce the CSQ query complexity

$$\mathsf{Q}_\star^{\mathsf{CSQ}}(\nu_d) = \min_{\ell \geq 1} \frac{n_{d,\ell}}{\|\xi_{d,\ell}^{\mathsf{CSQ}}\|_{L^2}^2},$$

where we defined

$$\xi_{d,\ell}^{\mathsf{CSQ}}(Y, R) := Y q_{\star,\ell}(R), \qquad q_{\star,\ell}(R) := \frac{1}{\|Y\|_{L^2}} \mathbb{E}_{\nu_d}[Y Q_\ell(Z)|R].$$

Adapting the proofs in Appendix C.1, we obtain the following query complexity lower bound:

**Proposition 1** (CSQ lower bound)**.** *Fix* $\nu_d \in \mathfrak{L}_d$. *If an algorithm* $\mathcal{A} \in \mathsf{CSQ}(q, \tau)$ *succeeds at distinguishing* $\mathbb{P}_{\nu_d, \boldsymbol{w}}$ *from* $\mathbb{P}_{\nu_d, 0}$, *then we must have*

$$q/\tau^2 \geq \mathsf{Q}_\star^{\mathsf{CSQ}}(\nu_d). \tag{32}$$

Using the non-linearity $\mathcal{T}_\ell(Y, R) := Y q_{\star,\ell}(R)$ in our algorithms (SP-Alg), (SGD-Alg) and (TU-Alg) described in Section 3, we can prove the same Theorems 2, 3, and 4 with sample complexities replaced by $\sqrt{n_{d,\ell}}/\|\xi_{d,\ell}^{\mathsf{CSQ}}\|_{L^2}^2$ and runtime complexities replaced by $n_{d,\ell}/\|\xi_{d,\ell}^{\mathsf{CSQ}}\|_{L^2}^2$ (one simply plug these nonlinearities in the theorems in Appendices D, E and F).

Specializing to the Gaussian case, one recover the exact same result as in Section 4, but now with $\mathsf{k_\star}$ (generative exponent) replaced by $\mathsf{k_I}$ (information exponent) of the Gaussian SIM $\rho$. In particular, for all $\ell \leq \mathsf{k_I}$, we have

$$\|\xi_{d,\ell}^{\mathsf{CSQ}}\|_{L^2}^2 \asymp d^{-(\mathsf{k_I}-\ell)/2} \text{ for } \ell \equiv \mathsf{k_I} \bmod 2 \quad \text{and} \quad \|\xi_{d,\ell}^{\mathsf{CSQ}}\|_{L^2}^2 \lesssim d^{-(\mathsf{k_I}-\ell+1)/2} \text{ for } \ell \not\equiv \mathsf{k_I} \bmod 2.$$

Similarly to the generative exponent case (and general SQ), the optimal degrees for learning Gaussian SIMs with CSQ algorithms are always achieves at $\mathsf{l}_{\mathsf{m},\star} = \mathsf{l}_{\mathsf{T},\star} \in \{1, 2\}$, with the spectral estimator (SP-Alg) achieving

$$\mathsf{m} = \Theta_d(d^{\mathsf{k_I}(\rho)/2}), \qquad \mathsf{T} = \Theta_d(d^{\mathsf{k_I}(\rho)/2+1}).$$

Similar results as in Section 4 hold for learning with CSQ algorithms without using the norm $\|\boldsymbol{x}\|_2$. We note, however, that here, non-CSQ algorithms can achieve much better performance (attaining the complexity predicted by the generative exponent).

# B Harmonic analysis on the sphere

In this section, we overview some basic properties of spherical harmonics, Gegenbauer polynomials, and Hermite polynomials. We refer the reader to [75, 24, 25] for additional background. In addition to these classical results, we provide an explicit harmonic decomposition of Hermite polynomials into Gegenbauer polynomials, which we use in our analysis of Gaussian single-index models.

## B.1 Spherical harmonics, Gegenbauer and Hermite polynomials

**Spherical Harmonics.** Consider the $d$-dimensional sphere $\mathbb{S}^{d-1} := \{z : \|z\|_2 = 1\}$ with uniform probability measure $\tau_d \equiv \mathrm{Unif}(\mathbb{S}^{d-1})$, and its associated function space $L^2(\mathbb{S}^{d-1}) := L^2(\mathbb{S}^{d-1}, \tau_d)$ equipped with the inner product:

$$\langle f, g \rangle_{L^2(\mathbb{S}^{d-1})} = \int_{z \in \mathbb{S}^{d-1}} f(z)g(z)\, \tau_d(\mathrm{d}z), \quad \text{for any } f, g \in L^2(\mathbb{S}^{d-1}).$$

We will denote $\langle \cdot, \cdot \rangle_{L^2} := \langle \cdot, \cdot \rangle_{L^2(\mathbb{S}^{d-1})}$ and $\|f\|_{L^2} = \langle f, f \rangle_{L^2(\mathbb{S}^{d-1})}^{1/2}$ when clear from context.

For $\ell \in \mathbb{N}$, consider $\tilde{V}_{d,\ell}$ be the space of degree $\ell$ homogeneous harmonic polynomials (i.e. homogeneous polynomial $q : \mathbb{R}^d \to \mathbb{R}$ with $\Delta q(\cdot) \equiv 0$). Let $V_{d,\ell}$ be the space of functions by restricting the domain to $\mathbb{S}^{d-1}$ of functions in $\tilde{V}_{d,\ell}$, that is degree-$\ell$ spherical harmonics on $\mathbb{S}^{d-1}$. We have the following orthogonal decomposition

$$L^2(\mathbb{S}^{d-1}) = \bigoplus_{\ell=0}^{\infty} V_{d,\ell}. \tag{33}$$

The dimension of each subspace is given by:

$$\dim(V_{d,\ell}) = n_{d,\ell} = \frac{d + 2\ell - 2}{d - 2}\binom{d + \ell - 3}{\ell}.$$

For each $\ell \in \mathbb{N}$, we further fix $\{Y_{\ell i}^{(d)} : i \in n_{d,\ell}\}$ an orthonormal basis on $V_{d,\ell}$:

$$\langle Y_{\ell i}^{(d)}, Y_{kj}^{(d)} \rangle_{\tau_d} = \delta_{\ell k}\delta_{ij}.$$

**Remark B.1.** If one considers the unitary representation $\rho : \mathcal{O}_d \to U(L^2(\mathbb{S}^{d-1}))$ of the orthogonal group $\mathcal{O}_d = \{R \in \mathbb{R}^{d \times d} : R^\mathsf{T} R = I_d\}$ given by

$$\rho(R)\, f(z) = f(R^\mathsf{T} z).$$

The decomposition (33) corresponds to the irreducible decomposition of this representation, that is, the decomposition of $L^2(\mathbb{S}^{d-1})$ into a direct sum of irreducible representations of $\mathcal{O}_d$ (see [25] for a detailed treatment on the subject).

**Gegenbauer Polynomials.** Let $\tau_{d,1}$ denote the marginal distribution of the first coordinate $\langle z, e_1 \rangle$ with $z \sim \tau_d$. We consider the family of Gegenbauer polynomials on $L^2([-1,1], \tau_{d,1})$, denoted by $\{Q_\ell^{(d)} : \ell \in \mathbb{N}\}$, where $Q_\ell^{(d)}$ is the degree-$\ell$ polynomial satisfying

$$\int_{-1}^{1} Q_\ell^{(d)}(z)Q_k^{(d)}(z)\, \tau_{d,1}(\mathrm{d}z) = \int_{\mathbb{S}^{d-1}} Q_\ell^{(d)}(\langle z, e_1 \rangle)Q_k^{(d)}(\langle z, e_1 \rangle)\, \tau_d(\mathrm{d}z) = \delta_{\ell k}.$$

A relationship between the spherical harmonics and Gegenbauer polynomials is as follows:

$$Q_\ell^{(d)}(\langle z, z' \rangle) = \frac{1}{\sqrt{n_{d,\ell}}} \sum_{s \in [n_{d,\ell}]} Y_{\ell i}^{(d)}(z)Y_{\ell i}^{(d)}(z'), \text{ for all } z, z' \in \mathbb{S}^{d-1}. \tag{34}$$

Another important relationship is for any $w, v \in \mathbb{S}^{d-1}$:

$$\left\langle Q_\ell^{(d)}(\langle \cdot, w \rangle), Q_k^{(d)}(\langle \cdot, v \rangle) \right\rangle_{\tau_{d,1}} = \frac{\delta_{\ell k} \cdot Q_\ell^{(d)}(\langle w, v \rangle)}{Q_\ell^{(d)}(1)}, \tag{35}$$

where $Q_\ell^{(d)}(1) = \sqrt{n_{d,\ell}}$ as derived in Eq. (45) in the next section. We note that the normalization of Gegenbauer polynomials considered here is such that $\|Q_\ell^{(d)}\|_{L^2(\tau_{d,1})} = 1$ holds. Another popular choice is such that the value at 1 evaluates to 1 which we shall explicitly refer by the family of polynomials $\{P_\ell^{(d)}\}_{\ell \in \mathbb{N}}$. The normalizing factor is such that $P_\ell^{(d)}(\cdot) = Q_\ell^{(d)}(\cdot)/\sqrt{n_{d,\ell}}$.

The derivative of the $\ell^{\text{th}}$ Gegenbauer polynomial for $\ell \geq 1$ can be expressed as

$$\frac{\mathrm{d}}{\mathrm{d}z} Q_\ell^{(d)}(z) = Q_\ell^{(d)}(z)' = \frac{\ell(\ell + d - 2)\sqrt{n_{d,\ell}}}{(d-1)\sqrt{B(d+2,\ell-1)}} Q_{k-1}^{d+2}(z) = C(d,\ell) \, Q_{k-1}^{(d+2)}(z), \qquad (36)$$

where for a fixed constant $\ell$ and growing $d$, we have $C(d,\ell) = \Theta_d(\sqrt{d})$. Let $f \in L^2(\mathbb{S}^{d-1}, \tau_d)$ such that $f$ is invariant by the action of $\mathcal{O}_{\boldsymbol{w}^\perp} = \{\boldsymbol{W} \in \mathcal{O}_d : \boldsymbol{W}^\mathsf{T} \boldsymbol{w} = \boldsymbol{w}\}$ which is the set of orthogonal matrices which keeps the direction $\boldsymbol{w}$ fixed, i.e. $f$ only depends on the projection $\langle \boldsymbol{w}, \boldsymbol{z} \rangle$. Then $f$ admits the following decomposition

$$f(\boldsymbol{z}) = \sum_{\ell=0}^{\infty} \alpha_\ell Q_\ell^{(d)}(\langle \boldsymbol{w}, \boldsymbol{z} \rangle). \qquad (37)$$

**Hermite polynomials.** Consider the probabilist's Hermite polynomials $\{\mathrm{He}_k : k \in \mathbb{N}\}$, in the normalization that form an orthonormal basis of $L^2(\mathbb{R}, \gamma)$, where $\gamma(\mathrm{d}x) = \frac{e^{-x^2/2}}{\sqrt{2\pi}}$ is the standard Gaussian measure. $\mathrm{He}_k$ is a polynomial of degree $k$ and

$$\mathbb{E}_{G \sim \mathsf{N}(0,1)} [\mathrm{He}_j(G)\mathrm{He}_k(G)] = \delta_{jk}.$$

As a consequence, for any $g \in L^2(\mathbb{R}, \gamma)$, we have the following decomposition

$$g(x) = \sum_{k=0}^{\infty} \mu_k(g) \, \mathrm{He}(x), \quad \mu_k(g) = \mathbb{E}_{G \sim \mathsf{N}(0,1)}[g(G)\mathrm{He}_k(G)].$$

## B.2 Harmonic decomposition of Hermite into Gegenbauer polynomials

The Gaussian distribution $\boldsymbol{x} \sim \mathsf{N}(0, \mathbf{I}_d)$ admits the polar decomposition

$$\boldsymbol{x} = \|\boldsymbol{x}\|_2 \cdot \frac{\boldsymbol{x}}{\|\boldsymbol{x}\|_2}, \quad \text{where } \|\boldsymbol{x}\|_2 =: r \sim \chi_d \text{ and } \frac{\boldsymbol{x}}{\|\boldsymbol{x}\|_2} =: \boldsymbol{z} \sim \mathrm{Unif}(\mathbb{S}^{d-1}) \text{ are independent.}$$

Therefore, $x_1 = r \cdot u_1$, where $x_1 \sim \mathsf{N}(0,1)$ and $z_1 \sim \tau_{d,1}$. In what follows, we denote $x = x_1$ and $z = z_1$ for convenience. The Gegenbauer polynomials are an orthonormal basis for $\tau_{d,1}$ and Hermite polynomials are (unnormalized) orthogonal basis for $\mathsf{N}(0,1)$. Our goal is to explicitly express $\mathrm{He}_k(x) = \mathrm{He}_k(r \cdot z)$ in terms Gegenbauer polynomials $\{Q_\ell^{(d)}(z)\}$, formalized in the following proposition.

**Proposition 2** (Decomposing Hermite into Gegenbauer). *For any $k \in \mathbb{N}$, we have*

$$\mathrm{He}_k(r \cdot z) = \sum_{\ell=0}^{\infty} \beta_{k,\ell}(r) Q_\ell^{(d)}(z), \qquad (38)$$

*where $\beta_{k,\ell}(r) = 0$ if $(\ell > k)$ or $(\ell \not\equiv k \mod 2)$, and otherwise*

$$\beta_{k,\ell}(r) := \frac{\sqrt{k!}\sqrt{K(d,\ell)}}{(N!)\,2^N} \left( \sum_{i=0}^{N} \frac{\binom{N}{i}(-1)^{N-i}\, r^{\ell+2i}}{\prod_{j=0}^{i+\ell-1}(d+2j)} \right), \qquad (39)$$

*where $N = (k-\ell)/2$ and $K(d,\ell) \asymp d^\ell$ as $d \to \infty$ and $\ell$ is constant, i.e. $K(d,\ell) = \Theta_d(d^\ell)$.*

Essentially, we are decomposing the Hermite basis into the Gegenbauer polynomials, which is the correct basis for the "directional" component $z$, and explicitly computing the coefficients that depend on the radial component $r$. Such relationship is derived by technical algebraic manipulations, and in similar spirit to [60], relating Hermite and Gegenbauer polynomials. However, the precise expression is sensitive to the normalization used for Gegenbauer polynomials. Thus, we provide an explicit calculation of this decomposition in Section B.3.

For our upper and lower bound analyses, in order to measure correlation of $\mathrm{He}_k(r \cdot z)$ with $Q_\ell^{(d)}(z)$, using both type of queries (with or without norm), the asymptotic bounds on the following moments of these coefficients will play an important role.

**Lemma 3.** *For any fixed $\ell \leq k \in \mathbb{N}$ with same parity, i.e. $k \equiv \ell \mod 2$, we have*

$$\mathbb{E}_{r \sim \chi_d}[\beta_{k,\ell}(r)^2] \asymp d^{-\frac{(k-\ell)}{2}} \quad \text{and} \quad \mathbb{E}_{r \sim \chi_d}[\beta_{k,\ell}(r)]^2 \asymp d^{-(k-\ell)}.$$

In order to show this lemma, we will use the following well-known facts.

**Fact 1** (Moments of $\chi_d$ distribution). *For any $p \in \mathbb{N}$, the even and odd moments of $\chi_d$ distribution are given by,*

$$\mathbb{E}_{r \sim \chi_d}[r^{2p}] = \prod_{j=0}^{p-1}(d+2j),$$

$$\mathbb{E}_{r \sim \chi_d}[r^{2p+1}] = \mathbb{E}[r]\prod_{j=0}^{p-1}(d+2j+1) = \frac{\sqrt{2}\,\Gamma(\frac{d+1}{2})}{\Gamma(d/2)}\prod_{j=0}^{p-1}(d+2j+1). \tag{40}$$

*Therefore, for any fixed $m \in \mathbb{N}$, the asymptotic behavior of the $m^{\text{th}}$ moment as $d \to \infty$ is given by*

$$\mathbb{E}_{r \sim \chi_d}[r^m] \asymp \left(\sqrt{d} \cdot \mathbf{1}\{m \equiv 1 \mod 2\}\right) \prod_{j=0}^{\lfloor m/2 \rfloor - 1}(d+2j).$$

**Fact 2.** *For any univariate polynomial $g : \mathbb{R} \to \mathbb{R}$, the $n^{\text{th}}$ forward finite difference of $g$ at any value $u$ is given by*

$$\Delta^n g(u) := \sum_{i=0}^{n}\binom{n}{i}(-1)^{n-i}g(u+i).$$

*For any polynomial $g$, the $n^{\text{th}}$ forward finite difference $\Delta^n g(u) \equiv 0$ if $\deg(g) < n$ and $\Delta^n g(u)$ is a non-zero constant if $\deg(g) = n$. Moreover, for any polynomial given by shifted binomial coefficient of degree $n$, with shift $u_0 \in \mathbb{R}$*

$$g_n(u) = \frac{(u+u_0)(u+u_0-1)\cdots(u+u_0-n+1)}{n!} =: \binom{u+u_0}{n},$$

*the constant value of $n^{\text{th}}$ forward finite difference is unity, i.e. $\Delta^n g(u) = 1$.*

*Proof of Lemma 3.* We will use the above facts directly along with $K(d,\ell) = d^\ell$ and $N, k, \ell$ are constants (not dependent on $d$) from Proposition 2 throughout the proof. We start by the first part of the claim

$$\mathbb{E}_{r \sim \chi_d}[\beta_{k,\ell}(r)^2] = \frac{(k!)}{(N!)^2\,2^{2N}}K(d,\ell)\left(\sum_{n=0}^{N}\sum_{m=0}^{N}\frac{\binom{N}{n}\binom{N}{m}(-1)^{2N-n-m}\,\mathbb{E}[r^{2\ell+2n+2m}]}{\prod_{j=0}^{n+\ell-1}(d+2j)\prod_{j=0}^{m+\ell-1}(d+2j)}\right)$$

$$\asymp d^\ell\left(\sum_{n=0}^{N}\sum_{m=0}^{N}\frac{\binom{N}{n}\binom{N}{m}(-1)^{2N-n-m}\prod_{j=0}^{\ell+m+n-1}(d+2j)}{\prod_{j=0}^{n+\ell-1}(d+2j)\prod_{j=0}^{m+\ell-1}(d+2j)}\right) \qquad \text{(using Fact 1)}$$

$$\asymp \frac{d^\ell}{\prod_{j=0}^{N+\ell-1}(d+2j)}\left(\sum_{n=0}^{N}\sum_{m=0}^{N}\binom{N}{n}\binom{N}{m}(-1)^{2N-n-m}\prod_{j=m+\ell}^{\ell+m+n-1}(d+2j)\prod_{j=n+\ell}^{N+\ell-1}(d+2j)\right)$$

$$\asymp d^{-N}\left(\sum_{n=0}^{N}\sum_{m=0}^{N}\binom{N}{n}\binom{N}{m}(-1)^{2N-n-m}\prod_{j=m+\ell}^{\ell+m+n-1}(d+2j)\prod_{j=n+\ell}^{N+\ell-1}(d+2j)\right)$$

$$= d^{-\frac{(k-\ell)}{2}}\left(\sum_{n=0}^{N}\binom{N}{n}(-1)^{N-n}\prod_{j=n+\ell}^{N+\ell-1}(d+2j)\left(\sum_{m=0}^{N}\binom{N}{m}(-1)^{N-m}\prod_{j=m+\ell}^{\ell+m+n-1}(d+2j)\right)\right)$$

$$= d^{-\frac{(k-\ell)}{2}}\left(\sum_{n=0}^{N}\binom{N}{n}(-1)^{N-n}\prod_{j=n+\ell}^{N+\ell-1}(d+2j)\left(\sum_{m=0}^{N}\binom{N}{m}(-1)^{N-m}\binom{\frac{d}{2}+m+n-1}{n}2^n n!\right)\right).$$

We are now going to show that term inside the parenthesis is some constant independent of $d$. A priori it may seem that it depends on $d$, however, we have a sum with alternative positive and negative signs and we will show that all the terms that depend on $d$ mutually cancel out.

To show this we first observe that $g(m) = \binom{\frac{d}{2}+m+n-1}{n}$ is a polynomial of degree $n$ given by binomial coefficient. And, therefore, the $N^{\text{th}}$ forward finite difference $\Delta^N g(m) \equiv 0$ for any $n < N$, or simply 1 for $n = N$ using Fact 2. Formally,

$$\sum_{m=0}^{N} \binom{N}{m}(-1)^{N-m}\binom{\frac{d}{2}+m+n-1}{n}2^n n! = 2^n \, n! \cdot \delta_{Nn},$$

where the scaling factor of $2^n n!$ of the polynomial $g(m)$ can be taken out as the forward finite difference operator $\Delta^N$ is linear. We conclude that

$$\mathbb{E}[\beta_{k,\ell}(r)^2]$$

$$\asymp d^{-\frac{(k-\ell)}{2}}\left(\sum_{n=0}^{N}\binom{N}{n}(-1)^{N-n}\prod_{j=n+\ell}^{N+\ell-1}(d+2j)\left(\sum_{m=0}^{N}\binom{N}{m}(-1)^{N-m}\binom{\frac{d}{2}+m+n-1}{n}2^n n!\right)\right)$$

$$= d^{-\frac{(k-\ell)}{2}}\left(\sum_{n=0}^{N}\binom{N}{n}(-1)^{N-n}\prod_{j=n+\ell}^{N+\ell-1}(d+2j)\cdot\delta_{Nn}\cdot 2^n n!\right) = 2^N N! \, d^{-\frac{(k-\ell)}{2}}$$

$$\asymp d^{-\frac{(k-\ell)}{2}}.$$

We now show the second part that

$$\mathbb{E}[\beta_{k,\ell}(r)]^2 \asymp d^{-(k-\ell)}.$$

Let us consider any $k, \ell$ such that they have the same parity $k \equiv \ell \mod 2$. We have

$$\mathbb{E}[\beta_{k,\ell}(r)] = \frac{\sqrt{(k!)\,K(d,\ell)}}{(N!)\,2^N}\left(\sum_{i=0}^{N}\frac{\binom{N}{i}(-1)^{N-i}\,\mathbb{E}[r^{\ell+2i}]}{\prod_{j=0}^{\ell+i-1}(d+2j)}\right)$$

$$\asymp d^{\ell/2}\left(\sqrt{d}\cdot\mathbf{1}\{\ell\equiv 1\bmod 2\}\right)\left(\sum_{i=0}^{N}\binom{N}{i}(-1)^{N-i}\frac{\prod_{j=0}^{\lfloor\ell/2\rfloor+i-1}(d+2j)}{\prod_{j=0}^{\ell+i-1}(d+2j)}\right) \quad \text{(using Fact 1)}$$

$$\asymp d^{\lceil\ell/2\rceil}\left(\sum_{i=0}^{N}\binom{N}{i}(-1)^{N-i}\frac{1}{\prod_{j=\ell/2+i}^{\ell+i-1}(d+2j)}\right)$$

$$\asymp d^{\lceil\ell/2\rceil}\sum_{i=0}^{N}\binom{N}{i}(-1)^{N-i}\frac{\prod_{j=i}^{N+i-1}(d+2j)}{\prod_{j=i}^{N+i-1}(d+2j)\prod_{j=\lfloor\ell/2\rfloor+i}^{\ell+i-1}(d+2j)}$$

$$\asymp\frac{d^{\lceil\ell/2\rceil}}{d^{\lceil k/2\rceil}}\sum_{i=0}^{N}(1+o_d(1))\binom{N}{i}(-1)^{N-i}\prod_{j=i}^{N+i-1}(d+2j).$$

In the last line, we used the fact that $k \equiv \ell \mod 2$ and $N = (k-\ell)/2$ and thus for every $0 \le i \le N$,

$$\prod_{j=i}^{N+i-1}(d+2j)\prod_{j=\ell/2+i}^{i+\ell-1}(d+2j) = d^{N+\ell/2}\prod_{j=i}^{N+i-1}\left(1+\frac{2j}{d}\right)\prod_{j=\lfloor\ell/2\rfloor+i}^{\ell+i-1}\left(1+\frac{2j}{d}\right)$$

$$=(1+o_d(1))\,d^{N+\ell-\lfloor\ell/2\rfloor},$$

and

$$N+\ell-\lfloor\ell/2\rfloor = \frac{(k-\ell)}{2}+\ell-\lfloor\ell/2\rfloor = \frac{k+\ell}{2}-\lfloor\ell/2\rfloor = \lceil k/2\rceil.$$

Continuing to simplify the original expression

$$\mathbb{E}[\beta_{k,\ell}(r)] \asymp \frac{d^{\lceil \ell/2 \rceil}}{d^{\lceil k/2 \rceil}} \sum_{i=0}^{N} (1 + o_d(1)) \binom{N}{i} (-1)^{N-i} \prod_{j=i}^{N+i-1} (d + 2j)$$

$$= \frac{1}{d^{\frac{k-\ell}{2}}} \sum_{i=0}^{N} \binom{N}{i} (-1)^{N-i} \binom{\frac{d}{2} + i + N - 1}{N} 2^N N!$$

$$= d^{-\frac{(k-\ell)}{2}} 2^N N! \asymp d^{-\frac{(k-\ell)}{2}} .$$

Here the last line followed from the fact that the polynomial $g(i) = \binom{\frac{d}{2} + i + N - 1}{N}$ is of degree $N$ given by a shifted binomial coefficient, and thus, the $N^{\text{th}}$ forward finite difference of $g$ is constant, which in this case is just 1 by Fact 2. We finally conclude the proof by noting that $\mathbb{E}[\beta_{k,\ell}(r)]^2 \asymp d^{-(k-\ell)}$. □

### B.3  Proof of Proposition 2

We now return to the deferred proof of Proposition 2. First, we use the explicit expression of $\mathrm{He}_k$ so that $\|\mathrm{He}_k\|_{L^2} = 1$.

$$\mathrm{He}_k(r \cdot z) = \mathrm{He}_k(x) = \sqrt{k!} \sum_{m=0}^{\lfloor k/2 \rfloor} \frac{(-1)^m}{m!(k-2m)!} \frac{x^{k-2m}}{2^m} = \sqrt{k!} \sum_{m=0}^{\lfloor k/2 \rfloor} \frac{(-1)^m r^{k-2m}}{m!(k-2m)!} \frac{z^{k-2m}}{2^m} . \quad (41)$$

Our goal is to express $z^{k-2m}$ in terms of $Q_\ell^{(d)}(z)$ to get the final decomposition. To this end, we use the explicit expressions computed with (a different normalization) of Gegenbauer polynomials. In particular, using [39, Eq. 18.18.17]

$$z^n = \frac{n!}{2^n} \sum_{l=0}^{\lfloor n/2 \rfloor} \frac{\alpha + n - 2l}{\alpha} \frac{1}{l!(\alpha + 1)_{n-l}} C_{n-2l}^{(\alpha)}(z) , \quad (42)$$

where $(b)_a$ is the rising factorial and $C_\ell^{(\alpha)}(z)$ is an unnormalized Gegenbauer polynomial $Q_\ell^{(d)}(z)$ with $\alpha = (d-2)/2$ satisfying

$$\int_{-1}^{+1} C_\ell^{(\alpha)}(z) C_k^{(\alpha)}(z)(1-z^2)^{\alpha - \frac{1}{2}} \mathrm{d}z = \delta_{\ell k} \frac{\pi 2^{1-2\alpha} \Gamma(\ell + 2\alpha)}{\ell!(\ell + \alpha)[\Gamma(\alpha)]^2} \quad (43)$$

We can express $C_\ell^{(\alpha)}(z)/\sqrt{K(d,\ell)} = Q_\ell^{(d)}(z)$ where $K(d,\ell)$ can be computed using

$$1 = \frac{1}{K(d,\ell)} \int_{-1}^{+1} C_\ell^{(\alpha)}(z)^2 \, \tau_{d,1}(\mathrm{d}z) = \frac{1}{K(d,\ell) \mathsf{B}\left(\alpha + \frac{1}{2}, \frac{1}{2}\right)} \int_{-1}^{+1} C_\ell^{(\alpha)}(z)^2 \, (1-z^2)^{\alpha - \frac{1}{2}} \tau_{d,1}(\mathrm{d}z)$$

where $\mathsf{B}(\cdot, \cdot)$ is the standard Beta function. We can use Eq. (43) to compute

$$K(d,\ell) = \frac{\pi 2^{1-2\alpha} \Gamma(\ell + 2\alpha)}{\mathsf{B}\left(\alpha + \frac{1}{2}, \frac{1}{2}\right) \ell!(\ell + \alpha)[\Gamma(\alpha)]^2} \quad (44)$$

It is straight-forward to simplify

$$K(d,\ell) = \frac{\sqrt{\pi} \, 2^{1-2\alpha} \Gamma(\ell + 2\alpha) \Gamma(\alpha + 1)}{\Gamma(\alpha + \frac{1}{2}) \ell!(\ell + \alpha)[\Gamma(\alpha)]^2} = \frac{\sqrt{\pi} \, 2^{1-2\alpha} \Gamma(\ell + 2\alpha) \alpha}{\Gamma(\alpha + \frac{1}{2}) \ell!(\ell + \alpha) \Gamma(\alpha)}$$

$$= \frac{\alpha \Gamma(\ell + 2\alpha)}{\ell! \, (\alpha + \ell) \Gamma(2\alpha)} \qquad (\because \Gamma(\alpha)\gamma(\alpha + \tfrac{1}{2}) = \sqrt{\pi} 2^{1-2\alpha} \Gamma(2\alpha))$$

$$= \frac{(d-2)\Gamma(d-2+\ell)}{\ell! \, (d-2+2\ell)\Gamma(d-2)} = \frac{(d-2)}{(d+2\ell-2)} \binom{d+\ell-3}{\ell} = \Theta_d(d^\ell) .$$

Also substituting $C_\ell^{(\alpha)}(1)$ from [85], we obtain

$$Q_\ell^{(d)}(1) = \frac{C_\ell^{(\alpha)}(1)}{\sqrt{K(d,\ell)}} = \sqrt{\frac{\ell! \, (\alpha + \ell) \Gamma(2\alpha)}{\alpha \Gamma(2\alpha + \ell)} \cdot \frac{\Gamma(2\alpha + \ell)}{\Gamma(2\alpha) \ell!}} = \sqrt{\frac{d + 2\ell - 2}{d - 2} \binom{d + \ell - 3}{\ell}} = \sqrt{n_{d,\ell}} . \quad (45)$$

We are now ready to combine the equations derived and compute the desired decomposition Eq. (38). For any $M \in \mathbb{N}$, we let $\boxed{M} = \{m \in \mathbb{N} : m \leq M \text{ and } m \equiv M \mod 2\}$. Recall from (41)

$$\mathrm{He}_k(r \cdot u) = \sqrt{k!} \sum_{m=0}^{\lfloor k/2 \rfloor} \frac{(-1)^m r^{k-2m}}{m!(k-2m)!} \frac{u^{k-2m}}{2^m} = \sqrt{k!} \sum_{m \in \boxed{k}} \frac{(-1)^{(k-m)/2} r^m}{m!((k-m)/2)!} \frac{u^m}{2^{(k-m)/2}}$$

$$\text{(Change of variables)}$$

$$= \sqrt{k!} \sum_{m \in \boxed{k}} \frac{(-1)^{(k-m)/2} r^m}{m!((k-m)/2)!} \frac{m!}{2^m 2^{(k-m)/2}} \sum_{l=0}^{\lfloor m/2 \rfloor} \frac{\alpha+m-2l}{\alpha} \frac{1}{l!(\alpha+1)_{m-l}} C_{m-2l}^{(\alpha)}(u)$$

$$\text{(Using (42))}$$

$$= \sqrt{k!} \sum_{m \in \boxed{k}} \frac{(-1)^{(k-m)/2} r^m}{((k-m)/2)!} \frac{1}{2^{(k+m)/2}} \sum_{\ell \in \boxed{m}} \frac{\alpha+\ell}{\alpha} \frac{1}{((m-\ell)/2)!(\alpha+1)_{(m+\ell)/2}} \sqrt{K(d,\ell)} \, Q_\ell^{(d)}(u)$$

$$\text{(Changing } \ell = m - 2l \text{ and } C_\ell^{(\alpha)} = \sqrt{K(d,\ell)} \, Q_\ell^{(d)})$$

$$= \sqrt{(k!) \, K(d,\ell)} \sum_{\ell \in \boxed{k}} Q_\ell^{(d)}(u) \frac{\alpha+\ell}{\alpha} \left( \sum_{\substack{m=\ell \\ m \equiv k \bmod 2}}^{k} \frac{(-1)^{(k-m)/2} r^m}{((k-m)/2)!} \frac{1}{2^{(k+m)/2}((m-\ell)/2)!(\alpha+1)_{(m+\ell)/2}} \right)$$

$$:= \sum_{\ell=0}^{\infty} Q_\ell^{(d)}(u) \beta_{k,\ell}(r), \text{ where if } \ell \notin \boxed{k} \text{ then } \beta_{k,\ell}(r) = 0.$$

Otherwise, letting $N = (k-\ell)/2$

$$\beta_{k,\ell}(r) = \sqrt{k! \, K(d,\ell)} \frac{\alpha+\ell}{\alpha} \left( \sum_{\substack{m=\ell \\ m \equiv k \bmod 2}}^{k} \frac{(-1)^{(k-m)/2} r^m}{((k-m)/2)! \, 2^{(k+m)/2}((m-\ell)/2)!(\alpha+1)_{(m+\ell)/2}} \right)$$

$$= \sqrt{k! \, K(d,\ell)} \frac{\alpha+\ell}{\alpha} \left( \sum_{i=0}^{N} \frac{(-1)^{N-i} r^{\ell+2i}}{(N-i)! \, 2^{(k+\ell)/2+i}(i)!(\alpha+1)_{\ell+i}} \right) \quad \text{(changing } m = \ell + 2i)$$

$$= \frac{\sqrt{k! \, K(d,\ell)}}{(N!) \, 2^N} \frac{\alpha+\ell}{\alpha} \left( \sum_{i=0}^{N} \binom{N}{i} \frac{(-1)^{N-i} r^{\ell+2i}}{2^{\ell+i}(\alpha+1)_{\ell+i}} \right).$$

Recall that $\alpha = (d-2)/2$ here, and thus

$$2^{\ell+i}(\alpha+1)_{\ell+i} = 2^{\ell+i} \prod_{j=0}^{\ell+i-1} (\alpha+1+j) = 2^{\ell+i} \prod_{j=0}^{\ell+i-1} \left( \frac{d-2}{2} + 1 + j \right) = \prod_{j=0}^{\ell+i-1} (d+2j).$$

Thus, for $\ell \equiv k \mod 2$

$$\beta_{k,\ell}(r) = \frac{\sqrt{k! \, K(d,\ell)}}{(N!) \, 2^N} \frac{d+2\ell-2}{d-2} \left( \sum_{i=0}^{N} \binom{N}{i} \frac{(-1)^{N-i} r^{\ell+2i}}{\prod_{j=0}^{\ell+i-1}(d+2j)} \right).$$

The lemma follows by redefining $K(d,\ell)$ with $K(d,\ell)(d+2\ell-2)^2/(d-2)^2 = \Theta_d(d^\ell)$.

# C  Statistical Query (SQ) and Low-Degree Polynomial (LDP) lower bounds

In this appendix, we briefly review the statistical query (SQ) and low-degree polynomial (LDP) frameworks and present the proof of Theorem 1. In particular, we provide an interpretation of our lower bounds in terms of subproblems with queries restricted to the harmonic subspace $V_{d,\ell}$.

## C.1  Statistical Query lower bounds

The Statistical Query (SQ) framework, introduced by Kearns [53], models algorithms that interact with data only through expectations of query functions, rather than direct access to samples. The complexity of these algorithms is measured up to some worst-case tolerance on these expectations. While based on worst-case error rather than sampling error encountered in practice, the SQ framework has proven remarkably effective in analyzing the computational complexity of statistical problems, often yielding accurate predictions for algorithmic feasibility. We refer to [22, 71] for additional background.

Below, it will be useful to present a variant of SQ algorithms, called *query-restricted statistical query algorithms*, introduced in [50]. In this model, queries are restricted to a set $\mathcal{Q} \subseteq \mathbb{R}^{\mathcal{Y} \times \mathbb{R}^d}$ of measurable functions $\mathcal{Y} \times \mathbb{R}^d \to \mathbb{R}$. We denote $\mathcal{Q}\text{-}\mathsf{SQ}(q, \tau)$ this class of algorithms, with number of queries $q$ and tolerance $\tau > 0$. We will mainly consider the standard case of *unrestricted queries*, denoted $\mathsf{SQ}(q, \tau)$, where $\mathcal{Q}$ contains all measurable functions. In Appendix A.5 we discuss the case of *correlation statistical queries* (CSQ).

Our lower bounds hold for the detection problem (hypothesis testing) of distinguishing between

$$\{\mathbb{P}_{\nu_d, \boldsymbol{w}} : \boldsymbol{w} \in \mathbb{S}^{d-1}\} \qquad \text{v.s.} \qquad \{\mathbb{P}_{\nu_d, 0}\}. \tag{46}$$

Below, we describe $\mathcal{Q}\text{-}\mathsf{SQ}$ algorithms in this context.

**$\mathcal{Q}$-restricted SQ algorithm.**  For a number of queries $q$ and tolerance $\tau > 0$, a $\mathcal{Q}$-restricted SQ algorithm $\mathcal{A} \in \mathcal{Q}\text{-}\mathsf{SQ}(q, \tau)$ for detecting SIMs takes an input distribution $\mathbb{P}$ from (46) and operates in $q$ rounds where at each round $t \in \{1, \ldots, q\}$, it issues a query $\phi_t \in \mathcal{Q}$, and receives a response $v_t$ such that

$$\left| v_t - \mathbb{E}_{\mathbb{P}}[\phi_t(y, \boldsymbol{x})] \right| \leq \tau \sqrt{\mathrm{Var}_{\mathbb{P}_0}(\phi_t)}, \tag{47}$$

where we set $\mathbb{P}_0$ to be the null distribution (that is, $\mathbb{P}_{\nu_d, 0}$ here). The query $\phi_t$ can depend on the past responses $v_1, \ldots, v_{t-1}$. After issuing $q$ queries, the learner outputs $\mathcal{A}(\mathbb{P}) \in \{0, 1\}$. We say that $\mathcal{A}$ succeeds in distinguishing $\mathbb{P}_{\nu_d, \boldsymbol{w}}$ and $\mathbb{P}_{\nu_d, 0}$, if $\mathcal{A}(\mathbb{P}_{\nu_d, \boldsymbol{w}}) = 1$ for all $\boldsymbol{w} \in \mathbb{S}^{d-1}$, and $\mathcal{A}(\mathbb{P}_{\nu_d, 0}) = 0$.

**Remark C.1.**  The variance scaling on the right-hand side in (47) is non-standard in the SQ literature. It is introduced here as a convenient way to normalize queries, which is necessary for $\tau$ to be meaningful. We note that other normalizations are possible and refer to [50, Remark 3.1] for a discussion.

**General lower-bound.**  The following proposition is a simple, standard lower bound on the query complexity based on the second moment method (e.g., see [50]):

**Proposition 3** (General $\mathcal{Q}$-restricted SQ lower bound). *Fix $\nu_d \in \mathfrak{L}_d$. If an algorithm $\mathcal{A} \in \mathcal{Q}\text{-}\mathsf{SQ}(q, \tau)$ succeeds at distinguishing $\mathbb{P}_{\nu_d, \boldsymbol{w}}$ from $\mathbb{P}_{\nu_d, 0}$, then we must have*

$$q/\tau^2 \geq \left[ \sup_{\phi \in \mathcal{Q}} \frac{\mathrm{Var}_{\boldsymbol{w} \sim \tau_d}\{\mathbb{E}_{\mathbb{P}_{\nu_d, \boldsymbol{w}}} \phi\}}{\mathrm{Var}_{\mathbb{P}_{\nu_d, 0}}\{\phi\}} \right]^{-1}. \tag{48}$$

*Proof.*  Consider $\mathcal{A} \in \mathcal{Q}\text{-}\mathsf{SQ}(q, \tau)$ and denote $\phi_1, \ldots, \phi_q \in \mathcal{Q}$ the sequence of queries issued by $\mathcal{A}$ when it receives responses $v_t = \mathbb{E}_{\mathbb{P}_{\nu_d, 0}}[\phi_t], t \in [q]$. Here the responses are fixed and deterministic, and the queries $\{\phi_t\}_{t \in [q]}$ do not depend on the source distribution $\mathbb{P}_{\nu_d, \boldsymbol{w}}$, and in particular $\boldsymbol{w}$. By union bound and Markov's inequality,

$$\mathbb{P}_{\boldsymbol{w} \sim \tau_d}\left( \exists t \in [q], \ |\mathbb{E}_{\mathbb{P}_{\nu_d, \boldsymbol{w}}}[\phi_t] - v_t| > \tau \sqrt{\mathrm{Var}_{\mathbb{P}_{\nu_d, 0}}[\phi_t]} \right) \leq \frac{q}{\tau^2} \cdot \sup_{t \in [q]} \frac{\mathrm{Var}_{\boldsymbol{w} \sim \tau_d}\{\mathbb{E}_{\mathbb{P}_{\nu_d, \boldsymbol{w}}} \phi_t\}}{\mathrm{Var}_{\mathbb{P}_{\nu_d, 0}}\{\phi_t\}}$$

$$\leq \frac{q}{\tau^2} \cdot \sup_{\phi \in \mathcal{Q}} \frac{\mathrm{Var}_{\boldsymbol{w} \sim \tau_d}\{\mathbb{E}_{\mathbb{P}_{\nu_d, \boldsymbol{w}}} \phi\}}{\mathrm{Var}_{\mathbb{P}_{\nu_d, 0}}\{\phi\}}.$$

This implies that the $v_t = \mathbb{E}_{\mathbb{P}_{\nu_d,0}}[\phi_t]$ responses are compatible for all $q$ queries with positive probability over $\boldsymbol{w} \sim \tau_d$ whenever inequality (48) is not satisfied, and $\mathcal{A}$ fails the detection task in that case. This concludes the proof. $\qquad\square$

The query complexity bound in Theorem 1.(i) follows from Proposition 3 with unrestricted queries $\mathcal{Q}_{\mathsf{SQ}}$ and the following identity:

**Lemma 4.** *For $\nu_d \in \mathfrak{L}_d$ and $\mathcal{Q}_{\mathsf{SQ}}$ the class of unrestricted queries (all measurable functions), we have the identity*

$$\sup_{\phi \in \mathcal{Q}_{\mathsf{SQ}}} \frac{\mathrm{Var}_{\boldsymbol{w} \sim \tau_d}\{\mathbb{E}_{\mathbb{P}_{\nu_d,\boldsymbol{w}}} \phi\}}{\mathrm{Var}_{\mathbb{P}_{\nu_d,0}}\{\phi\}} = \sup_{\ell \geq 1} \frac{\|\xi_{d,\ell}\|_{L^2}^2}{n_{d,\ell}} =: [\mathsf{Q}_\star(\nu_d)]^{-1}. \tag{49}$$

We defer the proof of this lemma below to Section C.1.1. The identity (49) shows that the lower bound effectively decouples across the different harmonic subspaces. Below, we provide an interpretation of this result: if we restrict the queries $\mathcal{Q}$ to be in $V_{d,\ell}$, then the $\mathsf{SQ}$ lower bound becomes $n_{d,\ell}/\|\xi_{d,\ell}\|_{L^2}^2$. Specifically, for each $\ell \geq 1$, define $\mathcal{Q}_{\mathsf{SQ},\ell}$ to be the set of all queries $\phi(y, \boldsymbol{x})$ than can be written as

$$\phi(y, \boldsymbol{x}) = \sum_{s \in [n_{d,\ell}]} g_\ell(y, r) Y_{\ell s}(\boldsymbol{z}), \tag{50}$$

where $\{Y_{\ell s}\}_{s \in [n_{d,\ell}]}$ is a basis of $V_{d,\ell}$. Then by Proposition 3 and the proof of Lemma 4, we have:

**Corollary 3.** *Fix $\nu_d \in \mathfrak{L}_d$ and $\ell \geq 1$. If an algorithm $\mathcal{A} \in \mathcal{Q}_{\mathsf{SQ},\ell}\text{-}\mathsf{SQ}(q, \tau)$ succeeds at distinguishing $\mathbb{P}_{\nu_d,\boldsymbol{w}}$ from $\mathbb{P}_{\nu_d,0}$, then we must have*

$$q/\tau^2 \geq \left[ \sup_{\phi \in \mathcal{Q}_{\mathsf{SQ},\ell}} \frac{\mathrm{Var}_{\boldsymbol{w} \sim \tau_d}\{\mathbb{E}_{\mathbb{P}_{\nu_d,\boldsymbol{w}}} \phi\}}{\mathrm{Var}_{\mathbb{P}_{\nu_d,0}}\{\phi\}} \right]^{-1} = \frac{n_{d,\ell}}{\|\xi_{d,\ell}\|_{L^2}^2}. \tag{51}$$

For each $\ell \geq 1$, we show algorithms with queries restricted to $V_{d,\ell}$ as in (50) that matches this lower bound. Thus, effectively, the problem decouples into subproblems, one for each $V_{d,\ell}$: on each harmonic subspace, we have a matching upper and lower bound on the query complexity, and the optimal algorithm is obtained by choosing the optimal degree $\ell$ that attains the maximum in (49).

### C.1.1 Proof of Lemma 4

For clarity, we drop the subscript $\nu_d$ below and denote $\mathbb{P}_{\boldsymbol{w}} := \mathbb{P}_{\nu_d,\boldsymbol{w}}$ and $\mathbb{P}_0 := \mathbb{P}_{\nu_d,0}$. Note that a property of the null distribution is that

$$\mathbb{E}_{\boldsymbol{w}}\left[\mathbb{E}_{\mathbb{P}_{\boldsymbol{w}}}[\phi]\right] = \mathbb{E}_{\mathbb{P}_0}[\phi],$$

that is, $\mathbb{P}_0$ is the marginal distribution of $(y, \boldsymbol{x})$ under the uniform prior $\boldsymbol{w} \sim \tau_d$. Thus,

$$\mathrm{Var}_{\boldsymbol{w}}\{\mathbb{E}_{\mathbb{P}_{\boldsymbol{w}}} \phi\} = \mathbb{E}_{\boldsymbol{w}}\left[|\Delta_\phi(\boldsymbol{w})|^2\right], \quad \text{where} \quad \Delta_\phi(\boldsymbol{w}) = \mathbb{E}_{\mathbb{P}_{\boldsymbol{w}}}[\phi] - \mathbb{E}_{\mathbb{P}_0}[\phi].$$

Let's introduce the Radon-Nikodym derivative and write

$$\Delta_\phi(\boldsymbol{w}) = \mathbb{E}_{\mathbb{P}_0}\left[\left(\frac{\mathrm{d}\mathbb{P}_{\boldsymbol{w}}}{\mathrm{d}\mathbb{P}_0}(y_0, \boldsymbol{z}, r) - 1\right) \phi(y_0, \boldsymbol{z}, r)\right].$$

Recall that the likelihood ratio decomposes into Gegenbauer polynomials as (equality in $L^2(\mathbb{P}_0)$)

$$\frac{\mathrm{d}\mathbb{P}_{\boldsymbol{w}}}{\mathrm{d}\mathbb{P}_0}(y_0, \boldsymbol{z}, r) - 1 = \sum_{\ell=1}^\infty \xi_{d,\ell}(y_0, r) Q_\ell(\langle \boldsymbol{w}, \boldsymbol{z} \rangle), \qquad \xi_{d,\ell}(y_0, r) = \mathbb{E}_{\nu_d}[Q_\ell(Z)|Y = y_0, R = r].$$

Similarly, we can expand $\phi \in L^2(\mathbb{P}_0)$ as

$$\phi(y_0, \boldsymbol{z}, r) = \sum_{\ell=0}^\infty \sum_{s \in [n_{d,\ell}]} \alpha_{\ell s}(y_0, r) Y_{\ell s}(\boldsymbol{z}),$$

where $\{Y_{\ell s}\}_{\ell \geq 0, s \in [n_{d,\ell}]}$ is an orthonormal basis of spherical harmonics in $L^2(\mathbb{S}^{d-1})$. Using the identity $Q_\ell(\langle \boldsymbol{w}, \boldsymbol{z} \rangle) = n_{d,\ell}^{-1/2} \sum_{s \in [n_{d,\ell}]} Y_{\ell s}(\boldsymbol{w}) Y_{\ell s}(\boldsymbol{z})$, we obtain the decomposition

$$\Delta_\phi(\boldsymbol{w}) = \sum_{\ell=1}^{\infty} \sum_{s \in [n_{d,\ell}]} Y_{\ell s}(\boldsymbol{w}) \frac{\mathbb{E}_{\mathbb{P}_0}[\xi_{d,\ell}(y_0, r) \alpha_{\ell s}(y_0, r)]}{\sqrt{n_{d,\ell}}},$$

and thus,

$$\mathbb{E}_{\boldsymbol{w}}[|\Delta_\phi(\boldsymbol{w})|^2] = \sum_{\ell=1}^{\infty} \sum_{s \in [n_{d,\ell}]} \frac{\mathbb{E}_{\mathbb{P}_0}[\xi_{d,\ell}(y_0, r) \alpha_{\ell s}(y_0, r)]^2}{n_{d,\ell}}.$$

Denote $\mathsf{P}_\ell \phi = \sum_{s \in [n_{d,\ell}]} \alpha_{\ell s} Y_{\ell s}$ the projection on the degree-$\ell$ harmonics. We can decompose the supremum over $\phi \in \mathcal{Q}_{\mathsf{SQ}}$ as

$$\sup_{\phi \in \mathcal{Q}_{\mathsf{SQ}}} \frac{\mathbb{E}_{\boldsymbol{w}}[|\Delta_\phi(\boldsymbol{w})|^2]}{\mathrm{Var}_{\mathbb{P}_0}(\phi)}$$

$$= \sup_{\phi \in \mathcal{Q}_{\mathsf{SQ}}} \frac{1}{\sum_{\ell \geq 1} \|\mathsf{P}_\ell \phi\|_{L^2}^2} \sum_{\ell \geq 1} \frac{\|\mathsf{P}_\ell \phi\|_{L^2}^2}{n_{d,\ell}} \left[ \sum_{s \in [n_{d,\ell}]} \frac{\mathbb{E}_{\mathbb{P}_0}[\xi_{d,\ell}(y_0, r) \alpha_{\ell s}(y_0, r)]^2}{\|\mathsf{P}_\ell \phi\|_{L^2}^2} \right]$$

$$= \sup_{\phi \in \mathcal{Q}_{\mathsf{SQ}}} \frac{1}{\sum_{\ell \geq 1} \|\mathsf{P}_\ell \phi\|_{L^2}^2} \sum_{\ell \geq 1} \frac{\|\mathsf{P}_\ell \phi\|_{L^2}^2}{n_{d,\ell}} \left[ \sup_{\psi \in L^2(\nu_{d,Y,R})} \frac{\langle \xi_{d,\ell}, \psi \rangle_{L^2}^2}{\|\psi\|_{L^2}^2} \right]$$

$$= \sup_{\phi \in \mathcal{Q}_{\mathsf{SQ}}} \frac{\sum_{\ell \geq 1} \|\mathsf{P}_\ell \phi\|_{L^2}^2 \frac{\|\xi_{d,\ell}\|_{L^2}^2}{n_{d,\ell}}}{\sum_{\ell \geq 1} \|\mathsf{P}_\ell \phi\|_{L^2}^2} = \sup_{\ell \geq 1} \frac{\|\xi_{d,\ell}\|_{L^2}^2}{n_{d,\ell}},$$

which concludes the proof of this lemma.

### C.2 Low-Degree Polynomial lower bounds

We now consider sample complexity lower bounds within the Low-Degree Polynomial (LDP) framework, another powerful tool for studying computational hardness in statistical inference problems. We refer to [45, 55, 73, 83] for background.

Below we follow the presentation of [29]. The planted distribution with $m$ samples is generated by first drawing $\boldsymbol{w} \sim \tau_d$ (uniformly at random on the sphere), then sampling $m$ points $(y_i, \boldsymbol{x}_i) \sim_{iid} \mathbb{P}_{\nu_d, \boldsymbol{w}_*}$. The null distribution corresponds to $(y_i, \boldsymbol{x}_i) \sim_{iid} \mathbb{P}_{\nu_d, 0}$. The likelihood ratio in this model is given by

$$\mathcal{R}((y_i, \boldsymbol{x}_i)_{i \in [m]}) = \mathbb{E}_{\boldsymbol{w}} \left[ \prod_{i \in [m]} \frac{d\mathbb{P}_{\nu_d, \boldsymbol{w}}}{d\mathbb{P}_{\nu_d, 0}}(y_i, \boldsymbol{x}_i) \right].$$

We consider the orthogonal projection $\mathcal{P}_{\leq D}$ (in $L^2(\mathbb{P}_{\nu_d,0}^{\otimes m})$) onto degree at most $D$ polynomial in $\boldsymbol{z}_i$, that is, we allow arbitrary degree on the scalars $(y_i, r_i)$. We denote

$$\mathcal{R}_{\leq D}((y_i, \boldsymbol{x}_i)_{i \in [m]}) = \mathcal{P}_{\leq D} \mathcal{R}((y_i, \boldsymbol{x}_i)_{i \in [m]}). \tag{52}$$

Informally, the low-degree conjecture [45] states that for $D = \omega_d(\log d)$:

- *Weak detection hardness:* If $\|\mathcal{R}_{\leq D}\|_{L^2}^2 = 1 + o_d(1)$, then no polynomial time algorithm can achieve weak detection between $\mathbb{E}_{\boldsymbol{w}}[\mathbb{P}_{\nu_d, \boldsymbol{w}}^{\otimes m}]$ and $\mathbb{P}_{\nu_d, 0}^{\otimes m}$, that is, have a non-vanishing advantage compared to random guessing.

- *Strong detection hardness:* If $\|\mathcal{R}_{\leq D}\|_{L^2}^2 = O_d(1)$, then no polynomial time algorithm can achieve strong detection between $\mathbb{E}_{\boldsymbol{w}}[\mathbb{P}_{\nu_d, \boldsymbol{w}}^{\otimes m}]$ and $\mathbb{P}_{\nu_d, 0}^{\otimes m}$, that is, have vanishing type I and II errors.

Below, we state our results for weak and strong detection for a sequence of spherical SIMs $\{\nu_d\}_{d \geq 1}$ with $\nu_d \in \mathfrak{L}_d$. Recall that we defined:

$$\mathsf{M}_\star(\nu_d) = \inf_{\ell \geq 1} \frac{\sqrt{n_{d,\ell}}}{\|\xi_{d,\ell}\|_{L^2}^2}$$

Without loss of generality, we will assume that $M_\star(\nu_d) = O_d(\mathrm{poly}(d))$—that is, the model can be solved in polynomial time—as stated in the following assumption:

**Assumption 3.** *There exists $p \in \mathbb{N}$ such that the sequence $\{\nu_d\}_{d\geq 1}$ satisfies $M_\star(\nu_d) = O_d(d^{p/2})$.*

We can now state our bound on the low-degree projection of the likelihood ratio in this problem.

**Theorem 5.** *Let $\{\nu_d\}_{d\geq 1}$ be a sequence of spherical SIMs $\nu_d \in \mathfrak{L}_d$ satisfying Assumption 3 for some integer $p \in \mathbb{N}$. Consider the detection task with $m$ samples as defined above. There exists a constant $c > 0$ that only depends on the constants in Assumption 3 such that if $D \leq cd^{2/(p+4)}$, then*

$$\|\mathcal{R}_{\leq D}\|_{L^2}^2 - 1 \leq \sum_{s=1}^{D} \left( m\frac{D^{p/2-1}}{M_\star(\nu_d)}[e(p+1)] \right)^s. \tag{53}$$

*In particular,*

(i) *(Weak detection.) If $m = o_d\left( \frac{M_\star(\nu_d)}{D^{p/2-1}} \right)$, then $\|\mathcal{R}_{\leq D}\|_{L^2}^2 = 1 + o_d(1)$.*

(ii) *(Strong detection.) If $m = O_d\left( \frac{M_\star(\nu_d)}{D^{p/2-1}} \right)$, then $\|\mathcal{R}_{\leq D}\|_{L^2}^2 = O_d(1)$.*

The proof of this theorem can be found in Section C.2.1 below.

Combining this theorem with the low-degree conjecture stated above, we conclude that no-polynomial time algorithm can detect (and thus, estimate) the spherical single-index model $\mathbb{P}_{\nu_d,\boldsymbol{w}}$ unless

$$m \gtrsim M_\star(\nu_d).$$

We further remark that we recover the tight threshold $M_\star(\nu_d)/D^{p/2-1}$ from [29]. Indeed, consider the case of Gaussian SIM with information exponent $\mathsf{k}_\star$. We can set $p = \mathsf{k}_\star$, and our bound recover the (conjectured) optimal computational-statistical trade-off $d^{\mathsf{k}_\star/2}/D^{\mathsf{k}_\star/2-1}$ from [29], which matches the optimal known trade-off in tensor PCA [84].

**Decoupling across harmonic subspaces.** Again, we provide an interpretation of this lower bound as the optimal lower bound among subproblems indexed by $\ell \geq 1$. For each $\ell \geq 1$, we consider the task of detecting single-index models only using degree-$\ell$ spherical harmonics. Consider polynomials that are product of degree-$\ell$ spherical harmonics in $\boldsymbol{z}_i$, and denote $\mathcal{P}_{\leq D,\ell}$ the projection onto this subspace, that is

$$\mathcal{R}_{\leq D,\ell_\star}((y_i, \boldsymbol{x}_i)_{i\in[m]}) := \mathcal{P}_{\leq D,\ell}\mathcal{R}((y_i, \boldsymbol{x}_i)_{i\in[m]}) = \sum_{S\subset[m],|S|\leq \lfloor D/\ell \rfloor} \mathbb{E}_{\boldsymbol{w}}\left[ \prod_{i\in S} \xi_{d,\ell}(y_i, r_i)Q_\ell(\langle \boldsymbol{w}, \boldsymbol{z}_i \rangle) \right].$$

Then we have the following upper bound on the norm of this projected likelihood ratio.

**Corollary 4.** *Let $\{\nu_d\}_{d\geq 1}$ be a sequence of spherical SIMs $\nu_d \in \mathfrak{L}_d$ satisfying Assumption 3 for some integer $p \in \mathbb{N}$. Consider the detection task with $m$ samples as defined above and fix an integer $\ell_\star \geq 1$. Then for all $D \geq 1$, we have*

$$\|\mathcal{R}_{\leq D,\ell}\|_{L^2}^2 - 1 \leq \sum_{s=1}^{\lfloor D/\ell_\star \rfloor} \left( m\frac{eD^{\ell_\star/2-1}\|\xi_{d,\ell_\star}\|_{L^2}^2}{\sqrt{n_{d,\ell_\star}}} \right)^s. \tag{54}$$

By analogy with the low-degree conjecture, we expect that no polynomial-time algorithm only using degree-$\ell$ spherical harmonics will succeed at the detection task, unless

$$m \gtrsim \frac{\sqrt{n_{d,\ell_\star}}}{\|\xi_{d,\ell_\star}\|_{L^2}^2}. \tag{55}$$

It would be interesting to make this subspace-restricted low-degree polynomial statement more formal, and we leave it to future work. Our harmonic tensor unfolding estimator matches this heuristic sample lower bound (55) for each $\ell_\star \geq 3$.

### C.2.1 Proof of Theorem 5

Recalling the expansion of the likelihood function into Gegenbauer polynomials, we can write

$$\mathcal{R}_{\leq D}((y_i, r_i, \boldsymbol{z}_i)_{i \in [m]}) = \mathcal{P}_{\leq D} \mathbb{E}_{\boldsymbol{w}} \left[ \prod_{i \in [m]} \frac{\mathrm{d}\mathbb{P}_{\nu_d, \boldsymbol{w}}}{\mathrm{d}\mathbb{P}_{\nu_d, 0}}(y_i, r_i, \boldsymbol{z}_i) \right]$$

$$= \sum_{\ell_1 + \ldots + \ell_m \leq D} \mathbb{E}_{\boldsymbol{w}} \left[ \prod_{i \in [m]} \xi_{d, \ell_i}(y_i, r_i) Q_{\ell_i}(\langle \boldsymbol{w}, \boldsymbol{z}_i \rangle) \right].$$

The norm of this projection with respect to $\mathbb{P}_0^{\otimes m}$ is then given by

$$\|\mathcal{R}_{\leq D}\|_{L^2}^2 = \sum_{\ell_1 + \ldots + \ell_m \leq D} \mathbb{E}_{\boldsymbol{w}, \boldsymbol{w}'} \left[ \prod_{i \in [m]} \frac{\|\xi_{d, \ell_i}\|_{L^2}^2}{\sqrt{n_{d, \ell_i}}} Q_{\ell_i}(\langle \boldsymbol{w}, \boldsymbol{w}' \rangle) \right], \tag{56}$$

where we used

$$\mathbb{E}_{\boldsymbol{z}}[Q_\ell(\langle \boldsymbol{w}, \boldsymbol{z} \rangle) Q_k(\langle \boldsymbol{z}, \boldsymbol{w}' \rangle)] = \frac{\delta_{\ell k}}{\sqrt{n_{d, \ell}}} Q_\ell(\langle \boldsymbol{w}, \boldsymbol{w}' \rangle).$$

Let's separate the zero degrees from the non-zero degrees in (56):

$$\|\mathcal{R}_{\leq D}\|_{L^2}^2 - 1 = \sum_{s=1}^{D} \binom{m}{s} \sum_{\substack{1 \leq \ell_1, \ldots, \ell_s \leq D \\ \ell_1 + \ldots + \ell_s \leq D}} \mathbb{E}_{\boldsymbol{w}, \boldsymbol{w}'} \left[ \prod_{i \in [s]} \frac{\|\xi_{d, \ell_i}\|_{L^2}^2}{\sqrt{n_{d, \ell_i}}} Q_{\ell_i}(\langle \boldsymbol{w}, \boldsymbol{w}' \rangle) \right].$$

To upper bound the expectation, we will not be careful and simply use Hölder's inequality and the hypercontractivity (Lemma 20) of Gegenbauer polynomials,

$$\mathbb{E}_{\boldsymbol{w}, \boldsymbol{w}'} \left[ \prod_{i \in [s]} Q_{\ell_i}(\langle \boldsymbol{w}, \boldsymbol{w}' \rangle) \right] \leq \prod_{i \in [s]} \|Q_{\ell_i}\|_{L^s} \leq \prod_{i \in [s]} s^{\ell_i/2}.$$

We obtain the upper bound

$$\|\mathcal{R}_{\leq D}\|_{L^2}^2 - 1 \leq \sum_{s=1}^{D} \binom{m}{s} \sum_{\substack{1 \leq \ell_1, \ldots, \ell_s \leq D \\ \ell_1 + \ldots + \ell_s \leq D}} \prod_{i \in [s]} s^{\ell_i/2} \frac{\|\xi_{d, \ell_i}\|_{L^2}^2}{\sqrt{n_{d, \ell_i}}} \leq \sum_{s=1}^{D} \binom{m}{s} \rho(s, D)^s,$$

where

$$\rho(s, D) = \sum_{\ell=1}^{D} s^{\ell/2} \frac{\|\xi_{d, \ell}\|_{L^2}^2}{\sqrt{n_{d, \ell}}}.$$

Let's upper bound $\rho(s, D)$. By definition $\|\xi_{d,\ell}\|_{L^2}^2 / n_{d,\ell} \leq 1/\mathsf{M}_\star$ for all $\ell \geq 1$ (where we denote $\mathsf{M}_\star = \mathsf{M}_\star(\nu_d)$ for simplicity). Furthermore, by Assumption 3, we have $\mathsf{M}_\star \leq C d^{p/2}$. Using $\|\xi_{d,\ell}\|_{L^2} \leq 1$, we deduce a second upper bound

$$\frac{\|\xi_{d, \ell}\|_{L^2}^2}{\sqrt{n_{d, \ell}}} \leq C \frac{d^{p/2}}{\mathsf{M}_\star \sqrt{n_{d, \ell}}}.$$

Separating $\ell \leq p$ and $\ell > p$, we get

$$\rho(s, D) \leq \sum_{\ell=1}^{p} \frac{s^{\ell/2}}{\mathsf{M}_\star} + C \sum_{\ell=p+1}^{D} \frac{s^{\ell/2} d^{p/2}}{\mathsf{M}_\star \sqrt{n_{d, \ell}}} \leq \frac{s^{p/2}}{\mathsf{M}_*} \left[ p + C \sum_{\ell=p+1}^{D} \frac{D^{(\ell-p)/2} d^{p/2}}{\sqrt{n_{d, \ell}}} \right].$$

Using that for $\ell \leq D \leq \sqrt{d}$, we have $n_{d,\ell} \geq c\binom{d}{\ell} \geq c(d/\ell)^\ell$ for some constant $c > 0$, the sum simplifies to

$$\sum_{\ell=p+1}^{D} \frac{D^{(\ell-p)/2} d^{p/2}}{\sqrt{n_{d, \ell}}} \leq C' \sum_{\ell=p+1}^{d} \frac{D^{(\ell-p)/2} d^{p/2} \ell^{\ell/2}}{d^{\ell/2}} \leq C' D^{p/2} \sum_{\ell=1}^{\infty} \left( \frac{D^2}{d} \right)^\ell \leq C'' \frac{D^{p/2+2}}{d}.$$

Assuming that $D \leq d^{2/(p+4)}/\tilde{C}$, we deduce that

$$\rho(s, D) \leq \frac{s^{p/2}}{\mathsf{M}_\star}(p+1).$$

Thus, we obtain

$$\|\mathcal{R}_{\leq D}\|_{L^2}^2 - 1 \leq \sum_{s=1}^{D} \binom{m}{s} \frac{s^{sp/2}}{\mathsf{M}_\star^s}(p+1)^s \leq \sum_{s=1}^{D} \left( \frac{mD^{p/2-1}}{\mathsf{M}_\star}[e(p+1)] \right)^s,$$

which concludes the proof of Theorem 5.

**Restricted projection.** Consider the likelihood ratio projected onto product of degree-$\ell_*$ spherical harmonics. The proof is particularly simple in this case:

$$\|\mathcal{R}_{\leq D,\ell_*}\|_{L^2}^2 - 1 = \sum_{s=1}^{\lfloor D/\ell_* \rfloor} \binom{m}{s} \left( \frac{\|\xi_{d,\ell_*}\|_{L^2}^2}{\sqrt{n_{d,\ell_*}}} \right)^s \mathbb{E}_{\boldsymbol{w},\boldsymbol{w}'} \left[ Q_{\ell_*}(\langle \boldsymbol{w}, \boldsymbol{w}' \rangle)^s \right]$$

$$\leq \sum_{s=1}^{\lfloor D/\ell_* \rfloor} \left( \frac{em}{s} \right)^s \left( \frac{\|\xi_{d,\ell_*}\|_{L^2}^2}{\sqrt{n_{d,\ell_*}}} \right)^s s^{s\ell_*/2}$$

$$\leq \sum_{s=1}^{\lfloor D/\ell_* \rfloor} \left( m \frac{eD^{\ell_*/2-1}\|\xi_{d,\ell_*}\|_{L^2}^2}{\sqrt{n_{d,\ell_*}}} \right)^s,$$

which proves Corollary 4.

## D   Spectral estimators

In this section, we provide details for the spectral algorithm (SP-Alg) and prove Theorem 2.

**Requirement 1.** *We are going to implement our algorithms on $\mathcal{T}_\ell$ satisfying the following criteria.*

1. *$\|\mathcal{T}_\ell\|_2 = 1$ and $\mathbb{E}_{(y,r,z)\sim\nu_d}[\mathcal{T}_\ell(y,r)Q_\ell(z)] := \beta_{d,\ell} > 0$ (w.l.o.g.).*

2. *There exits $\kappa_\ell > 1$, $k \in \mathbb{N}$, such that, for any $p \geq 3$, we have $\|\mathcal{T}_\ell\|_p \leq \kappa_\ell \, p^{k/2}$*

Note that a transformation $\mathcal{T}_\ell$ satisfying Assumption 2 is a special case of this requirement with $k = 0$ and $\beta_{d,\ell} \geq \|\xi_{d,\ell}\|_{L^2}/\kappa_\ell$, and thus the theorem will follow by invoking the guarantee for this more general $\mathcal{T}_\ell$ satisfying Requirement 1. We first specify the spectral algorithm.

---

**Algorithm 1:** A spectral algorithm on the frequency $\ell = 1$ and $\ell = 2$.

**Input**   : An example set $S = \{(\boldsymbol{x}_i, y_i) : i \in [m]\} \sim_{iid} \mathbb{P}_{\boldsymbol{w}_*}$, the frequency $\ell \in \{1, 2\}$, and a transformation $\mathcal{T}_\ell$.

**Output** : An estimator $\hat{\boldsymbol{w}} \in \mathbb{R}^d$.

1 Decompose $\boldsymbol{x}_i = (r_i, \boldsymbol{z}_i)$.
2 **if** $\ell = 1$ **then**
3 $\quad$ Let $\hat{\boldsymbol{v}}_m := \frac{1}{m} \sum_{i \in [m]} \mathcal{T}_\ell(y_i, r_i) \sqrt{d} \, \boldsymbol{z}_i$.
4 $\quad$ $\hat{\boldsymbol{w}} = \frac{\hat{\boldsymbol{v}}_m}{\|\hat{\boldsymbol{v}_m}\|_2}$
5 **end**
6 **if** $\ell = 2$ **then**
7 $\quad$ Let $\boldsymbol{M}_m = \frac{1}{m} \sum_{i=1}^{m} \mathcal{T}_\ell(y_i, r_i)(d\boldsymbol{z}_i\boldsymbol{z}_i^\mathsf{T} - \mathbf{I}_d)$.
8 $\quad$ Let $\hat{\boldsymbol{w}} = \boldsymbol{v}_1(\boldsymbol{M}_m)$ be the eigenvector associated with the highest magnitude eigenvalue .
9 **end**
10 Return $\hat{\boldsymbol{w}}$ .

---

### D.1   Analysis of $\ell = 2$

Let $\boldsymbol{M}^* := \mathbb{E}[\boldsymbol{M}_m]$ and $(\lambda_i^*, \boldsymbol{v}_i^*)_{i \in [d]}$ be eigenpairs of $\boldsymbol{w}^*$ such that $|\lambda_1^*| \geq \cdots \geq |\lambda_d^*|$. We first show that the top eigenvector $\boldsymbol{v}_1^* = \boldsymbol{w}_*$ with $\lambda_1^* = (1 + o_d(1))\beta_{d,2}$ and the other eigenvalues are of vanishing order relative to $\lambda_1^*$.

**Lemma 5.** *We have that $M^*$ has top eigenvalue $\lambda_1^* = (1 + o_d(1))\beta_{d,2}$ with $v_1(M^*) = w_*$ and for any $2 \le i \le d$, we have $|\lambda_i^*| \lesssim \frac{\lambda_1^*}{d}$ .*

*Proof.* We have that $\mathbb{E}_{(y,r,z)\sim \mathsf{P}_{w_*}}[\mathcal{T}(y,r) \mid z] = \sum_{\ell=0}^{\infty} \beta_{d,\ell} Q_\ell(\langle w_*, z \rangle)$. We now analyze $M^*$ through its quadratic form: for any $w \in \mathbb{S}^{d-1}$, consider

$$
\begin{aligned}
w^\mathsf{T} M^* w &= \frac{1}{m} \sum_{i=1}^m \mathbb{E}_{(y_i, r_i, z_i) \sim \mathbb{P}_{w_*}}[\mathcal{T}(y_i, r_i) d w^\mathsf{T}(z_i z_i^\mathsf{T} - \mathbf{I}_d) w] \\
&= \mathbb{E}_{(y,r,z)\sim\mathbb{P}_{w_*}}[\mathcal{T}(y,r)(d\langle w, z\rangle^2 - 1)] \\
&= (1 + o_d(1))\mathbb{E}_{(y,r,z)\sim\mathbb{P}_{w_*}}[\mathcal{T}(y,r)Q_2^{(d)}(\langle w, z\rangle)] \\
&= (1 + o_d(1))\beta_{d,2}\frac{Q_2^{(d)}(\langle w, w_*\rangle)}{\sqrt{n_{d,2}}},
\end{aligned}
$$

where we used the fact that $Q_2^{(d)}(z) = (1 + o_d(1))(dz^2 - 1)$. As $Q_2^{(d)}(\cdot)$ has its maximum value at 1. Clearly, the $w^\mathsf{T} M^* w$ is maximized for $w = w_*$, and thus it is an eigenvector with the eigenvalue

$$
\lambda_1^* = (1 + o_d(1))\beta_{d,2}Q_2^{(d)}(1)/\sqrt{n_{d,2}} = (1 + o_d(1))\beta_{d,2}.
$$

It suffices to show that the other eigenvalues are of much lower magnitude. For any $w \perp w_*$, we have $Q_2^{(d)}(\langle w, w_*\rangle) = Q_2^{(d)}(0) = (1 + o_d(1))(-1)$, and thus, for $2 \le i \le d$, we have

$$
|\lambda_i^*| = (1 + o_d(1))\frac{\beta_{d,2}}{\sqrt{n_{d,2}}} \lesssim \frac{\lambda_1^*}{d},
$$

which concludes the proof of this lemma. $\qquad\square$

Our goal is to ensure that the top eigenvectors $\hat{w} = v_1(M_m)$ and $w_* = v_1(M^*)$ are close to each other, when $m$ is chose sufficiently large. By Davis-Kahan's theorem, it suffices to show the concentration between the empirically estimated matrix $M_m$, and its expectation, in the following sense.

**Lemma 6.** *There exists a constant $C > 0$ (only depending on $k$) such that, for any $\delta > 0$, with*

$$
m \ge C\frac{\kappa_\ell d}{\beta_{d,2}^2}\left(1 + \beta_{d,2}\log^{k/2+2}\left(\frac{d}{\delta\beta_{d,2}^2}\right)\right),
$$

*we have that with probability $1 - \delta$,*

$$
\|M_m - M^*\|_{\mathrm{op}} \le \frac{\lambda_1^*}{8},
$$

*where recall that $\lambda_1^*$ is the top eigenvalue of $M^*$ (see Lemma 5).*

*Proof.* Our goal is to use Lemma 25, with $Y_i = \frac{1}{m}\left(\mathcal{T}(y_i, r_i)(dz_i z_i^\mathsf{T} - \mathbf{I}_d) - M^*\right) \in \mathbb{R}^{d\times d}$ which are zero mean, and thus, $Y = M_m - M^*$. Let us bound the each quantity of interest

$$
\begin{aligned}
\sigma^2 &= \|\mathbb{E}[(M_m - M^*)^2]\|_2 = \frac{1}{m}\|\mathbb{E}[(M_1 - M^*)^2]\|_2 \le \frac{2}{m}\|\mathbb{E}[M_1^2]\|_2 \\
&\le \frac{2}{m}\sup_{w\in\mathbb{S}^{d-1}} w^\mathsf{T}\mathbb{E}[M_1^2]w \\
&= \frac{2}{m}\sup_{w\in\mathbb{S}^{d-1}}\mathbb{E}_{(y,r,z)\sim\mathbb{P}_{w_*}} w^\mathsf{T}\left[\mathcal{T}(y,r)^2\left((d^2 - 2d)zz^\mathsf{T} + \mathbf{I}_d\right)\right]w \\
&= \frac{2}{m}\sup_{w\in\mathbb{S}^{d-1}}\mathbb{E}_{(y,r,z)\sim\mathbb{P}_{w_*}}\left[\mathcal{T}(y,r)^2((d^2 - 2d)\langle w, z\rangle^2 + 1)\right].
\end{aligned}
$$

We now note that $g(\boldsymbol{z}) = (d^2 - 2d)\langle \boldsymbol{w}, \boldsymbol{z} \rangle^2 + 1$ is a polynomial of degree 2 with $\|g(\boldsymbol{z})\|_{L^2(\tau_d)} \lesssim d$. Therefore using spherical hypercontractivity (Lemma 20), we have $\|g(\boldsymbol{z})\|_{L^p(\tau_d)} \lesssim p \cdot d$. Applying Lemma 24 then gives us

$$\mathbb{E}_{(y,r,\boldsymbol{z}) \sim \mathbb{P}_{\boldsymbol{w}_*}} \left[ \mathcal{T}(y,r)^2 \cdot ((d^2 - 2d) \cdot \langle \boldsymbol{w}, \boldsymbol{z} \rangle^2 + 1) \right] \lesssim d \cdot \|\mathcal{T}\|_2^2 \cdot \max\left( 1, \log\left( \frac{\|\mathcal{T}\|_4}{\|\mathcal{T}\|_2} \right) \right) \lesssim d \log(\kappa_\ell)$$

where we used the fact that $\|\mathcal{T}\|_2^2 = 1$ and $\|\mathcal{T}\|_4 \lesssim \log(\kappa_\ell)$. We finally obtain that

$$\sigma \lesssim \sqrt{\frac{d \log \kappa_\ell}{m}} \, .$$

We next analyze the other variance term using similar idea:

$$\sigma_*^2 := \sup_{\boldsymbol{u}, \boldsymbol{v} \in \mathbb{S}^{d-1}} \mathbb{E}\left[ (\boldsymbol{u}^\mathsf{T}(\boldsymbol{M}_m - \boldsymbol{M}^*)\boldsymbol{v})^2 \right] = \frac{1}{m} \sup_{\boldsymbol{u}, \boldsymbol{v} \in \mathbb{S}^{d-1}} \mathbb{E}\left[ (\boldsymbol{u}^\mathsf{T}(\boldsymbol{M}_1 - \boldsymbol{M})\boldsymbol{v})^2 \right]$$

$$\leq \frac{4}{m} \sup_{\boldsymbol{u}, \boldsymbol{v} \in \mathbb{S}^{d-1}} \mathbb{E}[(\boldsymbol{u}^\mathsf{T}\boldsymbol{M}_1\boldsymbol{v})^2]$$

$$\leq \frac{4}{m} \sup_{\boldsymbol{u}, \boldsymbol{v} \in \mathbb{S}^{d-1}} \mathbb{E}[\mathcal{T}(y,r)^2 (d\langle \boldsymbol{u}, \boldsymbol{z} \rangle \langle \boldsymbol{v}, \boldsymbol{z} \rangle - \langle \boldsymbol{u}, \boldsymbol{v} \rangle)^2].$$

We would like to use Lemma 24 to bound the expectation, for which we will first obtain a tight bound on all the moments of $g(\boldsymbol{z}) := (d\langle \boldsymbol{u}, \boldsymbol{z} \rangle \langle \boldsymbol{v}, \boldsymbol{z} \rangle - \langle \boldsymbol{u}, \boldsymbol{v} \rangle)^2$. Let us compute

$$\|g\|_{L^2(\tau_d)} \lesssim \mathbb{E}[d^4 \langle \boldsymbol{u}, \boldsymbol{z} \rangle^4 \langle \boldsymbol{v}, \boldsymbol{z} \rangle^4 + 1]^{1/2} = (d^4 \mathbb{E}[\langle \boldsymbol{u}, \boldsymbol{z} \rangle^4 \langle \boldsymbol{v}, \boldsymbol{z} \rangle^4] + 1)^{1/2}$$

$$\leq (d^4 \sqrt{\mathbb{E}[\langle \boldsymbol{u}, \boldsymbol{z} \rangle^8] \mathbb{E}[\langle \boldsymbol{v}, \boldsymbol{z} \rangle^8]} + 1)^{1/2} = (d^4 \cdot \mathbb{E}[z_1^8] + 1)^{1/2} \lesssim 1 \, .$$

In the above, we used the Cauchy-Schwarz inequality, rotational invariance of $\tau_d$, and $\mathbb{E}[z_1^8] \lesssim 1/d^4$ respectively. Using hypercontractivity (Lemma 20), we have $\|g\|_{L^p(\tau_d)} \lesssim (p-1)^2$. We now use Lemma 24 to conclude that

$$\sigma_* \lesssim \sqrt{\frac{1}{m} \sup_{\boldsymbol{u}, \boldsymbol{v} \in \mathbb{S}^{d-1}} \mathbb{E}[\mathcal{T}(y,r)^2 (d\langle \boldsymbol{u}, \boldsymbol{z} \rangle \langle \boldsymbol{v}, \boldsymbol{z} \rangle - \langle \boldsymbol{u}, \boldsymbol{v} \rangle)^2]}$$

$$\lesssim \sqrt{\frac{\|\mathcal{T}\|_2^2 \cdot \max\left( 1, \log(\frac{\|\mathcal{T}\|_4}{\|\mathcal{T}\|_2}) \right)}{m}} \lesssim \sqrt{\frac{\log(\kappa_\ell)}{m}} \, ,$$

where again we used that $\|\mathcal{T}\|_4 \lesssim \kappa_\ell$. Our next goal is to compute $\bar{R} = \mathbb{E}\left[ \max_{i \in [n]} \|\boldsymbol{Y}_i\|_2^2 \right]^{1/2}$, where $\boldsymbol{Y}_i = \frac{1}{m}(\mathcal{T}(y_i, r_i)d\boldsymbol{z}_i\boldsymbol{z}_i^\mathsf{T} - \boldsymbol{M}^*)$. And thus, $\|\boldsymbol{Y}_i\|_2 \leq \frac{1}{m}(d|\mathcal{T}(y_i, r_i)| + \|\boldsymbol{M}^*\|_2) \lesssim \frac{1}{m}(d|\mathcal{T}(y_i, r_i)| + \beta_{d,2})$, using Lemma 5. For any $p \geq 3$, bounding the $p^{\text{th}}$ moment

$$\mathbb{E}[\|\boldsymbol{Y}_i\|_2^p]^{1/p} \lesssim \frac{1}{m} (d\|\mathcal{T}\|_p + \beta_{d,2}) \lesssim \frac{d \kappa_\ell p^{k/2}}{m} \, ,$$

Using Lemma 26, we have

$$\bar{R} = \mathbb{E}\left[ \max_{i \in [m]} \|\boldsymbol{Y}_i\|_2^2 \right]^{1/2} \lesssim \frac{d \cdot \kappa_\ell \log^{k/2} m}{m} \, .$$

The threshold for choosing $R$ is:

$$\sigma^{1/2}\bar{R}^{1/2} + \sqrt{2}\bar{R} \lesssim \sqrt{\left( \frac{d \log \kappa_\ell}{m} \right)^{1/2} \cdot \frac{d \kappa_\ell \log^{k/2} m}{m}} + \frac{d \kappa_\ell \log^{k/2} m}{m} \lesssim \frac{d \kappa_\ell \log^{k/2} m}{m} \, ,$$

where in the last step, we used the fact that we are in the regime $m \geq d$, only keeping the dominant term. Therefore, for any $\delta \geq 0$ we can choose some $R$ that satisfies

$$R \lesssim \frac{d \kappa_\ell \log^{k/2}(m/\delta)}{m} \qquad \text{and} \qquad \mathbb{P}(\max_{i \in [m]} \|\boldsymbol{Y}_i\|_2 \geq R) \leq \delta/2 \, ,$$

where in the last inequality, we used Lemma 26. Finally, we apply Lemma 25. With probability $1 - \delta/2 - de^{-t}$, we have

$$\|\boldsymbol{M}_m - \boldsymbol{M}^*\|_{\mathrm{op}} \lesssim \sqrt{\frac{d\log(\kappa_\ell)}{m}} + t^{1/2}\sqrt{\frac{\log\kappa_\ell}{m}} + \left(\frac{d\,\kappa_\ell\log^{k/2}(m/\delta)}{m} \cdot \frac{d\log\kappa_\ell}{m}\right)^{1/3} t^{2/3} + \frac{d\kappa_\ell\log^{k/2}(m/\delta)}{m}t\,.$$

Choosing $t = \log(2d/\delta)$, we obtain with probability $1 - \delta$,

$$\|\boldsymbol{M}_m - \boldsymbol{M}^*\|_{\mathrm{op}} \lesssim \sqrt{\frac{d\log\kappa_\ell}{m}} + \sqrt{\frac{\log(\kappa_\ell)\log(d/\delta)}{m}} + \left(\frac{d^2\kappa_\ell\log\kappa_\ell\,\log^{k/2}(m/\delta)\log^2(d/\delta)}{m^2}\right)^{1/3}$$

$$+ \frac{d\kappa_\ell\log^{k/2}(m/\delta)\log(d/\delta)}{m}\,.$$

Therefore, there exists a constant $C > 0$ such that, for

$$m_0 = \frac{C\,\kappa_\ell\,d}{\beta_{d,2}^2}\left(1 + \beta_{d,2}\log^{k/2+2}\left(\frac{d}{\delta\,\beta_{d,2}^2}\right)\right),$$

any $m \geq m_0$, with probability $1 - \delta$,

$$\|\boldsymbol{M}_m - \boldsymbol{M}^*\|_{\mathrm{op}} \leq \frac{\beta_{d,2}}{16} \leq \frac{\lambda_1^*}{8},$$

which concludes the proof. $\qquad\square$

**Proof of Theorem 2: Spectral algorithm, case $\ell = 2$:** For any transformation $\mathcal{T}_\ell$ satisfying requirement Requirement 1, we have that choosing m sufficiently large that is

$$\mathsf{m} \leq \frac{C\,\kappa_\ell\,d}{\beta_{d,2}^2}\left(1 + \beta_{d,2}\log^{k/2+2}\left(\frac{d}{\delta\,\beta_{d,2}^2}\right)\right)$$

by Davis and Kahn's theorem, we have

$$\min_{s\in\{\pm 1\}}\|s\boldsymbol{v}_1(\boldsymbol{M}_{\mathsf{m}}) - \boldsymbol{v}_1(\boldsymbol{M}^*)\|_2 \leq \frac{\|\boldsymbol{M}_{\mathsf{m}} - \boldsymbol{M}^*\|_{\mathrm{op}}}{|\lambda_1^* - \lambda_2^*|}\,.$$

According to Lemma 5, this corresponds to

$$\min_{s\in\{\pm 1\}}\|s\hat{\boldsymbol{w}} - \boldsymbol{w}_*\|_2 \leq \frac{\lambda_1^*/8}{(1+o(1))\,\lambda_1^*} \leq \frac{1}{4}\,.$$

Rearranging terms, we obtain $|\langle\hat{\boldsymbol{w}}, \boldsymbol{w}_*\rangle| \geq 1/4$. Finally, the above sample complexity bound of m simplifies to the one provided in Theorem 2 under the stronger Assumption 2.

### D.2 Analysis for $\ell = 1$

We now analyze the case $\ell = 1$ case, where the analysis is even simpler and follows by concentration of vector to its expected value. For simplicity, we will denote $\mathcal{T} = \mathcal{T}_1$. Let us first evaluate the expectation of the vector statistic $\boldsymbol{v}_m$ computed by the algorithm.

**Lemma 7.** *We have that $\mathbb{E}[\hat{\boldsymbol{v}}_m] = \beta_{d,1}\cdot\boldsymbol{w}_*$.*

*Proof.* For any $i \in [d]$,

$$\mathbb{E}[\boldsymbol{v}_m]_i = \mathbb{E}_{(y,r,\boldsymbol{z})\sim\mathbb{P}_{\boldsymbol{w}_*}}[\mathcal{T}(y,r)Q_1^{(d)}(z_i)] = \mathbb{E}_{(y,r,\boldsymbol{z})\sim\mathbb{P}_{\boldsymbol{w}_*}}[\mathcal{T}(y,r)Q_1^{(d)}(\langle\boldsymbol{z},\boldsymbol{e}_i\rangle)]$$

$$= \sum_{\ell=0}^{\infty}\beta_{d,\ell}\mathbb{E}_{(y,r,\boldsymbol{z})\sim\mathbb{P}_{\boldsymbol{w}_*}}[Q_\ell^{(d)}(\langle\boldsymbol{w}_*,\boldsymbol{z}\rangle)Q_1^{(d)}(\langle\boldsymbol{z},\boldsymbol{e}_i\rangle)] = \beta_{d,1}Q_1(\langle\boldsymbol{w}_*,\boldsymbol{e}_i\rangle)/\sqrt{n_{d,1}}$$

$$= \beta_{d,1}\cdot(\boldsymbol{w}_*)_i\,.$$

We conclude that $\mathbb{E}[\boldsymbol{v}_m] = \beta_{d,1}\boldsymbol{w}_*$. $\qquad\square$

We now show that for sufficiently large sample size, our final estimator $\hat{\boldsymbol{w}}$ has the desired overlap with the ground-truth $\boldsymbol{w}_*$ via concentration arguments.

**Lemma 8.** *There exists a universal constant $C > 0$ such that for any $\delta > 0$ and any*

$$m \geq \frac{C\kappa_\ell \sqrt{d}}{\beta_{d,1}^2} \left( 1 + \frac{1}{\sqrt{d}} \log^{(k+1)/2} \left( \frac{d\kappa_\ell}{\delta\beta_{d,1}} \right) \right) \quad and \quad m \geq \frac{C\kappa_\ell d}{\beta_{d,1}^2} \left( 1 + \frac{1}{d} \log^{(k+1)/2} \left( \frac{d\kappa_\ell}{\delta\beta_{d,1}} \right) \right),$$

*respectively, with probability $1 - \delta$, we have*

$$\frac{\langle \boldsymbol{v}_m, \boldsymbol{w}_* \rangle}{\|\boldsymbol{v}_m\|_2} \geq \frac{d^{-1/4}}{4} \quad and \quad \frac{\langle \boldsymbol{v}_m, \boldsymbol{w}_* \rangle}{\|\boldsymbol{v}_m\|_2} \geq \frac{1}{4}.$$

*Proof.* Denote $\boldsymbol{v}^* = \mathbb{E}[\boldsymbol{v}_m]$ and define $X_i := \frac{1}{m}(\mathcal{T}(y, r)\sqrt{d}\langle \boldsymbol{z}_i, \boldsymbol{w}_* \rangle - \langle \boldsymbol{v}^*, \boldsymbol{w}_* \rangle)$. Calculating the variance

$$\sigma^2 = \sum_{i=1}^m \mathbb{E}[X_i^2] \ \lesssim \ \frac{1}{m}\mathbb{E}_{(y,r,\boldsymbol{z})\sim\mathbb{P}_{\boldsymbol{w}_*}}[\mathcal{T}(y, r)^2 d\langle \boldsymbol{z}, \boldsymbol{w}_* \rangle^2] \ \lesssim \ \frac{\log \kappa_\ell}{m},$$

where in the last inequality we used Lemma 24 and the fact that $\|\mathcal{T}\|_2 = 1$ and $g(\boldsymbol{z}) = d\langle \boldsymbol{w}_*, \boldsymbol{z} \rangle^2$ is a polynomial of degree two with $\|g\|_2 \lesssim 1$. Moreover, for any $p \geq 2$

$$\|X_i\|_p \leq \frac{1}{m} \left( \mathbb{E}[|\mathcal{T}(y_i, r_i)|^p |Q_1(\langle \boldsymbol{w}_*, \boldsymbol{z}_i \rangle)|^p]^{1/p} + \beta_{d,1} \right) \lesssim \frac{(\|\mathcal{T}\|_{2p}\|Q_1\|_{2p} + \beta_{d,1})}{m} \lesssim \frac{\kappa_\ell \, p^{(k+1)/2}}{m},$$

where in the last inequality, we used $\langle \boldsymbol{v}^*, \boldsymbol{w}^* \rangle = \beta_{d,1} \leq 1$ (cf. Lemma 7) and $\|\mathcal{T}\|_{2p} \leq \kappa_\ell \, (2p)^{k/2}$ and $\|Q_1\|_{2p} \leq \sqrt{2p}$ by hypercontractivity (Lemma 20). Applying Lemma 27, with probability $1 - \delta$,

$$|\langle \boldsymbol{v}_m - \boldsymbol{v}^*, \boldsymbol{w}_* \rangle| \lesssim \sqrt{\frac{\log(\kappa_\ell)\log(1/\delta)}{m}} + \frac{\kappa_\ell \log(1/\delta) \log^{(k+1)/2}(m/\delta)}{m}.$$

Therefore, there is a constant $C > 0$ such that with any $m \geq C\kappa_\ell\sqrt{d}/\beta_{d,1}^2$, with probability $1 - e^{-d^c}$ for small enough $c > 0$

$$|\langle \boldsymbol{v}_m - \boldsymbol{v}^*, \boldsymbol{w}_* \rangle| \leq \frac{\beta_{d,1}}{4} \tag{57}$$

Now let $\boldsymbol{v}_m^\perp = \boldsymbol{v}_m - \langle \boldsymbol{v}_m, \boldsymbol{w}_* \rangle \boldsymbol{w}_*$ be the component of $\boldsymbol{v}_m$ orthogonal to $\boldsymbol{w}_*$. Our goal is to find a high probability bound on $\|\boldsymbol{v}_m^\perp\|_2$. Due to spherical symmetry, w.l.o.g., fix $\boldsymbol{w}_* = \boldsymbol{e}_1$ and so $S \sim \mathbb{P}_{\boldsymbol{e}_1}^m$, and the norm of the desired vector is given by

$$\|\boldsymbol{v}_m^\perp\|_2 = \sqrt{\sum_{j=2}^d (\boldsymbol{v}_m)_j^2} \text{ where } \boldsymbol{v}_m^\perp = \frac{1}{m}\sum_{i=1}^m \mathcal{T}(y_i, r_i)\sqrt{d}(\boldsymbol{z}_i)_{-1}.$$

Observe that $\boldsymbol{v}_m^\perp$ is a linear combination of $m$ i.i.d. vectors, however, the coefficients of linear combinations $\mathcal{T}(y_i, r_i)$ are not independent from the vectors $(\boldsymbol{z}_i)_{-1}$ themselves, since it is coupled with $z_{i,1}$. We will decouple the laws and make coefficients independent of the vectors. To this end, consider $(y, r, \boldsymbol{z}) \sim \mathbb{P}_{\boldsymbol{e}_1}$ and $\tilde{\boldsymbol{z}} \sim \tau_{d-1}$ independent of $(y, r, \boldsymbol{z})$. Then the following two random variables have identical laws:

$$\mathcal{T}(y, r)\sqrt{d}(\boldsymbol{z})_{-1} \equiv \frac{\mathcal{T}(y, r)}{\sqrt{1 - z_1^2}}\sqrt{d}\tilde{\boldsymbol{z}}$$

Using such argument for each of $m$ samples, and $\boldsymbol{v}_m^\perp$ viewed as a random vector of variables $(\sqrt{d}\tilde{\boldsymbol{z}}_i)_{i\in[m]}$ is sub-gaussian with variance parameter $\sigma_*^2 \leq \frac{1}{m^2}\sum_{i=1}^m \frac{\mathcal{T}(y_i,r_i)^2}{(1-z_{i,1}^2)}$. Thus, with probability $1 - 2e^{-d}$, we have $\|\boldsymbol{v}_m^\perp\|_2 \lesssim \sigma_*\sqrt{d}$. Therefore, it suffices to bound $\sigma_*$. By Lemma 27, for any $\delta > 0$ such that $\log(1/\delta) < cd$ for some $c > 0$, we have with probability $1 - \delta/2$,

$$\sigma_*^2 \lesssim \mathbb{E}[\sigma_*^2] + \frac{\sqrt{\kappa_\ell \log \kappa_\ell}}{m^{1.5}}\sqrt{\log(1/\delta)} + \frac{\kappa_\ell^2 \log(1/\delta) \log(m/\delta)^k}{m^2}.$$

Here, the condition $\log(1/\delta) < cd$ arises from the fact the function $g(z) = 1/(1 - z_1^2)$ has $\|g\|_p \lesssim 1$ only for some $p < cd$ for some universal constant $c > 0$.

$$\sigma_*^2 \lesssim \frac{\log \kappa_\ell}{m} + \frac{\sqrt{\kappa_\ell \log \kappa_\ell}}{m^{1.5}} \sqrt{\log(1/\delta)} + \frac{\kappa_\ell^2 \log(1/\delta) \log(m/\delta)^k}{m^2} \, .$$

Finally, for any $\delta < e^{-d^c}$, choosing sample size

$$m \geq \frac{C \kappa_\ell \sqrt{d}}{\beta_{d,1}^2} \left( 1 + \frac{1}{\sqrt{d}} \log^{(k+1)/2} \left( \frac{d \kappa_\ell}{\delta \beta_{d,1}} \right) \right) , \tag{58}$$

with probability $1 - \delta/2 - e^{-2d}$

$$\sigma_*^2 \lesssim \frac{\beta_{d,1}^2}{C \sqrt{d}}, \quad \text{and thus,} \quad \|\boldsymbol{v}_m^\perp\| \lesssim \sigma_* \sqrt{d} \lesssim \frac{\beta_{d,1} \sqrt{d}}{\sqrt{C \sqrt{d}}} \leq \frac{\beta_{d,1} d^{1/4}}{\sqrt{C}} \, .$$

For $C > 1$ sufficiently large, we obtain with probability $1 - \delta/2 - e^{-2d}$,

$$\|\boldsymbol{v}_m^\perp\|_2 \leq \beta_{d,1} d^{1/4}$$

Combining this with (57), with probability $1 - \delta/2 - e^{-2d} - e^{-d^c}$

$$\|\boldsymbol{v}_m - \boldsymbol{v}^*\|_2 \leq 2\beta_{d,1} d^{1/4} \, .$$

Therefore, we finally analyze our overlap combining with (57). For $C > 0$ sufficiently large, for any $\delta > 0$, for any $m \geq \frac{C \kappa_\ell \sqrt{d}}{\beta_{d,1}^2} \left( 1 + \frac{1}{\sqrt{d}} \log^{(k+1)/2} \left( \frac{\kappa_\ell d}{\beta_{d,1} \delta} \right) \right)$, with probability $1 - \delta$,

$$\frac{\langle \boldsymbol{v}_m, \boldsymbol{w}_* \rangle}{\|\boldsymbol{v}_m\|_2} \geq \frac{\langle \boldsymbol{v}^*, \boldsymbol{w}_* \rangle + \langle \boldsymbol{v}_m - \boldsymbol{v}^*, \boldsymbol{w}_* \rangle}{\|\boldsymbol{v}^*\|_2 + \|\boldsymbol{v}_m - \boldsymbol{v}^*\|_2} \geq \frac{\beta_{d,1} - \beta_{d,1}/4}{\beta_{d,1} + 2\beta_{d,1} d^{1/4}} \geq \frac{d^{-1/4}}{4} \, .$$

Similarly, if in Eq.(58), we instead chose a sample of size $m \geq \frac{C \kappa_\ell d}{\beta_{d,1}^2} \left( 1 + \frac{1}{d} \log^{(k+1)/2} \left( \frac{\kappa_\ell d}{\delta \beta_{d,1}} \right) \right)$ for sufficiently large $C > 1$, then we obtain a tighter control on $\|\boldsymbol{v}_m - \boldsymbol{v}^*\|_2$, and directly achieve weak recovery. With probability $1 - \delta/2 - e^{-2d} - e^{-d^c}$

$$\|\boldsymbol{v}_m^\perp\| \lesssim \sigma_* \sqrt{d} \lesssim \frac{\beta_{d,1} \sqrt{d}}{\sqrt{Cd}} \leq \frac{\beta_{d,1}}{\sqrt{C}} \quad \text{and} \quad \|\boldsymbol{v}_m - \boldsymbol{v}^*\|_2 \leq 2\beta_{d,1} \, .$$

Combining this with (57), with probability $1 - \delta$

$$\frac{\langle \boldsymbol{v}_m, \boldsymbol{w}_* \rangle}{\|\boldsymbol{v}_m\|_2} \geq \frac{\langle \boldsymbol{v}^*, \boldsymbol{w}_* \rangle + \langle \boldsymbol{v}_m - \boldsymbol{v}^*, \boldsymbol{w}_* \rangle}{\|\boldsymbol{v}^*\|_2 + \|\boldsymbol{v}_m - \boldsymbol{v}^*\|_2} \geq \frac{\beta_{d,1} - \beta_{d,1}/4}{\beta_{d,1} + 2\beta_{d,1}} = \frac{1}{4} \, .$$

$\square$

Note that the regime of interest where we can hope to succeed in polynomial sample and runtime is when $\|\xi_{d,1}\|_{L^2} \gg \text{poly}(d)^{-1}$ i.e. which corresponds to $\beta_{d,1} \gg \text{poly}(d)^{-1}$ for $\mathcal{T}$ from Requirement 1. In this regime, Lemma 8 establishes that with sample complexity

$$\mathsf{m} \leq \frac{C \kappa_\ell \sqrt{d}}{\beta_{d,1}^2} \sqrt{\log(1/\delta)} \quad \text{and} \quad \mathsf{m} \leq \frac{C \kappa_\ell d}{\beta_{d,1}^2} \sqrt{\log(1/\delta)}$$

one can achieve the overlaps

$$|\langle \hat{\boldsymbol{w}}, \boldsymbol{w}_* \rangle| \geq d^{-1/4}/4 \quad \text{and} \quad |\langle \hat{\boldsymbol{w}}, \boldsymbol{w}_* \rangle| \geq 1/4 \, .$$

This nearly finishes the proof of Theorem 2 for the case $\ell = 1$ (under stronger Assumption 2). Depending on the problem, the sample complexity bound either matches the one provided in Theorem 2, or it is suboptimal by a factor of $O(\sqrt{d})$. In the latter case, we can first get to $\Omega_d(d^{-1/4})$ overlap, followed by another boosting phase, as long as the following assumption holds.

**Assumption 4.** *There exists $\ell \geq 3$ and $c > 0$ such that* $\frac{\sqrt{d^\ell}}{\|\xi_{d,\ell}\|_{L^2}^2} \leq c \left( \frac{\sqrt{d}}{\|\xi_{d,1}\|_{L^2}^2} \vee d \right)$ .

Note that this assumption holds for Gaussian SIMs according to the discussion in Section 4, i.e. all $\ell \equiv \mathsf{k}_\star \mod 2$ are all optimal for samples. The next section is dedicated to showing the guarantee for boosting procedure.

### D.3 Boosting the overlap to achieve weak recovery

We now show how to boost the overlap from $\Omega_d(d^{-1/4})$ to $\Omega_d(1)$. We follow the boosting procedure introduced in [29]. The proof follows similarly, and we provide details for completeness.

---

**Algorithm 2:** A single step of the boosting algorithm on $\ell \geq 3$.

---

**Input** : An example set $S = \{(\boldsymbol{x}_i, y_i) : i \in [m]\} \sim_{iid} \mathbb{P}_{\boldsymbol{w}_*}$, the frequency $\ell \geq 3$, a transformation $\mathcal{T}_\ell$, and a vector $\boldsymbol{w}_0$ such that $\langle \boldsymbol{w}_0, \boldsymbol{w}_* \rangle \geq d^{-1/4}/4$.
**Output** : The vector $\hat{\boldsymbol{w}}$.
1 Let $\Upsilon = \lceil \log d \rceil$ be the total number of steps to be taken.
2 Divide the training set $S = \{S^{(t)}\}_{i \in [\Upsilon]}$ into disjoint collections of $\Upsilon$ steps, where $|S^{(t)}| = |S|/2^{t+2}$.
3 **for** $t = 1, \ldots, \Upsilon$ **do**
4     $\boldsymbol{w}_t = \text{boost-step}(\boldsymbol{w}_{t-1}, S^{(t)})$.
5 **end**
6 Rerurn $\hat{\boldsymbol{w}} = \boldsymbol{w}_\Upsilon$

---

**Algorithm 3:** boost-step

---

**Input** : An example set $S$ of size $m$ and a vector $\boldsymbol{v}$ such that $\langle \boldsymbol{v}, \boldsymbol{w}_* \rangle = \alpha \in [d^{-1/4}/4, 1/4]$.
**Output** : The new vector $\boldsymbol{v}_{\text{next}}$ with $\langle \boldsymbol{v}_{\text{next}}, \boldsymbol{w}_* \rangle = \alpha_{\text{next}}$ .
1 Compute $\hat{\boldsymbol{v}} = \frac{1}{m} \sum_{i=1}^m \mathcal{T}_\ell(y_i, r_i) Q'_\ell(\langle \boldsymbol{v}, \boldsymbol{z}_i \rangle) \boldsymbol{z}_i$.
2 Return $\boldsymbol{v}_{\text{next}} = \frac{\hat{\boldsymbol{v}}}{\|\hat{\boldsymbol{v}}\|_2}$ .

---

We have the following guarantee for one step of boosting algorithm.

**Lemma 9.** *There exists a constant $C = C(k, \ell)$ and $c = c(k, \ell)$ such that the following holds. For the input $\boldsymbol{v}$ such that $\langle \boldsymbol{v}, \boldsymbol{w}_* \rangle = \alpha \in [d^{-0.25}/4, 1/4]$ of the* boost-step *procedure (Algorithm 3), we have that for any*

$$m \geq C\kappa_\ell \frac{d}{\beta_{d,\ell}^2 \alpha^{2\ell-4}} \, ,$$

*with probability $1 - e^{-d^c}$, we have $\alpha_{\text{next}} = \langle \boldsymbol{v}_{\text{next}}, \boldsymbol{w}_* \rangle \geq 2\alpha$.*

Using this lemma we can obtain the following theorem on the performance of the boosting algorithm.

**Theorem 6.** *There exists a constant $C = C(k, \ell) > 1$ and $c = c(k, \ell) > 0$ such that the following holds. On the initialization $|\langle \boldsymbol{w}_0, \boldsymbol{w}_* \rangle| \geq d^{-1/4}/4$, the boosting algorithm (Algorithm 2) on the training set $S$ whose size is*

$$\mathsf{m} \leq C\kappa_\ell \frac{\sqrt{d^\ell}}{\beta_{d,\ell}^2} \, ,$$

*with probability $1 - e^{-d^c}$, we have $\langle \hat{\boldsymbol{w}}, \boldsymbol{w}_* \rangle \geq 1/4$.*

*Proof.* According to Lemma 9, choosing $|S^{(1)}| \gtrsim \frac{\kappa_\ell d}{\beta_{d,\ell}^2 \alpha_0^{2\ell-4}} \asymp \frac{\kappa_\ell \sqrt{d^\ell}}{\beta_{d,\ell^2}}$ (hiding constants in $k, \ell$), with probability $1 - e^{-d^c}$ we have $|\langle \boldsymbol{w}_1, \boldsymbol{w}_* \rangle| \geq 2\alpha_0$. The sample size threshold for subsequent iteration is strictly less than 1/2 of the previous one since $\alpha \in [d^{-1/4}, 1/4]$ and $\ell \geq 3$. Therefore, the overlap increases geometrically, and we have $\langle \boldsymbol{w}_\Upsilon, \boldsymbol{w}_* \rangle \geq 1/4$ in $\Upsilon = \log(d^{1/4}) \asymp \log d$ iterations. The probability of success is still $1 - e^{-d^c}$ by union bound over $\log d$ for smaller $c > 0$. For the total sample size it suffices to choose,

$$\mathsf{m} = |S| \leq \sum_{t=1}^{\Upsilon} |S^{(t)}| = |S^{(1)}| \sum_{t=1}^{\Upsilon} \frac{1}{2^{t-1}} \leq 2|S^{(1)}| \lesssim \kappa_\ell \frac{\sqrt{d^\ell}}{\beta_{d,\ell}^2} \, .$$

$\square$

We now return to the deferred proof that shows the overlap increases geometrically in the boost-step procedure.

*Proof of Lemma 9.* Recall from Appendix B that we use $P_\ell(\cdot) = Q_\ell(\cdot)/\sqrt{n_{d,\ell}}$ to denote Gegenbauer polynomial that is normalized to have $P_\ell(1) = 1$. We first note that

$$
\begin{aligned}
\boldsymbol{v}^* = \mathbb{E}[\hat{\boldsymbol{v}}] &= \mathbb{E}_{\mathbb{P}_{\boldsymbol{w}_*}}[\mathcal{T}_\ell(y,r)Q'_\ell(\langle \boldsymbol{v}, \boldsymbol{z}\rangle)\boldsymbol{z}] = \nabla_{\boldsymbol{v}}\mathbb{E}_{\mathbb{P}_{\boldsymbol{w}_*}}[\mathcal{T}_\ell(y,r)Q_\ell(\langle \boldsymbol{v}, \boldsymbol{z}\rangle)] \\
&= \beta_{d,\ell}\nabla_{\boldsymbol{v}}\mathbb{E}[Q_\ell(\langle \boldsymbol{w}_*, \boldsymbol{z}\rangle)Q_\ell(\langle \boldsymbol{v}, \boldsymbol{z}\rangle)] \\
&= \beta_{d,\ell}\nabla_{\boldsymbol{v}}P_\ell(\langle \boldsymbol{w}_*, \boldsymbol{v}\rangle) = \beta_{d,\ell}P'_\ell(\langle \boldsymbol{v}, \boldsymbol{w}_*\rangle)\boldsymbol{w}_* \\
&= \beta_{d,\ell}P'_\ell(\alpha)\boldsymbol{w}_* \, .
\end{aligned}
$$

Consider any fixed $\boldsymbol{w} \in \mathbb{S}^{d-1}$ and consider the following analysis. Define $X_i = \frac{1}{m}(\mathcal{T}_\ell(y_i, r_i)Q'_\ell(\langle \boldsymbol{v}, \boldsymbol{z}_i\rangle)\langle \boldsymbol{z}_i, \boldsymbol{w}\rangle - \langle \boldsymbol{v}_*, \boldsymbol{w}\rangle)$. Using Lemma 24, we can say that

$$
\mathbb{E}[X_i^2] \leq \frac{2}{m^2}\mathbb{E}_{\mathbb{P}_{\boldsymbol{w}}}[\mathcal{T}_\ell(y,r)^2\, Q'_\ell(\langle \boldsymbol{v}, \boldsymbol{z}\rangle)^2\langle \boldsymbol{z}, \boldsymbol{w}\rangle^2] \lesssim \frac{1}{m^2}\log^\ell(\kappa_\ell) \, ,
$$

where we used the fact that $g(\boldsymbol{z}) = Q'_\ell(\langle \boldsymbol{v}, \boldsymbol{z}\rangle)^2\langle \boldsymbol{z}, \boldsymbol{w}\rangle^2$ is a polynomial of degree $2\ell$ with $\|g\|_2 \lesssim 1$. Thus, by Lemma 20, we have $\|g\|_p \lesssim p^\ell$, which allows us to use Lemma 24. Similarly, computing

$$
\|X_i\|_p \lesssim \frac{1}{m}\mathbb{E}[|\mathcal{T}_\ell(y,r)\underbrace{Q'_\ell(\langle \boldsymbol{v}, \boldsymbol{z}\rangle)\langle \boldsymbol{z}, \boldsymbol{w}\rangle}_{:=g(\boldsymbol{z})}|^p]^{1/p} \leq \frac{1}{m}\|\mathcal{T}_\ell\|_{2p}\|g\|_{2p} \lesssim \frac{\kappa_\ell\, p^{(k+\ell)/2}}{m} \, ,
$$

where we used the facts that $\|\mathcal{T}_\ell\|_{2p} \lesssim \kappa_\ell p^{k/2}$ and $g(\boldsymbol{z})$ is a polynomial of degree $\ell$ with $\|g\|_2 \lesssim 1$, and thus, by hypercontractivity, we have $\|g\|_{2p} \lesssim p^{\ell/2}$. Using Lemma 27, with probability $1 - \delta$

$$
|\langle \hat{\boldsymbol{v}} - \boldsymbol{v}^*, \boldsymbol{w}\rangle| \lesssim \sqrt{\frac{\log^\ell \kappa_\ell \log(1/\delta)}{m}} + \frac{\kappa_\ell \log(1/\delta) \log^{(k+\ell)/2}(m/\delta)}{m} \, .
$$

Therefore, invoking this guarantee for $\boldsymbol{w} \in \{\boldsymbol{w}_*, \boldsymbol{v}\}$, we obtain that with probability $1 - e^{-d^c}$,

$$
|\langle \hat{\boldsymbol{v}} - \boldsymbol{v}^*, \boldsymbol{w}_*\rangle| + |\langle \hat{\boldsymbol{v}} - \boldsymbol{v}^*, \boldsymbol{v}\rangle| \lesssim \sqrt{\frac{\log^\ell \kappa_\ell \log(1/\delta)}{m}} + \frac{\kappa_\ell \log(1/\delta) \log^{(k+\ell)/2}(m/\delta)}{m} \, .
$$

Our next goal is to bound $\|\hat{\boldsymbol{v}}^\perp\|_2$, where $\hat{\boldsymbol{v}}^\perp$ is the component of $\hat{\boldsymbol{v}}$ orthogonal to $\mathrm{span}\{\boldsymbol{w}_*, \boldsymbol{v}\}$. Due to rotational symmetry, w.l.o.g., let us say $\boldsymbol{w}_* = \boldsymbol{e}_1$ and $\boldsymbol{v} = \alpha\boldsymbol{e}_1 + \sqrt{1 - \alpha^2}\boldsymbol{e}_2$. Then $\hat{\boldsymbol{v}}^\perp = \hat{\boldsymbol{v}} - (\hat{\boldsymbol{v}})_1\boldsymbol{e}_1 - (\hat{\boldsymbol{v}})_2\boldsymbol{e}_2$. So our goal is to bound

$$
\|\hat{\boldsymbol{v}}^\perp\|_2 \quad \text{where} \quad \hat{\boldsymbol{v}} = \frac{1}{m}\sum_{i=1}^m \mathcal{T}_\ell(y_i, r_i)Q'_\ell(\alpha z_{i,1} + \sqrt{1 - \alpha^2}z_{i,2})(\boldsymbol{z}_i)_{3:d} \, .
$$

Consider the following analysis similar to the proof of Lemma 8. For a single sample $(y, r, \boldsymbol{z}) \sim \mathbb{P}_{\boldsymbol{e}_1}$, one can define $\tilde{\boldsymbol{z}} \sim S^{d-3}$ independent of $(y, r, \boldsymbol{z})$. The following two random variables have identical distribution.

$$
\mathcal{T}_\ell(y,r)Q'_\ell(\alpha z_1 + \sqrt{1 - \alpha^2}z_2)(\boldsymbol{z}_i)_{3:d} \equiv \mathcal{T}_\ell(y,r)\frac{Q'_\ell(\alpha z_1 + \sqrt{1 - \alpha^2}z_2)}{\sqrt{1 - z_1^2 - z_2^2}}\tilde{\boldsymbol{z}} \, .
$$

Now let us define $\sqrt{d}\tilde{\boldsymbol{z}} \sim$ consider $\boldsymbol{z}_i^\perp$, the component of $\boldsymbol{z}_i$ that is orthogonal to . Using the identical argument from Lemma 7 that $\hat{\boldsymbol{v}}^\perp$ is sub-Gaussian in random variables $(\tilde{\boldsymbol{z}}_i)_{i \in [m]}$, with probability $1 - e^{-2d}$, we have $\|\hat{\boldsymbol{v}}^\perp\| \lesssim \sigma_*\sqrt{d}$, where the parameter

$$
\sigma_*^2 = \frac{1}{m^2}\sum_{i=1}^m \mathcal{T}_\ell(y_i, r_i)^2\frac{Q'_\ell(\alpha z_{i,1} + \sqrt{1 - \alpha^2}\, z_{i,2})^2}{d\,(1 - z_{i,1}^2 - z_{i,2}^2)} \, .
$$

Using exactly the same bounding strategy used in Lemma 8, for any $\delta \geq d^{-d^c}$, we have with probability $1 - e^{-d^c}$,

$$
\sigma_*^2 \lesssim \frac{\log \kappa_\ell}{m} + \frac{\sqrt{\kappa_\ell \log \kappa_\ell}}{m^{1.5}}\sqrt{\log(1/\delta)} + \frac{\kappa_\ell^2 \log(1/\delta) \log^{k+\ell}(m/\delta)}{m^2} \, .
$$

Overall, we can conclude that there exists some constant $C(k, \ell) > 1$ such that choosing any

$$m \geq C\kappa_\ell \frac{d}{\beta_{d,\ell}^2 \alpha^{2\ell-4}},$$

with probability $1 - e^{-d^c}$,

$$\|\hat{\boldsymbol{v}}^\perp\| \lesssim \sigma_* \sqrt{d} \lesssim \frac{\beta_{d,\ell}\alpha^{\ell-2}\sqrt{d}}{\sqrt{Cd}} \leq \frac{\alpha^{\ell-2}\beta_{d,\ell}}{16} \quad \text{and} \quad \langle \hat{\boldsymbol{v}} - \boldsymbol{v}^*, \boldsymbol{w}_* \rangle \leq \frac{\alpha^{\ell-1}\beta_{d,\ell}}{4} .$$

Combining we have

$$\|\hat{\boldsymbol{v}} - \boldsymbol{v}^*\|_2 \leq (1 + o(1))\frac{\alpha^{\ell-2}\beta_{d,\ell}}{8} .$$

Finally, analyzing the quantity of desired interest under this high probability event that happens with probability $1 - e^{-d^c}$:

$$\alpha_{\text{next}} = \langle \boldsymbol{v}_{\text{next}}, \boldsymbol{w}_* \rangle = \frac{\langle \hat{\boldsymbol{v}}, \boldsymbol{w}_* \rangle}{\|\hat{\boldsymbol{v}}\|_2} \geq \frac{\langle \boldsymbol{v}^*, \boldsymbol{w}_* \rangle + \langle \hat{\boldsymbol{v}} - \boldsymbol{v}^*, \boldsymbol{w}_* \rangle}{\|\boldsymbol{v}^*\|_2 + \|\hat{\boldsymbol{v}} - \boldsymbol{v}^*\|_2} \geq \frac{\beta_{d,\ell}P'_\ell(\alpha) - \beta_{d,\ell}\alpha^{\ell-1}/100}{\beta_{d,\ell}P'_\ell(\alpha) + \beta_{d,\ell}\alpha^{\ell-2}/50}$$

$$\geq (1 + o(1))\left(\frac{\alpha^{\ell-1} - \frac{\alpha^{\ell-1}}{100}}{\alpha^{\ell-1} + \frac{\alpha^{\ell-2}}{50}}\right) \geq \frac{98\alpha}{50\alpha + 1} \geq 2\alpha .$$

$\square$

# E    Online SGD estimator

In this section, we present the analysis of the online SGD on the harmonic loss which stated in Theorem 3. As in Section 3, we will implement the algorithm on $\mathcal{T}_\ell$ for $\ell > 2$. We work under the following assumption

**Assumption 5.** *For each $\ell \geq 1$, we assume that there exists $\mathcal{T}_\ell$ such that $\|\mathcal{T}_\ell\|_{L^2} = 1$, $\|\mathcal{T}_\ell\|_\infty \leq \kappa_\ell$, consider $\mathcal{T}_\ell := \xi_{d,\ell}/\|\xi_{d,\ell}\|_{L^2}$, and we have the following inequality*

$$\|\xi_{d,\ell}\|_{L^8} \leq C\|\xi_{d,\ell}\|_{L^4}^2, \tag{59}$$

*and we denote $\mathbb{E}[\mathcal{T}_\ell(y, r)Q_\ell(\langle \boldsymbol{w}, \boldsymbol{z} \rangle)] := \beta_{d,\ell} > 0$.*

We perform online stochastic gradient descent on the squared loss

$$L(\boldsymbol{w}; \boldsymbol{z}_i, y_i, r_i) = (\mathcal{T}_\ell(y_i, r_i) - Q_\ell(\langle \boldsymbol{w}, \boldsymbol{z}_i \rangle))^2 . \tag{60}$$

We consider spherical gradients and project at each step to keep $\boldsymbol{w}_t \in \mathbb{S}^{d-1}$.

---

**Algorithm 4:** Online SGD algorithm on the frequency $\ell$.

---

**Input**   : An example set $S = \{(\boldsymbol{x}_i, y_i) : i \in [m]\} \sim_{iid} \mathbb{P}_{\boldsymbol{w}_*}$, the frequency $\ell > 2$, and a transformation $\mathcal{T}_\ell$, a step size $\eta$ and a number of step-size.
**Output** : An estimator $\hat{\boldsymbol{w}} \in \mathbb{R}^d$ .
1 Decompose $\boldsymbol{x}_i = (r_i, \boldsymbol{z}_i)$.
2 Sample $\boldsymbol{w}_0 \in \mathbb{S}^{d-1}$.
3 **for** $i = 1, \dots, N$ **do**
4     Compute $L_i := L(\boldsymbol{w}_{i-1}; \boldsymbol{z}_i, y_i, r_i)$
5     Let $\boldsymbol{w}_i := \frac{\boldsymbol{w}_{i-1} - \eta \nabla_{\boldsymbol{w}_{i-1}}^{\mathbb{S}^{d-1}} L_i}{\|\boldsymbol{w}_{i-1} - \eta \nabla_{\boldsymbol{w}_{i-1}}^{\mathbb{S}^{d-1}} L_i\|}$
6 **end**
7 Return $\hat{\boldsymbol{w}_N}$ .

---

We state the formal statement for weak recovery of online SGD. We focus on weak recovery, and refer to the section for explanations on how to boost the weak to strong recovery. We also only focus on $\ell \geq 3$, however notice that the proof can be adapted to the cases $\ell \in \{1, 2\}$.

**Theorem 7** (Online SGD for learning $\nu_d$ ). *Let $(\boldsymbol{w}_t)_{t\geq 0}$ the iterates of the SGD dynamics with the loss given by Eq.(60), with $b > s_\ell$ (where $s_\ell$ only depends on $\ell$) then conditionally on $m_0 \geq \frac{b}{\sqrt{d}}$, we then have*

$$\tau_{1/2}^+ \leq \frac{\mathcal{C}_\ell d^{\ell-1}}{\beta_{d,\ell}^2},$$

*with probability at least $c > 0$ for some constant $c > 0$.*

The proof of this theorem will follow by adapting the arguments from [10, 88].

The good initialization probability is at least constant (see [88, Appendix A]). Thus, the above online SGD algorithm succeeds in total with probability at least constant bounded away from zero. We can then boost the confidence to $1 - \delta$ by just choosing the best estimator over multiple starts, and $O(\log(1/\delta))$ trails suffice.

Let introduce some notations for the following, denote $m_t = \langle \boldsymbol{w}_t, \boldsymbol{w}_* \rangle$, we define $\tau_c^- = \inf\{t \geq 0 \colon m_t \leq c\}$, and $\tau_c = \inf\{t \geq 0 \colon m_t \geq c\}$.

*Proof of theorem 7.* Let $\ell \geq 3$. Consider a transformation $\mathcal{T}_\ell$ given by assumption 5. We first state a lemma 10 on the population loss defined as

$$\mathcal{L}(\boldsymbol{w}) = \mathbb{E}[L(\boldsymbol{w}; \boldsymbol{z}, y, r)]. \tag{61}$$

**Lemma 10** (Population loss). *Let $\ell \geq 1$, consider the normalized tranformation $\mathcal{T}_\ell$ given by Assumption 5, we then have the following inequality*

$$\forall m \geq 2\sqrt{\frac{s^*}{d}}, \quad \langle \nabla_{\boldsymbol{w}}^{\mathbb{S}^{d-1}} \mathcal{L}(\boldsymbol{w}), \boldsymbol{w}_* \rangle \leq -2(1-m^2)\beta_{d,\ell} \frac{\ell(\ell+d-2)}{d-1} \langle \boldsymbol{w}, \boldsymbol{w}_* \rangle^{\ell-1}, \tag{62}$$

*where $s^* = \sqrt{\frac{(\ell-2)(\ell+d-3)}{(\ell-d/2-3)(\ell+d/2-2)}} \cos(\pi/\ell)$.*

**Discretization bounds.** In this part, we give bounds on the discretization error from the online SGD and the population loss gradient flow. Consider the SGD iterations

$$\boldsymbol{w}_{t+1} = \frac{\boldsymbol{w}_t - \eta \nabla_{\boldsymbol{w}}^{\mathbb{S}^{d-1}} L(\boldsymbol{w}_t; \boldsymbol{z}_t, r_t, y_t)}{\|\boldsymbol{w}_t - \eta \nabla_{\boldsymbol{w}}^{\mathbb{S}^{d-1}} L(\boldsymbol{w}_t; \boldsymbol{z}_t, r_t, y_t)\|},$$

initialized at $\boldsymbol{w}_0$ uniformly on the sphere $\mathbb{S}^{d-1}$.

**Proposition 4** (Discretization bound). *Consider*

$$p_{\eta,\mathcal{K}_1,\mathcal{K}_2,\mathcal{K}_3} = \frac{\mathcal{K}_2 dT\eta^2}{b} + \exp\left(-\frac{b^2}{2(\beta_{d,\ell}^2 + \mathcal{K}_2 d^2\eta^2)d\eta^2 T + 2bd^{1/2}\eta(\beta_{d,\ell} + \eta d^{3/2})}\right)$$

$$+ \frac{\mathcal{K}_3 Td^{1/2}\eta^3}{b} + \frac{\sqrt{\mathcal{K}_1 \mathcal{K}_2} T\eta^2 d^{-1}}{b},$$

*where we define*

$$\mathcal{K}_1 := K\ell \left(\frac{4e}{\ell}\right)^{\ell/2} \log(\|\mathcal{T}_\ell\|_4^2)^{\ell/2},$$

$$\mathcal{K}_2 := 4K\ell \left(\frac{4e}{4\ell}\right)^{4\ell/2} \|\mathcal{T}_\ell\|_4^4 \log\left(\frac{\|\mathcal{T}_\ell\|_8^2}{\|\mathcal{T}_\ell\|_4^4}\right)^{4\ell/2},$$

$$\mathcal{K}_3 := 2K \left(\frac{4e}{\ell}\right)^{\ell/2} \log(\|\mathcal{T}_\ell\|_4^2)^{\ell/2}.$$

*With probability at least $1 - p_{\eta,\mathcal{K}_1,\mathcal{K}_2,\mathcal{K}_3}$, we have*

$$m_T \geq \frac{m_0}{2} + \eta\beta_{\ell,d} \frac{\ell \cdot (\ell+d-2)}{d-1} \sum_{t=0}^{T-1} (1-m_t^2)m_t^{\ell-1}.$$

*Conditioned on the event $\{T \leq \tau_{1/2}^+ \wedge \tau_{2s^*/\sqrt{d}}^-\}$, we have the following inequality*

$$m_T \geq \frac{s^*}{\sqrt{d}} + \eta\beta_{\ell,d} \frac{\ell \cdot (\ell+d-2)}{(d-1)2^{\ell+1}} \sum_{t=0}^{T-1} m_t^{\ell-1}.$$

**Sample complexity for weak recovery** We are interested in the weak recovery setting i.e we want to bound $\tau_{1/2}$. We work under the event $\{T \le \tau_{1/2}^+ \wedge \tau_{2s^*/\sqrt{d}}^-\}$, we then can apply the same analysis as in [88]. We choose $\eta = c_\ell \beta_{\ell,d} d^{-\ell/2}$, we then have

$$\eta \tau_{1/2}^+ \le \frac{2d^{\ell/2-1}}{\beta_{\ell,d}\frac{\ell \cdot (\ell+d-2)}{(d-1)2^{\ell+1}}},$$

with probability at least $1 - \frac{\mathcal{K}_2 + \mathcal{K}_3/bd + \sqrt{\mathcal{K}_1\mathcal{K}_2}}{b} + \exp\left(-\frac{b^2}{2\beta_{d,\ell}^2 + Cbd^{-1/2}}\right)$. Rearranging this, it gives us

$$\tau_{1/2}^+ \le \frac{2d^{\ell-1}}{\beta_{\ell,d}^2 c_\ell \frac{\ell \cdot (\ell+d-2)}{(d-1)2^{\ell+1}}} \le \frac{2d^{\ell-1}}{\beta_{\ell,d}^2} \cdot \left(c_\ell \frac{\ell \cdot (\ell+d-2)}{(d-1)2^{\ell+1}}\right)^{-1}.$$

$\square$

**Proof of Lemma 10**

*Proof.* We have the expansion

$$\mathbb{E}[\mathcal{T}_\ell(y,r)|\boldsymbol{z}] = \sum_{i=0}^{+\infty} \beta_{d,i} Q_i^{(d)}(\langle \boldsymbol{w}_*, \boldsymbol{z}\rangle). \tag{63}$$

Consider the population mean-squared loss (we can also directly use the correlation loss)

$$\mathcal{L}(\boldsymbol{w}) = \mathbb{E}_{(y,r,\boldsymbol{z})}\left[\left(\mathcal{T}_\ell(y,r) - Q_\ell^{(d)}(\langle \boldsymbol{w}, \boldsymbol{z}\rangle)\right)^2\right] = 2 - 2\beta_{d,\ell}\mathbb{E}\left[\mathcal{T}_\ell(y,r)Q_\ell^{(d)}(\langle \boldsymbol{w}, \boldsymbol{z}\rangle)\right].$$

Using the above decomposition (63), and orthogonality of spherical harmonics

$$\mathcal{L}(\boldsymbol{w}) = 2 - 2\beta_{d,\ell}\mathbb{E}[Q_\ell^{(d)}(\langle \boldsymbol{w}_*, \boldsymbol{z}\rangle)Q_\ell^{(d)}(\langle \boldsymbol{w}, \boldsymbol{z}\rangle)]. \tag{64}$$

Using the identity (35) and plugging it into (64), we then have

$$\mathcal{L}(\boldsymbol{w}) = 2 - 2\frac{\beta_{d,\ell}}{\sqrt{n_{\ell,d}}}Q_\ell^{(d)}(\langle \boldsymbol{w}_*, \boldsymbol{w}\rangle).$$

Let denote $m := \langle \boldsymbol{w}_*, \boldsymbol{w}\rangle$, we can rewrite the loss in term of the overlap parameter. The spherical gradient of the population loss is given by

$$\langle \nabla_{\boldsymbol{w}}^{\mathbb{S}^{d-1}}\mathcal{L}(\boldsymbol{w}), \boldsymbol{w}^*\rangle = -(1-m^2)\ell'(m) = -2(1-m^2)\frac{\beta_{d,\ell}}{\sqrt{n_{d,\ell}}}Q_\ell^{(d)}(\langle \boldsymbol{w}_*, \boldsymbol{w}\rangle)'.$$

We can use the representation of the derivative of Gegenbauer i.e $Q_\ell^{(d)}(\boldsymbol{z})' = \frac{\ell(\ell+d-2)\sqrt{n_{d,\ell}}}{(d-1)\sqrt{n_{d+2,\ell-1}}}Q_{\ell-1}^{(d+2)}(\boldsymbol{z})$ ([88] Fact C.3). So, the loss can be written as

$$\langle \nabla_{\boldsymbol{w}}^{\mathbb{S}^{d-1}}\mathcal{L}(\boldsymbol{w}), \boldsymbol{w}^*\rangle = -(1-m^2)\ell'(m) = -2(1-m^2)\frac{\beta_{d,\ell}\ell(\ell+d-2)}{(d-1)\sqrt{n_{d+2,\ell-1}}}Q_{\ell-1}^{(d+2)}(\langle \boldsymbol{w}_*, \boldsymbol{w}\rangle).$$

We use the facts C.4 and C.5 from [88] (note that the Gegenbauer polynomials in [88] is normalized such that $P_\ell^{(d)}(1) = 1$, meanwhile we consider $Q_\ell^{(d)}(1) = \sqrt{B(d,\ell)}$) to state that

$$\forall m \ge 2\sqrt{\frac{s^*}{d}}, \quad \langle \nabla_{\boldsymbol{w}}^{\mathbb{S}^{d-1}}\mathcal{L}(\boldsymbol{w}), \boldsymbol{w}_*\rangle \le -2(1-m^2)\beta_{d,\ell}\frac{\ell(\ell+d-2)}{d-1}\langle \boldsymbol{w}, \boldsymbol{w}_*\rangle^{\ell-1}, \tag{65}$$

where $s^* = \sqrt{\frac{(\ell-2)(\ell+d-3)}{(\ell-d/2-3)(\ell+d/2-2)}}\cos(\pi/\ell)$. $\square$

**Proof of Proposition 4**

*Proof.* In the following, we denote $r_t = \|\boldsymbol{w}_t - \eta \nabla_{\boldsymbol{w}}^{\mathbb{S}^{d-1}} L(\boldsymbol{w}_t; \boldsymbol{z}_t, y_t)\|$ and the martingale part $\boldsymbol{M}_t = L(\boldsymbol{w}_t; \boldsymbol{z}_t, y_t) - \mathbb{E}[L(\boldsymbol{w}_t; \boldsymbol{z}_t, y_t)]$. We have the recursion

$$m_{t+1} = \frac{1}{r_t}\left(m_t - \eta\langle\nabla_{\boldsymbol{w}}^{\mathbb{S}^{d-1}}\mathcal{L}(\boldsymbol{w}_t), \boldsymbol{w}_*\rangle - \eta\langle\nabla_{\boldsymbol{w}}^{\mathbb{S}^{d-1}}\boldsymbol{M}_t, \boldsymbol{w}^*\rangle\right). \tag{66}$$

The strategy of the proof is to use the results from [88]. The proofs of these lemmas are classical bounds of martingale relying on some assumptions on the moments of the gradients. Notice here that $C$ is no longer a constant and extra care of the analysis is necessary for the proof of Lemma [88, Lemma B.4]. To use the lemmas [88, Lemmas B.2,B.3,B.4], we need to prove the bounds on the growth of gradients norms of [88, Lemma B.8]. We check this

$$
\begin{aligned}
\mathbb{E}_{(y,r,\boldsymbol{z})}[\|\nabla_{\boldsymbol{w}}^{\mathbb{S}^{d-1}} L(\boldsymbol{w}_t; \boldsymbol{z}_t, r_t, y_t)\|^2] &= \mathbb{E}_{(y,r,\boldsymbol{z})}[\|\mathsf{P}_{\boldsymbol{w}_t}(\boldsymbol{z}_t)(Q_\ell^{(d)})'(\langle\boldsymbol{w}, \boldsymbol{z}_t\rangle)\mathcal{T}_\ell(y,r)\|^2] \\
&\leq \mathbb{E}_{(y,r,\boldsymbol{z})}\left[\left|(Q_\ell^{(d)})'(\langle\boldsymbol{w}, \boldsymbol{z}_t\rangle)\mathcal{T}_\ell(y,r)\right|^2\right] \\
&\leq C(d)^2\mathbb{E}\left[\left|Q_{\ell-1}^{(d-2)}(\langle\boldsymbol{w}, \boldsymbol{z}\rangle)\right|^2\mathcal{T}_\ell(y,r)^2\right] \\
&\leq d\ell\left(\frac{4e}{\ell}\right)^{\ell/2}\mathbb{E}\left[\mathcal{T}_\ell(y,r)^2\right]\log(1/\mathbb{E}\left[\mathcal{T}_\ell(y,r)\right])^{\ell/2} \\
&\leq Kd\ell\left(\frac{4e}{\ell}\right)^{\ell/2}\log(\|\mathcal{T}_\ell\|_4^2)^{\ell/2} \\
&\leq d\mathcal{K}_1.
\end{aligned}
$$

where we have used Lemma 24, the identity and the hypercontractivity and Jensen inequality in the last line.

$$
\begin{aligned}
\mathbb{E}_{(y,r,z)}[\|\nabla_{\boldsymbol{w}}^{\mathbb{S}^{d-1}} L(\boldsymbol{w}_t; \boldsymbol{z}_t, r_t, y_t)\|^4] &= \mathbb{E}_{(y,r,z)}[\|\mathsf{P}_{\boldsymbol{w}_t}(\boldsymbol{z}_t)(Q_\ell^{(d)})'(\langle\boldsymbol{w}, \boldsymbol{z}_t\rangle)\mathcal{T}_\ell(y,r)\|^4] \\
&\leq \mathbb{E}_{(y,r,\boldsymbol{z})}\left[\left|(Q_\ell^{(d)})'(\langle\boldsymbol{w}, \boldsymbol{z}_t\rangle)\mathcal{T}_\ell(y,r)\right|^4\right] \\
&\leq C(d)^4\mathbb{E}\left[\left|Q_{\ell-1}^{(d-2)}(\langle\boldsymbol{w}, \boldsymbol{z}\rangle)\right|^4|\mathcal{T}_\ell(y,r)|^4\right] \\
&\leq 4C(d)^4\ell\left(\frac{4e}{4\ell}\right)^{4\ell/2}\|\mathcal{T}_\ell\|_4^4\log\left(\frac{\|\mathcal{T}_\ell\|_8^2}{\|\mathcal{T}\|_4^4}\right)^{4\ell/2}. \\
&\leq 4d^2K\ell\left(\frac{4e}{4\ell}\right)^{4\ell/2}\|\mathcal{T}_\ell\|_4^4\log\left(\frac{\|\mathcal{T}_\ell\|_8^2}{\|\mathcal{T}_\ell\|_4^4}\right)^{4\ell/2} \\
&\leq d^2\mathcal{K}_2.
\end{aligned}
$$

We have $\boldsymbol{M}(\boldsymbol{w}_t; \boldsymbol{z}_t, r_t, y_t) = L(\boldsymbol{w}_t; \boldsymbol{z}_t, r_t, y_t) - \mathbb{E}[L(\boldsymbol{w}_t; \boldsymbol{z}_t, r_t, y_t)]$, hence

$$
\begin{aligned}
\langle\nabla_{\boldsymbol{w}}^{\mathbb{S}^{d-1}}\boldsymbol{M}(\boldsymbol{w}_t; \boldsymbol{z}_t, r_t, y_t), \boldsymbol{w}_*\rangle &= \langle\nabla_{\boldsymbol{w}}^{\mathbb{S}^{d-1}}(L(\boldsymbol{w}_t; \boldsymbol{z}_t, r_t, y_t), \boldsymbol{w}_*\rangle - \mathbb{E}[\langle\nabla_{\boldsymbol{w}}^{\mathbb{S}^{d-1}}(L(\boldsymbol{w}_t; \boldsymbol{z}_t, r_t, y_t), \boldsymbol{w}^*\rangle] \\
&\quad - \langle\boldsymbol{w}, \nabla_{\boldsymbol{w}}^{\mathbb{S}^{d-1}}(L(\boldsymbol{w}_t; \boldsymbol{z}_t, r_t, y_t)\rangle m + \mathbb{E}[m\langle\boldsymbol{w}, \nabla_{\boldsymbol{w}}^{\mathbb{S}^{d-1}}(L(\boldsymbol{w}_t; \boldsymbol{z}_t, r_t, y_t)\rangle].
\end{aligned}
$$

We then have

$$
\begin{aligned}
&\mathbb{E}_{(y,r,\boldsymbol{z})}[\langle\nabla_{\boldsymbol{w}}^{\mathbb{S}^{d-1}}(\boldsymbol{M}(\boldsymbol{w}_t; \boldsymbol{z}_t, r_t, y_t), \boldsymbol{w}_*\rangle^2] \\
&\leq 2\mathrm{Var}_{(y,r,\boldsymbol{z})}(\nabla_{\boldsymbol{w}}^{\mathbb{S}^{d-1}}(L(\boldsymbol{w}_t; \boldsymbol{z}_t, r_t, y_t), \boldsymbol{w}_*\rangle) + 2\mathrm{Var}_{(y,r,\boldsymbol{z})}(\langle\boldsymbol{w}, \nabla_{\boldsymbol{w}}^{\mathbb{S}^{d-1}}(L(\boldsymbol{w}_t; \boldsymbol{z}_t, r_t, y_t)\rangle m) \\
&\leq 2\mathbb{E}[\langle\boldsymbol{z}_t, \boldsymbol{w}_*\rangle^2(Q_\ell^{(d)})'(\langle\boldsymbol{w}, \boldsymbol{z}_t\rangle)^2\mathcal{T}(y,r)^2] + 2\mathbb{E}[\langle\boldsymbol{z}_t, \boldsymbol{w}_*\rangle^2(Q_\ell^{(d)})'(\langle\boldsymbol{w}, \boldsymbol{z}_t\rangle)^2\mathcal{T}_\ell(y,r)^2] \\
&\leq 2K\left(\frac{4e}{\ell}\right)^{\ell/2}\log(\|\mathcal{T}_\ell\|_4^2)^{\ell/2} := \mathcal{K}_3,
\end{aligned}
$$

where we have used the inequality $(a+b)^2 \leq 2(a^2 + b^2)$, and hypercontractivity of Gegenbauer polynomials (lemma 20).

Conditioned on the event $\{T \leq \tau^-_{2s^*/\sqrt{d}}\}$, and using the inequality (65), we have

$$m_{t+1} \geq \frac{1}{r_t}\left(m_t + 2\eta(1-m_t^2)\beta_{\ell,d}\frac{\ell \cdot (\ell+d-2)}{d-1}m_t^{\ell-1} - \eta\langle\nabla^{\mathbb{S}^{d-1}}\boldsymbol{M}_t, \boldsymbol{w}^*\rangle\right).$$

Using the following bound on $r_t$, which is for all $t \in \mathbb{N}$, we have

$$1/r_t \geq 1 - \eta^2\|\nabla_{\boldsymbol{w}_t}L(\boldsymbol{w}_t; y_t, r_t, \boldsymbol{z}_t)\|^2,$$

and plugging this into previous inequality, we have

$$m_{t+1} \geq m_t + 2\eta(1-m_t^2)\beta_{\ell,d}\frac{\ell \cdot (\ell+d-2)}{d-1}m_t^{\ell-1} - \eta\langle\nabla^{\mathbb{S}^{d-1}}\boldsymbol{M}_t, \boldsymbol{w}^*\rangle - \eta^2|m_t|\,\|\nabla_{\boldsymbol{w}_t}L(\boldsymbol{w}_t; y_t, r_t, \boldsymbol{z}_t)\|^2 - \eta^3\xi_T,$$
(67)

where $\xi_T = \|\nabla_{\boldsymbol{w}}L(\boldsymbol{w}_T; y_T, r_T, \boldsymbol{z}_T)\|^2 \cdot |\langle\nabla_{\boldsymbol{w}}L(\boldsymbol{w}_T; y_T, r_T, \boldsymbol{z}_T), \boldsymbol{w}^*\rangle|^2$.

We use [88, Lemma B.3] to state that with probability at least $1 - \frac{\mathcal{K}_3 t\eta^2}{\lambda^2}$, we have for all $\lambda > 0$,

$$\sup_{t \leq T} \eta\left|\sum_{k=0}^{t-1}\left\langle\nabla^{\mathbb{S}^{d-1}}\boldsymbol{M}_t, \boldsymbol{w}^*\right\rangle\right| \leq \lambda.$$
(68)

We employ [88, Lemma B.6] to state that for all $\lambda > 0$, with probability at least $1 - \frac{\sqrt{\mathcal{K}_1\mathcal{K}_2}Td\eta^3}{\lambda}$, we have

$$\sup_{t \leq T} \eta^3\sum_{k=0}^{t-1}\xi_k \leq \lambda.$$
(69)

We sum the equation (67) to obtain

$$m_T \geq m_0 + 2\eta\frac{\beta_{\ell,d}\ell \cdot (\ell+d-2)}{d-1}\sum_{t=0}^{T-1}(1-m_t^2)m_t^{\ell-1} - \eta\sum_{t=0}^{T-1}\langle\nabla^{\mathbb{S}^{d-1}}\boldsymbol{M}_t, \boldsymbol{w}^*\rangle$$
$$- \eta^2\sum_{t=0}^{T-1}|m_t|\,\|\nabla_{\boldsymbol{w}_t}L(\boldsymbol{w}_t; y_t, r_t, \boldsymbol{z}_t)\|^2 - \sum_{t=0}^{T-1}\eta^3\xi_T,$$

We then use (68),(69) and plug it into (67), and use $\lambda = b/\sqrt{d}$, to state that with probability at least $1 - \frac{\mathcal{K}_3 Td^{1/2}\eta^3}{b^2} - \frac{\sqrt{\mathcal{K}_1\mathcal{K}_2}T\eta^2d^{-1}}{b}$, we have

$$m_T \geq \frac{7m_0}{10} + 2\eta\frac{\beta_{\ell,d}\ell \cdot (\ell+d-2)}{d-1}\sum_{t=0}^{T-1}(1-m_t^2)m_t^{\ell-1} - \eta^2\sum_{t=0}^{T-1}|m_t|\,\|\nabla_{\boldsymbol{w}_t}L(\boldsymbol{w}_t; y_t, r_t, \boldsymbol{z}_t)\|^2.$$
(70)

We now bound the term coming from the projection step in inequality (70). We adapt the proof to take account the dependency on $\|\mathcal{T}_\ell\|_2$.

**Lemma 11.** *For all $\lambda > 0$, if for all $t \leq T, m_t \in [2b/\sqrt{d}, 1/2]$, and $\eta \leq \frac{c\|\xi_{d,\ell}\|_2}{d}$, with probability at least $1 - \frac{\mathcal{K}_2 Td^{1/2}\eta^2}{\lambda} - \exp\left(-\frac{\lambda^2}{2(\beta_{\ell,d}^2 + \mathcal{K}_2 d^2\eta^2)\eta^2 T + 2\lambda\eta(\beta_{\ell,d}+\eta d^{3/2})}\right)$, we have*

$$\eta^2\sum_{t=0}^{T-1}|m_t|\,\|\nabla_{\boldsymbol{w}_t}L(\boldsymbol{w}_t; y_t, r_t, \boldsymbol{z}_t)\|^2 + \eta\sum_{t=0}^{T-1}(1-m_t^2)\beta_{d,\ell}\frac{\ell \cdot (\ell+d-2)}{d-1}m_t^{\ell-1} \leq 2\lambda.$$

*Proof.* The proof is a slight adaptation of [88, Lemma B.4,B.5]. An adaptation of the proof of Lemma B.4 gives us the following. For all $\lambda > 0$, if for all $t \leq T, m_t \in [2b/\sqrt{d}, 1/2]$, and $\eta > 0$, we have

$$\mathbb{P}(\eta\sum_{t=0}^{T-1}D_t \leq -\lambda) \leq \exp\left(-\frac{\lambda^2}{2(\beta_{\ell,d}^2 + \mathcal{K}_2 d^2\eta^2)\eta^2 T + 2\lambda\eta(\beta_{\ell,d}+\eta d^{3/2})}\right).$$

Besides, the adaptation of Lemma B.5 gives us

$$\mathbb{P}\left(\sup_{t \leq T}\eta^2\sum_{t=0}^{T-1}|m_t|\|\nabla_{\boldsymbol{w}}L\|^2 1_{\|\nabla_{\boldsymbol{w}}L\|>d^{3/2}} \geq \lambda\right) \leq \frac{\mathcal{K}_2 Td^{1/2}\eta^2}{\lambda}.$$

Combining the two inequalities, we end up the desired claim. $\square$

We then use $\lambda = b/\sqrt{d}$, and we obtain that with probability at least $1 - p_{\eta,\mathcal{K}_1,\mathcal{K}_2,\mathcal{K}_3}$ where

$$p_{\eta,\mathcal{K}_1,\mathcal{K}_2,\mathcal{K}_3}$$
$$= \frac{\mathcal{K}_2 dT\eta^2}{b} + \exp\left(-\frac{b^2}{2(\beta_{k,d}^2 + \mathcal{K}_2 d^2\eta^2)\eta^2 dT + 2bd^{1/2}\eta(\beta_{k,d} + \eta d^{3/2})}\right) + \frac{\mathcal{K}_3 T d^{1/2}\eta^3}{b^2} - \frac{\sqrt{\mathcal{K}_1\mathcal{K}_2}T\eta^2 d^{-1}}{b},$$

we have

$$m_T \geq \frac{m_0}{2} + \beta_{\ell,d}\frac{\ell \cdot (\ell + d - 2)}{d-1}\eta\sum_{t=0}^{T-1}(1 - m_t^2)m_t^{\ell-1}.$$

Conditioned on the event $\{T \leq \tau_{1/2}^+ \wedge \tau_{2s^*/\sqrt{d}}^-\}$, we have the following inequality

$$m_T \geq \frac{s^*}{\sqrt{d}} + \eta\beta_{\ell,d}\frac{\ell \cdot (\ell + d - 2)}{(d-1)2^{\ell+1}}\sum_{t=0}^{T-1}m_t^{\ell-1}.$$

$\square$

# F   Harmonic tensor unfolding

In this appendix, we analyze the harmonic tensor unfolding estimators (TU-Alg-b) and (TU-Alg), with integer $\ell \geq 3$, and prove the guarantees in Theorem 4. For simplicity, we assume throughout that the transformation $\mathcal{T}_\ell : \mathcal{Y} \times \mathbb{R}_{\geq 0} \to \mathbb{R}$ is bounded, with

$$\|\mathcal{T}_\ell\|_{L^2} = 1, \quad \|\mathcal{T}_\ell\|_\infty \leq \kappa_\ell, \qquad \mathbb{E}_{\nu_d}[\mathcal{T}_\ell(Y,R)Q_\ell(Z)] = \beta_{d,\ell}. \tag{71}$$

Without loss of generality, we take $\beta_{d,\ell} > 0$. We describe in Remark F.1 how to relax this condition.

## F.1   Algorithms and guarantees

Consider first the naive tensor unfolding algorithm. Compute the empirical tensor

$$\hat{\boldsymbol{T}} := \frac{1}{\mathsf{m}} \sum_{i \in [\mathsf{m}]} \mathcal{T}_\ell(y_i, r_i)\mathcal{H}_\ell(\boldsymbol{z}_i) \in (\mathbb{R}^d)^{\otimes \ell},$$

where $\mathcal{H}_\ell(\boldsymbol{z})$ is the degree-$\ell$ harmonic tensor (see Section F.3). Consider the 'unfolded' matrix

$$\textbf{Mat}_{I,J}(\hat{\boldsymbol{T}}) \in \mathbb{R}^{d^I \times d^J}, \quad \text{with } I = J = \ell/2 \text{ if } \ell \text{ even, and } I = \lfloor \ell/2 \rfloor, J = \lfloor \ell/2 \rfloor + 1 \text{ o.w.,}$$

and compute

$$\boldsymbol{s}_1(\textbf{Mat}_{I,J}(\hat{\boldsymbol{T}})) \in \mathbb{R}^{d^{\lfloor \ell/2 \rfloor}},$$

the top left singular vector of $\textbf{Mat}_{I,J}(\hat{\boldsymbol{T}})$. We then estimate $\hat{\boldsymbol{w}}$ via

$$\hat{\boldsymbol{w}} := \textbf{Vec}\left(\boldsymbol{s}_1\left(\textbf{Mat}_{I,J}(\hat{\boldsymbol{T}})\right)\right), \tag{TU-Alg-b}$$

where the mapping $\textbf{Vec} : \mathbb{R}^{d^k} \to \mathbb{S}^{d-1}$ applied to $\boldsymbol{u} \in \mathbb{R}^{d^k}$ returns the top left eigenvector of the folded matrix $\textbf{Mat}_{1,k-1}(\boldsymbol{u}) \in \mathbb{R}^{d \times d^{k-1}}$, that is,

$$\textbf{Vec}(\boldsymbol{u}) = \arg\max_{\boldsymbol{w} \in \mathbb{R}^d} \boldsymbol{w}^\mathsf{T}[\textbf{Mat}_{1,k-1}(\boldsymbol{u})\textbf{Mat}_{1,k-1}(\boldsymbol{u})^\mathsf{T}]\boldsymbol{w}.$$

We first show the following guarantee.

**Theorem 8** (Balanced Harmonic tensor unfolding, $\ell$ even)**.** *Let $\nu_d \in \mathfrak{L}_d$ be a spherical SIM and $\ell$ be an even integer. Consider $\mathcal{T}_\ell$ a transformation satisfying (71). Their exists a universal constant $C_\ell > 0$ such that the following holds. For $\delta > 0$, given $m$ samples $(y_i, \boldsymbol{x}_i) \sim_{iid} \mathbb{P}_{\nu_d, \boldsymbol{w}_*}$ with*

$$m \geq C_\ell \kappa_\ell \frac{d^{\ell/2}}{\beta_{d,\ell}^2}\left[1 + \beta_{d,\ell}\log(d/\delta)\right], \tag{72}$$

*the estimator $\hat{\boldsymbol{w}}$ in (TU-Alg-b) satisfies $|\langle \hat{\boldsymbol{w}}, \boldsymbol{w}_* \rangle| \geq 1/4$ with probability at least $1 - \delta$. Furthermore, $\hat{\boldsymbol{w}}$ can be computed with power iteration in $O_d(md^{\ell/2}\log(d))$ runtime.*

We prove the sample guarantee of Theorem 8 in Section F.4 and the runtime guarantee in Section F.6. Thus, when $\ell$ is even and choosing $\mathcal{T}_\ell$ such that $\beta_{d,\ell} = \Theta(\|\xi_{d,\ell}\|_{L^2})$, the algorithm (TU-Alg-b) achieves almost optimal sample and runtime on $V_{d,\ell}$:

$$\mathsf{m} \asymp \frac{d^{\ell/2}}{\|\xi_{d,\ell}\|_{L^2}^2}\log(d), \qquad \mathsf{T} \asymp \frac{d^\ell}{\|\xi_{d,\ell}\|_{L^2}^2}\log^2(d).$$

When $\|\xi_{d,\ell}\|_{L^2} = O(1/\log(d))$, then this algorithm achieves optimal sample complexity $\mathsf{m} \asymp d^{\ell/2}/\|\xi_{d,\ell}\|_{L^2}^2$. When $\ell$ is odd, however, the algorithm (TU-Alg-b) requires $\mathsf{m} \asymp d^{\lceil \ell/2 \rceil}/\|\xi_{d,\ell}\|_{L^2}^2$ and is suboptimal by a factor $d^{1/2}$. This is due to the covariance structure of the Harmonic tensor $\mathcal{H}_\ell(\boldsymbol{z})$: a similar problem, with same suboptimality, arises for tensor PCA with symmetric noise [67] (if the noise is not symmetric and all entries are independent, then optimal complexity is achieved by the naive tensor unfolding algorithm [48]).

Here, we modify (TU-Alg-b) by removing the diagonal elements. Consider integers $a, b \geq 1$ such that $a < b$ and $a + b = \ell$. Introduce the matrices

$$\hat{M}_1 = \frac{1}{m} \sum_{i \in [m]} \mathcal{T}_\ell(y_i, r_i) \mathbf{Mat}_{a,b}(\mathcal{H}_\ell(z_i)) \in \mathbb{R}^{d^a \times d^b},$$

$$\hat{M}_2 = \frac{1}{m^2} \sum_{i \in [m]} \mathcal{T}_\ell(y_i, r_i)^2 \mathbf{Mat}_{a,b}(\mathcal{H}_\ell(z_i)) \mathbf{Mat}_{b,a}(\mathcal{H}_\ell(z_i)) \in \mathbb{R}^{d^a \times d^a},$$

and

$$\hat{M} = \hat{M}_1 \hat{M}_1^\mathsf{T} - \hat{M}_2 = \frac{1}{m^2} \sum_{i \neq j} \mathcal{T}_\ell(y_i, r_i) \mathcal{T}_\ell(y_j, r_j) \mathbf{Mat}_{a,b}(\mathcal{H}_\ell(z_i)) \mathbf{Mat}_{b,a}(\mathcal{H}_\ell(z_j)) \in \mathbb{R}^{d^a \times d^a}.$$

Note that

$$\mathbb{E}[M] = (1 - m^{-1}) \mathbb{E}[\mathcal{T}_\ell(y, r) \mathbf{Mat}_{a,b}(\mathcal{H}_\ell(z))] \mathbb{E}[\mathcal{T}_\ell(y, r) \mathbf{Mat}_{a,b}(\mathcal{H}_\ell(z))]^\mathsf{T} \approx [w_*^{\otimes a}][w_*^{\otimes a}]^\mathsf{T}.$$

Thus, we define our tensor unfolding estimator to be

$$\hat{w} := \mathbf{Vec}\left(s_1\left(\hat{M}\right)\right), \tag{TU-Alg}$$

where $s_1(\hat{M})$ is the top left eigenvector of $\hat{M}$.

**Theorem 9** (Harmonic tensor unfolding). *Let $\nu_d \in \mathfrak{L}_d$ be a spherical SIM and let $a, b \geq 1$ be two integers such that $a < b$ and $a + b = \ell$. Consider $\mathcal{T}_\ell$ a transformation satisfying (71). There exist universal constants $c_\ell, C_\ell > 0$ that only depend on $\ell$ such that the following holds. Given $m$ samples $(y_i, x_i) \sim_{iid} \mathbb{P}_{\nu_d, w_*}$ with*

$$m \geq C_\ell \kappa_\ell^2 \frac{d^{\ell/2}}{\beta_{d,\ell}^2}, \tag{73}$$

*the estimator $\hat{w}$ in (TU-Alg) satisfies $|\langle \hat{w}, w_* \rangle| \geq 1/4$ with probability at least $1 - e^{-d^{c_\ell}}$. Furthermore, $\hat{w}$ can be computed with power iteration in $O_d(m d^b \log(d))$ runtime.*

We prove the sample guarantee of Theorem 9 in Section F.5 and the runtime guarantee in Section F.6. Thus choosing $\mathcal{T}_\ell$ such that $\beta_{d,\ell} = \Theta(\|\xi_{d,\ell}\|_{L^2})$, the algorithm (TU-Alg) achieves optimal sample complexity

$$\mathsf{m} \asymp \frac{d^{\ell/2}}{\|\xi_{d,\ell}\|_{L^2}^2},$$

for all $1 \leq a < b$. Choosing $a < b$ with $a + b = \ell$ with smallest runtime, we obtain the following learning guarantees:

$\ell$ even: $\quad \mathsf{m} \asymp \dfrac{d^{\ell/2}}{\|\xi_{d,\ell}\|_{L^2}^2} \quad$ and $\quad \mathsf{T} \asymp \dfrac{d^{\ell+1}}{\|\xi_{d,\ell}\|_{L^2}^2} \log(d) \quad$ by taking $a = \ell/2 - 1$ and $b = \ell/2 + 1$,

$\ell$ odd: $\quad \mathsf{m} \asymp \dfrac{d^{\ell/2}}{\|\xi_{d,\ell}\|_{L^2}^2} \quad$ and $\quad \mathsf{T} \asymp \dfrac{d^{\ell+1/2}}{\|\xi_{d,\ell}\|_{L^2}^2} \log(d) \quad$ by taking $a = \lfloor \ell/2 \rfloor$ and $b = \lceil \ell/2 \rceil$.

## F.2 Notations

Below, we introduce some notations from tensor calculus that will be useful throughout our proofs. Let $A, B \in (\mathbb{R}^d)^{\otimes \ell}$ be two $\ell$-tensors. We define the inner-product

$$\langle A, B \rangle = \sum_{i_1, \ldots, i_\ell \in [d]} A_{i_1, \ldots, i_\ell} B_{i_1, \ldots, i_\ell}. \tag{74}$$

In particular, the Frobenius norm of the tensor is $\|A\|_F = \langle A, A \rangle^{1/2}$. For $A \in (\mathbb{R}^d)^{\otimes \ell}$ and $B \in (\mathbb{R}^d)^{\otimes k}$ with $k \leq \ell$, we define the contraction $A[B] \in (\mathbb{R}^d)^{\otimes (\ell - k)}$ to be the $(\ell - k)$-tensor with entries given by

$$A[B]_{i_1, \ldots, i_{\ell-k}} := \sum_{i_{\ell-k+1}, \ldots, i_\ell \in [d]} A_{i_1, \ldots, i_\ell} B_{i_{\ell-k+1}, \ldots, i_\ell}. \tag{75}$$

In particular, if $k = \ell$, we have $\boldsymbol{A}[\boldsymbol{B}] = \boldsymbol{B}[\boldsymbol{A}] = \langle \boldsymbol{A}, \boldsymbol{B} \rangle$. We further introduce partial contraction $\boldsymbol{A} \otimes_r \boldsymbol{B}$ of $\boldsymbol{A} \in (\mathbb{R}^d)^{\otimes \ell}$ and $\boldsymbol{B} \in (\mathbb{R}^d)^{\otimes k}$, with $r \leq \min(\ell, k)$, given by

$$(\boldsymbol{A} \otimes_r \boldsymbol{B})_{i_1,\ldots,i_{\ell-r},j_{r+1},\ldots,j_k} = \sum_{s_1,\ldots,s_r \in [d]} \boldsymbol{A}_{i_1,\ldots,i_{\ell-r},s_1,\ldots,s_r} \boldsymbol{B}_{s_1,\ldots,s_r,j_{r+1},\ldots,j_k}. \tag{76}$$

Given a permutation $\pi \in \mathfrak{S}_\ell$ and $\boldsymbol{A} \in (\mathbb{R}^d)^{\otimes \ell}$, we define $\pi(\boldsymbol{A})$ to be the tensor obtained by permuting the coordinates of $\boldsymbol{A}$ with permutation $\pi$, that is

$$\pi(\boldsymbol{A})_{i_1,\ldots,i_\ell} = \boldsymbol{A}_{i_{\pi(1)},\ldots,i_{\pi(\ell)}}. \tag{77}$$

We define the symmetrization operator $\mathrm{Sym} : (\mathbb{R}^d)^{\otimes \ell} \to (\mathbb{R}^d)^{\otimes \ell}$ such that for each tensor $\boldsymbol{A} \in (\mathbb{R}^d)^{\otimes \ell}$, it outputs its symmetrized version

$$\mathrm{Sym}(\boldsymbol{A}) = \frac{1}{\ell!} \sum_{\pi \in \mathfrak{S}_\ell} \pi(\boldsymbol{A}). \tag{78}$$

We denote $\mathrm{Sym}((\mathbb{R}^d)^{\otimes \ell})$ the image of this operator, that is, the space of symmetric $\ell$-tensors.

We denote the unfolded matrix of the tensor $\boldsymbol{T} \in (\mathbb{R}^d)^{\otimes \ell}$ as $\mathbf{Mat}_{q,\ell-q}(\boldsymbol{T}) \in \mathbb{R}^{d^q} \times \mathbb{R}^{d^{\ell-q}}$ with entries given by

$$(\mathbf{Mat}_{q,\ell-q}(\boldsymbol{T}))_{(i_1,\ldots,i_q),(j_1,\ldots,j_{\ell-q})} = \boldsymbol{T}_{i_1,\ldots,i_\ell,j_1,\ldots,j_\ell},$$

where we identify $(i_1, \ldots, i_q)$ with $1 + \sum_{k=1}^{q} (i_k - 1) d^{k-1}$, and $(j_1, \ldots, j_{\ell-q})$ with $1 + \sum_{k=1}^{\ell-q} (j_k - 1) d^{k-1}$. For clarity, we will drop the dependency on $q$ in the proofs, since the unfolding parameter will be made clear. Similarly, with a slight overloading, we denote $\mathbf{Mat}_{q,\ell-q} : \mathbb{R}^{d^\ell} \to \mathbb{R}^{d^q \times d^{\ell-q}}$ a folding operation that takes a vector $\boldsymbol{u} \in \mathbb{R}^{d^\ell}$ and associate the matrix

$$(\mathbf{Mat}_{q,\ell-q}(\boldsymbol{u}))_{(i_1,\ldots,i_q),(j_1,\ldots,j_{\ell-q})} = u_{i_1,\ldots,i_q,j_1,\ldots,j_{\ell-q}},$$

where we identify the index $(i_1, \ldots, i_\ell)$ with $1 + \sum_{k=1}^{\ell} (i_k - 1) d^{k-1}$.

Throughout the proofs, we denote $C$ an absolute constant and $C_\ell$ a constant that only depends on $\ell$. In particular, the value of these constants are allowed to change from line to line.

### F.3 Harmonic tensors and their properties

We start by defining harmonic tensors and present some basic properties about them.

**Definition 2** (Harmonic Tensors). *For every $\ell > 0$, we define $\mathcal{H}_\ell : \mathbb{S}^{d-1} \to \mathrm{Sym}((\mathbb{R}^d)^{\otimes \ell})$ the unique symmetric tensor such that for all $\boldsymbol{z}, \boldsymbol{w} \in \mathbb{S}^{d-1}$, we have*

$$Q_\ell^{(d)}(\langle \boldsymbol{z}, \boldsymbol{w} \rangle) = \langle \mathcal{H}_\ell(\boldsymbol{z}), \boldsymbol{w}^{\otimes \ell} \rangle, \tag{79}$$

*where $Q_\ell^{(d)}$ is the degree-$\ell$ (normalized) Gegenbauer polynomial as defined in Appendix B.*

In words, harmonic tensors can be seen as the projection of $\boldsymbol{z}^{\otimes \ell}$ into the space of traceless symmetric tensors. Note that the uniqueness of (79) follows simply by stating that $\langle \mathcal{H}_\ell(\boldsymbol{z}) - \mathcal{H}'_\ell(\boldsymbol{z}), \boldsymbol{w}^{\otimes \ell} \rangle$ is a degree-$\ell$ polynomial in $\boldsymbol{w}$ that is identically zero, and therefore $\mathcal{H}_\ell(\boldsymbol{z}) = \mathcal{H}'_\ell(\boldsymbol{z})$, where we use that $\mathcal{H}_\ell(\boldsymbol{z})$ is assumed to be symmetric.

Further note that these tensors are equivariant with respect to rotations: let $\boldsymbol{O} \in \mathcal{O}_d$, then $\mathcal{H}_\ell(\boldsymbol{O}\boldsymbol{z}) = \boldsymbol{O}^{\otimes \ell}[\mathcal{H}_\ell(\boldsymbol{z})]$, where the contraction is along one coordinate for each $\boldsymbol{O}$, that is

$$\mathcal{H}_\ell(\boldsymbol{O}\boldsymbol{z})_{i_1,\ldots,i_\ell} = \sum_{j_1,\ldots,j_\ell \in [d]} \left( \prod_{s \in [\ell]} O_{i_s j_s} \right) \mathcal{H}_\ell(\boldsymbol{z})_{j_1,\ldots,j_\ell}.$$

(This follows simply by noting that $Q_k(\langle \boldsymbol{w}, \boldsymbol{O}\boldsymbol{z} \rangle) = Q_k(\langle \boldsymbol{O}^\mathsf{T}\boldsymbol{w}, \boldsymbol{z} \rangle)$.)

Recall the zonal property of Gegenbauer polynomials:

**Lemma 12** (Zonal property of Gegenbauer polynomials [25])**.** *Let $f \in L^2(\mathbb{S}^{d-1})$. Consider the projection $\mathsf{P}_{V_{d,\ell}}$ of $f$ onto the subspace $V_{d,\ell}$ of degree-$\ell$ spherical harmonics. This projection can be written as*

$$\mathsf{P}_{V_{d,\ell}} f(\boldsymbol{x}) = \sqrt{n_{d,\ell}} \cdot \mathbb{E}_{\boldsymbol{z} \sim \tau_d}[f(\boldsymbol{z}) Q_\ell^{(d)}(\langle \boldsymbol{z}, \boldsymbol{x} \rangle)]. \tag{80}$$

The following lemma follows directly from this property:

**Lemma 13** (Reproducing property of harmonic tensors)**.** *Let $\ell, k \in \mathbb{N}$. We have the identity*

$$\mathbb{E}_{\boldsymbol{z} \sim \tau_d}\left[ Q_\ell^{(d)}(\langle \boldsymbol{w}, \boldsymbol{z} \rangle) \mathcal{H}_\ell(\boldsymbol{z}) \right] = \frac{\delta_{\ell k}}{\sqrt{n_{d,\ell}}} \mathcal{H}_\ell(\boldsymbol{w}). \tag{81}$$

This property will be key to our analysis of our tensor unfolding algorithm. Indeed, recalling the definition $\beta_{d,\ell} = \mathbb{E}_{\nu_d}[\mathcal{T}_\ell(Y, R) Q_\ell(Z)]$, we have

$$\mathbb{E}[\mathcal{T}_\ell(y, r) \mathcal{H}_\ell(\boldsymbol{z})] = \frac{\beta_{d,\ell}}{\sqrt{n_{d,\ell}}} \mathcal{H}_\ell(\boldsymbol{w}_*). \tag{82}$$

We further list some useful properties of harmonic tensors below. In particular, the last property states that the principal component of $\mathcal{H}_\ell(\boldsymbol{w}_*)$ is $\Theta_d(d^{\ell/2}) \cdot \boldsymbol{w}_*^{\otimes \ell}$, that is the principal component of the expectation of our empirical tensor is $\Theta_d(1) \cdot \boldsymbol{w}_*^{\otimes \ell}$.

**Proposition 5** (Properties of harmonic tensors)**.** *Let $\ell \in \mathbb{N}$ and $\mathcal{H}_\ell(\boldsymbol{z})$ the harmonic tensor from Definition 2.*

*(i) We have the following explicit formula:*

$$\mathcal{H}_\ell(\boldsymbol{z}) = \sum_{j=0}^{\lfloor \ell/2 \rfloor} c_{\ell,j} \, \mathrm{Sym}(\boldsymbol{z}^{\otimes(\ell-2j)} \otimes \mathbf{I}_d^{\otimes j}), \tag{83}$$

*where*

$$c_{\ell,j} = (-1)^j 2^{\ell-2j} \frac{\ell!}{j!(\ell-2j)!} \frac{(d/2-1)_{\ell-j}}{(d-2)_\ell} \sqrt{n_{d,\ell}},$$

*with $(a)_p = a(a+1)\cdots(a+p-1)$ the (rising) Pochhammer symbol. In particular, we have $c_{\ell,j} = \Theta_d(d^{\ell/2-j})$.*

*(ii) Conversely, we have the identity*

$$\boldsymbol{z}^{\otimes \ell} = \sum_{j=0}^{\lfloor \ell/2 \rfloor} b_{\ell,j} \, \mathrm{Sym}(\mathcal{H}_{\ell-2j}(\boldsymbol{z}) \otimes \mathbf{I}_d^{\otimes j}), \tag{84}$$

*where*

$$b_{\ell,j} = 2^{-\ell} \frac{\ell!}{j!(\ell-2j)!} \frac{(d/2+\ell-2j-1)(d-2)_{\ell-2j}}{(d/2-1)(d/2)_{\ell-j}} \frac{1}{\sqrt{n_{d,\ell-2j}}}.$$

*In particular, we have $b_{\ell,j} = \Theta_d(d^{-\ell/2})$.*

*(iii) The harmonic tensors satisfy the following recurrence relation:*

$$\widetilde{\mathcal{H}}_{\ell+1}(\boldsymbol{z}) = a_{d,\ell}^{(1)} \, \mathrm{Sym}(\widetilde{\mathcal{H}}_\ell(\boldsymbol{z}) \otimes \boldsymbol{z}) - a_{d,\ell}^{(2)} \, \mathrm{Sym}(\widetilde{\mathcal{H}}_{\ell-1}(\boldsymbol{z}) \otimes \mathbf{I}_d), \tag{85}$$

*where we denoted $\widetilde{\mathcal{H}}_\ell(\boldsymbol{z}) := \mathcal{H}_\ell(\boldsymbol{z})/\sqrt{n_{d,\ell}}$ and*

$$a_{d,\ell}^{(1)} = \frac{2\ell+d-2}{d-2+\ell}, \qquad a_{d,\ell}^{(2)} = \frac{\ell}{d+\ell-2}.$$

*(iv) The leading principal component of $\mathcal{H}_\ell(\boldsymbol{z})$ satisfies*

$$\|\mathcal{H}_\ell(\boldsymbol{z}) - c_{\ell,0} \boldsymbol{z}^{\otimes \ell}\|_F = O_d(d^{\ell/2-1/2}), \tag{86}$$

*where we recall that $\|c_{\ell,0} \boldsymbol{z}^{\otimes \ell}\|_F = c_{\ell,0} = \Theta_d(d^{\ell/2})$.*

*Proof.* The identities in parts (i), (ii) and (iii) simply follows from standard identities on Gegenbauer polynomials. To prove part (iv), using identity (83), we have

$$\|\mathcal{H}_\ell(\boldsymbol{z}) - c_{\ell,0}\boldsymbol{z}^{\otimes\ell}\|_F \le \sum_{j=1}^{\lfloor\ell/2\rfloor} |c_{\ell,j}| \|\mathrm{Sym}(\boldsymbol{z}^{\otimes(\ell-2j)} \otimes \mathbf{I}_d^{\otimes j})\|_F$$

$$\le \sum_{j=1}^{\lfloor\ell/2\rfloor} |c_{\ell,j}| \|\boldsymbol{z}\|_2^{\ell-2j} \|\mathbf{I}_d\|_F^j = \sum_{j=1}^{\lfloor\ell/2\rfloor} |c_{\ell,j}| d^{j/2} = \Theta_d(d^{\ell/2-1/2}),$$

where we used that $|c_{\ell,j}| = \Theta_d(d^{\ell/2-j})$. $\square$

Additionally, it is interesting to introduce the following tensor:

$$\Sigma_\ell^{(2)} = \sqrt{n_{d,\ell}}\, \mathbb{E}_{\boldsymbol{z}\sim\tau_d}\Big[\mathcal{H}_\ell(\boldsymbol{z}) \otimes \mathcal{H}_\ell(\boldsymbol{z})\Big] \in (\mathbb{R}^d)^{\otimes 2\ell}. \qquad (87)$$

In particular, by the reproducing property, we have for all $\boldsymbol{u}, \boldsymbol{w} \in \mathbb{S}^{d-1}$,

$$\mathbb{E}[Q_\ell(\langle\boldsymbol{u}, \boldsymbol{z}\rangle)Q_\ell(\langle\boldsymbol{w}, \boldsymbol{z}\rangle)] = \Big\langle \mathbb{E}[\mathcal{H}_\ell(\boldsymbol{z}) \otimes \mathcal{H}_\ell(\boldsymbol{z})], \boldsymbol{w}^{\otimes\ell} \otimes \boldsymbol{u}^{\otimes\ell}\Big\rangle$$

$$= \frac{1}{\sqrt{n_{d,\ell}}}\langle\Sigma_\ell^{(2)}, \boldsymbol{w}^{\otimes\ell} \otimes \boldsymbol{u}^{\otimes\ell}\rangle = \frac{1}{\sqrt{n_{d,\ell}}}Q_\ell(\langle\boldsymbol{u}, \boldsymbol{w}\rangle),$$

and we can write

$$\langle\mathcal{H}_\ell(\boldsymbol{z}), \boldsymbol{w}^{\otimes\ell}\rangle = \langle\Sigma_\ell^{(2)}, \boldsymbol{z}^{\otimes\ell} \otimes \boldsymbol{w}^{\otimes\ell}\rangle.$$

Note that $\Sigma_\ell^{(2)}$ is only partially symmetric. Using Proposition 5.(i), we can decompose this tensor explicitly into

$$\Sigma_\ell^{(2)} = \sum_{j=0}^{\lfloor\ell/2\rfloor} c_{\ell,j} \cdot \mathrm{Sym}_A\left(\mathbf{I}_d^{\otimes(\ell-2j)} \otimes (\mathbf{I}_d \otimes \mathbf{I}_d)^{\otimes j}\right), \qquad (88)$$

where we introduced an alternate symmetrizer $\mathrm{Sym}_A$ such that

$$\mathrm{Sym}_A\left(\mathbf{I}_d^{\otimes(\ell-2j)} \otimes (\mathbf{I}_d \otimes \mathbf{I}_d)^{\otimes j}\right)$$

$$= \sum_{r_1,\ldots,r_{\ell-2j}\in[d]} \mathrm{Sym}(\boldsymbol{e}_{r_1} \otimes \ldots \otimes \boldsymbol{e}_{r_{\ell-2j}} \otimes \mathbf{I}_d^{\otimes j}) \otimes \mathrm{Sym}(\boldsymbol{e}_{r_1} \otimes \ldots \otimes \boldsymbol{e}_{r_{\ell-2j}} \otimes \mathbf{I}_d^{\otimes j}),$$

with $(\boldsymbol{e}_s)_{s\in[d]}$ the canonical basis in $\mathbb{R}^d$.

## F.4   Proof of Theorem 8

*Proof of Theorem 8.* Denote $\ell = 2p$ and introduce the matrices

$$\boldsymbol{Z}_i = \mathcal{T}_\ell(y_i, r_i)\boldsymbol{H}(\boldsymbol{z}_i) \in \mathbb{R}^{d^p \times d^p}, \qquad \text{where} \quad \boldsymbol{H}(\boldsymbol{z}_i) = \mathbf{Mat}_{p,p}(\mathcal{H}_\ell(\boldsymbol{z}_i)) \in \mathbb{R}^{d^p \times d^p},$$

and the centered matrices

$$\overline{\boldsymbol{Z}}_i = \boldsymbol{Z}_i - \boldsymbol{E}, \qquad \text{where} \ \ \boldsymbol{E} = \mathbb{E}[\boldsymbol{Z}_i] = \mathbb{E}[\mathcal{T}_\ell(y, r)\boldsymbol{H}(\boldsymbol{z})].$$

By the reproducing property (82) and Proposition 5.(iv), we have

$$\boldsymbol{E} = \beta_{d,\ell}\frac{c_{\ell,0}}{\sqrt{n_{d,\ell}}}[\boldsymbol{w}_*^{\otimes p}][\boldsymbol{w}_*^{\otimes p}]^\mathsf{T} + \beta_{d,\ell}\boldsymbol{\Delta}_E, \qquad (89)$$

where $\|\boldsymbol{\Delta}_E\|_{\mathrm{op}} \le C_\ell d^{-1/2}$ and $c_{\ell,0}/\sqrt{n_{d,\ell}} = \Theta_d(1)$.

We consider the symmetric matrix

$$\hat{\boldsymbol{M}} = \frac{1}{m}\sum_{i\in[m]} \boldsymbol{Z}_i,$$

and bound $\|\hat{M} - E\|_{\mathrm{op}}$ using Lemma 25. First, note that by applying Lemma 14 with $A = u \otimes v$, we have

$$
\begin{aligned}
\sigma_*(\hat{M} - E)^2 &= \sup_{u,v \in \mathbb{S}^{d^p-1}} \mathbb{E}[\langle u, (\hat{M} - E)v \rangle^2] \\
&\leq \frac{\kappa_\ell^2}{m} \sup_{u,v \in \mathbb{S}^{d^p-1}} \mathbb{E}[\langle u, H(z_i)v \rangle^2] \leq C_\ell \frac{\kappa_\ell^2}{m}.
\end{aligned}
\tag{90}
$$

Further, note that

$$
\sigma(\hat{M} - E)^2 = \|\mathbb{E}[(\hat{M} - E)(\hat{M} - E)^\mathsf{T}]\|_{\mathrm{op}} \leq d^p \sigma_*(\hat{M} - E)^2 \leq C_\ell \kappa_\ell^2 \frac{d^p}{m}.
\tag{91}
$$

Using that $\|H(z_i)\|_F \leq C_\ell d^{\ell/2}$ deterministically and combining the above displays into Lemma 25, we obtain with probability at least $1 - \delta$ that

$$
\|\hat{M} - E\|_{\mathrm{op}} \leq C_\ell \kappa_\ell \sqrt{\frac{d^p}{m}} \left[ 1 + \left( \frac{d^p}{m} \log^2(d/\delta) \right)^{1/2} \vee 1 \right].
$$

where we assumed without loss of generality that $\delta \geq e^{-d}$ to avoid carrying additional terms.

From Eq. (89) and by Davis-Kahan theorem, the leading eigenvector $s$ of $\hat{M}$ satisfy

$$
|\langle s, w_*^{\otimes p} \rangle| \geq 1 - \eta,
$$

with probability at least $1 - \delta$ when

$$
m \geq C_\ell \frac{\kappa_\ell}{\eta^2} \frac{d^{\ell/2}}{\beta_{d,\ell}^2} \left[ 1 + \beta_{d,\ell} \log(d/\delta) \right].
$$

The estimator $\hat{w} = \mathbf{Vec}(s)$ is obtained by taking the top eigenvector of

$$
\begin{aligned}
&\mathbf{Mat}_{1,p-1}(s)\mathbf{Mat}_{1,p-1}(s)^\mathsf{T} \\
&= w_* w_*^\mathsf{T} + \mathbf{Mat}_{1,p-1}(s - w_*^{\otimes p})\mathbf{Mat}_{1,p-1}(s)^\mathsf{T} + \mathbf{Mat}_{1,p-1}(w_*^{\otimes p})\mathbf{Mat}_{1,p-1}(s - w_*^{\otimes p})^\mathsf{T},
\end{aligned}
$$

so that

$$
\|\mathbf{Mat}_{1,p-1}(s)\mathbf{Mat}_{1,p-1}(s)^\mathsf{T} - w_* w_*^\mathsf{T}\|_{\mathrm{op}} \leq 2\|s - w_*^{\otimes p}\|_F \leq 2\sqrt{\eta}.
$$

Thus, taking $\eta$ constant small enough, we obtain $|\langle \hat{w}, w_* \rangle| \geq 1/4$ by Davis-Kahan theorem. $\qquad \square$

**Lemma 14.** *There exists a constant $C_\ell > 0$ such that for all $A \in (\mathbb{R}^d)^{\otimes \ell}$, we have*

$$
\mathbb{E}_z \left[ \langle \mathcal{H}_\ell(z), A \rangle^2 \right] \leq C_\ell \|A\|_F^2.
$$

*Proof.* Using the identity for the quadratic tensor (88), we decompose

$$
\begin{aligned}
\mathbb{E}_z \left[ \langle \mathcal{H}_\ell(z), A \rangle^2 \right] &= \left\langle \mathbb{E}_z[\mathcal{H}_\ell(z) \otimes \mathcal{H}_\ell(z)], A \otimes A \right\rangle \\
&= \frac{1}{\sqrt{n_{d,\ell}}} \langle \Sigma_\ell^{(2)}, A \otimes A \rangle \\
&= \frac{1}{\sqrt{n_{d,\ell}}} \sum_{j=0}^{\lfloor \ell/2 \rfloor} c_{\ell,j} \frac{1}{(\ell!)^2} \sum_{\pi,\pi' \in \mathfrak{S}_\ell} \langle \pi(A)[\mathbf{I}_d^{\otimes j}], \pi'(A)[\mathbf{I}_d^{\otimes j}] \rangle \\
&\leq \sum_{j=0}^{\lfloor \ell/2 \rfloor} \frac{|c_{\ell,j}|}{\sqrt{n_{d,\ell}}} d^j \|A\|_F^2 \\
&\leq C_\ell \|A\|_F^2,
\end{aligned}
$$

where we used that $\|A[\mathbf{I}_d^{\otimes j}]\|_F^2 \leq d^j \|A\|_F^2$ and $|c_{\ell,j}| \leq C_\ell d^{\ell/2-j}$. $\qquad \square$

**Remark F.1.** To relax the boundedness assumption in the above proof, the only changes are in the bound on $\sigma_*(\hat{M} - E)$ and $\|Z_i\|_{\mathrm{op}}$. First, note that for all $\eta > 1$, we have by Hölder's inequality

$$
\begin{aligned}
\sigma_*(\hat{M} - E) &\leq \frac{1}{m} \sup_{u \in \mathbb{S}^{d^p-1}} \mathbb{E}[\mathcal{T}_\ell(y,r)^2 \langle u, H(z)v\rangle^2] \\
&\leq \frac{1}{m} \mathbb{E}[\mathcal{T}_\ell(y,r)^{2+\eta}]^{1/(1+\eta/2)} \sup_{u \in \mathbb{S}^{d^p-1}} \mathbb{E}[\langle u, H(z)v\rangle^{2+4/\eta}]^{1/(1+2/\varepsilon)} \\
&\leq \frac{1}{m} \|\mathcal{T}_\ell\|_{L^{2+\eta}}^2 (1 + 2/\eta)^\ell \sup_{u \in \mathbb{S}^{d^p-1}} \mathbb{E}[\langle u, H(z)v\rangle^2] \\
&\leq C_\ell (1 + 2/\eta)^\ell \frac{\|\mathcal{T}_\ell\|_{L^{2+\eta}}^2}{m}.
\end{aligned}
$$

where we used hypercontractivity of degree-$\ell$ spherical harmonics. Thus as long as $\|\mathcal{T}_\ell\|_{L^{2+\eta}}^2 = \Theta_d(1)$ for some $\eta = \Theta_d(1)$, the bound does not change. Furthermore, for all integer $q$

$$
\mathbb{P}(\max_{i \in [m]} \|Z_i\|_{\mathrm{op}} \geq R) \leq \mathbb{P}(\max_{i \in [m]} |\mathcal{T}_\ell(y_i, r_i)| \geq c_\ell R m/d^p) \leq \left( C_\ell \frac{d^p}{Rm} m^{1/q} \mathbb{E}[\mathcal{T}_\ell^q]^{1/q} \right)^q.
$$

If we assume that $\|\mathcal{T}_\ell\|_{L^q} \leq q^C$ for all $q$, we can set $q = \log(m)$ and obtain essentially the same guarantees as above. For the second algorithm (TU-Alg), we can be less careful and simply set $q = C_\ell$ and only assume $\|\mathcal{T}_\ell\|_{L^q} \leq C_\ell$.

## F.5 Proof of Theorem 9

*Proof of Theorem 9.* **Step 1: Decomposing $\|\hat{M} - \mathbb{E}[\hat{M}]\|_{\mathrm{op}}$.** Recall that we fix $\ell = a + b$ with positive integers $a \leq b$. Introduce the matrices

$$
Z_i = \mathcal{T}_\ell(y_i, r_i) H(z_i) \in \mathbb{R}^{d^a \times d^b}, \qquad \text{where} \quad H(z_i) = \mathbf{Mat}_{a,b}(\mathcal{H}_\ell(z_i)) \in \mathbb{R}^{d^a \times d^b},
$$

and the centered matrices

$$
\overline{Z}_i = Z_i - E, \qquad \text{where} \quad E = \mathbb{E}[Z_i] = \mathbb{E}[\mathcal{T}_\ell(y,r)H(z)].
$$

Recall that by the reproducing property (82) and Proposition 5.(iv), we have

$$
E = \beta_{d,\ell} \frac{c_{\ell,0}}{\sqrt{n_{d,\ell}}} [w_*^{\otimes a}][w_*^{\otimes b}]^\mathsf{T} + \beta_{d,\ell} \Delta_E, \tag{92}
$$

where $\|\Delta_E\|_{\mathrm{op}} \leq C_\ell d^{-1/2}$ and $c_{\ell,0}/\sqrt{n_{d,\ell}} = \Theta_d(1)$. For convenience, we consider a slight change of normalization and define

$$
\hat{M} = \frac{1}{m(m-1)} \sum_{i \neq j} Z_i Z_j^\mathsf{T},
$$

such that

$$
\mathbb{E}[\hat{M}] = E E^\mathsf{T} = \beta_{d,\ell}^2 \frac{c_{\ell,0}^2}{n_{d,\ell}} [w_*^{\otimes a}][w_*^{\otimes a}]^\mathsf{T} + \beta_{d,\ell}^2 \Delta_0, \qquad \|\Delta_0\|_{\mathrm{op}} \leq C_\ell d^{-1/2}. \tag{93}
$$

By a standard decoupling argument (see [80, 33] Chapter 6.1), there exists an absolute constant $C > 0$ such that

$$
\mathbb{P}\left( \left\| \hat{M} - E E^\mathsf{T} \right\|_{\mathrm{op}} \geq t \right) \leq C \mathbb{P}\left( \left\| \tilde{M} - E E^\mathsf{T} \right\|_{\mathrm{op}} \geq t \right),
$$

where we defined

$$
\tilde{M} = \frac{1}{m(m-1)} \sum_{i \neq j} Z_i \tilde{Z}_j^\mathsf{T},
$$

with $\tilde{Z}_j = \mathcal{T}_\ell(\tilde{y}_j, \tilde{r}_j) H(\tilde{z}_j)$ and $(\tilde{y}_j, \tilde{r}_j, \tilde{z}_j)_{j \in [m]}$ are iid independent of $(y_i, r_i, z_i)_{i \in [m]}$. Thus, it is enough to study the concentration of $\tilde{M}$:

$$
\tilde{M} = \frac{1}{m} \sum_{i \in [m]} Z_i B_i^\mathsf{T}, \qquad \text{where} \quad B_i = \frac{1}{m-1} \sum_{j \neq i} \tilde{Z}_j.
$$

Thus we decompose $\tilde{\boldsymbol{M}} - \boldsymbol{E}\boldsymbol{E}^\mathsf{T} = \boldsymbol{\Delta}_1 + \boldsymbol{\Delta}_2$ where

$$\boldsymbol{\Delta}_1 = \frac{1}{m} \sum_{i \in [m]} \overline{\boldsymbol{Z}}_i \boldsymbol{B}_i^\mathsf{T},$$

$$\boldsymbol{\Delta}_2 = \frac{1}{m} \sum_{i \in [m]} \boldsymbol{E}\{\boldsymbol{B}_i - \boldsymbol{E}\}^\mathsf{T} = \frac{1}{m} \sum_{i \in [m]} \boldsymbol{E}(\tilde{\boldsymbol{Z}}_i - \boldsymbol{E})^\mathsf{T}.$$

(94)

We bound the operator norm of these two matrices below.

**Step 2: Bound on $\|\boldsymbol{B}_i\|_{\mathrm{op}}$.** For all $i \in [m]$, we have

$$\boldsymbol{B}_i = \tilde{\boldsymbol{S}} + \frac{m}{m-1}\boldsymbol{E} - \frac{1}{m-1}\tilde{\boldsymbol{Z}}_i, \qquad \text{where } \tilde{\boldsymbol{S}} = \frac{1}{m-1} \sum_{i \in [m]} \tilde{\boldsymbol{Z}}_j - \boldsymbol{E}.$$

Note that $\|\boldsymbol{Z}_i\|_{\mathrm{op}} \le \|\mathcal{T}_\ell\|_\infty \|\boldsymbol{H}(\tilde{\boldsymbol{z}}_i)\|_F \le C_\ell \kappa_\ell d^{\ell/2}$, so that

$$\|\boldsymbol{B}_i\|_{\mathrm{op}} \le \|\tilde{\boldsymbol{S}}\|_{\mathrm{op}} + C_\ell + C_\ell \kappa_\ell \frac{d^{\ell/2}}{m}.$$

(95)

We use Lemma 25 to bound $\|\tilde{\boldsymbol{S}}\|_{\mathrm{op}}$. Applying Lemma 14 with $\boldsymbol{A} = \boldsymbol{u} \otimes \boldsymbol{v}$,

$$\sigma_*(\tilde{\boldsymbol{S}})^2 \le \frac{C}{m} \sup_{\boldsymbol{u} \in \mathbb{S}^{d^a-1}, \boldsymbol{v} \in \mathbb{S}^{d^b-1}} \mathbb{E}[\langle \boldsymbol{u}, \tilde{\boldsymbol{Z}}_j \boldsymbol{v} \rangle^2]$$

$$\le \frac{C}{m} \kappa_\ell^2 \sup_{\boldsymbol{u} \in \mathbb{S}^{d^a-1}, \boldsymbol{v} \in \mathbb{S}^{d^b-1}} \mathbb{E}[\langle \mathcal{H}_\ell(\boldsymbol{z}), \boldsymbol{u} \otimes \boldsymbol{v} \rangle^2]$$

$$\le C_\ell \frac{\kappa_\ell^2}{m}.$$

To bound $\sigma(\tilde{\boldsymbol{S}})$, we simply use

$$\|\mathbb{E}[\tilde{\boldsymbol{S}}\tilde{\boldsymbol{S}}^\mathsf{T}]\|_{\mathrm{op}} = \sup_{u \in \mathbb{S}^{d^a-1}} \mathbb{E}[\boldsymbol{u}^\mathsf{T}\tilde{\boldsymbol{S}}\tilde{\boldsymbol{S}}^\mathsf{T}\boldsymbol{u}] \le d^b \sup_{\boldsymbol{u} \in \mathbb{S}^{d^a-1}, \boldsymbol{v} \in \mathbb{S}^{d^b-1}} \mathbb{E}[\langle \boldsymbol{u}, \tilde{\boldsymbol{S}}\boldsymbol{v} \rangle^2] = d^b \sigma_*(\tilde{\boldsymbol{S}})^2,$$

(96)

so that

$$\sigma(\tilde{\boldsymbol{S}})^2 \le \max(d^a, d^b)\sigma_*(\tilde{\boldsymbol{S}})^2 \le C_\ell \kappa_\ell^2 \frac{d^b}{m}.$$

Combining the above displays into Lemma 25, we get with probability at least $1 - de^{-t}$,

$$\|\tilde{\boldsymbol{S}}\|_{\mathrm{op}} \le C_\ell \kappa_\ell \left[ \frac{d^{b/2}}{m^{1/2}} + \frac{t^{1/2}}{m^{1/2}} + \frac{d^{b/3+\ell/6}t^{2/3}}{m^{2/3}} + \frac{d^{\ell/2}}{m}t \right].$$

We deduce that with probability at least $1 - \delta/3$,

$$\sup_{i \in [m]} \|\boldsymbol{B}_i\|_{\mathrm{op}} \le C_\ell \kappa_\ell \sqrt{\frac{d^b}{m}} \left[ 1 + \left( \frac{d^a \log^4(d/\delta)}{m} \right)^{1/2} \vee 1 \right],$$

(97)

where we assumed without loss of generality that $\delta > e^{-d}$ to avoid carrying extra terms.

**Step 3: Bound on $\|\boldsymbol{\Delta}_1\|_{\mathrm{op}}$.** Let's bound $\boldsymbol{\Delta}_1$ conditioned on $(\tilde{y}_i, \tilde{r}_i, \tilde{\boldsymbol{z}}_i)_{i \in [m]}$. Using Lemma 14 with $\boldsymbol{A} = \boldsymbol{B}_i^\mathsf{T}\boldsymbol{v} \otimes \boldsymbol{u}$,

$$\sigma_*(\boldsymbol{\Delta}_1)^2 \le \frac{1}{m^2} \sum_{i \in [m]} \sup_{\boldsymbol{u}, \boldsymbol{v} \in \mathbb{S}^{d^a-1}} \mathbb{E}[\langle \boldsymbol{u}, \overline{\boldsymbol{Z}}_i \boldsymbol{B}_i^\mathsf{T}\boldsymbol{v} \rangle^2]$$

$$\le \frac{C}{m^2} \sum_{i \in [m]} \kappa_\ell^2 \sup_{\boldsymbol{u}, \boldsymbol{v} \in \mathbb{S}^{d^a-1}} \mathbb{E}[\langle \mathcal{H}_\ell(\boldsymbol{z}_i), (\boldsymbol{B}_i^\mathsf{T}\boldsymbol{v}) \otimes \boldsymbol{u} \rangle^2] \le C_\ell \frac{\kappa_\ell^2}{m} \sup_{i \in [m]} \|\boldsymbol{B}_i\|_{\mathrm{op}}^2.$$

(98)

Furthermore, similarly to Eq. (96),

$$\sigma(\boldsymbol{\Delta}_1)^2 \le d^a \sigma_*(\boldsymbol{\Delta}_1)^2 \le C_\ell \kappa_\ell^2 \frac{d^a}{m} \sup_{i \in [m]} \|\boldsymbol{B}_i\|_{\mathrm{op}}^2.$$

(99)

Next, for all $p \geq 1$, we have

$$\mathbb{E}[\|\overline{\boldsymbol{Z}}_i \boldsymbol{B}_i^\mathsf{T}\|_{\mathrm{op}}^{2p}]^{1/p} \leq d^{2a} \sup_{\boldsymbol{u},\boldsymbol{v} \in \mathbb{S}^{d^a-1}} \mathbb{E}[\langle \boldsymbol{u}, \overline{\boldsymbol{Z}}_i \boldsymbol{B}_i^\mathsf{T} \boldsymbol{v}\rangle^{2p}]^{1/p}$$

$$\leq d^{2a} p^\ell \sup_{\boldsymbol{u},\boldsymbol{v} \in \mathbb{S}^{d^a-1}} \mathbb{E}[\langle \boldsymbol{u}, \overline{\boldsymbol{Z}}_i \boldsymbol{B}_i^\mathsf{T} \boldsymbol{v}\rangle^2] \leq C_\ell d^{2a} p^\ell \kappa_\ell^2 \sup_{i \in [m]} \|\boldsymbol{B}_i\|_{\mathrm{op}}^2,$$

where we used the triangular inequality in the first line and hypercontractivity of degree-$\ell$ spherical harmonics on the second line. In particular,

$$\frac{1}{m} \mathbb{E}[\sup_{i \in [m]} \|\overline{\boldsymbol{Z}}_i \boldsymbol{B}_i^\mathsf{T}\|_{\mathrm{op}}^2]^{1/2} \leq \frac{1}{m} \left( \sum_{i \in [m]} \mathbb{E}\left[ \|\overline{\boldsymbol{Z}}_i \boldsymbol{B}_i^\mathsf{T}\|_{\mathrm{op}}^{2p} \right] \right)^{1/2p} \leq C_\ell \kappa_\ell \frac{d^a}{m} m^{1/2p} p^{\ell/2} \sup_{i \in [m]} \|\boldsymbol{B}_i\|_{\mathrm{op}}.$$

Taking $p = \log(m)$, we can set

$$\bar{R} = C_\ell \kappa_\ell \frac{d^a}{m} \log^{\ell/2}(m) \sup_{i \in [m]} \|\boldsymbol{B}_i\|_{\mathrm{op}}.$$

Similarly,

$$\mathbb{P}\left( \max_{i \in [m]} \frac{1}{m} \|\overline{\boldsymbol{Z}}_i \boldsymbol{B}_i^\mathsf{T}\|_{\mathrm{op}} \geq R \right) \leq \left( C_\ell \kappa_\ell \frac{d^a}{Rm} m^{1/2p} p^{\ell/2} \sup_{i \in [m]} \|\boldsymbol{B}_i\|_{\mathrm{op}} \right)^{2p} = \delta.$$

Taking $p = C \log(m/\delta)$, we can choose

$$R = C_\ell \kappa_\ell \frac{d^a}{m} \log^{\ell/2}(m/\delta) \sup_{i \in [m]} \|\boldsymbol{B}_i\|_{\mathrm{op}}. \tag{100}$$

Combining Eqs. (98), (99) and (100) into Lemma 25, we obtain with probability at least $1 - \delta/3$

$$\|\boldsymbol{\Delta}_1\|_{\mathrm{op}} \leq C_\ell \kappa_\ell \left[ \sup_{i \in [m]} \|\boldsymbol{B}_i\|_{\mathrm{op}} \right] \sqrt{\frac{d^a}{m}} \left[ 1 + \left( \frac{d^a \log^{\ell+4}(d/\delta)}{m} \right)^{1/2} \vee 1 \right], \tag{101}$$

where we assumed without loss of generality that $\delta > e^{-d}$ to avoid carrying extra terms.

**Step 4: Bound on $\|\boldsymbol{\Delta}_2\|_{\mathrm{op}}$.** Following the exact same argument as for $\boldsymbol{\Delta}_1$ and recalling that $\|\boldsymbol{E}\|_{\mathrm{op}} \leq C_\ell \beta_{d,\ell}$, we directly get with probability at least $1 - \delta/3$,

$$\|\boldsymbol{\Delta}_2\|_{\mathrm{op}} \leq C_\ell \beta_{d,\ell} \kappa_\ell \sqrt{\frac{d^a}{m}} \left[ 1 + \left( \frac{d^a \log^{\ell+4}(d/\delta)}{m} \right)^{1/2} \vee 1 \right]. \tag{102}$$

**Step 4: Concluding.** Combining the bounds (93), (97), (101) and (102), we obtain with probability at least $1 - \delta$,

$$\left\| \hat{\boldsymbol{M}} - \beta_{d,\ell}^2 \frac{c_{\ell,0}^2}{n_{d,\ell}} [\boldsymbol{w}_*^{\otimes a}][\boldsymbol{w}_*^{\otimes a}]^\mathsf{T} \right\|_{\mathrm{op}} \leq \beta_{d,\ell}^2 \|\boldsymbol{\Delta}_0\|_{\mathrm{op}} + \|\boldsymbol{\Delta}_1\|_{\mathrm{op}} + \|\boldsymbol{\Delta}_2\|_{\mathrm{op}}$$

$$\leq C_\ell \beta_{d,\ell}^2 d^{-1/2} + C_\ell \kappa_\ell^2 \frac{d^{\ell/2}}{m} \left[ 1 + \left( \frac{d^a \log^{\ell+4}(d/\delta)}{m} \right) \vee 1 \right],$$

where we used that $a + b = \ell$. Thus by Davis-Kahan theorem, the leading eigenvector $\boldsymbol{s}$ of $\hat{\boldsymbol{M}}$ satisfy

$$|\langle \boldsymbol{s}, \boldsymbol{w}_*^{\otimes a}\rangle| \geq 1 - \eta,$$

with probability at least $1 - \delta$ when

$$m \geq C_\ell \frac{\kappa_\ell^2}{\eta^2} \frac{d^{\ell/2}}{\beta_{d,\ell}^2} \left[ 1 + \frac{\beta_{d,\ell}}{d^{\ell/4-a/2}} \log^{\ell/2+2}(d/\delta) \right].$$

The theorem follows by the same argument as in Section F.4. $\qquad \square$

## F.6 Runtime of the tensor unfolding algorithm

The overall runtime of the algorithm depends on the runtime for matrix-vector multiplication of the matrices $\boldsymbol{H}(\boldsymbol{z}) = \mathbf{Mat}_{a,b}(\mathcal{H}_\ell(\boldsymbol{z}))$. We show that the total runtime if $\Theta(\max(d^a, d^b))$, that is, one does not need to compute the $d^a \times d^b = d^\ell$ entries of $\boldsymbol{H}(\boldsymbol{z})$ to do matrix-vector multiplication with the (unfolded) harmonic tensor. The total runtime of algorithms (TU-Alg-b) and (TU-Alg) in Theorems 8 and 9 follows by recalling that the leading eigenvector can be obtained with $\Theta_d(\log(d))$ iterations of the power method.

**Lemma 15.** *For integers $a, b \geq 1$ with $\ell = a + b$, there exist $C_\ell$ that only depends on $\ell$ such that matrix-vector multiplication with matrix $\boldsymbol{H}(\boldsymbol{z}) = \boldsymbol{Mat}_{a,b}(\mathcal{H}_\ell(\boldsymbol{z}))$ requires at most $C_\ell(d^a + d^b)$ elementary operations.*

*Proof.* Using the identity (83) in Proposition 5, we can decompose

$$\boldsymbol{H}(\boldsymbol{z}) = \sum_{j=0}^{\lfloor \ell/2 \rfloor} c_{\ell,j} \, \mathbf{Mat}_{a,b} \left( \mathrm{Sym}(\boldsymbol{z}^{\otimes(\ell-2j)} \otimes \mathbf{I}_d^{\otimes j}) \right).$$

Thus, we can decompose $\boldsymbol{H}(\boldsymbol{z})$ into $C_\ell$ matrices of the following form (without loss of generality):

$$\sum_{\substack{i_1,\ldots,i_{s_1} \in [d], \\ j_1,\ldots,j_{s_2} \in [d], \\ k_1,\ldots,k_u \in [d]}} \left[ \boldsymbol{z}^{\otimes p_1} \otimes \bigotimes_{r \in [s_1]} \boldsymbol{e}_{i_r}^{\otimes 2} \otimes \bigotimes_{l \in [u]} \boldsymbol{e}_{e_{k_l}} \right] \left[ \boldsymbol{z}^{\otimes p_2} \otimes \bigotimes_{r \in [s_2]} \boldsymbol{e}_{j_r}^{\otimes 2} \otimes \bigotimes_{l \in [u]} \boldsymbol{e}_{e_{k_l}} \right]^{\mathsf{T}}$$

$$= \sum_{k_1,\ldots,k_u \in [d]} \left[ \sum_{i_1,\ldots,i_{s_1} \in [d]} \boldsymbol{z}^{\otimes p_1} \otimes \bigotimes_{r \in [s_1]} \boldsymbol{e}_{i_r}^{\otimes 2} \otimes \bigotimes_{l \in [u]} \boldsymbol{e}_{e_{k_l}} \right] \left[ \sum_{j_1,\ldots,j_{s_2} \in [d]} \boldsymbol{z}^{\otimes p_2} \otimes \bigotimes_{r \in [s_2]} \boldsymbol{e}_{j_r}^{\otimes 2} \otimes \bigotimes_{l \in [u]} \boldsymbol{e}_{e_{k_l}} \right]^{\mathsf{T}}$$

where $p_1 + p_2 = \ell - 2j$, $s_1 + s_2 + u = j$, $p_1 + 2s_1 + u = a$ and $p_2 + 2s_2 + u = b$. The total number of operations to multiply this matrix by a vector is then given by

$$O_d \left( d^u (d^{p_1+s_1} + d^{p_2+s_2}) \right) = O_d \left( d^{u+p_1+s_1} + d^{u+p_2+s_2} \right) = O_d(d^a + d^b),$$

which concludes the proof of this lemma. $\qquad\square$

## F.7 Additional discussions

In this section, we provide an additional discussion on the use of the Harmonic tensor. First, note that all the properties discussed in Section F.3 can be derived using the following Wick's formula for spherical measure and tedious calculations:

**Lemma 16** (Wick's formula). *Let $\boldsymbol{z}$ be a uniform vector on $\mathbb{S}^{d-1}$. We have*

$$\mathbb{E}_{\boldsymbol{z}} \left[ \prod_{j=1}^{2p} z_{k_j} \right] = \frac{1}{d(d+2)\cdots(d+2p-2)} \sum_{\text{pairings } \pi} \prod_{(a,b) \in \pi} \delta_{k_a k_b}, \tag{103}$$

*where the sum runs over all $(2p - 1)!!$ perfect pairings $\pi$ of $\{1, 2, \ldots, 2k\}$.*

We note that for our tensor unfolding algorithm, it is enough to consider a simplified tensor $\mathcal{K}_\ell$ that only keeps the off-diagonal entries of the harmonic tensor.

**Definition 3** (Elementary symmetric tensors). *For every $\ell$, we define $\mathcal{K}_\ell : \mathbb{S}^{d-1} \to \mathrm{Sym}((\mathbb{R}^d)^{\otimes \ell})$ the tensor obtained from $\mathcal{H}_\ell$ by putting $0$ in the entries with repeated indices: for all $\boldsymbol{z} \in \mathbb{S}^{d-1}$, the tensor $\mathcal{K}_\ell(\boldsymbol{z})$ has entries*

$$\mathcal{K}_\ell(\boldsymbol{z})_{i_1,\ldots,i_\ell} = \begin{cases} \mathcal{H}_\ell(\boldsymbol{z})_{i_1,\ldots,i_\ell} = c_{\ell,0} z_{i_1} z_{i_2} \ldots z_{i_\ell} & \text{if } i_1 \neq i_2 \neq \ldots \neq i_\ell, \\ 0 & \text{otherwise.} \end{cases} \tag{104}$$

For convenience, we will denote $\mathcal{I}_{d,j}$ the set of all subset of $j$ indices in $[d]$ with no repetitions. From the reproducing property of $\mathcal{H}_\ell(\boldsymbol{z})$, we have similarly

$$\mathbb{E}[\mathcal{T}_\ell(y,r)\mathcal{K}_\ell(\boldsymbol{z})] = \frac{\beta_{d,\ell}}{\sqrt{n_{d,\ell}}}\mathcal{K}_\ell(\boldsymbol{w}_*). \tag{105}$$

Apply a random rotation to all the input vectors $\boldsymbol{z}_i$. Equivalently, this amounts to having $\boldsymbol{w}_* \sim \tau_d$. This guarantees that $\boldsymbol{w}_*$ is not aligned with any coordinate vector with high probability.

**Lemma 17.** *Assume $\boldsymbol{w} \sim \tau_d$. Define $\Delta(\boldsymbol{w}) = \mathcal{K}_\ell(\boldsymbol{w})/c_{\ell,0} - \boldsymbol{w}^{\otimes\ell}$. Then there exist universal constants $c, C > 0$ such that for all $t \geq 0$,*

$$\mathbb{P}(\|\Delta(\boldsymbol{w})\|_F^2 \geq \ell^2(C+t)/d) \leq 2\exp(-cdt^{1/2}). \tag{106}$$

*Proof.* Simply use that the non-zero entries in $\Delta(\boldsymbol{w})$ have at least one repeated index:

$$\|\Delta(\boldsymbol{w})\|_F^2 \leq \ell^2 \sum_{i_1,\ldots,i_{\ell-1}\in[d]} w_{i_1}^2\cdots w_{i_{\ell-2}}^2 \cdot w_{i_{\ell-1}}^4 \leq \ell^2\|\boldsymbol{w}\|_4^4.$$

Then the tail bound (106) follows from standard concentration argument. $\square$

From this lemma, we deduce that

$$\mathbf{Mat}_p(\mathbb{E}[\mathcal{T}_\ell(y,r)\mathcal{K}_\ell(\boldsymbol{z})]) = \frac{\beta_{d,\ell}}{\sqrt{n_{d,\ell}}}c_{\ell,0}\left([\boldsymbol{w}_*^{\otimes p}][\boldsymbol{w}_*^{\otimes(\ell-p)}]^\mathsf{T} + \boldsymbol{\Delta}\right), \tag{107}$$

where with probability at least $1 - e^{-cd}$ over the random rotation, we have $\|\boldsymbol{\Delta}\|_{\mathrm{op}} \leq C_\ell d^{-1/2}$. Thus, it is enough to consider the tensor $\mathcal{K}_\ell$ which has slightly simpler properties. For example,

$$\mathbf{Mat}_\ell\left(\mathbb{E}[\mathcal{K}_\ell(\boldsymbol{z}) \otimes \mathcal{K}_\ell(\boldsymbol{z})]\right) = \frac{c_{\ell,0}}{\sqrt{n_{d,\ell}}}\frac{1}{\ell!}\sum_{\sigma\in\mathfrak{S}_\ell}\boldsymbol{P}_\sigma,$$

where $\boldsymbol{P}_\sigma$ is the permutation matrix with non-zero entry at row $i$ and column $\sigma(i)$.

**Lemma 18.** *For integer $\ell \geq 2$ and $p = \lfloor\ell/2\rfloor$, there exist $C_\ell$ that only depends on $\ell$ such that matrix-vector multiplication with matrix $\mathbf{Mat}_p(\mathcal{K}_\ell(\boldsymbol{z}))$ requires at most $C_\ell d^{\lceil\ell/2\rceil}$ elementary operations.*

*Proof.* Let us use Newton–Girard identities to decompose $\mathcal{K}_\ell(\boldsymbol{z})$. Note that $\mathcal{K}_\ell(\boldsymbol{z})$ corresponds to the $\ell$-th elementary symmetric polynomial (in tensor form):

$$\mathcal{K}_\ell(\boldsymbol{z}) = \ell!c_{\ell,0}\sum_{i_1<\ldots<i_\ell} z_{i_1}\ldots z_{i_\ell}\mathrm{Sym}(\boldsymbol{e}_{i_1}\otimes\cdots\otimes\boldsymbol{e}_{i_\ell}).$$

Denote $\lambda \vdash \ell$ a partition of $\ell$ with $\lambda = 1^{a_1}2^{a_2}\cdots\ell^{a_\ell}$ so that $\sum_{j\in[\ell]}a_j j = \ell$ and $|\lambda| = a_1 + \ldots + a_\ell$. Then there exist coefficients $c_\lambda$ such that (see Lemma 19 below)

$$\mathcal{K}_\ell(\boldsymbol{z}) = \ell!c_{\ell,0}\sum_{\lambda\vdash\ell}(-1)^{\ell-|\lambda|}c_\lambda\mathrm{Sym}\left(\prod_{j\in[\ell]}\left(\sum_{k_j\in[d]}z_k^j\boldsymbol{e}_k^{\otimes j}\right)^{\otimes a_j}\right). \tag{108}$$

Let's decompose $\mathbf{Mat}_p(\mathcal{K}_\ell(\boldsymbol{z}))$ as a sum of matrices associated to each $\lambda$ (the partition) and $\sigma \in \mathfrak{S}_\ell$ (the permutation in the symmetrization operator). The number of such matrices only depends on $\ell$. Let us bound the runtime for doing a matrix-vector multiplication for each of these matrices. For each $j \in [\ell]$ and $t \in [a_j]$, the tensor $\boldsymbol{e}_k^{\otimes j}$ has its indices $e_{j,t} + f_{j,t} = j$ split into $e_{j,t}$ left indices and $f_{j,t}$ right indices. Denote $\mathcal{E}$ the set of $j \in [\ell], t \in [a_j]$ with $a_j > 0$, and the subsets $\mathcal{E}_M\cup\mathcal{E}_L\cup\mathcal{E}_R = \mathcal{E}$ that contains respectively the indices with $\{e_{j,t} > 0, f_{j,t} > 0\}$, $\{e_{j,t} > 0, f_{j,t} = 0\}$ and $\{e_{j,t} = 0, f_{j,t} > 0\}$. Then we can write the matrix (up to permutation of the indices)

$$\sum_{\{k_{j,t}\}_{(j,t)\in\mathcal{E}}}\prod_{(j,t)\in\mathcal{E}}z_{k_{j,t}}^j\left(\bigotimes_{(j,t)\in\mathcal{E}}\boldsymbol{e}_{k_{t,j}}^{\otimes e_{j,t}}\right)\left(\bigotimes_{(j,t)\in\mathcal{E}}\boldsymbol{e}_{k_{t,j}}^{\otimes f_{j,t}}\right)^\mathsf{T}$$

$$= \sum_{\{k_{j,t}\}_{(j,t)\in\mathcal{E}_M}}\prod_{(j,t)\in\mathcal{E}_M}z_{k_{j,t}}^j\left(\sum_{\{k_{j,t}\}_{(j,t)\in\mathcal{E}_L}}\prod_{(j,t)\in\mathcal{E}_L}z_{k_{j,t}}^j\bigotimes_{(j,t)\in\mathcal{E}}\boldsymbol{e}_{k_{t,j}}^{\otimes e_{j,t}}\right)\left(\sum_{\{k_{j,t}\}_{(j,t)\in\mathcal{E}_R}}\prod_{(j,t)\in\mathcal{E}_R}z_{k_{j,t}}^j\bigotimes_{(j,t)\in\mathcal{E}}\boldsymbol{e}_{k_{t,j}}^{\otimes f_{j,t}}\right)^\mathsf{T}.$$

For each $\{k_{j,t}\}_{(j,t)\in\mathcal{E}_M}$, the left and right vectors take $O_d(d^{|\mathcal{E}_L|})$ and $O_d(d^{|\mathcal{E}_R|})$ operations to compute. Therefore the total runtime is

$$O_d\left(d^{|\mathcal{E}_M|}\left(d^{|\mathcal{E}_L|}+d^{|\mathcal{E}_R|}\right)\right)=O_d\left(d^{|\mathcal{E}_M|+|\mathcal{E}_L|}+d^{|\mathcal{E}_M|+|\mathcal{E}_R|}\right)=O_d\left(d^p+d^{\ell-p}\right)=O_d\left(d^{\lceil\ell/2\rceil}\right),$$

where we used that $|\mathcal{E}_M+\mathcal{E}_L|\leq p$ the number of indices in the left side of the matrix, and $|\mathcal{E}_M|+|\mathcal{E}_R|\leq\ell-p$ the number of indices on the right side of the matrix. $\quad\square$

**Lemma 19** (Newton-Girard identities)**.** *Denote for each integers $\ell,k\geq 1$,*

$$Q_\ell(\boldsymbol{z})=\sum_{1\leq i_1<\ldots<i_\ell\leq d}z_{i_1}\cdots z_{i_\ell},\qquad P_k(\boldsymbol{z})=\sum_{i\in[d]}z_i^k,$$

*the elementary symmetric and power-sum symmetric polynomials respectively. We have*

$$Q_\ell(\boldsymbol{z})=\sum_{\lambda\vdash\ell}(-1)^{\ell-|\lambda|}c_\lambda P_1(\boldsymbol{z})^{a_1}P_2(\boldsymbol{z})^{a_2}\cdots P_\ell(\boldsymbol{z})^{a_\ell},\tag{109}$$

*where $\lambda=1^{a_1}2^{a_2}\cdots\ell^{a_\ell}$ and*

$$c_\lambda=\frac{1}{a_1!a_2!\cdots a_\ell!1^{a_1}2^{a_2}\cdots\ell^{a_\ell}}.$$

Using Eq. (109), we have the following polynomial identity: for all $\boldsymbol{u}\in\mathbb{R}^d$,

$$\langle\mathcal{K}_\ell(\boldsymbol{z}),\boldsymbol{u}^{\otimes\ell}\rangle=\ell!c_{\ell,0}Q_\ell(\boldsymbol{z}\odot\boldsymbol{u})=\ell!c_{\ell,0}\sum_{\lambda\vdash\ell}(-1)^{\ell-|\lambda|}c_\lambda P_1(\boldsymbol{z}\odot\boldsymbol{u})^{a_1}\cdots P_\ell(\boldsymbol{z}\odot\boldsymbol{u})^{a_\ell}.$$

Matching the coefficients in these polynomials in $\boldsymbol{u}$ yields the identity (108).

# G  Proofs for Gaussian SIMs

In this, we shall prove the results from Section 4. We first start by showing the rates on $L^2$ norm of coefficients $\xi_{d,\ell}$ for a Gaussian SIM of generative exponent $\mathsf{k}_\star$ (cf. Lemma 1).

**Proof of Lemma 1:**  We start by recalling that, from [29], the generative exponent $\mathsf{k}_\star(\rho)$ is only defined for $\rho$ whose $\nu_d$ is such that $\nu_d\ll\bar{\nu}_{d,0}$, where $\bar{\nu}_{d,0}:=\nu_{d,Y}\otimes\chi_d\otimes\tau_{d,1}$ is completely decoupled null. In particular, $\|\frac{\mathrm{d}\nu_d}{\mathrm{d}\bar{\nu}_{d,0}}\|_{L^2(\bar{\nu}_{d,0})}$ is bounded by a constant independent of $d$. Let $\{\nu_d\}_{d\geq 1}$ be the sequence of spherical SIMs associated to the Gaussian SIM $\rho$, i.e. $\nu_R=\chi_d$ and $\nu_d(Y\mid Z,R)=\rho(Y\mid Z\cdot R)=\rho(Y\mid X)$. Let $\bar{\nu}_{d,0}=\nu_Y\otimes\nu_R\otimes\nu_Z$. Let us consider the likelihood ratio decomposition in $L^2(\bar{\nu}_{d,0})$ identical to the one in [29, Lemma D.1]

$$\frac{\mathrm{d}\nu_d}{\mathrm{d}\bar{\nu}_{d,0}}(y,r,z)-1\overset{L^2(\bar{\nu}_{d,0})}{=}\sum_{k\geq\mathsf{k}_\star}\zeta_k(y)\mathrm{He}_k(r\cdot z),\quad\zeta_k(y)=\mathbb{E}_{(Y,R,Z)\sim\nu_d}[\mathrm{He}_k(R\cdot Z)\mid Y=y].$$

Denote $\lambda_k=\|\zeta_k\|_{\nu_Y}$, which are completely determined the model $\rho$ independent of $d$, and by definition of generative exponent (2) we have $\lambda_{\mathsf{k}_\star}^2>0$. We now use the decomposition of Hermite into Gegenbauer polynomials from Proposition 2 to rewrite the above as

$$\frac{\mathrm{d}\nu_d}{\mathrm{d}\bar{\nu}_{d,0}}(y,r,z)-1\overset{L^2(\bar{\nu}_{d,0})}{=}\sum_{k\geq\mathsf{k}_\star}\zeta_k(y)\sum_{\ell=0}^{k}\beta_{k,\ell}(r)Q_\ell^{(d)}(z)=\sum_{\ell=0}^{\infty}Q_\ell^{(d)}(z)\sum_{k\geq\mathsf{k}_\star}\beta_{k,\ell}(r)\zeta_k(y).\tag{110}$$

We can now expand the same directly in the Gegenbauer basis

$$\frac{\mathrm{d}\nu_d}{\mathrm{d}\bar{\nu}_{d,0}}(y,r,z)\overset{L^2(\bar{\nu}_{d,0})}{=}\bar{\xi}_{d,0}(y,r)+\sum_{\ell\geq 1}\bar{\xi}_{d,\ell}(y,r)Q_\ell^{(d)}(z),\tag{111}$$

where

$$\bar{\xi}_{d,0}(y,r)=\frac{\mathrm{d}\nu_{d,Y,R}}{\mathrm{d}\nu_{d,Y}\otimes\nu_{d,R}}(y,r)\quad\text{and}\quad\xi_{d,\ell}(y,r)=\mathbb{E}_{(Y,R,Z)\sim\nu_d}[Q_\ell^{(d)}(Z)\mid Y=y,R=r]$$

Equating both (110) and (111), we have the following equalities in $L^2(\bar{\nu}_{d,0})$

$$\bar{\xi}_{d,0}(y,r) = \frac{\mathrm{d}\nu_{d,Y,R}}{\mathrm{d}\nu_{d,Y} \otimes \nu_{d,R}}(y,r) = 1 + \sum_{k \geq \mathsf{k}_\star} \zeta_k(y)\beta_{k,0}(r) := 1 + \psi(y,r) \quad \text{for} \quad \psi(y,r) = \sum_{k \geq \mathsf{k}_\star} \zeta_k(y)\beta_{k,0}(r) \,,$$

and for $\ell \geq 1$

$$\bar{\xi}_{d,\ell}(y,r) \overset{L^2(\bar{\nu}_{d,0})}{=} \bar{\xi}_{d,0}(y,r)\xi_{d,\ell}(y,r) = \sum_{k \geq \mathsf{k}_\star} \zeta_k(y)\beta_{k,\ell}(r) = \sum_{k \in \mathcal{I}_\ell} \zeta_k(y)\beta_{k,\ell}(r) \,,$$

where $\mathcal{I}_\ell := \{k \geq \mathsf{k}_\star : k \equiv \ell \mod 2\}$. In the last equality, we used the fact that $\beta_{k,\ell}(r) = 0$ for $\ell \not\equiv k \mod 2$. Our goal is to bound

$$\mathbb{E}_{\nu_d}[\xi_{d,\ell}(y,r)^2] = \mathbb{E}_{\bar{\nu}_{d,0}}\Big[\frac{\mathrm{d}\nu_{d,0}}{\mathrm{d}\bar{\nu}_{d,0}}(y,r)\xi_{d,\ell}(y,r)^2\Big] = \mathbb{E}_{\bar{\nu}_{d,0}}[\bar{\xi}_{d,0}(y,r)\xi_{d,\ell}(y,r)^2]$$

Denoting $\mathcal{K}(y,r) := \bar{\xi}_{d,0}(y,r)\xi_{d,\ell}(y,r)^2$, we are interested in calculating $\mathbb{E}_{\bar{\nu}_{d,0}}[\mathcal{K}(y,r)]$:

$$\mathbb{E}_{\bar{\nu}_{d,0}}[\mathcal{K}(y,r)] = \mathbb{E}_{\bar{\nu}_{d,0}}[\mathcal{K}(y,r)(1 + \psi(y,r) - \psi(y,r))]$$
$$= \mathbb{E}_{\bar{\nu}_{d,0}}[\mathcal{K}(y,r)\bar{\xi}_{d,0}(y,r)] - \mathbb{E}_{\bar{\nu}_{d,0}}[\mathcal{K}(y,r)\psi(y,r)] \,, \tag{112}$$

We now have the following claim.

**Claim 1.** *We have that for a constant $d$ sufficiently large (only in terms of $\mathsf{k}_\star$),*

$$|E_{\bar{\nu}_{d,0}}[\mathcal{K}(y,r)\psi(y,r)]| \leq \mathbb{E}_{\bar{\nu}_{d,0}}[\mathcal{K}(y,r)]/2 \,.$$

The proof is deferred; for now we use the claim and proceed with the following simplification. It is straightforward to see that, combining Eq. (112) with Claim 1, for $d$ sufficiently large (that only depends on $\mathsf{k}_\star$), we have

$$\frac{1}{2}\mathbb{E}_{\bar{\nu}_{d,0}}\big[\mathcal{K}(y,r)\bar{\xi}_{d,0}(y,r)\big] \leq \mathbb{E}_{\bar{\nu}_{d,0}}[\mathcal{K}(y,r)] \leq \frac{3}{2}\mathbb{E}_{\bar{\nu}_{d,0}}\big[\mathcal{K}(y,r)\bar{\xi}_{d,0}(y,r)\big] \,. \tag{113}$$

Thus, it suffices to obtain rates on $\mathcal{K}(y,r)\bar{\xi}_{d,0}(y,r) = \bar{\xi}_{d,0}(y,r)^2\xi_{d,\ell}(y,r)^2 = \bar{\xi}_{d,\ell}(y,r)^2$ by definition. We finally have the following claim which we will show separately to finish the proof of the lemma.

**Claim 2.** *For all $\ell \leq \mathsf{k}_\star$, we have*

$$\mathbb{E}_{\bar{\nu}_{d,0}}[\bar{\xi}_{d,\ell}(y,r)^2] \asymp d^{-(\mathsf{k}_\star-\ell)/2} \text{ for } \ell \equiv \mathsf{k}_\star \mod 2 \quad \text{and} \quad \mathbb{E}_{\bar{\nu}_{d,0}}[\bar{\xi}_{d,\ell}(y,r)^2] \lesssim d^{-(\mathsf{k}_\star-\ell+1)/2} \text{ for } \ell \not\equiv \mathsf{k}_\star \mod 2 \,.$$

Observe that Claim 2 along with Eq. (113) establishes the desired rates on $\mathbb{E}_{\bar{\nu}_{d,0}}[\mathcal{K}(y,r)] = \mathbb{E}_{\nu_d}[\xi_{d,\ell}(y,r)^2] = \|\xi_{d,\ell}\|_{L^2(\nu_d)}$ concluding the proof of the lemma. $\qquad\square$

We now return to the deferred proofs. In order to show Claim 1, the following bounds on the moments of $\psi(y,r)$ will be useful. The idea is to then directly apply Lemma 24 to conclude that $|\mathbb{E}_{\bar{\nu}_{d,0}}[\mathcal{K}(y,r)\psi(y,r)]|$ is vanishing as compared to $\mathbb{E}_{\bar{\nu}_{d,0}}[\mathcal{K}(y,r)]$, which is sufficient to establish Claim 1.

**Claim 3.** *Let $\psi(y,r) = \sum_{k \geq \mathsf{k}_\star} \zeta_k(y)\beta_{k,0}(r)$, then there exists a universal constant $C > 0$ such that for $d \geq Cp^4$, we have*

$$\|\psi\|_{L^p(\bar{\nu}_{d,0})} \leq C\left(\frac{p}{d^{1/4}}\right)^{\mathsf{k}_\star} \,.$$

*Proof.* For the ease of notation we shall denote $\|\cdot\|_p := \|\cdot\|_{L^p(\bar{\nu}_{d,0})}$ and $\mathcal{I} = \{k \in \mathbb{N} : k \geq \mathsf{k}_\star, k \equiv 0 \mod 2\}$. Recall from Proposition 2 that $\beta_{k,0}(r)$ is a polynomial of degree $k$ with only even degree terms when $k$ is even and zero otherwise. Using this we have:

$$\|\psi\|_p = \mathbb{E}_{\bar{\nu}_{d,0}}[|\psi(y,r)|^p]^{1/p} \leq \sum_{k \in \mathcal{I}} \|\zeta_k\|_p \|\beta_{k,0}\|_p \leq \sum_{k \in \mathcal{I}} \|\mathrm{He}_k\|_p \|\beta_{k,0}\|_p \quad \text{(Jensen's inequality)}$$

$$\leq \sum_{k \in \mathcal{I}} (p-1)^{k/2}(p-1)^{k/2}\|\beta_{k,0}\|_2 \quad\quad \text{(Hypercontractivity Lemmas 21 and 22)}$$

$$\lesssim \sum_{k \in \mathcal{I}} \frac{(p-1)^k}{d^{k/4}}, \quad\quad\quad\quad\quad\quad\quad\quad \text{(using Lemma 3)}$$

hiding a universal constant. We now note that the summation forms a geometric sequence with ratio $(p-1)^2/\sqrt{d}$. Therefore, when $d \geq Cp^4$ for sufficiently large constant $C$, we have that

$$\|\psi\|_p \leq C\left(\frac{p}{d^{1/4}}\right)^{\min_{k\in\mathcal{I}} k} \leq C\left(\frac{p}{d^{1/4}}\right)^{k_\star}.$$

$\square$

Using the above claim, we are now ready to prove Claim 1.

*Proof of Claim 1.* Again let $\|\cdot\|_p$ denote $\|.\|_{L^p(\bar{\nu}_{d,0})}$. We first evaluate

$$\|\mathcal{K}\|_2 = \mathbb{E}_{\bar{\nu}_{d,0}}[\mathcal{K}(y,r)^2]^{1/2} = \mathbb{E}_{\bar{\nu}_{d,0}}\left[\frac{\mathrm{d}\nu_d}{\mathrm{d}\bar{\nu}_{d,0}}(y,r)^2\xi_{d,\ell}(y,r)^4\right]^{1/2}$$

$$\leq \mathbb{E}_{\bar{\nu}_{d,0}}\left[\frac{\mathrm{d}\nu_d}{\mathrm{d}\bar{\nu}_{d,0}}(y,r)^2Q_\ell(y,r)^4\right]^{1/2} \qquad \text{(Jensen's inequality)}$$

$$\lesssim d^\ell \left\|\frac{\mathrm{d}\nu_d}{\mathrm{d}\bar{\nu}_{d,0}}\right\|_{L^2(\bar{\nu}_{d,0})} \lesssim d^\ell,$$

where we used the fact that $\|Q_\ell\|_\infty \leq \sqrt{n_{d,\ell}} = \sqrt{d^\ell}$ and that $\frac{\mathrm{d}\nu_d}{\mathrm{d}\bar{\nu}_{d,0}}$ has $L^2(\bar{\nu}_{d,0})$ norm bounded by a universal constant by definition of Gaussian SIMs $\rho$. Therefore, we have $\frac{2}{k_\star}\log\left(\frac{\|\mathcal{K}\|_2}{\|\mathcal{K}\|_1}\right) \lesssim \log(d)$. Thus for $d$ greater than sufficiently large universal constant, we will indeed have $d \geq Cp^{1/4}$ for all $p \leq \frac{2}{k_\star}\log(\|\mathcal{K}\|_2/\|\mathcal{K}\|_1)$. Using Claim 3, for $d$ greater than sufficiently large universal constant

$$\|\psi\|_p \leq C\left(\frac{p^{k_\star}}{d^{k_\star/4}}\right).$$

Invoking Lemma 24 along with Claim 3, we obtain that

$$|\mathbb{E}_{\bar{\nu}_{d,0}}[\mathcal{K}(y,r)\psi(y,r)]| \lesssim \frac{\|\mathcal{K}\|_1}{d^{k_\star/4}}(2e)^{k_\star}\log^{k_\star}(d).$$

Note that whenever $k_\star \geq 1$, for $d$ greater than some sufficiently large constant (completely determined in terms of $k_\star$), we have

$$|\mathbb{E}_{\bar{\nu}_{d,0}}[\mathcal{K}(y,r)\psi(y,r)]| \leq \|\mathcal{K}\|_1/2 = \mathbb{E}_{\bar{\nu}_{d,0}}[\mathcal{K}(y,r)]/2,$$

as desired. The last equality follows from the fact that $\mathcal{K}(y,r) \geq 0$ a.s. under $\bar{\nu}_{d,0}$. $\square$

Finally, we prove Claim 2.

*Proof of Claim 2.* Let us expand $\bar{\xi}_{d,\ell}(y,r)^2$ under $\bar{\nu}_{d,0}$

$$\bar{\xi}_{d,\ell}(y,r)^2 = \sum_{k\in\mathcal{I}_\ell}\beta_{k,\ell}(r)^2\zeta_k^2(y) + 2\sum_{(k_2>k_1)\in\mathcal{I}_\ell}\zeta_{k_1}(y)\zeta_{k_1}(y)c_{k_1,\ell}(r)c_{k_2,\ell}(r).$$

$$\mathbb{E}_{\bar{\nu}_{d,0}}[\bar{\xi}_{d,\ell}(y,r)^2] = \sum_{k\in\mathcal{I}_\ell}\mathbb{E}[\beta_{k,\ell}(r)^2]\lambda_k^2 + 2\sum_{(k_2>k_1)\in\mathcal{I}_\ell}\mathbb{E}[\zeta_{k_1}(y)\zeta_{k_1}(y)]\mathbb{E}[c_{k_1,\ell}(r)c_{k_2,\ell}(r)]. \quad (114)$$

We now use the bound from Lemma 3 and find the rates for each term in the above.
**Case (a)** $\ell < k_\star$ with $\ell \equiv k_\star \mod 2$ : The first term

$$\sum_{k\in\mathcal{I}_\ell}\mathbb{E}[\beta_{k,\ell}(r)^2]\lambda_k^2 \asymp \lambda_{k_\star}^2 d^{-(k_\star-\ell)/2} + \sum_{k_\star<k\in\mathcal{I}_\ell}\lambda_k^2 d^{-(k-\ell)/2} \asymp \lambda_{k_\star}^2 d^{-(k_\star-\ell)/2},$$

where in the last step we noticed that the latter term forms a geometric series with decaying ratio whose leading term is of smaller order than the former term. We now show that the sum arising from

the cross terms from (114) is of smaller order in the absolute value.

$$
|\sum_{(k_2>k_1)\in\mathcal{I}_\ell} \mathbb{E}[\zeta_{k_1}(y)\zeta_{k_1}(y)]\mathbb{E}[c_{k_1,\ell}(r)c_{k_2,\ell}(r)]| \leq \sum_{(k_2>k_1)\in\mathcal{I}_\ell} |\mathbb{E}[\zeta_{k_1}(y)\zeta_{k_1}(y)]| \cdot |\mathbb{E}[c_{k_1,\ell}(r)c_{k_2,\ell}(r)]|
$$

$$
\leq \sum_{(k_2>k_1)\in\mathcal{I}_\ell} |\lambda_{k_1}\lambda_{k_2}| \cdot \sqrt{\mathbb{E}[c_{k_1,\ell}(r)^2]\cdot\mathbb{E}[c_{k_2,\ell}(r)^2]}
$$

$$
\lesssim \sum_{(k_2>k_1)\in\mathcal{I}_\ell} d^{-\left(\frac{k_1+k_2}{4}-\frac{\ell}{2}\right)} \lesssim \sum_{k_1\in\mathcal{I}_\ell} d^{-\left(\frac{k_1-\ell}{2}+1\right)}
$$

$$
\lesssim d^{-\left(\frac{k_\star-\ell}{2}+1\right)}.
$$

Substituting this bound in (114), we conclude that for any $\ell < k_\star$ with $\ell \equiv k_\star \mod 2$, we have

$$
\mathbb{E}_{\bar{\nu}_{d,0}}[\bar{\xi}_{d,\ell}(y,r)^2] \asymp d^{-(k_\star-\ell)/2}.
$$

**Case (b)** $\ell < k_\star$ with $\ell \not\equiv k_\star \mod 2$: We now do similar simplifications.

$$
\sum_{k\in\mathcal{I}_\ell} \mathbb{E}[\beta_{k,\ell}(r)^2]\lambda_k^2 \asymp \min_{\substack{k>k_\star \\ k\in\mathcal{I}_\ell}} \lambda_k^2\, d^{-(k-\ell)/2} \lesssim d^{-(k_\star+1-\ell)/2}.
$$

Bounding the contribution of the cross terms

$$
|\sum_{(k_2>k_1)\in\mathcal{I}_\ell} \mathbb{E}[\zeta_{k_1}(y)\zeta_{k_1}(y)]\mathbb{E}[c_{k_1,\ell}(r)c_{k_2,\ell}(r)]| \leq \sum_{(k_2>k_1)\in\mathcal{I}_\ell} |\lambda_{k_1}\lambda_{k_2}| \cdot \sqrt{\mathbb{E}[c_{k_1,\ell}(r)^2]\cdot\mathbb{E}[c_{k_2,\ell}(r)^2]}
$$

$$
\lesssim \sum_{(k_2>k_1)\in\mathcal{I}_\ell} d^{-\left(\frac{k_1+k_2}{4}-\frac{\ell}{2}\right)} \lesssim \sum_{k_1\in\mathcal{I}_\ell} d^{-\left(\frac{k_1-\ell}{2}+1\right)}
$$

$$
\lesssim d^{-\left(\frac{k_\star+1-\ell}{2}+1\right)}.
$$

Putting the bounds in (114), for any $\ell < k_\star$ such that $\ell \not\equiv k_\star \mod 2$, we have

$$
\mathbb{E}_{\bar{\nu}_{d,0}}[\bar{\xi}_{d,0}(y,r)\xi_{d,\ell}(y,r)^2] \lesssim d^{-(k_\star+1-\ell)/2}.
$$

$\square$

We next prove Lemma 2 which we use to characterize the complexity when one is only allowed to use the directional component $z$.

**Proof of Lemma 2:** The proof is very similar to and, in fact, simpler than that of Lemma 1. Our goal is to provide rates for the $L^2$ norm $\xi_{d,\ell}$. However, as $r = 1$ always as we only observe $(y, z)$, both completely decoupled null and "partially" decoupled null (where $(Y, R)$ is decoupled from $Z$) are identical. Therefore, the change-of-measure argument required to used in the proof of Lemma 1 is no longer needed. The proof then follows from the calculations similar to the one done in the proof of Claim 2.

For clarity, we will continue to denote the original problem $(y, \boldsymbol{x})$ with $\nu_d$, and the new spherical single index model where one only observes $(y, \boldsymbol{z})$ by $\upsilon_d$.

Again, we have $\{\nu_d\}_d$ with $\nu_R = \chi_d$ and $\rho(Y \mid X) = \rho(Y \mid Z \cdot R) = \nu_d(Y \mid Z, R)$. Let $\bar{\nu}_{d,0} = \nu_Y \otimes \nu_R \otimes \nu_Z$ be the completely decoupled null. We will also let $\{\upsilon_d\}_d$ be the sequence of problem associated with $\{\nu_d\}_d$ where we only observe $(y, \boldsymbol{z})$. We have

$$
\frac{\mathrm{d}\nu_d}{\mathrm{d}\bar{\nu}_{d,0}}(y,r,z) - 1 \stackrel{L^2(\bar{\nu}_{d,0})}{=} \sum_{k\geq k_\star} \zeta_k(y)\mathrm{He}_k(r\cdot z), \text{ where } \zeta_k(y) = \mathbb{E}_{(Y,R,Z)\sim\nu_d}[\mathrm{He}_k(R\cdot Z) \mid Y = y]
$$

$$
\stackrel{L^2(\bar{\nu}_{d,0})}{=} \sum_{k\geq k_\star} \zeta_k(y)\sum_{\ell=0}^{k} \beta_{k,\ell}(r)Q_\ell^{(d)}(z) = \sum_{\ell=0}^{\infty} Q_\ell^{(d)}(z)\sum_{k\geq k_\star} \beta_{k,\ell}(r)\zeta_k(y),
$$

where in the second line, we used the harmonic decomposition of Hermite from Proposition 2. We marginalize the radius to explicitly write the likelihood ratio of only $(y, z)$ part under $\nu_d$ and $\bar{\nu}_{d,0}$, is identical to that of $\upsilon_d$ and $\upsilon_{d,0} = \nu_Y \otimes \tau_{d,1}$.

$$
\frac{\mathrm{d}\upsilon_d}{\mathrm{d}\upsilon_{d,0}}(y,z) - 1 \stackrel{L^2(\upsilon_{d,0})}{=} \sum_{\ell=0}^{\infty} Q_\ell^{(d)}(z)\sum_{k\geq k_\star} \mathbb{E}[\beta_{k,\ell}(r)]\zeta_k(y).
$$

We can also expand the log likelihood ratio of $(y, z)$ directly in the Gegenbauer basis

$$\frac{dv_d}{dv_{d,0}}(y,z) - 1 \overset{L^2(v_{d,0})}{=} \sum_{\ell \geq 1} \xi_{d,\ell}(y) Q_\ell^{(d)}(z) \,, \text{ where } \xi_{d,\ell}(y) = \mathbb{E}_{(Y,Z)\sim v_d}[Q_\ell^{(d)}(Z) \mid Y = y].$$

Equating both, we have for any $\ell \geq 1$

$$\xi_{d,\ell}(y) \overset{L^2(v_{d,0})}{=} \sum_{k \geq \mathsf{k}_\star} \zeta_k(y)\mathbb{E}[\beta_{k,\ell}(r)] = \sum_{k \in \mathcal{I}_\ell} \zeta_k(y)\mathbb{E}[\beta_{k,\ell}(r)] \text{ where } \mathcal{I}_\ell := \{k \geq \mathsf{k}_\star : k \equiv \ell \mod 2\}\,.$$

Squaring both sides

$$\xi_{d,\ell}(y)^2 = \sum_{k \geq \mathcal{I}_\ell} \mathbb{E}[\beta_{k,\ell}(r)]^2 \zeta_k^2(y) + 2 \sum_{(k_2 > k_1) \in \mathcal{I}_\ell} \zeta_{k_1}(y)\zeta_{k_1}(y)\mathbb{E}[c_{k_1,\ell}(r)]\mathbb{E}[c_{k_2,\ell}(r)]\,.$$

$$\|\xi_{d,\ell}\|_{L^2(v_Y)}^2 = \sum_{k \in \mathcal{I}_\ell} \mathbb{E}[\beta_{k,\ell}(r)]^2 \lambda_k^2 + 2 \sum_{(k_2 > k_1) \in \mathcal{I}_\ell} \mathbb{E}[\zeta_{k_1}(y)\zeta_{k_1}(y)]\mathbb{E}[c_{k_1,\ell}(r)]\mathbb{E}[c_{k_2,\ell}(r)]\,. \quad (115)$$

We now use the rates on $\mathbb{E}_{r\sim\chi_d}[\beta_{k,\ell}(r)]^2$ from Lemma 3 to carry out the simplification similar to the one done in the proof of Claim 2.
**Case (a)** $\ell < \mathsf{k}_\star$ with $\ell \equiv \mathsf{k}_\star \mod 2$:

$$\sum_{k \in \mathcal{I}_\ell} \mathbb{E}[\beta_{k,\ell}(r)]^2 \lambda_k^2 \asymp \lambda_{\mathsf{k}_\star}^2 \, d^{-(\mathsf{k}_\star - \ell)} + \sum_{\mathsf{k}_\star < k \in \mathcal{I}_\ell} \lambda_k^2 \, d^{-(k-\ell)} \asymp \lambda_{\mathsf{k}_\star} d^{-(\mathsf{k}_\star-\ell)}\,,$$

where the step followed by observing that it is a sum of geometric series whose rate is dominated by the first term. We now show the bound on the magnitude of the cross terms

$$\left|\sum_{(k_2 > k_1) \in \mathcal{I}_\ell} \mathbb{E}[\zeta_{k_1}(y)\zeta_{k_1}(y)]\mathbb{E}[c_{k_1,\ell}(r)]\mathbb{E}[c_{k_2,\ell}(r)]\right| \leq \sum_{(k_2 > k_1) \in \mathcal{I}_\ell} |\mathbb{E}[\zeta_{k_1}(y)\zeta_{k_1}(y)]| \cdot |\mathbb{E}[c_{k_1,\ell}(r)]\mathbb{E}[c_{k_2,\ell}(r)]|$$

$$\leq \sum_{(k_2 > k_1) \in \mathcal{I}_\ell} |\lambda_{k_1}\lambda_{k_2}| \cdot \sqrt{\mathbb{E}[c_{k_1,\ell}(r)]^2 \cdot \mathbb{E}[c_{k_2,\ell}(r)]^2}$$

$$\lesssim \sum_{(k_2 > k_1) \in \mathcal{I}_\ell} d^{-\left(\frac{k_1+k_2}{2} - \ell\right)} \lesssim \sum_{k_1 \in \mathcal{I}_\ell} d^{-(k_1 - \ell + 1)}$$

$$\lesssim d^{-(\mathsf{k}_\star - \ell + 1)}\,.$$

Combining these rates with (115), we obtain for any $\ell < \mathsf{k}_\star$ with $\ell \equiv \mathsf{k}_\star \mod 2$,

$$\|\xi_{d,\ell}\|_{L^2(v_Y)}^2 = \mathbb{E}_v[\xi_{d,\ell}(y)^2] \asymp d^{-(\mathsf{k}_\star-\ell)}\,.$$

**Case (b)** $\ell < \mathsf{k}_\star$ such that $\ell \not\equiv \mathsf{k}_\star \mod 2$: We do similar calculation in the other case.

$$\sum_{k \geq \mathcal{I}_\ell} \mathbb{E}[\beta_{k,\ell}(r)]^2 \lambda_k^2 \asymp \min_{\substack{k > \mathsf{k}_\star \\ k \in \mathcal{I}_\ell}} \lambda_k^2 \, d^{-(k-\ell)} \lesssim d^{-(\mathsf{k}_\star - \ell + 1)}\,.$$

Bounding the cross terms

$$\left|\sum_{(k_2 > k_1) \in \mathcal{I}_\ell} \mathbb{E}[\zeta_{k_1}(y)\zeta_{k_1}(y)]\mathbb{E}[c_{k_1,\ell}(r)]\mathbb{E}[c_{k_2,\ell}(r)]\right| \leq \sum_{(k_2 > k_1) \in \mathcal{I}_\ell} |\lambda_{k_1}\lambda_{k_2}| \cdot |\mathbb{E}[c_{k_1,\ell}(r)] \cdot \mathbb{E}[c_{k_2,\ell}(r)]|$$

$$\lesssim \sum_{(k_2 > k_1) \in \mathcal{I}_\ell} d^{-\left(\frac{k_1+k_2}{2} - \ell\right)} \lesssim \sum_{k_1 \in \mathcal{I}_\ell} d^{-(k_1 - \ell + 1)} \lesssim d^{-(\mathsf{k}_\star - \ell + 2)}\,.$$

Substituting these bounds in (115), for any $\ell < \mathsf{k}_\star$ such that $\ell \not\equiv \mathsf{k}_\star \mod 2$, we have

$$\mathbb{E}_{\bar{\nu}_{d,0}}[\xi_{d,\ell}(y,r)^2] \lesssim d^{-(\mathsf{k}_\star - \ell + 1)}\,.$$

# H Information-theoretic sample complexity

## H.1 Information theoretic lower-bound

Below we derive an information-theoretic lower bound for recovering $\boldsymbol{w}_*$ in single-index models under an illustrative assumption. This information-theoretic result completes the low-degree polynomial and SQ lower bounds for the detection problem. Recall that when $\|\xi_{d,1}\|_{L^2} = \Theta_d(1)$, the LDP lower bounds scales as $\sqrt{d}$: indeed, detection can be achieved with this many samples by taking the test statistics obtained by projecting the likelihood ratio onto samplewise-$(t, 1)$ polynomials with $t = \omega_d(1)$. However, the information-theoretic lower bound for recovery scales as $\Omega(d)$ (that is, there is a detection-recovery gap in this model). This is well understood in the Gaussian case (e.g., see [66, 67, 29]) and we provide a short proof for spherical SIMs below for completeness.

We will consider the following illustrative regularity assumptions on the link function.

**Assumption 6.** *We assume that for all $s, t \in \mathbb{R}_{\geq 0}$, we have*

$$\mathsf{KL}(\nu_d(\cdot|r,t)||\nu_d(\cdot|r,s)) \leq dLK(r)(t-s)^2,$$

*where $L > 0$ is a constant, and $K \in L^1(\mu_r)$.*

**Remark H.1.** In the case of a Gaussian noise i.e $y = f(\langle \boldsymbol{w}, \boldsymbol{x} \rangle) + \sigma \boldsymbol{Z}$, where $\boldsymbol{Z} \sim \mathcal{N}(0, \sigma^2)$, we have $\mathsf{KL}(\mathbb{P}_{\boldsymbol{w}}||\mathbb{P}_{\boldsymbol{w}'}) = \frac{1}{2\sigma^2}\mathbb{E}[(f(\langle \boldsymbol{w}, \boldsymbol{x} \rangle) - f(\langle \boldsymbol{w}', \boldsymbol{x} \rangle))^2]$ which satisfies the assumption 6.

We now state our minimax lower bound under this assumption.

**Theorem 10.** *Let $\nu_d \in \mathfrak{L}_d$ that satisfies Assumption 6 and let $\mathbb{P}_{\nu_d, \boldsymbol{w}}$ be the associated family of SIM distributions indexed by $\boldsymbol{w} \in \mathbb{S}^{d-1}$. Then, for any estimator $\hat{\boldsymbol{w}}$ based on $\mathsf{m}$ observations from $\mathbb{P}_{\nu_d, \boldsymbol{w}}$, we have:*

$$\inf_{\hat{\boldsymbol{w}}} \sup_{\boldsymbol{w} \in \mathbb{S}^{d-1}} \mathbb{E}_{\mathbb{P}_{\nu_d, \boldsymbol{w}}}[\|\hat{\boldsymbol{w}} - \boldsymbol{w}\|^2] \geq \frac{(d-1)\log(C)}{8\mathsf{m}LC^2},$$

*where $C$ is a constant.*

*Proof of theorem 10.* We define the planted distribution denoted by $\mathbb{P}_{\nu_d, \boldsymbol{w}}^n$: given $\boldsymbol{w} \in \mathbb{S}^{d-1}$, we sample $n$ points $(y_i, \boldsymbol{x}_i) \sim_{idd} \mathbb{P}_{\nu_d, \boldsymbol{w}}$. We construct the set of hypotheses $\mathcal{H} = \{\boldsymbol{w}_1, \ldots, \boldsymbol{w}_M\}$ as a $2\delta$-packing of $\mathbb{S}^{d-1}$ ($\delta$ is chosen small enough such that the $2\delta$-packing is not a singleton), and such that $\forall i \neq j, \|\boldsymbol{w}_i - \boldsymbol{w}_j\| \leq \mathcal{C}\delta$. Using Fano's lower bound [81, Chapter 15] to the problem of estimating $\boldsymbol{w}$ from the observations $(y_i, \boldsymbol{x}_i)_{i=1}^n$ yields

$$\inf_{\hat{\boldsymbol{w}}} \sup_{\boldsymbol{w} \in \mathbb{S}^{d-1}} \mathbb{E}_{\mathbb{P}_{\nu_d, \boldsymbol{w}}}[\|\hat{\boldsymbol{w}} - \boldsymbol{w}\|^2] \geq \frac{1}{2}\delta^2 \left(1 - \frac{I(\boldsymbol{Z};J) + \log(2)}{\log(M)}\right), \tag{116}$$

where $J$ is uformly distributed over $\{1, \ldots, M\}$, and $\boldsymbol{Z}$ is a random variable distributed according to $\mathbb{P}_{\nu_d, \boldsymbol{w}_J}$, where $J$ is independent of $\boldsymbol{Z}$. By the convexity of the Kullback-Leibler divergence and additivity of the KL divergence, we have

$$I(\boldsymbol{Z};J) \leq \frac{1}{M^2} \sum_{i,j=1}^M \mathsf{KL}(\mathbb{P}_{\nu_d, \boldsymbol{w}_i}^n, \mathbb{P}_{\nu_d, \boldsymbol{w}_j}^n) \leq \frac{n}{M^2} \sum_{i,j=1}^M \mathsf{KL}(\mathbb{P}_{\nu_d, \boldsymbol{w}_i}||\mathbb{P}_{\nu_d, \boldsymbol{w}_j}). \tag{117}$$

Using Assumption 6, we bound the KL divergence between two distributions $\mathbb{P}_{\nu_d, \boldsymbol{w}_i}$ and $\mathbb{P}_{\nu_d, \boldsymbol{w}_j}$

$$\begin{aligned}
\mathsf{KL}(\mathbb{P}_{\nu_d, \boldsymbol{w}_i}, \mathbb{P}_{\nu_d, \boldsymbol{w}_j}) &= \mathbb{E}_{\boldsymbol{z}, r, \boldsymbol{y} \sim \mathbb{P}_{\nu_d, \boldsymbol{w}_i}} \left[\log\left(\frac{d\mathbb{P}_{\nu_d, \boldsymbol{w}_i}}{d\mathbb{P}_{\nu_d, \boldsymbol{w}_j}}(y, r, \boldsymbol{z})\right)\right] \\
&\leq \mathbb{E}_{\boldsymbol{z}} \left[\mathbb{E}_{\boldsymbol{y}, r \sim \mathbb{P}_{\nu_d, \boldsymbol{w}_i}} \left[\log\left(\frac{d\mathbb{P}_{\nu_d, \boldsymbol{w}_i}}{d\mathbb{P}_{\nu_d, \boldsymbol{w}_j}}(y, r, \boldsymbol{z})\right) |\boldsymbol{z}, r\right]\right] \\
&\leq L\|\boldsymbol{w}_i - \boldsymbol{w}_j\|^2 \mathbb{E}_r[K(r)] \leq LC\delta^2,
\end{aligned}$$

where $L$ is a constant from assumption 6, and we used that for all $j \neq i, \|\boldsymbol{w}_i - \boldsymbol{w}_j\| \leq \mathcal{C}\delta$ for some constant $\mathcal{C} > 0$. We then have

$$I(\boldsymbol{Z};J) \leq nLC\delta^2. \tag{118}$$

We bound the cardinality of $M$, using a classical volume argument: let define $\mathcal{S}_\delta(\boldsymbol{x}) = \{\boldsymbol{y} \in \mathbb{S}^{d-1}\colon \langle \boldsymbol{x}, \boldsymbol{y}\rangle \geq 1 - \delta^2\} = \{\boldsymbol{y} \in \mathbb{S}^{d-1}\colon \|\boldsymbol{x} - \boldsymbol{y}\| \leq 2\delta\}$, we then have

$$\mathcal{P}_\delta(\mathbb{S}^{d-1}) \geq \mathcal{N}_\delta(\mathbb{S}^{d-1}) \geq \frac{\mathrm{Vol}(\mathbb{S}^{d-1} \bigcap \mathcal{S}_{\mathcal{C}\delta}(\theta_1))}{\mathrm{Vol}(\mathcal{S}_\delta)} \geq C\mathcal{C}^{d-1},$$

where $C$ is an universal constant, and we denote that $\mathcal{N}_\delta(\mathbb{S}^{d-1})$ is the covering number and $\mathcal{P}_\delta(\mathbb{S}^{d-1})$ is the packing number, and we used $\mathcal{N}_\delta(\mathbb{S}^{d-1}) \leq \mathcal{P}_\delta(\mathbb{S}^{d-1})$ [80, Prop 4.2.1], and that the homogenity of the volume on the sphere $\mathbb{S}^{d-1}$. With the choice of $\mathcal{H}$ described above, and plugging this into equation (116) and equation (117), we obtain

$$\inf_{\hat{\boldsymbol{w}}} \sup_{\boldsymbol{w} \in \mathbb{S}^{d-1}} \mathbb{E}_{\mathbb{P}_{\nu_d, \boldsymbol{w}}}[\|\hat{\boldsymbol{w}} - \boldsymbol{w}\|^2] \geq \frac{\delta^2}{2}\left(1 - \frac{nLC\delta^2 + \log(2)}{(d-1)\log(\mathcal{C}) + \log(c)}\right). \tag{119}$$

We choose $\delta^2 = \frac{(d-1)}{2nLC} \leq 1$, and plugging this into the previous inequality equation (119), we obtain the following lower bound for $d$ sufficiently large

$$\inf_{\hat{\boldsymbol{w}}} \sup_{\boldsymbol{w} \in \mathbb{S}^{d-1}} \mathbb{E}_{\mathbb{P}_{\nu_d, \boldsymbol{w}}}[\|\hat{\boldsymbol{w}} - \boldsymbol{w}\|^2] \geq \frac{d-1}{8nLC}. \tag{120}$$

$\square$

## H.2 Information theoretic upper-bound

We complement our information-theoretic lower bound with a sample complexity upper bound. We exhibit an estimator (which is not computable in polynomial time) that achieves strong recovery with $O(d)$ samples for general spherically symmetric measure under a mild assumption on the sequence $\{\nu_d\}_{d \geq 1}$ that there exists $\ell \geq 1$ such that $\|\xi_{d,\ell}\|_{L^2} = \Omega_d(1)$ (cf. Appendix A). The proof is directly adapted from [29, Theorem 6.1]. We focus on proving weak recovery of the ground truth since we can boost it to obtain strong recovery: there exists a (non-polynomial time) algorithm such that for all $\varepsilon > 0$, it outputs $\hat{\boldsymbol{w}}$ with $|\langle\hat{\boldsymbol{w}}, \boldsymbol{w}_*\rangle| \geq 1 - \varepsilon$ with probability $1 - o(1)$ and sample complexity

$$\mathsf{m} = O(d/\varepsilon).$$

**Proposition 6.** *Under the assumption that there exists $\ell \geq 1$ such that $\|\xi_{d,\ell}\|_{L^2} = \Theta_d(1)$, there exists an estimator (non polynomially computable) that returns $\hat{\boldsymbol{w}} \in \mathbb{S}^{d-1}$ that satisfies $|\langle\hat{\boldsymbol{w}}, \boldsymbol{w}_*\rangle| \geq 1/2$, with probability at least $1 - 2e^{-d}$ with information theoretic sample complexity $\mathsf{m} = O(d)$, hiding constants in $\ell$ and $\|\xi_{d,\ell}\|_{L^2}$.*

*Proof.* For any $\delta > 0$, let $\mathcal{N}_\delta$ be a $\delta$-net of $\mathbb{S}^{d-1}$, and we can choose $\mathcal{N}_\delta$ such that $|\mathcal{N}_\delta| \leq \left(\frac{3}{\delta}\right)^d$. Consider the following $g(y, r, z) = \xi_{d,\ell}(y, r)Q_\ell(z)$. For simplicity, let us denote $\beta_{d,\ell} = \|\xi_{d,\ell}\|_{L^2}$. Fix a truncation $R > 0$, and denote $L_n(\boldsymbol{w})$ defined as

$$L_n(\boldsymbol{w}) := \frac{1}{n}\sum_{i=1}^n g(\langle\boldsymbol{w}, \boldsymbol{z}_i\rangle, y_i, r_i)\mathbf{1}_{|g(\langle\boldsymbol{w},\boldsymbol{z}_i\rangle,y_i,r_i)|\leq R}.$$

We consider the min-max estimator

$$\hat{\boldsymbol{w}} \in \operatorname*{arg\,min}_{\hat{\boldsymbol{w}} \in \mathbb{S}^{d-1}} \max_{\boldsymbol{w} \in \mathcal{N}_\delta} \left|L_n(\boldsymbol{w}) - \frac{\beta_{d,\ell}^2 Q_\ell(\langle\boldsymbol{w}, \hat{\boldsymbol{w}}\rangle)}{\sqrt{n_{d,\ell}}}\right|.$$

Using Lemma 24, we have

$$\mathbb{E}[g(\langle\boldsymbol{w}, \boldsymbol{z}\rangle, y, r)^2] = \mathbb{E}[\xi_{d,\ell}(\langle\boldsymbol{w}, \boldsymbol{z}\rangle)^2 Q_\ell(\langle\boldsymbol{w}, \boldsymbol{z}\rangle^2)] \leq \beta_{d,\ell}^2 \log(3/\beta_{d,\ell}^2)^{k/2}.$$

Using Bernstein's lemma, we have for any $\boldsymbol{w} \in \mathbb{S}^{d-1}$, with probability at least $1 - 2e^{-t}$

$$|L_n(\boldsymbol{w}) - \mathbb{E}[L_n(\boldsymbol{w})]| \leq \sqrt{\frac{\beta_{d,\ell}^2 \log\left(\frac{3}{\beta_{d,\ell}^2}\right)^{\ell/2} t}{n}} + \frac{Rt}{n}.$$

By union bound and setting $t = d\left(\log\left(\frac{3}{\delta}\right) + 1\right)$, we then have with probability at least $1 - 2e^{-d}$,

$$\sup_{\boldsymbol{w}\in\mathcal{N}_\delta} |L_n(\boldsymbol{w}) - \mathbb{E}[L_n(\boldsymbol{w})]| \lesssim \sqrt{\frac{\beta_{d,\ell}^2 \log\left(\frac{3}{\beta_{d,\ell}^2}\right)^{k/2} d\log\left(\frac{3}{\delta}\right)}{n}} + \frac{Rd\left(\log\left(\frac{1}{\delta}\right)\right)}{n}. \tag{121}$$

We bound the effect of the truncation

$$
\begin{aligned}
|\mathbb{E}[L_n(\boldsymbol{w}) - \mathbb{E}[g(\langle \boldsymbol{w},\boldsymbol{z}\rangle, y, r)]]| &= |\mathbb{E}[g(\langle \boldsymbol{w},\boldsymbol{z}\rangle, y, r)\mathbf{1}_{|g(\langle \boldsymbol{w},\boldsymbol{z}\rangle,y,r)|\geq R}]| \\
&\leq \sqrt{\mathbb{E}[g(\langle \boldsymbol{w},\boldsymbol{z}\rangle, y, r)^2]\mathbb{P}(|g(\langle \boldsymbol{w},\boldsymbol{z}\rangle, y, r)| > R)} \\
&\leq \beta_{d,\ell}\sqrt{\mathbb{P}(|g(\langle \boldsymbol{w},\boldsymbol{z}\rangle, y, r)| > R)}.
\end{aligned}
$$

We then have the following control on the moments of $|g(\langle \boldsymbol{w},\boldsymbol{z}\rangle, y, r))|$

$$\mathbb{E}[|g(\langle \boldsymbol{w},\boldsymbol{z}\rangle, y, r)|^p]^{1/p} \leq \mathbb{E}[|Q_\ell(\langle \boldsymbol{w},\boldsymbol{z}\rangle)|^p |\xi_{d,\ell}(y,r)|^p]^{1/p} \leq (2p)^\ell,$$

by using Jensen inequality and spherical hypercontractivity. By taking $R \geq (2e)^\ell$, and $\delta = R^{1/\ell}/2e$

$$\mathbb{P}(|g(\langle \boldsymbol{w},\boldsymbol{z}\rangle, y, r)| > R) \leq \frac{2p^{\ell p}}{R^p} \leq \exp\left(-\frac{\ell}{2e}R^{1/\ell}\right)$$

Combining the two inequalities gives

$$|\mathbb{E}[L_n(\boldsymbol{w})] - \mathbb{E}[g(\langle \boldsymbol{w},\boldsymbol{z}\rangle, y, r)]| \leq \beta_{d,\ell}\exp\left(-\frac{\ell}{4e}R^{1/\ell}\right).$$

Combining the above inequalities, we then have

$$
\begin{aligned}
&\max_{\boldsymbol{w}\in\mathcal{N}_\delta}\left|\frac{\beta_{d,\ell}^2 Q_\ell(\langle \boldsymbol{w},\boldsymbol{w}_*\rangle)}{\sqrt{n_{d,\ell}}} - \frac{\beta_{d,\ell}^2 Q_\ell(\langle \boldsymbol{w},\hat{\boldsymbol{w}}\rangle)}{\sqrt{n_{d,\ell}}}\right| \\
&\leq \max_{\boldsymbol{w}\in\mathcal{N}_\delta}\left|L_n(\boldsymbol{w}) - \frac{\beta_{d,\ell}^2 Q_\ell(\langle \boldsymbol{w},\boldsymbol{w}_*\rangle)}{\sqrt{n_{d,\ell}}}\right| + \max_{\boldsymbol{w}\in\mathcal{N}_\delta}\left|L_n(\boldsymbol{w}) - \frac{\beta_{d,\ell}^2 Q_\ell(\langle \boldsymbol{w},\hat{\boldsymbol{w}}\rangle)}{\sqrt{n_{d,\ell}}}\right| \\
&\leq 2\max_{\boldsymbol{w}\in\mathcal{N}_\delta}\left|L_n(\boldsymbol{w}) - \frac{\beta_{d,\ell}^2 Q_\ell(\langle \boldsymbol{w},\boldsymbol{w}_*\rangle)}{\sqrt{n_{d,\ell}}}\right| \\
&\leq 2\max_{\boldsymbol{w}\in\mathcal{N}_\delta}|L_n(\boldsymbol{w}) - \mathbb{E}[g(\langle \boldsymbol{w},\boldsymbol{z}\rangle, y, r)]| \\
&\leq \max_{\boldsymbol{w}\in\mathcal{N}_\delta}|L_n(\boldsymbol{w}) - \mathbb{E}[L_n(\boldsymbol{w})]| + 3^\ell\exp\left(-\frac{\ell}{4e}R^{1/\ell}\right) \\
&\leq \sqrt{\frac{\beta_{d,\ell}^2 \log\left(\frac{3}{\beta_{d,\ell}^2}\right)^{k/2} t}{n}} + \frac{Rt}{n} + 3^\ell\exp\left(-\frac{\ell}{4e}R^{1/\ell}\right).
\end{aligned}
$$

We have the following

$$
\begin{aligned}
\left|\frac{Q_\ell(\langle \boldsymbol{w},\boldsymbol{w}_*\rangle) - Q_\ell(\langle \boldsymbol{w},\hat{\boldsymbol{w}}\rangle)}{\sqrt{n_{d,\ell}}}\right| &= \frac{1}{\sqrt{n_{d,\ell}}}\left|\sum_{q=0}^{\lfloor \ell/2\rfloor} c_{\ell,q}\left(\langle \boldsymbol{w},\boldsymbol{w}_*\rangle^{\ell-2q} - \langle \boldsymbol{w},\hat{\boldsymbol{w}}\rangle^{\ell-2q}\right)\right| \\
&= \frac{c_{\ell,0}}{\sqrt{n_{d,\ell}}}\left|\langle \boldsymbol{w},\boldsymbol{w}_*\rangle^\ell - \langle \boldsymbol{w},\hat{\boldsymbol{w}}\rangle^\ell\right| + O(d^{-1}).
\end{aligned}
$$

We then deduce that

$$\max_{\boldsymbol{w}\in\mathcal{N}_\delta}\left|\frac{\beta_{d,\ell}^2 Q_\ell(\langle \boldsymbol{w},\boldsymbol{w}_*\rangle)}{\sqrt{n_{d,\ell}}} - \frac{\beta_{d,\ell}^2 Q_\ell(\langle \boldsymbol{w},\hat{\boldsymbol{w}}\rangle)}{\sqrt{n_{d,\ell}}}\right| = \frac{\beta_{d,\ell}^2 c_{\ell,0}}{\sqrt{n_{d,\ell}}}\max_{\boldsymbol{w}\in\mathcal{N}_\delta}\left|\langle \boldsymbol{w},\boldsymbol{w}_*\rangle^\ell - \langle \boldsymbol{w},\hat{\boldsymbol{w}}\rangle^\ell\right| + O(d^{-1}).$$

Using [41, Lemma 25], we then have

$$\frac{\beta_{d,\ell}^2 c_{\ell,0}}{\sqrt{n_{d,\ell}}}\min_{s\in\{\pm 1\}}\|s\hat{\boldsymbol{w}} - \boldsymbol{w}_*\| \lesssim \beta_{d,\ell}^2\left(\max_{\boldsymbol{w}\in\mathcal{N}_\delta}\left|\frac{Q_\ell(\langle \boldsymbol{w},\boldsymbol{w}_*\rangle)}{\sqrt{n_{d,\ell}}} - \frac{Q_\ell(\langle \boldsymbol{w},\hat{\boldsymbol{w}}\rangle)}{\sqrt{n_{d,\ell}}}\right| + \delta + O(d^{-1})\right)$$

Using this inequality above, and plugging the inequality, we have

$$\min_{s\in\{\pm1\}} \|s\hat{\boldsymbol{w}} - \boldsymbol{w}_*\|_2 \lesssim \sqrt{\frac{\beta_{d,\ell}^2 \log(\frac{3}{\beta_{d,\ell}^2})^{\ell/2} d \log\left(\frac{3}{\delta}\right)}{n}} + \delta + \frac{Rd\log\left(\frac{3}{\delta}\right)}{n} + 3^\ell \exp\left(-\frac{\ell}{4e}R^{1/\ell}\right)$$

Choosing $R = (4e\log(3/\delta))^\ell$, it yields

$$\frac{\beta_{d,\ell}^2 c_{\ell,0}}{\sqrt{n_{d,\ell}}} \min_{s\in\{\pm1\}} \|s\hat{\boldsymbol{w}} - \boldsymbol{w}_*\| \lesssim \delta + \sqrt{\frac{\beta_{d,\ell}^2 \log(\frac{3}{\beta_{d,\ell}^2})^{\ell/2} d \log\left(\frac{3}{\delta}\right)}{n}} + \frac{Rd\log\left(\frac{3}{\delta}\right)}{n} + 3^\ell \exp\left(-\frac{\ell}{4e}R^{1/\ell}\right).$$

Taking $\delta = O(\beta_{d,\ell}^2)$ concludes the proof. $\qquad\square$

# I  Additional technical results

The uniform distribution $\tau_d = \mathrm{Unif}(\mathbb{S}^{d-1})$ and the isotropic Gaussian distribution $\mathsf{N}(0, \mathbf{I}_d)$ satisfy the following hypercontractivity properties:

**Lemma 20** (Spherical Hypercontractivity [13]). *For any $\ell \in \mathbb{N}$ and $f \in L^2(\tau_d)$ which is a degree $\ell$ polynomial, for any $p \geq 2$, we have*

$$\|f\|_{L^p(\tau_d)} \leq (p-1)^{\ell/2} \|f\|_{L^2(\tau_d)}.$$

**Lemma 21** (Gaussian Hypercontractivity). *For any $\ell \in \mathbb{N}$ and $f \in L^2(\mathsf{N}(\mathbf{0}, \mathbf{I}_d))$ which is a degree $\ell$ polynomial, for any $p \geq 2$, we have*

$$\|f\|_{L^p} := \mathbb{E}_{\boldsymbol{x}\sim\mathsf{N}(\mathbf{0},\mathbf{I}_d)}[|f(\boldsymbol{x})|^p]^{1/p} \leq (p-1)^{\ell/2} \|f\|_{L^2}.$$

As a corollary of this, we also have the following property for the $\chi_d$ distribution.

**Lemma 22.** *For any even $\ell \in \mathbb{N}$ and $f \in L^2(\chi_d)$ which is a polynomial of degree $\ell$ with only even degree terms (i.e. $f(r) = \sum_{i=0}^{\ell/2} a_i r^{2i}$), for any $p \geq 2$ we have*

$$\|f\|_{L^p(\chi_d)} \leq (p-1)^{\ell/2}\|f\|_{L^2(\chi_d)}.$$

This lemma follows from Lemma 21 by noting that $f(r) = f(\|\boldsymbol{x}\|_2) = f(\sqrt{x_1^2 + \cdots + x_d^2})$ is a polynomial of $\boldsymbol{x}$ of degree $\ell$, where $\boldsymbol{x} \sim \mathsf{N}(0, \mathbf{I}_d)$. To obtain high probability tail bounds from bounds on the moments, we will often use the above hypercontractivity properties with the following standard tail-bounds:

**Lemma 23** (Lemma 24 in [28]). *Let $\delta \geq 0$ and $X$ be a mean zero random variable satisfying*

$$\mathbb{E}[|X|^p]^{1/p} \leq B\, p^{k/2} \text{ for } p = \frac{2\log(1/\delta)}{k},$$

*for some $k$. Then with probability $1 - \delta$, we have $|X| \leq B\, p^{k/2}$.*

Similar to [28], we will use the following lemma to bound $\mathbb{E}[XY]$ instead of standard Cauchy-Schwarz, when we have a tight bound $\|X\|_1$ and all moments $\|Y\|_p$ but a very loose bound on $\|X\|_2$.

**Lemma 24** (Lemma 23 in [28]). *Let $X, Y$ be random variables with $\|Y\|_p \leq B\, p^{k/2}$. Then*

$$\mathbb{E}[XY] \leq \|X\|_1 \cdot B \cdot (2e)^{k/2} \cdot \max\left(1, \frac{2}{k}\log\left(\frac{\|X\|_2}{\|X\|_1}\right)\right)^{k/2}.$$

**Lemma 25** (Lemma I.5 in [29]). *Let $\boldsymbol{Y} = \sum_{i=1}^n \boldsymbol{Z}_i$, where $\boldsymbol{Z}_i \in \mathbb{R}^{p\times q}$ are mean zero independent matrices. Define*

$$\sigma := \sigma(\boldsymbol{Y}) = \max\left(\|\mathbb{E}[\boldsymbol{Y}\boldsymbol{Y}^\mathsf{T}]\|_{\mathrm{op}}^{1/2}, \|\mathbb{E}[\boldsymbol{Y}^\mathsf{T}\boldsymbol{Y}]\|_{\mathrm{op}}^{1/2}\right),$$

$$\sigma_* := \sigma_*(\boldsymbol{Y}) = \sup_{\boldsymbol{v}\in\mathbb{S}^{p-1}, \boldsymbol{u}\in\mathbb{S}^{q-1}} \mathbb{E}[(\langle\boldsymbol{v}, \boldsymbol{Y}\boldsymbol{w}\rangle)^2]^{1/2},$$

$$\bar{R} = \mathbb{E}[\max_{i\in[n]} \|\boldsymbol{Z}_i\|_{\mathrm{op}}^2]^{1/2}.$$

*Then for*

$$R \geq \bar{R}^{1/2}\sigma^{1/2} + \sqrt{2}\bar{R},$$

*and $t \geq 0$, denoting $\delta = \mathbb{P}(\max_{i \in [n]} \|\boldsymbol{Z}_i\| \geq R)$, we have with probability at least $1 - \delta - de^{-t}$,*

$$\|\boldsymbol{Y} - \mathbb{E}[\boldsymbol{Y}]\|_{\mathrm{op}} \leq 2\sigma + \sigma_* t^{1/2} + R^{1/3}\sigma^{2/3}t^{2/3} + Rt.$$

**Lemma 26.** *Let $\{Z_i\}_{i \in [n]}$ be a sequence of independent random variables with polynomial tails, i.e. there exists $B, k$ such that $\mathbb{E}[|Z_i|^p]^{1/p} \leq Bp^{k/2}$. Define $R = \max_{i \in [n]} Z_i$. Then for any $p \leq \log n/k$, we have $\mathbb{E}[|R|^p]^{1/p} \leq B \log^{k/2}(n)$ and for any $\delta \geq 0$, with probability at least $1 - \delta$, $R \leq B \log^{k/2}(n/\delta)$.*

**Lemma 27** (Lemma I.3 from [29])**.** *Let $X_1, \ldots, X_n$ be independent mean zero random variables such that for all $p \geq 2$, we have $\|X_i\|_p \leq Bp^{k/2}$ for some $k$ and let $\sigma^2 = \sum_{i=1}^n \mathbb{E}[X_i^2]$. Let $Y = \sum_{i=1}^n X_i$. Then with pribability at least $1 - \delta$,*

$$|Y| \lesssim_k \sigma\sqrt{\log(1/\delta)} + B \log(1/\delta) \log(n/\delta)^{k/2}.$$

**Lemma 28** ([78])**.** *Let $\boldsymbol{X}_k$ be i.i.d random matrices of dimensions $d_1 \times d_2$. Assume that each matrix is bounded by*

$$\forall k, \|\boldsymbol{X}_k - \mathbb{E}[\boldsymbol{X}_k]\|_{\mathrm{op}} \leq L.$$

*Consider $v(\boldsymbol{Z}) = \max\{\|\sum_{k=1}^n \mathbb{E}[(\boldsymbol{X}_k - \mathbb{E}[\boldsymbol{X}_k])(\boldsymbol{X}_k - \mathbb{E}[\boldsymbol{X}_k])^\mathsf{T}]\|, \|\sum_{k=1}^n \mathbb{E}[(\boldsymbol{X}_k - \mathbb{E}[\boldsymbol{X}_k])^\mathsf{T}(\boldsymbol{X}_k - \mathbb{E}[\boldsymbol{X}_k])]\|\}$, then with probability at least $1 - \delta$,*

$$\left\|\sum_{k=1}^n (\boldsymbol{X}_k - \mathbb{E}[\boldsymbol{X}_k])\right\|_{\mathrm{op}} \leq \frac{L}{3} \log\left(\frac{d_1 + d_2}{\delta}\right) + \sqrt{4v \log\left(\frac{d_1 + d_2}{\delta}\right)}.$$

