# OpenReview forum: "Learning single index models via harmonic decomposition"
_NeurIPS.cc/2025/Conference — NeurIPS 2025 poster_

### Official Review · Reviewer_91Gz · 2025-06-30

**Clarity:** 2
**Significance:** 3
**Originality:** 4
**Rating:** 5
**Confidence:** 4

**Summary:**

This paper develops a harmonic-decomposition perspective on high-dimensional single-index models (SIMs).  SIMs have attracted intense attention in recent years, with many complementary papers already charting phase transitions, statistical-computational gaps, and learning dynamics.By expanding the likelihood in spherical harmonics rather than Hermite polynomials, the authors (i) derive tight Statistical-Query and Local-Differential-Privacy lower bounds for every harmonic level in the signal; (ii) propose two estimators—harmonic tensor unfolding (which is sample-optimal) and online projected SGD (which is runtime-optimal)—that match those bounds up to logarithmic factors; and (iii) show that the classical Gaussian SIM results emerge as a special case of their framework. The analysis clarifies why landscape-smoothing and partial-trace spectral methods succeed, highlights when vanilla SGD is provably sub-optimal, and unifies several lines of earlier work.

**Questions:**

Extension to staircase links: Could the harmonic-decomposition framework be adapted to “staircase’’ regressions, i.e. settings where the link function is piece-wise constant rather than smooth?

Multi-index generalisation: Is there a principled way to lift the analysis from single- to multi-index models that involve several latent directions?

Noise robustness: How sensitive are the proposed lower bounds and algorithms to additive observation noise—especially of the type analysed in arXiv:2502.06443v1—and can the theory be modified to accommodate such perturbations?

**Ethical Concerns:**

["NO or VERY MINOR ethics concerns only"]

**Final Justification:**

This paper makes a strong and original contribution by introducing a harmonic-decomposition framework that unifies lower bounds, optimal algorithms, and spectral methods in single-index models. The rebuttal and discussions confirmed the correctness and novelty of the main results, increasing my confidence in the work. The main unresolved issue is the omission of several key citations, which affects scholarly completeness but not the technical contribution. Overall, the paper is rigorous, impactful, and I recommend acceptance.

**Limitations:**

yes

**Quality:**

3

**Strengths And Weaknesses:**

Strenghts: This is a technically strong and unusually original paper: by viewing single-index models through the lens of harmonic decomposition, it unifies a patchwork of open questions—tight lower bounds, sample-optimal versus runtime-optimal algorithms, and the peculiar success of spectral “partial-trace’’ tricks—within a single, coherent framework. The quality of the work is high: the Statistical-Query and Local-DP lower bounds are rigorously proved, the two complementary algorithms meet those bounds up to log factors, and the analysis cleanly recovers the classical Gaussian results as a special case, thereby subsuming several separate lines of literature

Weaknesses: The exposition suffers from a notable gap in citations. The discussion of the easy/hard transition in SIMs references overlooks earlier rigorous work that first established the information-theoretic limit and computational gap; recent rigorous papers on learning dynamics are omitted as well, and the genealogy of spectral methods is incomplete. These omissions currently temper the otherwise excellent contribution.

While the paper offers an original and compelling perspective on single-index models through harmonic decomposition, I believe several important references are missing, which would significantly improve the completeness and scholarly context of the discussion. First and foremost, the paper “Optimal Errors and Phase Transitions in High-Dimensional Generalized Linear Models” (arXiv:1708.03395) deserves to be cited when discussing the GE=2 case and the statistical-to-computational gap. This work rigorously established the information-theoretic limits and the phase transition at Hermite degree 2—what is now often called the “Hermite-2 threshold”—and provided an early and formal treatment of the Parisi–Franz potential and the associated computational gap. While MM18 proposed an important spectral method to approach this threshold, the existence of the transition itself and the structure of the gap were already clearly laid out in this earlier paper. As a point of reference, this work is cited in [DPVLB24] and in several other recent contributions.

In addition, two more recent papers—arXiv:2202.00293 and arXiv:2406.02157—provide rigorous and contemporary analyses of SIM learning under Gaussian assumptions, focusing in particular on gradient-based algorithms and the implicit regularization they induce. These works are on par with other cited results (e.g., [AGJ21], [DNGL24]) and would be highly relevant to include in the broader discussion. On the spectral side, the paper also omits a key antecedent to MM18: Yue Lu’s 2017 work (arXiv:1702.06435), which provides early results on spectral thresholds in high-dimensional nonconvex estimation problems, including phase retrieval

---

> ### Author Rebuttal · Authors · 2025-07-30
>
> We thank the reviewer for acknowledging the contribution of the work! **We agree that our original submission omitted several important references**. We are grateful to the reviewers for pointing out these works. We have now incorporated all the suggested citations, which better situate our paper within the existing literature and significantly strengthen the presentation. Specifically, we **added several references in both the introduction and the technical sections**. Additionally, we have written a "Related Work" section now. All these updates will be available in the final version.
>
> Answers to the questions are in order.
>
> ---
>
> * **Extension to non-smooth link functions:** We allow for **arbitrary link functions in our spherical SIM definition (Definition 1)**, which includes the situation which includes “staircase” type link functions (as long as Radon Nikodym derivative w.r.t. null is bounded in L^2 norm, as defined in Definition 1).
>
> * **Multi-index functions:** We believe that some of our results can be adapted straightforwardly to the multi-index setting. Our SQ and LDP lower bounds directly exploit the $\mathcal{O}_d$ invariance, and the **Gegenbauer polynomials can be replaced by multivariate spherical harmonics** with little modification. The **harmonic tensor unfolding can be directly applied to the multi-index setting**:  this is in fact the algorithm proposed in a recent work on learning multi-index models [Damian et al. 2025], which appeared after this neurips deadline.
>
> * **Effect of noise:** Our lower bounds are for any spherical SIM over a rotationally symmetric input distribution. In particular, Gaussian SIM $\nu$ with noise can be seen as the new Gaussian SIM $\tilde{\nu}$. The results of [Cornacchia et al.2025] can be interpreted as, in the Gaussian case, under mild assumption most single index model $\nu$ with “high complexity”, their corresponding $\tilde{\nu}$ have low complexity. Whether such separations are there more generally beyond Gaussian cases remains to be seen, but our lower and upper bounds cover them by definition. We believe it is interesting to investigate this question further.
>
> ---
> [Damian et al 2025] The Generative Leap : Sharp Sample Complexity of Efficiently Learning Gaussian Multi-Index Models  , Damian, Bruna, Lee
>
> [Cornacchia et al 2025] Low-dimensional functions are efficiently learnable under randomly biased distributions, E Cornacchia, D Mikulincer, E Mosse

---

> > ### Comment · Reviewer_91Gz · 2025-08-07
> >
> > I appreciate the authors’ thorough clarifications and the inclusion of the missing citations. These updates satisfactorily address my earlier concerns, and I maintain my positive recommendation for acceptance of the work.

---

### Official Review · Reviewer_vFaG · 2025-06-30

**Clarity:** 3
**Significance:** 4
**Originality:** 4
**Rating:** 5
**Confidence:** 4

**Summary:**

The paper studied the problem of learning single index models under rotation-invariant distributions, where the data vector x satisfies $x=rz, r\sim\mu_r$ and $z\sim\tau_d$, and $\tau_d = \mathrm{Unif}(\mathbb{S}^{d-1})$. The labels y satisfy $y | r,z \sim \nu_d(. | r,w^* z)$ where $w^*$ is the unit-norm hidden direction. It is assumed $\nu_d$ is known. The authors then used spherical harmonics decomposition of the link function rather than hermite polynomials to study the computational complexity. The main contributions of this paper are three-fold:
1. The authors showed that under the spherical model, using harmonic decomposition, there is a gap between the SQ lower bounds and LDP lower bounds, indicating a 'sample-runtime' trade-off phenomenon. In detail, SQ lower bounds indicate a runtime complexity lower bound that is higher than the sample complexity lower bound implied by LDPs. The authors proposed a characterization of the sample and runtime complexity using the degree of the Gegenbauer polynomial that has the maximum signal from the harmonic decomposition.
2. The authors provided sample optimal and runtime optimal algorithm respectively, using tensor unfolding and online SGD. The authors conjectured that there is no algorithm that can achieve optimal sample complexity and runtime complexity at the same time.
3. The authors applied the framework to Gaussian SIMs, i.e., they studied learning SIMs with link function $y|r,z=\rho(w*\cdot z)$ where $\rho$ has generative exponent $k*$ and $r\sim\chi_d$ and $z\sim \mathrm{Unif}(\mathbb{S}^{d-1})$. They showed that the optimal harmonic degree is either 1 or 2 (depending on the parity of k^*) and recovers the optimal sample and runtime complexity of learning Gaussian SIMs in prior literature.

**Questions:**

1. In line 113-119, the authors claimed that the SGD is suboptimal, possibly due to the fact that it does not exploit the radial component $r=||x||$ and claimed that algorithms ignoring r must have runtime $d^{k*}$. Can you provide a pointer to where this claim is justified? I skimmed through section E and did not spot the supporting lemma.
2. Furthermore, can the authors elaborate on how to exploit the radial exponent?

**Ethical Concerns:**

["NO or VERY MINOR ethics concerns only"]

**Final Justification:**

The authors cleared my confusion and I maintain my evaluation.

**Limitations:**

no negative limitations

**Paper Formatting Concerns:**

no concerns

**Quality:**

3

**Strengths And Weaknesses:**

Strengths
1. I think the observations and findings of this paper are very interesting. The paper goes beyond Gaussian distribution and studied the computational complexity of learning SIMs under more general rotational invariant distributions, and discovered a complexity measure that recovers prior complexity results on learning Gaussian SIMs. It is also very interesting to see that there is a gap between SQ and LDPs under the spherical model, indicating a sample-runtime trade-off.
2. The technical aspect of this paper is solid and strong.
3. The paper is mostly clearly and fluently written.

Weakness
1. I think in the introduction section where the authors compared prior works on learning Gaussian SIMs is somewhat confusing, e.g., line 40-47. [Damian et al Neurips] closes the gap in terms of the information exponent (i*) rather than the generative exponent (k*), and I don't think [Damian et al COLT] simply removes the polylog terms. Further, my understanding is that [D et al Neurips] and [D et al COLT] are essentially very different algorithms that tackle different computational barriers.

2. In the comparison to prior work part, I am afraid the authors missed another intensive thread of research on learning SIMs under general or Gaussian distributions with misspecification/agnostic noise, e.g., [DGKK+20], [DKTZ22], [WZDD23],[GGKS23], [ZWDD24], [WZDD24], [GV25], [ZWDD25] and so on. It would be more well-rounded if this thread of research were mentioned in the comparison to prior works part.

[DGKK+20]: Diakonikolas I, Goel S, Karmalkar S, Klivans R, Soltanolkotabi M. Approximation Schemes for ReLU Regression, in COLT 2020
[DKTZ22]: Diakonikolas I, Kontonis V, Tzamos C, Zarifis N. Learning a single neuron with adversarial label noise via gradient descent, in COLT 2022
[WZDD23]: Wang P, Zarifis N, Diakonikolas I, Diakonikolas J. Robustly Learning a Single Neuron via Sharpness, in ICML 2023
[GGKS23]: Gollakota A, Gopalan P, Klivans A, Stavropoulos K. Agnostically learning single-index models using omnipredictors, in Neurips 2023
[ZWDD24]: Zarifis N, Wang P, Diakonikolas I, Diakonikolas J. Robustly learning single-index models via alignment sharpness, in ICML  2024.
[WZDD24]: Wang P, Zarifis N, Diakonikolas I, Diakonikolas J. Sample and computationally efficient robust learning of gaussian single-index models, in Neurips 2024.
[GV25]: Guo A, Vijayaraghavan A. Agnostic Learning of Arbitrary ReLU Activation under Gaussian Marginals.
[ZWDD25]: Zarifis N, Wang P, Diakonikolas I, Diakonikolas J. Robustly Learning Monotone Generalized Linear Models via Data Augmentation.

---

> ### Author Rebuttal · Authors · 2025-07-30
>
> We thank the reviewers for acknowledging the contribution of the work and positive feedback! Below we provide clarifications to the weaknesses.
>
> ---
>
> * **A clarification on Damian et al. landscape smoothing vs generative exponent papers:** We are aware that prior to the [Damian et al, 2024] paper, the literature mostly revolved around information exponent (corresponds to CSQ). As mentioned in Lines 32-37, the primary contribution of [Damian et al, 2024] is to **introduce the generative exponent** as the fundamental barrier for computational sample complexity. However, from an **algorithmic** standpoint, the main contribution of [Damian et al, 2024], according to us, is in **matching the complexity removing log factors**—in particular, in our opinion, although information exponent is a fundamentally different complexity measure, it is straightforward to observe that, the landscaped smoothed SGD [Damian et al. 2023], on applying appropriate label transformations, achieves the guarantees where information exponent can be replaced with generative exponent. **This results in basically the optimal sample complexity up to log factors, and from here on the goal of the partial trace estimator is to remove them.**
>
> * Our goal in the main text is to directly focus on generative exponent complexity measure and how different methods like online SGD/ landscape smoothing/ partial trace (after label transformations) achieves different complexities, and to point out that there is **a gap in our understanding despite having tight results**. We tried to credit the original source and analysis of these algorithmic primitives, although this comes with a caveat that the online SGD and the landscape smoothing were only stated for information exponents and a label transformation is required to bring them down to generative exponent. We briefly touch upon this in footnote 1, with more discussion in Appendix A.3.
>
> * **Missing references:** Apologies for missing out on some important lines of work. **We agree that our original submission omitted several important references**. We are grateful to the reviewers for pointing out these works. We have now incorporated all the suggested citations, which better situate our paper within the existing literature and significantly strengthen the presentation. Specifically, we **added several references in both the introduction and the technical sections**. Additionally, we have written a "Related Work" section now. All these updates will be available in the final version.
>
> ---
>
> We now answer the questions.
> * **suboptimality of SGD:** All the prior algorithms are informally revisited in Appendix A.2, including the online SGD without norm. We provide a semi-formal justification of these informal claims in the same appendix. (We might further expand them in the final version.)
>
> * **Exploiting the norm:** One obvious way of exploiting the norm is to run online SGD according to Equation SGD-Alg (lines 199-203). This is one of the powers of harmonic decomposition that we can **create a transformation of radial component and label explicitly, and run online SGD with neuron $Q_{\ell}(\langle w,z\rangle )$**. There might be other “natural” ways of exploiting the norm, which we have not invested thoughts in. The purpose of discussing vanilla SGD is to provide a new explanation as to how this method could fall short in achieving the optimal complexity.
>
> ---
> [Damian et al, 2024] Computational-statistical gaps in gaussian single-index models Alex Damian, Loucas Pillaud-Vivien, Jason D Lee, Joan Bruna
>
> [Damian et al, 2023] Smoothing the landscape boosts the signal for sgd: Optimal sample complexity for learning single index models, Alex Damian, Eshaan Nichani, Rong Ge, Jason D Lee

---

> > ### Comment · Reviewer_vFaG · 2025-08-05
> >
> > I thank the authors for the informative responses that cleared my confusion. I would keep my score and recommend acceptance.

---

### Official Review · Reviewer_KRnG · 2025-06-30

**Clarity:** 3
**Significance:** 4
**Originality:** 3
**Rating:** 5
**Confidence:** 4

**Summary:**

The authors study learning in a **single index model** of the form $x \mapsto g(w^* \cdot x)$, where the input distribution over $x$ is **rotation-invariant**. Their analysis leverages the **spherical harmonic decomposition** of the signal to:

- **Characterize the problem difficulty**, via the spectral structure of the signal; (decomposition into the spherical harmonics)
- **Define the optimal transformation** of the learning task to recover the planted direction $w^*$;
- **Establish fundamental limits** through minimax **statistical complexity bounds**:
  - **SQ lower bound** ,
  - **LDP lower bound**
- **Analyze the complexity** of different algorithms:
  -  Spectral algorithm and Tensor Unfolding
  - **Stochastic Gradient Descent (SGD)** ;
- **Recover known results in the Gaussian case** when the *generative exponent* rules the complexity of the algorithms derived in the article.

**Questions:**

- I find the following sentence unclear: "The SGD algorithm of [2] is dominated by the high-frequency harmonics $V_{d,k}$, while smoothing [10] reweights the loss landscape toward low-frequency harmonics $V_{d,1}$ or $V_{d,2}$." It is not obvious to me how to interpret this, since the runtime complexity of SGD depends explicitly on $k^*$, whereas the spaces $V_{d,1}$ and $V_{d,2}$ do not seem to play a direct role in the complexity. Could the authors clarify what is meant here?

- Could you provide some insight on Assumption 1? As stated, it appears to imply that a single harmonic component dominates the others in terms of statistical or algorithmic complexity. Is this the intended meaning?

- In line 200, SGD is defined with respect to some parameter $l$, and a transformation $T_{l}$ that seems to extract the coefficient of the response in the $V_{d,l}$ harmonic. Is this interpretation correct? Also, which value of $l$ is chosen in Corollary 2? Presumably, it should be related to $k^*$; but this is not clearly stated. Could the authors make this connection more explicit?

**Ethical Concerns:**

["NO or VERY MINOR ethics concerns only"]

**Final Justification:**

The authors provides a clear article on single index models providing valuable insights. I thank the authors for their additional comments and the answer they give to the other reviewers. I keep my score.

**Limitations:**

No limit, I would be nice to have clearer results.

**Quality:**

4

**Strengths And Weaknesses:**

### Strengths

- **Conceptual clarity**: The use of spherical harmonics provides a principled and elegant framework to analyze single index models under rotational symmetry.
- **Theoretical depth**: The paper offers nontrivial lower bounds (SQ and LDP) that clarify the statistical hardness of the problem.
- **Unified perspective**: The approach recovers known results in Gaussian settings and discuss the fundamental limits of SGD.
- **Technically solid**: Proofs appear to be rigorous and well-structured, with clear connections between harmonic properties and learning performance.

### Weaknesses

- **Overloaded presentation**: While the authors make a commendable effort to define concepts precisely and clearly, the paper feels overly dense, with too many results introduced in rapid succession. This makes it hard to follow the central narrative, and the reader may struggle to grasp the key insights. A better organization of the material would help, although it's not immediately obvious how to restructure it. As it stands, the paper sometimes gives the impression of going in too many directions at once.

- **Unclear resolution of the central question**: The paper raises the important question of why SGD-based algorithms are suboptimal, exhibiting a runtime of $d^k$ instead of the optimal $d^{k/2 + 1}$. However, the answer to this question remains hard to extract. In particular, it is unclear where it is precisely stated (or proved) that a regularity assumption on the distribution over $\nu_{d, R}$ is necessary to halve the complexity exponent. Also, is there a clear intuition or conceptual reason behind this improvement? Making this point more explicit would strengthen the paper.

---

> ### Author Rebuttal · Authors · 2025-07-31
>
> We thank the reviewer for their time and positive feedback! We answer your question in order:
>
> * The landscape smoothing leads to different harmonic decompositions, which can be seen in Observation 3, page 24, and in line 843. So, the landscape smoothing reweights all the signals in the first two frequencies, and adds a normalizing factor depending on the harmonic decomposition of the loss. The spaces of degree 2 spherical harmonics are the effective harmonic decompositions, and our analysis relies on these harmonic decompositions.
> * **Asssumption 1**: We thank the reviewer for this remark. This assumption was indeed confusing and unnecessary for the proof of the lower bound. We have replaced it with a mild (necessary) assumption that the SIM is learnable in polynomial time. Formally
>
> **Assumption 1 (updated):** There exists $p\in \mathbb{N}$, such that the sequence $\{\nu_d \}$ satisifes $\mathsf{M}_\star(\nu_d)=O_d(d^{p/2})$.
>
> * This interpretation is correct. The frequency parameter ensures the transformation has a non-zero $\ell$-frequency. In Corollary 2, we applied Theorem 2 with any $\ell$ that has the same parity as $k_*$. That can be seen as a consequence of Theorem 1 and Lemma 2, which actually computes the asymptotics of $\lVert \xi_{d,\ell}\rVert^{2}$ that appears in the harmonic decomposition.

---

> > ### Comment · Reviewer_KRnG · 2025-08-05
> > **Answer**
> >
> > I thank the authors for their additional comments and the answer they give to the other reviewers. I keep my score.

---

### Official Review · Reviewer_VyBx · 2025-07-02

**Clarity:** 3
**Significance:** 4
**Originality:** 3
**Rating:** 5
**Confidence:** 3

**Summary:**

The author considers the setting of learning single index models with Gaussian marginals.
In particular, the noise model they consider here is kind of between realizable and agnostic,
in the sense that the distribution of y can only depends on the important direction but need not to labeled by a single target function (equation 1).
The authors gave a very novel and insightful observation that the hardness can be more naturally analyzed by spherical harmonics instead of Hermite polynomials (which is the typical choice of basis in previous literature).
This can be intuitively explained by that the spherical harmonics are the irreducible representations for the rotation symmetry.

Based on this, the authors proposed the spherical single index model framework (which generalizes the regular single index model).
While the authors did not provide any useful problem/application that fits into this framework but not the previous regular single index model framework.
They did show that by analyzing the SQ and LDP complexity of the spherical single index model, not only did they recover the previous results of generative exponents for single index models.
But also give some explanation about why online SGP types of algorithm failed to give the right sample complexity in SIM.

**Questions:**

1. Can the authors give some further high-level comments on why LDP and SQ gives different information-computation trade-off?
Especially, if there is any intuition about why they predicts the right sample/time complexity for tensor unfolding and SGD respectively.

2. I'm a bit confused about Lemma 1, from my understanding Lemma 1 only assume that p has generative exponent k^*.
Combining this with the decomposition of Hermite polynomial into Gegenbauer polynomials.
Shouldn't this only implies that there exists one spherical harmonic (in this decomposition) that has large coefficients?
Instead, Lemma states that coefficients for all spherical harmonic (in this decomposition) are this stated value (up to constant factor).

**Ethical Concerns:**

["NO or VERY MINOR ethics concerns only"]

**Limitations:**

Yes.

**Quality:**

3

**Strengths And Weaknesses:**

Strength:
The paper offers a very novel and insightful perspective on the single index model.

Weakness:
From what I know, there is also a line of work on agnostic single-index modes, which I think is relevant here and should be cited.
For example, here some that I found (I'm sure I'm missing some):

1. I. Diakonikolas, D. M. Kane, and N. Zarifis. Near-optimal SQ lower bounds for agnostically learning
halfspaces and ReLUs under Gaussian marginals.
2. S. Goel, A. Gollakota, and A. R. Klivans. Statistical-query lower bounds via functional gradients.
3. I. Diakonikolas, D. M. Kane, T. Pittas, and N. Zarifis. The optimality of polynomial regression for
agnostic learning under Gaussian marginals in the SQ model.
4. I. Diakonikolas, D. M. Kane, and L. Ren. Near-optimal cryptographic hardness of agnostically learning
halfspaces and ReLU regression under Gaussian marginals.
5. P. Manurangsi and D. Reichman. The computational complexity of training ReLU(s).
6. I. Diakonikolas, D. Kane, P. Manurangsi, and L. Ren. Hardness of learning a single neuron with
adversarial label noise.
7. I. Diakonikolas, S. Goel, S. Karmalkar, A. R. Klivans, and M. Soltanolkotabi. Approximation schemes
for ReLU regression.
8. I. Diakonikolas, V. Kontonis, C. Tzamos, and N. Zarifis. Learning a single neuron with adversarial label
noise via gradient descent.
9. P. Awasthi, A. Tang, and A. Vijayaraghavan. Agnostic learning of general ReLU activation using
gradient descent.
10. P. Wang, N. Zarifis, I. Diakonikolas, and J. Diakonikolas. Robustly learning a single neuron via
sharpness.
11. A. Gollakota, P. Gopalan, A. R. Klivans, and K. Stavropoulos. Agnostically learning single-index models
using omnipredictors.
12. N. Zarifis, P. Wang, I. Diakonikolas, and J. Diakonikolas. Robustly learning single-index models via
alignment sharpness.
13. A. Guo and A. Vijayaraghavan. Agnostic learning of arbitrary ReLU activation under Gaussian
marginals.
14. N. Zarifis, P. Wang, I. Diakonikolas, and J. Diakonikolas. Robustly Learning Monotone Generalized Linear Models
via Data Augmentation

---

> ### Author Rebuttal · Authors · 2025-07-30
>
> We thank the reviewer for their interest in our work and for the interesting questions.Below we address their concerns in the weaknesses.
>
>
> ---
> **Missing references:**
>
>  (a) **We agree that our original submission omitted several important references**. We are grateful to the reviewers for pointing out these works. We have now incorporated all the suggested citations, which better situate our paper within the existing literature and significantly strengthen the presentation. Specifically, we **added several references in both the introduction and the technical sections**. Additionally, we have written a "Related Work" section now. All this updates will be available in the final version.
>
> (b) We thank the reviewer for suggesting adding the “robust-learning” line of work to our related work section.
>
>
> ---
>
> **Answers to the questions are in order.**
>
> **LDP and SQ**:
> *  **LDP:** LDP is used to derive a **lower bound on the sample-size** required to have a polynomial-time algorithm (that is, derive the `computational threshold’ in the sample size). SQ is used to derive a **lower bound on the query complexity**, which we heuristically equate to a **lower bound on the runtime of the algorithm** (see answer to Reviewer 33AS).
>
> *  **SQ:** We can prove an upper bound on the tolerance in SQ—which is used as evidence for a *computational threshold* in the sample size—similarly as in [Damian et al, 2024]. Both predicts $d^{\ell/2}/\lVert \xi_{d,\ell}\rVert_2^2$ lower bound on the sample size when restricted to a given harmonic subspace.
>
> * LDP is not designed to give (fine-grained) runtime lower bounds, so we use SQ for this purpose, and we obtain $d^\ell/\lVert \xi_{d,\ell} \rVert_2^2$. **LDP and SQ are used to capture two different things and are not in contradiction.**
>
> * When looking for sample-optimal or runtime-optimal algorithms, we consider the subspace that minimizes the sample or runtime lower bound. This might be **two different subspaces**, which we interpret as providing evidence that no single algorithm can be both sample and runtime optimal.
>
> * **Algorithms as LDP and SQ:** Tensor unfolding is a **low-degree polynomial of the data and spectral algorithms fall in the LDP framework**. Thus, it is unsurprising that LDP predicts the correct sample complexity for tensor unfolding. (But there is a priori no reason to expect no detection-recovery gaps).
> **Online SGD is an SQ algorithm**, and it was noticed in several previous papers that **SQ predicts the correct runtime for these algorithms**. We do not know of a priori reason why tensor unfolding should have runtime nearly matching the SQ complexity lower bound.
>
> **Lemma 1:**
> In the Hermite-to-Gegenbauer expansion, $\xi_{d,\ell}$, for $\ell \leq k_*$ will **only depend on the $k_*$-th  hermite coefficient to leading order**. This is due to the fact that the projection of an Hermite-k on Gegenbauer of degree $\ell < k$ has $L^2$-norm vanishing as $d^{-(k-\ell)/2}$. Thus a likelihood ratio with GE = $k_*$ will have vanishing projections on spherical harmonics of degree $\ell < k_*$, and Lemma 1 bounds how fast they are vanishing as polynomials in $d$. **Despite the projection vanishing, for Gaussians, it is always optimal to use degree-$1$ or $2$ spherical harmonics.**
>
> [Damian et al, 2024] Computational-statistical gaps in gaussian single-index models A Damian, L Pillaud-Vivien, JD Lee, J Bruna

---

### Official Review · Reviewer_33AS · 2025-07-07

**Clarity:** 2
**Significance:** 3
**Originality:** 2
**Rating:** 5
**Confidence:** 3

**Summary:**

The paper analyzes the sample complexity of learning single-index models under spherically symmetric input distributions. Unlike most existing results that characterize the sample-complexity (when restricted to a class of algorithms) for Gaussian inputs through Hermite decomposition of targets/likelihood ratio, this work provides such a characterization through a decomposition along radial components and spherical harmonics. This, in turn, subsumes the existing results on Gaussian inputs.

The main contributions are twofold:
 * Lower bounds for SQ and low-degree polynomial methods.
* Spectral and online-SGD based algorithms achieving weak-recovery of the target direction with optimal sample-complexity and runtime, respectively.

Both the above contributions are based on the generalization of existing techniques, such as the analysis of the partial trace estimator and online-SGD from Gaussian to general spherical distributions. By providing run-time lower bounds for SQ and sample-complexity lower bounds through the low-degree polynomial framework, the paper further highlights a trade-off between achieving optimal sample-complexity vs optimal runtime.

**Questions:**

* Assumption 1 doesn't appear quite transparent. Inclusion of interpretation and examples behind the assumption would strengthen the paper.
* Definition 1 says $\mathcal{Y}$ can be an arbitrary-measurable space. Aren't the targets restricted to be real-valued (as specified in equation 1 in the introduction)?
* Concerning section 4 and observation 2, how does label smoothing exploit the radial components?
* Why is the SQ-based run-time lower bound expected to hold for a general class of algorithms? The paper would benefit from a brief discussion of the evidence behind this heuristic, as well as a more transparent presentation of it as a conjecture.

**Ethical Concerns:**

["NO or VERY MINOR ethics concerns only"]

**Final Justification:**

I recommend acceptance since the work contributes important theoretical insights and directions to the theory of learning in structured data. I expect the reviewers to incorporate the clarifications mentioned in their response in the revised version.

**Limitations:**

The work is primarily theoretical and does not pose direct societal risks. The paper would benefit from a more detailed discussion of the limitations I've listed in the review, such as the conjectural status of the run-time lower-bound prediction from the SQ framework.

**Paper Formatting Concerns:**

I haven't noticed any major formatting issues in the paper.

**Quality:**

3

**Strengths And Weaknesses:**

# Strengths:

* The paper is presented and organized well. The definitions are clean, and the setting is quite general. For instance, the non-zero Gegenbauer coefficients are not restricted to be $\Theta(1)$. Appendix A provides a nice interpretable summary of the central ideas behind the lower bounds and estimators.
* The ideas behind the extension of the results for Gaussian inputs to general spherically symmetric distributions, in particular the decomposition along irreducible representations of the symmetry group, could allow extension to other interesting distributions with symmetries such as permutationally-invariant data.
* Certain conclusions, such as the limitations of algorithms not exploiting the radial component, and the sample-complexity/runtime tradeoffs, are insightful and could benefit future related work.

# Weaknesses

* Limited novelty and scope: The analysis is largely based on extensions of existing techniques from Gaussian to spherically symmetric distributions. The paper would benefit from a discussion of the novel insights gained by this extension and the novel ideas/challenges in the proofs. For instance, the proofs for Theorems 1 and the spectral/tensor-based algorithms largely follow Damian et al. 2024, while the analysis of the SGD algorithm is largely based on Zweig et al. 2023.

* Inadequate discussion of literature: The paper does not cite or discuss a number of relevant works:

  Learning multi-index models in polynomial time through an SGD-based algorithm:

  Chen, Sitan, and Raghu Meka. "Learning polynomials in few relevant dimensions." Conference on Learning Theory. PMLR, 2020.

  Another missing line of work shows how the sample complexity of online-SGD can be improved to match the SQ sample-complexity through the re-use of data:

  Dandi, Y., Troiani, E., Arnaboldi, L., Pesce, L., Zdeborova, L., &; Krzakala, F. The Benefits of Reusing Batches for Gradient Descent in Two-Layer Networks: Breaking the Curse of Information and Leap Exponents. In Forty-first International Conference on Machine Learning.

  Lee, J. D., Oko, K., Suzuki, T., &amp; Wu, D. (2024). Neural network learns low-dimensional polynomials with sgd near the information-theoretic limit. Advances in Neural Information Processing Systems, 37, 58716-58756.

  Arnaboldi, L., Dandi, Y., Krzakala, F., Pesce, L., &amp; Stephan, L. (2024). Repetita iuvant: Data repetition allows sgd to learn high-dimensional multi-index functions. arXiv preprint arXiv:2405.15459.

---

> ### Author Rebuttal · Authors · 2025-07-30
>
> We thank the reviewer for their time in reviewing our work and overall positive feedback! Below we address their concerns in the weaknesses.
>
> ---
>
> We kindly disagree that this tensor-based method follows largely from Damian et al. (2024).
> This algorithm is designed for the cases when the sample optimal frequency $l_{m,\star}\geq 3$, which is a novel algorithm inspired from tensor PCA literature and does not follow from partial trace.  In particular, the partial trace algorithm always projects the Hermite tensor onto harmonic frequencies $\ell=1$ and $\ell=2$, depending on GE is odd or even. Thus, this algorithmic idea is not sufficient to achieve optimal sample complexity when $l_{m,\star}\geq 3$. (Note that this never happens in the Gaussian case cf. lines 105-108, but happens for general single-index models, e.g., with data uniform over the sphere.)  **Our estimator is different: it is based on tensor unfolding and not partial trace.** The dependency structure is much more intricate and requires a different and more involved analysis than for the partial trace estimator. Furthermore, our tensor unfolding estimator is non-standard and requires removing the diagonal. We further introduce **a harmonic tensor, which, to our knowledge, is novel in the machine learning literature and might be of independent interest.**
>
> **Novelty and scope:** We believe there might be some misunderstanding in evaluating our contributions in this work. Below, we list the contributions that we believe are novel and of interest to the community:
>
> * The **main conceptual novelty of this work is the introduction of the harmonic decomposition to study single-index models**. To the best of our knowledge, this is the first work that studies Gaussian single-index models through this lens: the generative exponent and the earlier information exponent are defined with respect to Hermite polynomials. Many analyses, including many works cited by the reviewers and less recent works on GE=2 (phase retrieval), are exploiting this Hermite basis in their proofs.
> We argue in this paper that the harmonic decomposition is the natural basis to study single index-models as:
>     * **It explicitly exploits the spherical symmetry of the problem.** We believe that this is an important and general observation, which will have important consequences when studying other similar `equivariant’ problems. (See point below on LDP and SQ lower bounds.)
>     * **It gives a more transparent derivation of optimal algorithms in the Gaussian setting**, by making explicit optimal statistics in terms of direction and norm of the Gaussian vector. (See point below on landscape smoothing and partial trace.)
>
> * As a consequence of the harmonic decomposition, we can study **general spherically symmetric input distributions**. To our knowledge, a fine-grained analysis of learning SIMs under general spherical distribution is novel. While this might appear incremental compared to the Gaussian distribution, we argue in the paper that this is not the case. It requires **novel analyses to study (harmonic tensor, Gegenbauer algebra) and it uncovers new and interesting phenomena that don’t appear with Gaussian data:** (1) novel statistical-computational trade-offs, (2) the importance of the norm of the input, (3) exploiting different degree spherical harmonics.
>
> * **Novel statistical-computational trade-offs in SIMs:** we provide evidence that for non-Gaussian input distributions, new statistical-computational trade-offs appear when optimal harmonic-degree for SQ and LDP are different. In this case, one can either be sample-optimal or runtime-optimal but not both. We further emphasize that this is a different phenomenon than typical smooth trade-offs discussed in the literature, e.g., in tensor PCA, where one can smoothly trade-off the polynomial degree of the estimator with sample complexity (as $m = d^{k} / D^{k/2 - 1}$).
> * **Importance of the norm:** In our opinion, this is the most surprising and novel observation. We provide evidence that In order to learn Gaussian SIM with optimal runtime, **one needs to exploit the norm, despite the norm not carrying any information on  $w_*$ and concentrating on $1$ as $d \to \infty$**. Thus, **normalizing the input data—which is a typical preprocessing step in statistics and machine learning—leads to suboptimal performance when learning Gaussian SIMs**. We emphasize that our harmonic analysis framework is necessary to uncover this phenomenon. As a side note, to prove this result required a novel hermite-to-Gegenbauer expansion (Appendix B.2).
>
> * **SQ and LDP lower bounds:** we kindly disagree with the reviewer’s assessment that Theorem 1 largely follows from Damian et al. 2024. While SQ and LDP lower bounds have now become routine computations and we do not deviate from the standard approach, our proofs are based on exploiting Gegenbauer properties and hypercontractivity. Importantly, we present a **novel decoupling of these lower bounds across harmonic subspaces**. To the best of our knowledge, this is a novel phenomenon and we heavily rely on this decomposition to design our optimal algorithms for learning SIMs. The fundamental structure behind this decoupling is that harmonic subspaces are **irreducible representations of the orthogonal group** (e.g., Gegenbauer polynomials can be seen as matrix coefficients of this representation, and the proof reduces to Schur orthogonality relations). Thus, our proof of LDP and SQ lower bounds directly generalizes to abstract groups because of this structure, which is not the case with a proof based on Hermite polynomials.
>
> * **Landscape smoothing and partial trace:** we show that even for Gaussian inputs, our framework offers novel and more transparent justifications for the success of existing algorithms.
>     * The partial trace estimator simply corresponds to projecting on the optimal degree-$1$ or $2$ harmonic subspaces, and is simply a spectral algorithm with the right transformation $T(y,r)$. While this does not modify the algorithm, we argue that **the harmonic decomposition is a particularly transparent derivation of this algorithm that does not require any tensors, Gaussian identities, or knowledge about the partial trace method**.
>     * **Landscape smoothing reweights the harmonic decomposition of the loss.** To the best of our knowledge, this is a novel interpretation of this algorithm that originates in the tensor PCA literature.
>     * Online SGD on hermite neurons **does not seem to be able to exploit the norm of the Gaussian input. It has optimal runtime among algorithms that do not exploit the norm.**
>
> **Despite this model being well-studied, we believe that our paper offers a novel and interesting perspective, rooted in harmonic analysis.** We believe that several aspects of our work—**symmetry-centric decomposition, decoupling into subproblems across irreducible subspaces, and harmonic tensor—will be of independent interest and apply beyond learning SIMs.**
>
> **Missing references:**  We thank the reviewer for providing these references, which we have included in the paper and it will be updated during the final version. A remark about reusing the batch line of work is that, while reusing batches might improve the sample complexity in online-SGD, it is conjectured that it can only improve from the information exponent (CSQ) to the generative exponent (SQ), which is what these three papers show. In our paper, we only consider the SQ: in this case, it is unclear whether multi-pass SGD can improve the sample complexity of our online SGD algorithm. We believe that this is an exciting research direction.
>
>
> **Questions:**
>
> * Assumption 1:  This assumption was confusing and unnecessary for the proof of the lower bound. We have replaced it with a mild (necessary) assumption that the SIM is learnable in polynomial time. Kindly see the response to Reviewer KRnG for the precise assumption.
> * Throughout the paper, we allow the label to live in a general measurable space (the label only appears in equations through $T(y)$, with general transformation $T:\mathcal{Y} \to \mathbb{R}$). We will modify the line below Equation 1, and not restrict $y$ to be a real number, even in this first equation.
> * Label Smoothing will change the decomposition of the link function into Spherical Harmonics. See the discussion in Appendix A.3, especially lines 844-848. For the Gaussian case, this operator essentially reweighs the frequency in the harmonic decomposition of the link function—smooths the higher-degree Gegenbauer and amplifies the contribution from lower-degree Gegenbauer.  Therefore, smoothing the population loss $L(w)=\mathbb{E}[T(y)He_k(<w,x>)]=\mathbb{E}[T(y) \sum_{\ell=0}^{k} \beta_{k,\ell}(r) Q_{\ell}(< w,z>)) ]$ turns out to be “roughly equivalent” to $\tilde{L}(w)=\mathbb{E}[T(Y)\beta_{k,2}(r)Q_2(<w,z>)]$ or  $\tilde{L}(w)=\mathbb{E}[T(y)\beta_{k,1}(r)Q_1(< w,z>)]$ depending on even or odd $k$. This is how it is able to exploit the norm. However, this turns out to be a highly indirect way of doing so -–the fact that the smoothing has a filtering effect and that it is able to exploit the norm nearly optimally is due to the Gaussianity and the relationship between Hermite and Gegenbauer basis. However, such a phenomenon does not hold more generally. Our online SGD algorithm, thus, directly try to fit the transformation $T(y,r)$ which exploits the norm more directly.
> * Indeed, this is an important point. We only briefly touch on it at the end of Section 2. This heuristic is based on the belief that $1/\tau^2$ runtime is necessary to compute one query to tolerance $\tau$. Similar to many other work in ML (which we will cite), we use these lower bounds as an indication of the best achievable runtime. We will expand and clarify this heuristic in the main text.

---

> > ### Comment · Reviewer_33AS · 2025-08-07
> >
> > I thank the authors for their detailed responses. In light of the clarifications regarding the novel aspects of the work, updated assumption, and inclusion of missing references, I’ve increased my score.

---

### Decision · Program_Chairs · 2025-09-17

**Decision:**

Accept (poster)

**Comment:**

This paper studies the task of learning single-index models under spherically symmetric distributions. Prior work had essentially resolved the complexity of this task under the Gaussian distribution, and this work provides a non-trivial generalization to a broader class of distributions. Specifically, the authors provide both Statistical Query lower bounds (giving evidence of inherent information-computation tradeoffs) and efficient SGD-style algorithms for weak recovery. Overall, this work was deemed interesting and technically worthy for acceptance by the reviewers.